# Benign Overfitting in Token Selection of Attention Mechanism

**Keitaro Sakamoto** [1]   **Issei Sato** [1]

## Abstract

Attention mechanism is a fundamental component of the transformer model and plays a significant role in its success. However, the theoretical understanding of how attention learns to select tokens is still an emerging area of research. In this work, we study the training dynamics and generalization ability of the attention mechanism under classification problems with label noise. We show that, with the characterization of signal-to-noise ratio (SNR), the token selection of attention mechanism achieves "benign overfitting", i.e., maintaining high generalization performance despite fitting label noise. Our work also demonstrates an interesting delayed acquisition of generalization after an initial phase of overfitting. Finally, we provide experiments to support our theoretical analysis using both synthetic and real-world datasets.

## 1. Introduction

The transformer models (Vaswani et al., 2017) have greatly succeeded across a wide range of fields, including natural language (Devlin et al., 2018; Brown et al., 2020), vision (Dosovitskiy et al., 2021; Touvron et al., 2021), and have become a foundational architecture in modern machine learning. The key component that characterizes the transformer model is the attention mechanism, which was originally introduced in recurrent neural network (RNN) and long short-term memory (LSTM) to capture the long-range structure of sequence (Bahdanau, 2014; Xu, 2015). The attention architecture can process variable-length input sequences and flexibly select important tokens based on the input. The seminal work on the training dynamics of this token selection mechanism (Tarzanagh et al., 2023a;b; Vasudeva et al., 2024; Sheen et al., 2024) studied the implicit bias of gradient descent to the max-margin token separator. However,

it remains unclear whether such max-margin solutions actually generalize well. In contrast, theoretical studies such as (Jelassi et al., 2022; Li et al., 2023a; Jiang et al., 2024) analyzed the generalization ability of attention mechanisms, but in the data models assumed by these works, token selection proceeds in the same direction for all training samples. Under annotation errors or adversarial label flips, the behavior of token selection in both clean and noisy samples and its impact on generalization remain unclear. These existing studies raise the following research questions: (Q1) *How do the training dynamics of token selection evolve under label noise?* (Q2) *Does the obtained solutions generalize well?*

To this end, we analyze the token selection of attention mechanisms in the context of benign overfitting studies, which allows us to study the training dynamics of gradient descent and the generalization performance. Modern overparameterized neural networks achieve a high generalization while perfectly fitting to the training data (Zhang et al., 2021). This "benign overfitting" phenomenon has attracted attention over the past few years because it contrasts with the conventional wisdom that achieving better generalization requires balancing training error and model complexity. There are lines of studies analyzing the benign overfitting phenomenon in various settings, including linear classification and two-layer neural networks, but the analysis of benign overfitting is mostly limited to these types of architecture. In this paper, following the common setup in existing attention work (Tarzanagh et al., 2023a;b; Oymak et al., 2023; Sheen et al., 2024), we analyze a one-layer attention network $f(\mathbf{X}; \mathbf{W}, \mathbf{p}) = \boldsymbol{\nu}^\top \mathbf{X}^\top \mathbb{S}(\mathbf{X}\mathbf{W}^\top \mathbf{p}) \in \mathbb{R}$, where $\mathbb{S}(\cdot)$ is the softmax function, $\mathbf{X} = (\mathbf{x}_1, \ldots, \mathbf{x}_T)^\top \in \mathbb{R}^{T \times d}$ is the sequence of input tokens, $\mathbf{W} \in \mathbb{R}^{d \times d}$ is the trainable key-query matrix, $\mathbf{p} \in \mathbb{R}^d$ is a tunable token, and $\boldsymbol{\nu} \in \mathbb{R}^d$ is a pretrained linear head, on a binary classification task. Here, $\mathbf{p}$ corresponds to the trainable [CLS] token (Devlin et al., 2018; Dosovitskiy et al., 2021) or prompt tuning (Li & Liang, 2021; Lester et al., 2021) in the application of transformers, and $f$ represents the output at that token position. To isolate the role of benign overfitting in token selection, we fix the linear classifier $\boldsymbol{\nu}$, and focus our analysis on the training dynamics and generalization behavior that arise solely from optimizing the attention mechanism.

[1]Department of Computer Science, The University of Tokyo, Tokyo, Japan. Correspondence to: Keitaro Sakamoto <sakakei-1999@g.ecc.u-tokyo.ac.jp>, Issei Sato <sato@g.ecc.u-tokyo.ac.jp>.

*Proceedings of the 42^{nd} International Conference on Machine Learning*, Vancouver, Canada. PMLR 267, 2025. Copyright 2025 by the author(s).

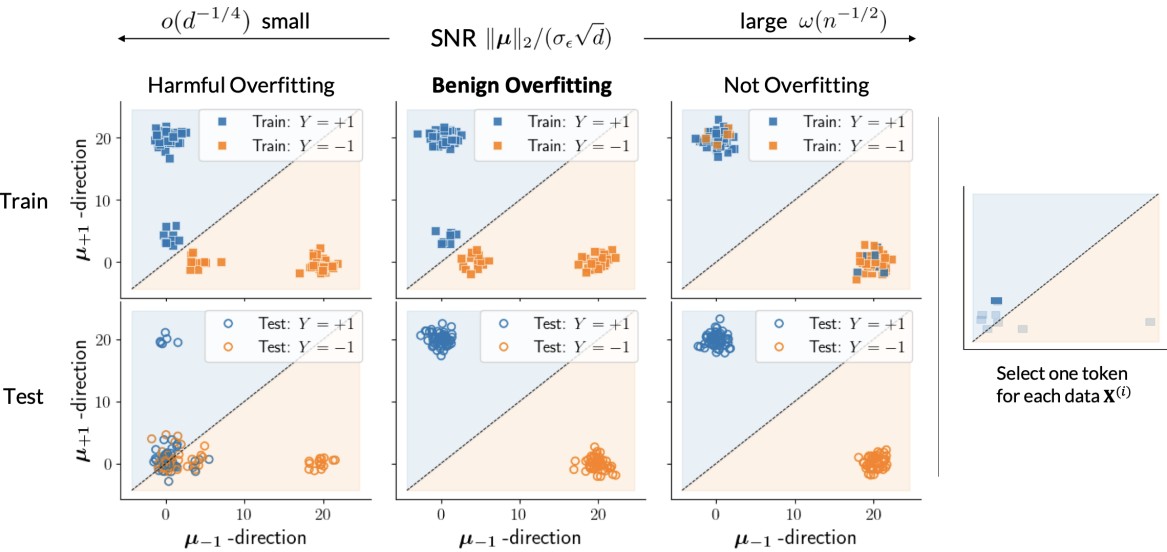

Figure 1: Projection of one selected token per sequence. Each point indicates $\mathbf{x}_t^{(i)}$ selected by attention from each input $\mathbf{X}^{(i)} = (\mathbf{x}_1^{(i)}, \dots, \mathbf{x}_T^{(i)})^{\top}$ in the direction of class signals $\boldsymbol{\mu}_{+1}$ and $\boldsymbol{\mu}_{-1}$ for the three scenarios of harmful overfitting, benign overfitting, and not overfitting. **Top:** Training data with label noise. **Bottom:** Test data. The decision boundary is common because the head $\boldsymbol{\nu}$ is fixed, but **the model can select an appropriate token that belongs to the desired output region.** Here, $\|\boldsymbol{\mu}\|_2$ denotes the strength of the class signal, $d$ is the dimension of the data and parameters, $\sigma_\epsilon^2$ is the variance of the input noise, and $n$ is the size of the training set.

**Contribution.** We show that, under the condition based on SNR, benign overfitting occurs in the token selection mechanism when optimizing $\mathbf{W}$ and $\mathbf{p}$ with gradient descent from random initialization. Specifically, when SNR is high, the model disregards fitting to noisy samples and fits only clean samples, and the generalization ability is high (not overfitting case). As SNR decreases, the model performs distinct token selection for clean and noisy training samples, fitting both while still generalizing well even in the presence of label noise (benign overfitting case). This reflects a balance where sufficient class signal strength is required for generalization, while noise memorization is necessary for fitting noisy samples. Figure 1 illustrates the benign overfitting in token selection.

Technically, we provide a method to evaluate the evolution of the softmax probabilities for each training example, both from above and below. We need it because the learning direction of the class signal is not determined solely by the number ratio of clean to noisy data but requires careful evaluation of the softmax probability ratio at each time step. Moreover, our results demonstrate that grokking (Power et al., 2022; Nanda et al., 2023) occurs in the benign overfitting case. While fitting the training data is achieved at a relatively early stage of training, further training in exponential order is required to learn the class signals sufficiently and reduce the generalization error. In this paper, we consider binary classification for simplicity of discussion, but as shown in Appendix E.1, it can be extended to the multi-class

setting without fundamentally modifying the argument.

## 2. Related Work

**Token-selection in Attention.** The theoretical analyses on the attention mechanism from the perspectives of generalization ability (Jelassi et al., 2022; Li et al., 2023a;b; 2024a) and training dynamics (Tian et al., 2023; 2024) have been progressing in recent years. The most related line of work to ours deals with the implicit bias of gradient descent for a one-layer attention model (Tarzanagh et al., 2023a;b; Li et al., 2024b; Vasudeva et al., 2024; Sheen et al., 2024). While our work follows these studies in terms of model settings, their primary focus is the training dynamics without generalization analysis. Furthermore, our work is also influenced by the problem setting of (Oymak et al., 2023). However, they analyze the initial few steps of training under the data model without label noise. As for the technical differences in proof, we show in Appendix A that our results cannot simply be obtained by adapting their analysis to the context of benign overfitting. Table 1 provides a concise comparison of the analysis focus with the existing studies of the attention mechanism for classification.

**Benign Overfitting.** The success of modern overparameterized models has led to numerous studies attempting to understand why and when benign overfitting occurs. The analysis is interesting because the standard generaliza-

Table 1: Comparison of the theoretical works of the attention mechanism on the classification task. "Label Noise" indicates whether distant token selections occur depending on the training example.

| Paper | Dynamics | Generalization | Label Noise | Token Type |
|---|---|---|---|---|
| (Tarzanagh et al., 2023a;b) | ✓ | | - | One optimal, the others non-optimal |
| (Vasudeva et al., 2024) | ✓ | | - | One optimal, the others non-optimal |
| (Sheen et al., 2024) | ✓ | | - | One optimal, the others non-optimal |
| (Oymak et al., 2023) | | ✓ | | Relevant and irrelevant tokens |
| (Li et al., 2023a) | ✓ | ✓ | | Relevant and irrelevant tokens |
| (Jiang et al., 2024) | ✓ | ✓ | | One relevant, the others irrelevant |
| (Magen et al., 2024) | | ✓ | ✓ | One relevant, the others irrelevant |
| **Ours** | ✓ | ✓ | ✓ | One relevant, the others weakly or irrelevant |

tion bound based on uniform convergence does not explain this phenomenon well (Nagarajan & Kolter, 2019). For comprehensive surveys of the literature on this topic, please see work such as Bartlett et al. (2021); Belkin (2021).

Benign overfitting in regression has been studied in the linear model (Bartlett et al., 2020; Hastie et al., 2022) and kernel regression (Liang & Rakhlin, 2020; Tsigler & Bartlett, 2023). It is complicated to analyze the classification task because the explicit formula for the min-norm separator is not obtained. One approach is to track the training dynamics with gradient descent in the linear classifier and two-layer neural network (Chatterji & Long, 2021; Frei et al., 2022; Xu & Gu, 2023; Cao et al., 2022; Kou et al., 2023; George et al., 2023; Meng et al., 2024; Xu et al., 2024). Another line of work is built on the results of implicit bias to the max-margin solution (Cao et al., 2021; Wang et al., 2021; Wang & Thrampoulidis, 2022; Frei et al., 2023a). These studies base their discussion of convergence on existing research on implicit bias (Soudry et al., 2018; Ji & Telgarsky, 2019; Lyu & Li, 2020; Frei et al., 2023b). Specifically, they analyze the properties of the solution through the KKT conditions of the max-margin problem in order to examine whether the solution at convergence shows benign overfitting or not. Furthermore, there is a research direction in investigating the degree of benignity of overfitting (Mallinar et al., 2022; Wen et al., 2023; Kornowski et al., 2024).

Recently, Jiang et al. (2024) studied benign overfitting in a simplified vision transformer model. A major difference is that our analysis is under label noise, where the direction of token selection differs between clean and noisy data in the same training run, and signal learning for generalization competes between clean and noisy data. Concurrently, Magen et al. (2024) has also analyzed benign overfitting in a similar model setting; however, our analysis is beyond the initial few training steps and is conducted on a more general data model. Furthermore, our research put more focus on the token selection mechanism than these studies, demonstrating that benign overfitting can be achieved

solely through optimization within the attention mechanism. Please refer to Appendix A for further details.

## 3. Problem Setting

In this section, we introduce the notation and the problem settings in the rest of the paper.

### 3.1. Notations

Let $[n]$ be a shorthand for the set $\{1, \ldots, n\}$. We denote a multivariate Gaussian distribution with mean vector $\boldsymbol{\eta}$ and covariance matrix $\boldsymbol{\Sigma}$ by $N(\boldsymbol{\eta}, \boldsymbol{\Sigma})$. Denote by $\mathbb{S} : \mathbb{R}^T \to \mathbb{R}^T, \mathbb{S}(\mathbf{v})_t = \exp(\mathbf{v}_t) / \sum_{t' \in [T]} \exp(\mathbf{v}_{t'})$ the softmax function. The standard Big-O notations $O(\cdot), \Theta(\cdot), \Omega(\cdot), o(\cdot)$, and $\omega(\cdot)$ are used to hide absolute constants, and we denote inequality ignoring constant factors by $\gtrsim, \lesssim$.

### 3.2. Data Model

In the analysis of benign overfitting, we typically need to consider the specific shape of the data distribution to evaluate the generalization error without using a uniform convergence argument. We consider the following data distribution $P$ defined over $(\mathbf{X}, Y) \in \mathbb{R}^{T \times d} \times \{\pm 1\}$. In this paper, we consider binary classification for simplicity of discussion, but the same argument applies to the multi-class case. Please refer to Section E.1 in the appendix for more details.

**Definition 3.1.** Let $\boldsymbol{\mu}_{+1}, \boldsymbol{\mu}_{-1} \in \mathbb{R}^d$ be fixed class signal vectors satisfying $\|\boldsymbol{\mu}\|_2 = \|\boldsymbol{\mu}_{+1}\|_2 = \|\boldsymbol{\mu}_{-1}\|_2$ and $\langle \boldsymbol{\mu}_{+1}, \boldsymbol{\mu}_{-1} \rangle = 0$. The input $\mathbf{X} = [\mathbf{x}_1, \ldots, \mathbf{x}_T]^\top \in \mathbb{R}^{T \times d}$ has $T$ tokens that are split into three groups: *relevant token* $\mathcal{R} = \{1\}$ containing the strong signal for true class, *weakly relevant token* $\mathcal{W} \subseteq [T] \setminus \mathcal{R}$ containing weak class signals, and *irrelevant token* $\mathcal{I} = [T] \setminus (\mathcal{R} \cup \mathcal{W})$ containing only noise. Let clean distribution $P^*$ be the distribution over $\mathbb{R}^{T \times d} \times \{\pm 1\}$ such that $(\mathbf{X}, Y^*)$ is sampled as follows:

1. The clean label $Y^*$ is sampled from $\text{Unif}(\{\pm 1\})$.

2. The noise vectors $(\boldsymbol{\epsilon}_t)_{t \in [T]}$ are sampled independently

from $N(\mathbf{0}, \sigma_\epsilon^2 \mathbf{I}_d)$.

3. The relevant token is given by $\mathbf{x}_1 = \boldsymbol{\mu}_{Y^*} + \boldsymbol{\epsilon}_1$.

4. The weakly relevant tokens $\mathbf{x}_u$ for $u \in \mathcal{W}$ are given by $\mathbf{x}_u = \rho \boldsymbol{\mu}_{w_u} + \boldsymbol{\epsilon}_u$, where $\rho \ll 1$ is a small scale parameter and $w_u \in \{\pm 1\}$ for $u \in \mathcal{W}$. For simplicity, we assume single confusing token $\mathbf{x}_2 = \rho \boldsymbol{\mu}_{-Y^*} + \boldsymbol{\epsilon}_2$ and other tokens $\mathbf{x}_u = \rho \boldsymbol{\mu}_{Y^*} + \boldsymbol{\epsilon}_u$ for $u \in \mathcal{W} \setminus \{2\}$.

5. The irrelevant tokens are given by $\mathbf{x}_v = \boldsymbol{\epsilon}_v$ for $v \in \mathcal{I}$.

We use the notation $\mathcal{W}_{+1} = \{u \in \mathcal{W} \mid w_u = +1\}$ and $\mathcal{W}_{-1} = \{u \in \mathcal{W} \mid w_u = -1\}$. The data distribution $P$ is defined as the label-corrupted version of $P^*$ with the level of label noise $\eta > 0$. The data point $(\mathbf{X}, Y)$ from $P$ is generated by first sampling $(\mathbf{X}, Y^*)$ from clean distribution $P^*$ and then setting $Y = -Y^*$ with probability $\eta$ and $Y = Y^*$ with probability $1 - \eta$. We denote signal-to-noise ratio by $\mathrm{SNR} = \|\boldsymbol{\mu}\|_2 / (\sigma_\epsilon \sqrt{d})$. For simplicity of notation, we assumed a fixed scale $\rho$ for weakly relevant tokens, but the analysis holds even if the scale varies across examples.

Training data $S = (\mathbf{X}^{(i)}, Y^{(i)})_{i=1}^n$ are sampled i.i.d. from $P$. We denote the clean data $\{i \in [n] \mid Y^{(i)} = Y^{*(i)}\}$ and noisy data $\{i \in [n] \mid Y^{(i)} \neq Y^{*(i)}\}$ by $\mathcal{C}$ and $\mathcal{N}$, respectively. The set of data $\{i \in \mathcal{C} \mid Y^{(i)} = 1\}$ are denoted as $\mathcal{C}_+$, and $\{i \in \mathcal{C} \mid Y^{(i)} = -1\}$ are denoted as $\mathcal{C}_-$. The same notation is applied to $\mathcal{N}$. The superscript $(i)$ denotes that the variable corresponds to the training data $i \in [n]$. For instance, we write $\mathbf{x}_t^{(i)}$ for $t \in [T]$, $\mathcal{W}^{(i)}$ and $\mathcal{I}^{(i)}$.

Data models based on signal and noise widely appear in the existing benign overfitting studies (Chatterji & Long, 2021; Frei et al., 2022; Cao et al., 2022; Jiang et al., 2024; Meng et al., 2024). Such data models based on signal and noise are not limited to benign overfitting works but are also commonly observed in other analyses of attention architecture (Jelassi et al., 2022; Li et al., 2023a; Oymak et al., 2023).

**Remark 3.2** (Weakly relevant token and label noise)**.** Weakly relevant tokens represent tokens with weak signal strength and confusing class information, reflecting a more realistic scenario than a clean separation into relevant and irrelevant tokens. Furthermore, such weak class information, including that which is confusing, is likely to lead to lower annotation quality, making it more plausible to consider the presence of label noise.

### 3.3. Attention Model

Given a sequential input $\mathbf{X} = (\mathbf{x}_1, \ldots \mathbf{x}_T)^\top \in \mathbb{R}^{T \times d}$, a single-head self-attention layer $f_{\mathrm{SA}} : \mathbb{R}^{T \times d} \to \mathbb{R}^{T \times m}$ is

$$f_{\mathrm{SA}}(\mathbf{X}) = \mathbb{S}(\mathbf{X} \mathbf{W}_Q \mathbf{W}_K^\top \mathbf{X}^\top) \mathbf{X} \mathbf{W}_V \mathbf{W}_o,$$

with trainable weights $\mathbf{W}_Q, \mathbf{W}_K, \mathbf{W}_V \in \mathbb{R}^{d \times d}$, and $\mathbf{W}_o \in \mathbb{R}^{d \times m}$. Here, the softmax function $\mathbb{S}(\cdot)$ is applied row-wise with the abuse of notation.

In practice, an additional tunable token $\mathbf{p} \in \mathbb{R}^d$ is concatenated to the input, and this position is used for the model prediction. This setup is widely used in, for example, the classification token [CLS] in BERT (Devlin et al., 2018) and ViT (Dosovitskiy et al., 2021), and prompt-tuning technique (Li & Liang, 2021; Lester et al., 2021). Let the concatenated input be $\mathbf{X}_{\mathbf{p}} := [\mathbf{p}, \mathbf{X}^\top]^\top \in \mathbb{R}^{(T+1) \times d}$; then the cross-attention feature between $\mathbf{X}_{\mathbf{p}}$ and $\mathbf{X}$ is given by

$$\begin{bmatrix} f(\mathbf{X})^\top \\ f_{\mathrm{SA}}(\mathbf{X}) \end{bmatrix} = \mathbb{S}(\mathbf{X}_{\mathbf{p}} \mathbf{W} \mathbf{X}^\top) \mathbf{X} \mathbf{W}_V \mathbf{W}_o$$

$$= \begin{bmatrix} \mathbb{S}(\mathbf{p}^\top \mathbf{W} \mathbf{X}^\top) \\ \mathbb{S}(\mathbf{X} \mathbf{W} \mathbf{X}^\top) \end{bmatrix} \mathbf{X} \mathbf{W}_V \mathbf{W}_o,$$

where we use $\mathbf{W}$ to denote a key-query weight matrix $\mathbf{W}_Q \mathbf{W}_K^\top$, and the output corresponding to the position of $\mathbf{p}$ is denoted by $f(\mathbf{X}) = \mathbf{W}_o^\top \mathbf{W}_V^\top \mathbf{X}^\top \mathbb{S}(\mathbf{X} \mathbf{W}^\top \mathbf{p}) \in \mathbb{R}^m$. In this work, we use the model output for binary classification, leading to the output dimension being $m = 1$, and we denote the value prediction head by $\boldsymbol{\nu} = \mathbf{W}_V \mathbf{W}_o \in \mathbb{R}^d$. Therefore, the model under our analysis is of the form

$$f(\mathbf{X}) = \boldsymbol{\nu}^\top \mathbf{X}^\top \mathbb{S}(\mathbf{X} \mathbf{W}^\top \mathbf{p}). \tag{1}$$

The output can be regarded as an affine combination of the token scores $\{\gamma_t := \boldsymbol{\nu}^\top \mathbf{x}_t\}_{t \in [T]}$, using the learned softmax probabilities. Let $\mathbf{s} \in \mathbb{R}^T$ be a shorthand for the softmax vector $\mathbb{S}(\mathbf{X} \mathbf{W}^\top \mathbf{p})$.

To clarify the role of token selection in the attention mechanism with respect to benign overfitting, we fix the linear classifier $\boldsymbol{\nu}$, for which benign overfitting has already been extensively studied (Bartlett et al., 2020; Chatterji & Long, 2021), and analyze the training dynamics and generalization behavior arising solely from the optimization of the token selection mechanism inside the softmax. To formulate the token selection problem based on given token scores $\{\gamma_t\}_{t \in [T]}$, we assume a pretrained head $\boldsymbol{\nu}$ that assigns appropriate scores to each token. Specifically, we consider $\boldsymbol{\nu}$ such that

$$k \cdot \cos \theta_k > \Theta(1), \tag{2}$$

for $k \in \{\pm 1\}$, where $\theta_k$ denotes the angle between $\boldsymbol{\nu}$ and $\boldsymbol{\mu}_k$. Jointly training $\boldsymbol{\nu}$ is itself an intriguing setting, but it would obscure whether benign overfitting occurs in the softmax weights or in the linear classifier. The analytical setup in this paper enables a stronger claim: that noise memorization and benign overfitting can occur solely within the token selection mechanism. We further discuss the rationale behind Equation (2) in Appendix E.2.

**Remark 3.3** (Relevance to practical scenarios). The analytical setup is relevant to practical scenarios, especially in the context of parameter-efficient fine-tuning. For example, prompt-tuning (Li & Liang, 2021; Lester et al., 2021) trains only the tunable input tokens, and LoRA (Hu et al., 2022) focuses on training only the attention weights. The results on the training dynamics and the generalization performance provide a guarantee regarding the required training time and the generalization when applying the model to low-quality downstream tasks containing label noise.

### 3.4. Gradient-descent Training

The learnable parameters $(\mathbf{W}, \mathbf{p})$ are trained to minimize the empirical risk objective:

$$\widehat{\mathcal{L}}(\mathbf{W}, \mathbf{p}) = \frac{1}{n} \sum_{i=1}^{n} \ell\left(Y^{(i)} \cdot f(\mathbf{X}^{(i)})\right), \qquad (3)$$

$$\ell(z) = \log(1 + \exp(-z)), \qquad (4)$$

where $\ell : \mathbb{R} \to \mathbb{R}$ is a binary cross-entropy loss.

Let each element of $\mathbf{W}$ and $\mathbf{p}$ be initialized as $\mathbf{W}(0)_{i,j} \sim N(0, \sigma_w^2)$ and $\mathbf{p}(0)_i \sim N(0, \sigma_p^2)$, respectively. The parameters are optimized by gradient descent with a step size $\alpha > 0$:

$$\mathbf{W}(\tau + 1) = \mathbf{W}(\tau) - \alpha \nabla_{\mathbf{W}} \widehat{\mathcal{L}}(\mathbf{W}(\tau), \mathbf{p}(\tau)), \quad (5)$$

$$\mathbf{p}(\tau + 1) = \mathbf{p}(\tau) - \alpha \nabla_{\mathbf{p}} \widehat{\mathcal{L}}(\mathbf{W}(\tau), \mathbf{p}(\tau)). \qquad (6)$$

Let $f_\tau$ be the $f$ after the $\tau$ gradient descent step and $\widehat{\mathcal{L}}(\tau)$ be a shorthand for $\widehat{\mathcal{L}}(\mathbf{W}(\tau), \mathbf{p}(\tau))$. The weight updates with specifically calculated loss gradients are provided in Appendix B.1.

### 3.5. Assumption on Parameters

In this section, we first discuss the necessity of the assumptions on parameters and compare them with existing studies, followed by a list of the assumptions used in our work.

The condition on $d$ in Assumption A1 is necessary for training in an over-parameterized setting, and similar terms $n\|\boldsymbol{\mu}\|_2^2$ can be found in (Chatterji & Long, 2021; Frei et al., 2022; Xu & Gu, 2023; Kou et al., 2023). Assumption A2 is required for generalization. The lower bound with $d^{1/4}$ appears in (Xu & Gu, 2023), and $n^{1/4}d^{1/4}$ and $nd^{1/4}$ are found in (Xu et al., 2024; Meng et al., 2024), respectively. While these conditions may seem intricate, the key aspect in the analysis is the relationship among $d$, $\|\boldsymbol{\mu}\|_2$, and $n$. Assumption A3 is a natural setting to represent weak class information. The lower bound is set such that the weak signal strength is on a larger scale than the inner product of the class signal and a random noise vector. Assumption A4 is for a sufficiently small learning rate, which is widely set

in the existing studies, including Frei et al. (2022); Cao et al. (2022); Jiang et al. (2024). Assumptions A5 and A6 are the common assumptions to evaluate the class balance in the training data and the amount of noisy data. For example, $n = \Omega(\text{polylog}(d))$ is assumed in (Cao et al., 2022; Jiang et al., 2024). Assumption A7 is put to focus on the dependencies on $d$, $\|\boldsymbol{\mu}\|_2$ and $n$ in the asymptotic notation, following (Jiang et al., 2024). Finally, Assumption A8 ensures that the attention probabilities at initial weights are reasonably uniform, and this type of condition is observed in existing studies, including other architectures (Cao et al., 2022; Kou et al., 2023; Meng et al., 2024; Jiang et al., 2024).

Now, we state our assumptions in the following. Let $\hat{\sigma}_\epsilon = \max\{\sigma_\epsilon, 1/\sigma_\epsilon\}$. Given a small failure probability $\delta > 0$ and a large enough universal constant $C$, we make the assumptions for each parameter as follows:

$$d \geq C\hat{\sigma}_\epsilon n\|\boldsymbol{\mu}\|_2^{4/3} \log^3(Tn/\delta), \qquad (A1)$$

$$\|\boldsymbol{\mu}\|_2 \geq C\sigma_\epsilon d^{3/8} \log(Tn/\delta), \qquad (A2)$$

$$C\|\boldsymbol{\mu}\|_2^{-1}\sigma_\epsilon \log(Tn/\delta) \leq \rho \leq 1/C, \qquad (A3)$$

$$\alpha \leq \max\{\|\boldsymbol{\mu}\|_2\sqrt{d}, \sigma_\epsilon d\}^{-1}/C, \qquad (A4)$$

$$n \geq C \log(d/\delta), \qquad (A5)$$

$$\eta \leq 1/C, \qquad (A6)$$

$$T = \Theta(1), \qquad (A7)$$

$$\sigma_w^2, \sigma_p^2 = \Theta\left(\max\{\|\boldsymbol{\mu}\|_2\sqrt{d}, \sigma_\epsilon d\}^{-1} \log^{-2}(Tn/\delta)\right). \qquad (A8)$$

## 4. Main Results

In this section, we provide the main results regarding the training dynamics and generalization of attention mechanisms. The key techniques used in the proof are presented in Section 4.1. By dividing ranges for the signal-to-noise ratio, we obtain the following results for the convergence.

**Theorem 4.1.** *Suppose that the norm of the linear head scales as $\|\boldsymbol{\nu}\|_2 = O(1/\|\boldsymbol{\mu}\|_2)$. Under the parameter assumptions in Section 3.5, we have*

1. *(Not Overfitting) If $\text{SNR}^2 = \omega(n^{-1})$, then with probability at least $1 - \delta$, there exists a time step $\tau = \Theta\left(\frac{1}{\alpha\|\boldsymbol{\nu}\|_2\|\boldsymbol{\mu}\|_2^3 d \max\{\sigma_w^2, \sigma_p^2\}}\right)$ such that the weights $(\mathbf{W}(\tau), \mathbf{p}(\tau))$ fit only the clean data, and the test loss is sufficiently low:*

$$\forall i \in \mathcal{C}, \ f_\tau(\mathbf{X}^{(i)}) = Y^{(i)},$$

$$\forall j \in \mathcal{N}, \ f_\tau(\mathbf{X}^{(j)}) \neq Y^{(j)},$$

$$\Pr_{(\mathbf{X}, Y^*) \sim P^*}\left[\text{sign}\left(f_\tau(\mathbf{X})\right) \neq Y^*\right] < \delta.$$

2. *(Benign Overfitting) If $\text{SNR}^2 = o(n^{-1})$, then with probability at least $1 - \delta$, there exists a time step*

$\tau = \Theta\left(\frac{\exp(n^{-1}\mathrm{SNR}^{-2})}{\alpha n^{-1}\sigma_\epsilon^2 \|\boldsymbol{\nu}\|_2 \|\boldsymbol{\mu}\|_2 d^2 \max\{\sigma_w^2, \sigma_p^2\}}\right)$ *such that the weights $(\mathbf{W}(\tau), \mathbf{p}(\tau))$ overfit the training data, and the test loss is sufficiently low:*

$$\forall i \in [n],\ f_\tau(\mathbf{X}^{(i)}) = Y^{(i)},$$
$$\Pr_{(\mathbf{X}, Y^*)\sim P^*}\left[\mathrm{sign}\left(f_\tau(\mathbf{X})\right) \neq Y^*\right] < \delta.$$

The assumption $\|\boldsymbol{\nu}\|_2 = O(1/\|\boldsymbol{\mu}\|_2)$ in the theorem ensures that the token scores remain bounded by a constant. This condition can be easily satisfied by appropriately scaling down the model output.

This theorem demonstrates that under the parameter assumptions in Section 3.5, the model does not fit noisy data and achieves high generalization accuracy with high SNR, and the model shows benign overfitting with low SNR. In both cases, high generalization is achieved despite the presence of label noise. Since both the signal required for generalization and the noise memorization for fitting label noise are significant, their balance is essential for benign overfitting. An upper bound on this balance is provided in the statement of the theorem, while the lower bound is determined by the parameter assumption A2. Specifically, by extracting the relationship among $\|\boldsymbol{\mu}\|_2$, $d$, and $n$, we see that $\mathrm{SNR}^2 = \Omega(d^{-1/4})$ holds in this benign overfitting case.

**Remark 4.2** (Harmful Overfitting). Theorem 4.1 presents cases of not overfitting and benign overfitting, where the test loss is sufficiently low. With a lower SNR that violates Assumption A2, specifically $\mathrm{SNR}^2 = o(d^{-1/2})$, even inner products among the random noises in the input dominate the selection of class signal (see Lemma B.1 in the appendix for details). It becomes inherently challenging to select class-relevant tokens and generalize effectively. The low SNR case corresponds to the left column of Figure 1.

**Remark 4.3** (Implication for Grokking). The time step demonstrated in Theorem 4.1 is also an important result. In the case of benign overfitting, fitting the training set requires a similar time step order to the not-overfitting case; however, to sufficiently learn the signal components and achieve high generalization, the exponential term of the SNR in the numerator is necessary. This delayed generalization ability after fitting the training set implies a connection to the phenomenon of grokking (Power et al., 2022; Nanda et al., 2023). For the necessity of this exponential term, please refer to the end of Section 4.1.

We illustrate the differences between the two scenarios of the main theorem using Figure 2. Selecting the class-relevant token $\mathbf{x}_1$, i.e., increasing $s_1$ towards 1, leads to fitting clean data but fails to adapt to noisy labels. For clean data, both signal learning and noise memorization contribute to increasing $s_1$, whereas for noisy data, these two processes are in competition. The learning direction is determined by

the strength of SNR, which leads to the different cases in Theorem 4.1. The middle in Figure 2 corresponds to the not-overfitting case, where signal learning dominates and the model does not fit noisy data. The right figure illustrates the benign overfitting case, where noise memorization becomes dominant. The vertical axis in Figure 2 represents the value of $s_1(1 - s_1)$, which determines the amount of parameter updates. As attention probabilities become more concentrated around 0 or 1, this value decreases, reducing the influence of the example on parameter updates. This makes analyzing the token selection dynamics inherently challenging. Our analysis under the label noise setting must account for *two competing training directions* within the same training run: 1) between signal learning and noise memorization (Figure 2), and 2) between clean and noisy samples for learning the class signals. These balances depend on softmax probabilities and are not determined by pre-training quantities such as SNR or label noise $\eta$. This is a specific difficulty with the attention mechanism, which is absent in existing benign overfitting studies. For instance, depending on the convergence speed—how quickly $s(\tau)$ approaches 0 or 1—it is possible that even when label noise $\eta$ is quite small, the actual contribution to the weight updates at some time step can be dominated by noisy samples. This motivates us to carefully analyze the dynamics of softmax probabilities to evaluate the direction of these competing relationships.

### 4.1. Key Techniques

In this section, we present the key techniques used in the proof of Theorem 4.1. The whole proof is provided in Appendix C. The proof for the not-overfitting case in the theorem is more straightforward and is presented first in the complete proof. In this section, to capture the essence of the proof in the main text, we focus on the more non-trivial benign overfitting case, i.e., $\mathrm{SNR}^2 = o(n^{-1})$.

In the analysis of benign overfitting, it is necessary to track the model behavior on the training data while also its generalization ability. We begin by presenting the results on training dynamics, and the generalization result is shown at the end of this section in Lemma 4.9. To this end, we introduce the following values that represent the relative strength of token selection of the learned weights. This quantity is useful for evaluating the softmax probability at each time step $\tau$ for a given training example.

**Definition 4.4** (Attention gap). We define the attention gap between a significant token and other tokens as follows:

$$\Lambda_{i,t}(\tau) := \left(\mathbf{x}_1^{(i)} - \mathbf{x}_t^{(i)}\right)^\top \mathbf{W}(\tau)^\top \mathbf{p}(\tau),$$
$$\Gamma_{i,u}(\tau) := \left(\mathbf{x}_2^{(i)} - \mathbf{x}_u^{(i)}\right)^\top \mathbf{W}(\tau)^\top \mathbf{p}(\tau),$$

for $i \in [n]$, $t \in [T] \setminus \{1\}$, and $u \in [T] \setminus \{2\}$.

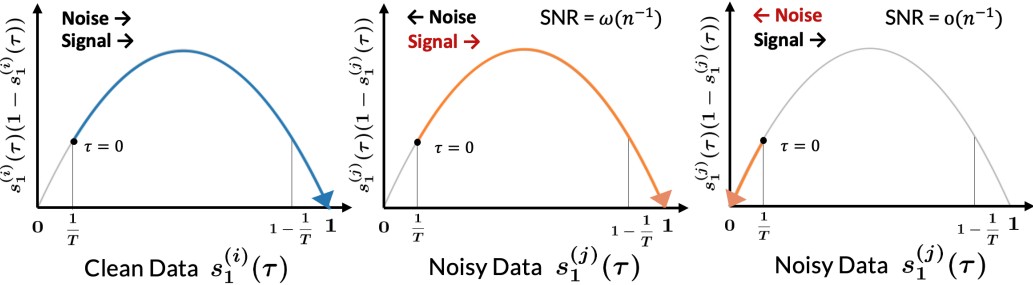

Figure 2: Illustration for the training dynamics of the probability assigned to relevant tokens $s_1$ in clean data $i \in \mathcal{C}$ and noisy data $j \in \mathcal{N}$. The y-axis shows $s_1(\tau)(1 - s_1(\tau))$, which determines the magnitude of the gradient descent update. This value converges to 0 as $s_1(\tau)$ approaches 0 or 1, and consequently, the contribution of this training example to the gradient update diminishes. The middle figure corresponds to the not-overfitting case in Theorem 4.1, and the right figure represents the benign overfitting case.

In this paper, we first show that the training dynamics of the attention gap can be described by the monotonically increasing function $g(x) = 2x + 2\sinh(x - \log T)$. This function naturally arises when expressing the evolution of the attention gap using the weight updates in Equations (5) and (6) and evaluating the dynamics via the quadrature method. Please refer to Appendix C for further details on the derivation. By using this function, the token selection of relevant token in clean data $i \in \mathcal{C}$ evolves as follows:

**Lemma 4.5** (Attention gap dynamics of clean data). *For any* $T_2 = \Theta\left(\frac{\exp(n^{-1}\mathrm{SNR}^{-2})}{\alpha n^{-1}\sigma_\epsilon^2 \|\boldsymbol{\nu}\|_2 \|\boldsymbol{\mu}\|_2 d^2 \max\{\sigma_w^2, \sigma_p^2\}}\right)$, *with probability at least* $1 - \delta$, *we have for all time step* $\tau \in [0, T_2]$ *that*

$$g\left(\Lambda_{i,t}(\tau)\right) = g\left(\Lambda_{i,t}(0)\right) \\ + \tau \cdot \Theta\left(\alpha n^{-1}\sigma_\epsilon^2 \|\boldsymbol{\nu}\|_2 \|\boldsymbol{\mu}\|_2 d^2 \max\{\sigma_w^2, \sigma_p^2\}\right),$$

*for any* $t \in [T] \setminus \{1\}$.

This lemma is shown by tracking the gradient descent dynamics and conducting an induction argument with several desirable properties. Due to the shape of the hyperbolic sine function, the function $g$ is close to linear around $x = \log T$, and gradually becomes exponential as $x$ moves away from $\log T$. This lemma states that while the value of $g$ evolves linearly over time, the actual evolution of the attention gap becomes increasingly a logarithmic factor. This finding corresponds to the left in Figure 2, where $s_1^{(i)}(\tau)$ approaches 1, and the value of $s_1^{(i)}(\tau)(1 - s_1^{(i)}(\tau))$, which determines the scale of gradient updates, diminishes accordingly. This part is also illustrated in Figure 4 in the appendix.

In contrast, the behavior of noisy data $j \in \mathcal{N}$ is described across two training stages. During the initial Stage 1, the softmax probability assigned to the relevant token $\mathbf{x}_1^{(j)}$ is suppressed. In this stage, as illustrated in the right column of Figure 2, the influence of noise memorization dominates, and the training progresses in a way that avoids selecting the relevant token, unlike the clean data case. In the subsequent

Stage 2, learning progresses to select the confusing token $\mathbf{x}_2^{(j)}$ that can fit the label noise.

The next lemma shows that in Stage 1, the value of $g$ decreases linearly over time, leading to a decrease in the attention gap $\Lambda_{j,t}$ in contrast to Lemma 4.5.

**Lemma 4.6** (Attention gap dynamics of noisy data in Stage 1). *For some* $T_1 = \Theta\left(\frac{\rho^{-1}}{\alpha n^{-1}\sigma_\epsilon^2 \|\boldsymbol{\nu}\|_2 \|\boldsymbol{\mu}\|_2 d^2 \max\{\sigma_w^2, \sigma_p^2\}}\right)$, *with probability at least* $1 - \delta$, *we have for all time step* $\tau \in [0, T_1]$ *that*

$$g\left(\Lambda_{j,t}(\tau)\right) = g\left(\Lambda_{j,t}(0)\right) \\ - \tau \cdot \Theta\left(\alpha n^{-1}\sigma_\epsilon^2 \|\boldsymbol{\nu}\|_2 \|\boldsymbol{\mu}\|_2 d^2 \max\{\sigma_w^2, \sigma_p^2\}\right),$$

*for any* $t \in [T] \setminus \{1\}$.

Subsequently, the training proceeds to select $\mathbf{x}_2^{(j)}$, that is, to increase the attention gap $\Gamma_{j,u}$.

**Lemma 4.7** (Attention gap dynamics of noisy data in Stage 2). *Let* $T_1$ *be the time step in Lemma 4.6. For any* $T_2 = \Theta\left(\frac{\exp(n^{-1}\mathrm{SNR}^{-2})}{\alpha n^{-1}\sigma_\epsilon^2 \|\boldsymbol{\nu}\|_2 \|\boldsymbol{\mu}\|_2 d^2 \max\{\sigma_w^2, \sigma_p^2\}}\right)$, *with probability at least* $1 - \delta$, *we have for all time step* $\tau \in [T_1, T_2]$ *that*

$$g\left(\Gamma_{j,1}(\tau) - \log \rho^{-1}\right) = g\left(\Gamma_{j,1}(0) - \log \rho^{-1}\right) \\ + \tau \cdot \Theta\left(\rho \alpha n^{-1}\sigma_\epsilon^2 \|\boldsymbol{\nu}\|_2 \|\boldsymbol{\mu}\|_2 d^2 \max\{\sigma_w^2, \sigma_p^2\}\right),$$
$$g\left(\Gamma_{j,u}(\tau)\right) = g\left(\Gamma_{j,u}(0)\right) \\ + \tau \cdot \Theta\left(\rho \alpha n^{-1}\sigma_\epsilon^2 \|\boldsymbol{\nu}\|_2 \|\boldsymbol{\mu}\|_2 d^2 \max\{\sigma_w^2, \sigma_p^2\}\right),$$

*for any* $u \in [T] \setminus \{1, 2\}$.

Similarly to Lemma 4.5, these lemmas are proved by using the parameter updates to track the evolution of the attention gap and the softmax probabilities assigned to each token. At this point, it is necessary to carefully evaluate several factors: the softmax probabilities, the norms and inner products between the model weights and both signal and noise

vectors, as well as the relative contributions of other training examples. In particular, the presence of label noise complicates the analysis involving softmax probabilities in the attention mechanism. Since the gradient updates depend on the softmax probabilities of training examples, the learning direction of the class signal cannot be determined solely by the ratio of clean to noisy data sample sizes, which is different from the existing benign overfitting work on linear or two-layer neural networks. Therefore, it is essential to evaluate the ratio of softmax probabilities across different training examples at each time step. This technique is useful for the analysis when dealing with distinct token selection behaviors among training samples, not limited to the label noise setting. The whole proofs based on mathematical induction are provided in Appendix C.

**Remark 4.8** (Two-stage Analysis). In the two-stage analysis in Lemmas 4.6 and 4.7, it is implicitly considered that $\exp(n^{-1}\mathrm{SNR}^{-2})$ is of larger order than $\rho^{-1}$, meaning that the $\log \rho^{-1} > 0$ scale is neglected on the benign overfitting condition $\mathrm{SNR}^2 = o(n)$. However, even if this is not the case, the benign overfitting result can still be obtained by replacing the numerator of $T_2$ with $\rho^{-2}$.

Finally, we present results related to generalization. The attention value of the class signal at time step $\tau$ is bounded from lower as follows:

**Lemma 4.9** (Attention value of class signals). *Let $T_2$ be the time step in Lemmas 4.5 and 4.7. With probability at least $1 - \delta$, we have for all time step $\tau \in [0, T_2]$ and $k \in \{\pm 1\}$ that*

$$\boldsymbol{\mu}_k^\top \mathbf{W}(\tau)^\top \mathbf{p}(\tau) \gtrsim \boldsymbol{\mu}_k^\top \mathbf{W}(0)^\top \mathbf{p}(0)$$
$$+ n\mathrm{SNR}^2 \cdot \log \left( \tau \cdot \alpha n^{-1} \sigma_\epsilon^2 \|\boldsymbol{\nu}\|_2 \|\boldsymbol{\mu}\|_2 d^2 \max\{\sigma_w^2, \sigma_p^2\} \right).$$

Please note that under the current condition $\mathrm{SNR}^2 = o(n^{-1})$, the value of $n\mathrm{SNR}^2$ is of a small order within the range that satisfies the parameter assumptions. To select the class relevant token for unseen examples and achieve a high generalization performance, $\boldsymbol{\mu}_k^\top \mathbf{W}(\tau)^\top \mathbf{p}(\tau)$ is required to sufficiently exceed $\boldsymbol{\epsilon}^\top \mathbf{W}(\tau)^\top \mathbf{p}(\tau)$, where $\boldsymbol{\epsilon}$ is a random noise vector. Lemma 4.9 states that this scenario is satisfied if the training time $\tau$ becomes exponentially large, thereby compensating for the smallness of $n\mathrm{SNR}^2$. This is why we need a time step of exponential order in Theorem 4.1. Such an exponential time scale is not necessary merely for fitting training examples, which implies that a significantly longer training period is required to achieve generalization.

## 5. Experiments

In this section, we provide experimental results to support our analysis. The code is available on GitHub [1].

[1] https://github.com/keitaroskmt/benign-attention

**Synthetic experiments.** We train the same model $f$ as defined in Equation (1) with gradient descent on $\mathbf{W}$ and $\mathbf{p}$, using the same data model as Definition 3.1 with $\sigma_\epsilon = 1$. Specifically, we consider the setting with $n = 20$, $T = 8$, $\eta = 0.2$, $\rho = 0.1$ and $\alpha = 5\mathrm{e}{-3}$, changing the value of the dimension $d$ and the signal size $\|\boldsymbol{\mu}\|_2$. For simplicity, we set $|\mathcal{W}_{+1}| = |\mathcal{W}_{-1}| = 1$ and $\boldsymbol{\nu} \propto \boldsymbol{\mu}_{+1} - \boldsymbol{\mu}_{-1}$, which aligns with our problem setup.

Figure 3 shows the dynamics of softmax probabilities for clean and noisy training samples from the initial weights. The left column represents the harmful overfitting case, where the dimension $d$ is larger compared to $\|\boldsymbol{\mu}\|_2$. The bottom figure shows that confusing weakly relevant token $\mathbf{x}_2$ is not always selected for noisy data; instead, irrelevant tokens that fit the label noise can be picked. This model can fit the training data with noise components, which hinders signal learning and reduces generalization ability. The middle figure shows the case where the balance between signal and noise is achieved, and benign overfitting is observed. In this figure, selecting the weakly relevant token $2 \in \mathcal{W}_{-Y^{*(j)}}$ for the noisy data $j \in \mathcal{N}$ aligns with our analysis in Theorem 4.1. In the right figure, where signal norm $\|\boldsymbol{\mu}\|_2$ is large, fitting the noisy data does not happen. The model is trained to select relevant token $\mathbf{x}_1$ for both clean and noisy samples, supporting the not-overfitting case in Theorem 4.1. For additional synthetic experiments, please refer to Appendix F.1. We provide the loss heat-map when varying $d$ and $\|\boldsymbol{\mu}\|_2$, which supports the SNR boundary established in the main theorem. Furthermore, we conduct similar experiments using a more general one-layer transformer encoder, providing results beyond our analytical setting.

**Real-world experiments** We further conducted real-world experiments on image and natural language datasets for classification. For each task, we used the pre-trained ViT (Dosovitskiy et al., 2021) and BERT (Devlin et al., 2018) models. To align as closely as possible with our analysis setup, we initialized the weights within the final attention layer and trained only these weights using a dataset with label noise. This setup corresponds to treating the fixed backbone model up to the last layer as a feature extractor and its output as the input to a one-layer attention model. We used datasets from various types: 10-class image classification with MNIST (LeCun et al., 2010) and CIFAR-10 (Krizhevsky et al., 2009), anomaly detection in medical image with PneumoniaMNIST and BreastMNIST (Yang et al., 2023), topic classification of text with AG-news (Zhang et al., 2015), and question type classification with TREC (Li & Roth, 2002). For detailed descriptions of these datasets, please refer to Appendix F.2.

Table 2 presents the training loss and test accuracy when varying the training size $n$. Since the SNR cannot be controlled due to its dependency on the dataset and the pre-

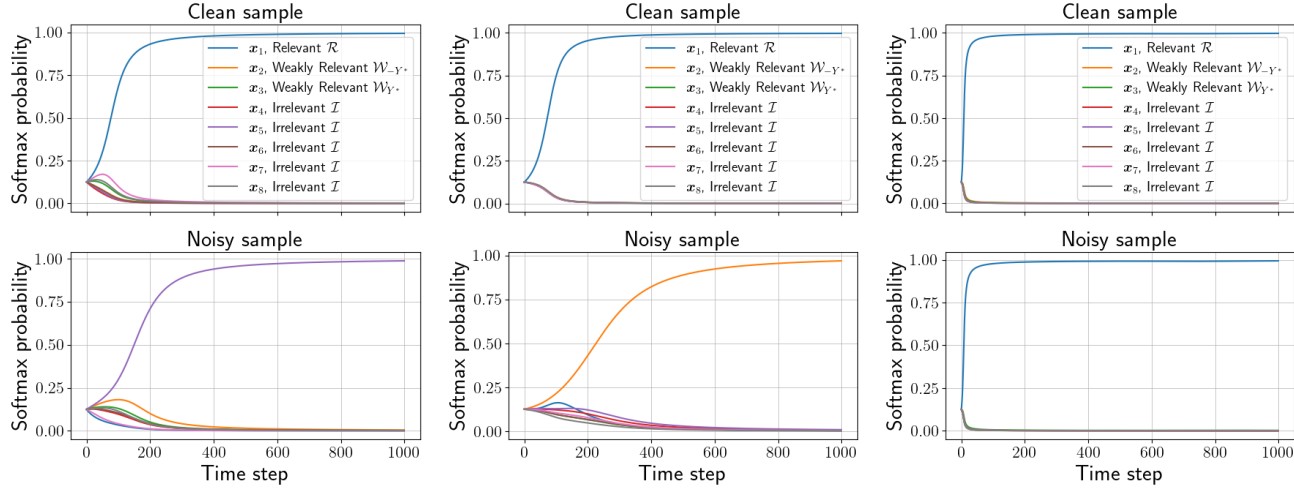

(a) Large noise setting: $d = 5000$, $\|\boldsymbol{\mu}\|_2 = 5$. Final training accuracy is $\mathbf{1.0}$ and test accuracy is $\mathbf{0.87}$ (Harmful overfitting).

(b) Balanced setting: $d = 2000$, $\|\boldsymbol{\mu}\|_2 = 20$. Final training accuracy is $\mathbf{1.0}$ and test accuracy is $\mathbf{1.0}$ (Benign overfitting).

(c) Large signal setting: $d = 1000$, $\|\boldsymbol{\mu}\|_2 = 100$. Final training accuracy is $\mathbf{0.8}$ and test accuracy is $\mathbf{1.0}$ (Not overfitting).

Figure 3: Dynamics of softmax probability. The top represents a clean sample, while the bottom represents a noisy sample. From left to right, the figures correspond to the cases of harmful overfitting, benign overfitting, and not overfitting.

Table 2: Training loss and test accuracy when only the final attention weights are trained on a label-noisy sub-dataset ($\eta = 0.1$) for 2000 epochs. The results show the average over three different runs with the standard deviation.

| Dataset | Eval | Training Size $n$ | | |
|---|---|---|---|---|
| | | 20 | 200 | 1000 |
| MNIST | train | $0.00_{\pm 0.00}$ | $0.01_{\pm 0.00}$ | $0.03_{\pm 0.00}$ |
| | test | $88.5_{\pm 3.2}$ | $90.0_{\pm 0.4}$ | $91.4_{\pm 0.2}$ |
| CIFAR-10 | train | $0.00_{\pm 0.01}$ | $0.07_{\pm 0.03}$ | $0.12_{\pm 0.01}$ |
| | test | $95.9_{\pm 0.4}$ | $95.1_{\pm 0.1}$ | $95.1_{\pm 0.1}$ |
| Pneumonia MNIST | train | $0.00_{\pm 0.00}$ | $0.00_{\pm 0.00}$ | $0.02_{\pm 0.00}$ |
| | test | $76.2_{\pm 4.1}$ | $78.5_{\pm 3.2}$ | $81.0_{\pm 1.5}$ |
| Breast MNIST | train | $0.07_{\pm 0.00}$ | $0.14_{\pm 0.01}$ | − |
| | test | $74.1_{\pm 2.0}$ | $77.1_{\pm 1.7}$ | − |
| AG-news | train | $0.00_{\pm 0.00}$ | $0.00_{\pm 0.00}$ | $0.00_{\pm 0.00}$ |
| | test | $83.6_{\pm 1.9}$ | $82.2_{\pm 1.1}$ | $77.6_{\pm 1.1}$ |
| TREC | train | $0.00_{\pm 0.00}$ | $0.02_{\pm 0.02}$ | $0.05_{\pm 0.02}$ |
| | test | $80.1_{\pm 1.0}$ | $78.3_{\pm 2.3}$ | $78.5_{\pm 0.2}$ |

trained model, we varied the training size $n$ to observe the scenario transitions shown in Theorem 4.1. Please recall that under our parameter and data model setup, Theorem 4.1 states that reducing $n$ for fixed SNR causes a transition from not-overfitting to benign overfitting. In the range of $n$ shown in Table 2, datasets such as CIFAR-10 and TREC exhibit such an increased fitting to the training set as $n$ decreases. MNIST and PneumoniaMNIST maintain very low training losses across all $n$, and the increase in noisy samples with larger $n$ does not negatively impact test accuracy. This scenario is closer to benign overfitting rather than harmful overfitting. In contrast, BreastMNIST does not overfit the training data, and AG-news demonstrates overfitting to the training set but with test accuracy negatively affected by noisy samples, suggesting a harmful overfitting case.

## 6. Conclusion

In this paper, we analyze the training dynamics and the generalization performance of the token selection of the attention architecture. Specifically, we showed that benign overfitting occurs in the token selection mechanism conditioning on SNR, and the model sufficiently learns class-relevant signals after overfitting. Furthermore, we supported the analysis with experiments. In general, this study is helpful in analyzing the balance of convergence speeds when the model selects different tokens for different training examples, including the label noise setting. As a natural next step, extending the analysis setup to next-token prediction or a more general data model can be considered.

## Acknowledgements

This work was supported by JSPS KAKENHI Grant Number JP24H00709.

## Impact Statement

This paper presents work whose goal is to advance the field of Machine Learning. There are many potential societal consequences of our work, none which we feel must be specifically highlighted here.

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

# Appendix

## Table of Contents

## A. Comparison with Existing Work

In this section, we highlight the novelty of our analysis and clarify the position of this study through a comparison with existing studies. Before presenting the existing work, we first describe the analytical challenges inherent in our setting, which are addressed in this study. This serves to clarify the novelty of our contribution.

### A.1. Difficulty of Our Label Noise Setting

The analytical setting that incorporates label noise distinctively characterizes our study, as shown in Table 1. By introducing variability in token selection across training examples, the presence of label noise enables a more general analysis of the token selection mechanism. Specifically, label noise introduces significant challenges due to the existence of competing training directions within the same training run. This results in two key difficulties:

- Signal learning vs memorization in token selection of each example (see Figure 2).

- Clean samples vs noisy samples in the learning direction of class signals, i.e., $\boldsymbol{\mu}_{+1}$ and $\boldsymbol{\mu}_{-1}$.

These challenges are further complicated by the fact that the weight updates depend nontrivially on the softmax probability, as will be shown later in Equations (9) and (12). Therefore, the training direction is not statically determined by pre-training quantities such as SNR or label noise ratio $\eta$. This is a fundamental difficulty that does not appear in previous analyses of benign overfitting, including two-layer NNs. For instance, depending on the convergence speed—how quickly the softmax probability $s(t)$ converges to 1 or 0—it is possible that, even when label noise is very small and the number of noisy samples is negligible, noisy samples can still dominate the weight updates at certain time step. Thus, it is crucial to track the whole training dynamics of each token to analyze the above two relationships, making the analysis inherently difficult. A large portion of this paper is dedicated to addressing and resolving this issue.

In the following, we present specific related studies and clarify the differences and novelty of our work.

### A.2. Comparison with Implicit Bias Studies

We compare with the previous papers on the implicit bias of the attention mechanism (Tarzanagh et al., 2023a;b; Li et al., 2024b; Vasudeva et al., 2024; Sheen et al., 2024). The two major differences from ours are:

(a) Analysis of a fixed training set $S$, rather than considering the underlying distribution.
(b) Single optimal token, with all other tokens having identical token scores.

In our setting, (b) corresponds to considering a single relevant token, irrelevant tokens $\mathbf{x}_t \sim N(0, \boldsymbol{\Sigma} - \boldsymbol{\mu}_{+1}\boldsymbol{\mu}_{+1}^\top/\|\boldsymbol{\mu}\|_2^2 - \boldsymbol{\mu}_{-1}\boldsymbol{\mu}_{-1}^\top/\|\boldsymbol{\mu}\|_2^2)$, and the linear head $\boldsymbol{\nu} \propto \boldsymbol{\mu}_{+1} - \boldsymbol{\mu}_{-1}$. In this setting, the noise vectors are orthogonal to the class signals and have an identical token score $\gamma_t = (\boldsymbol{\mu}_{+1} - \boldsymbol{\mu}_{-1})^\top \mathbf{x}_t = 0$, which aligns with (b). However, we consider a more general data model, including an intermediate state termed weakly relevant tokens and non-orthogonality of noise and signals. Furthermore, for data with label noise, selecting any irrelevant token results in the model with output 0, and the model cannot fit the training data. Therefore, this data model is too simple to analyze the distinct behavior of the token selection, which is the focus of our study. While we follow the model setting, our analytical methods differ entirely.

### A.3. Comparison with Benign Overfitting Studies

**Comparison with Jiang et al. (2024).** The major differences from ours are:

(a) Absence of label noise.
(b) Differences in the data model.
(c) Focus on vision transformer (ViT) model.

Their setup assumes no label noise in the target $Y$, meaning all training data is clean. In contrast, we focus on the benign overfitting concerning label noise, in addition to the input noises $\{\boldsymbol{\epsilon}_t\}_t$. As explained in Appendix A.1, the ability to fit label noise of the attention mechanism has not been analyzed before, and even when possible, the training direction is not straightforwardly determined due to the intrinsic softmax difficulties. These challenges are specific to the attention mechanism and are newly addressed in our work. For (b), they assume orthogonality between class and noise vectors because the noise covariance matrix is given by $\boldsymbol{\Sigma} = \mathbf{I} - \boldsymbol{\mu}_{+1}\boldsymbol{\mu}_{+1}^\top/\|\boldsymbol{\mu}\|_2^2 - \boldsymbol{\mu}_{-1}\boldsymbol{\mu}_{-1}^\top/\|\boldsymbol{\mu}\|_2^2$. This assumption overlooks the interactions between learned parameters and training samples, and it becomes a more unrealistic assumption in a multi-class setting in Appendix E.1. Additionally, our data model newly considers intermediate states as weakly relevant tokens, which reflects the real setting more closely. Finally, for (c), they explored the advantage of ViT compared to linear models and CNNs from the perspective of benign overfitting. Their analysis includes the optimization of self-attention and value matrix, which involve challenging interactions during training. In contrast, our focus is on the training dynamics and generalization ability of token selection mechanisms. By focusing on this aspect, we enable analyses in the above general settings. Additionally, our work demonstrates that benign overfitting can be achieved solely through the token selection mechanism within the softmax. This is demonstrated for the first time in our analytical setting. Training the linear head $\boldsymbol{\nu}$ would obscure whether noise memorization and benign overfitting originate from the attention mechanism or from the linear classifier, the latter of which has already been extensively studied in the literature.

**Comparison with Magen et al. (2024).** This work also studies the benign overfitting of the attention mechanism with the same type of architecture, under label noise setting, but it largely differs from our work in terms of the following points:

(a) Differences in the data model.

(b) Two-step gradient descent of $\mathbf{p}$ and $\boldsymbol{\nu}$.

For (a), they simplified the data model, such as signal-noise orthogonality and the clean separation of signal and noise vectors. Our analytical setup fills this gap by introducing noise into the relevant tokens, and by incorporating intermediate states between signal and noise vectors, which we refer to as weakly relevant tokens. We also do not assume the signal-noise orthogonality. For (b), we present general gradient descent results for $\mathbf{p}$ and key-query matrix $\mathbf{W}$. This allows us to discuss differences in the convergence of gradient descent conditioning on SNR, as well as the difference in the time scale. Instead, they provide the characterization of SNR for the max-margin solution.

We also comment on the differences in experimental validation. Their work conducted the empirical validation under the data model in their theoretical setting, not only for the model architecture analyzed but also for other settings such as self-attention and multiple layers. In contrast, we validate our theoretical analysis using synthetic data for both the analytical model architecture and a one-layer transformer encoder. Furthermore, we present results on real-world data, highlighting the connection to the analytical setting in our paper. We also provide a visualization of the heat map for the synthetic data.

### A.4. Comparison with Other Studies

**Comparison with Oymak et al. (2023).** We emphasize that while we followed the analysis setups from this work, our results cannot be obtained by simply applying their results to the context of benign overfitting. The key differences from our study are as follows:

(a) Absence of label noise.

(b) Not focusing on overfitting.

(c) Three optimization steps $(\boldsymbol{\nu}(1), \mathbf{p}(1), \boldsymbol{\nu}(2))$. In particular, the learning within the softmax is a single-step optimization of $\mathbf{p}$.

(d) Minor differences in input distribution.

For (a) and (b), we must analyze the competing training directions due to the presence of label noise, as explained in Appendix A.1. This largely complicates the analysis of softmax probabilities because it is not a simple selection of the relevant token. For (c), the analysis is in the very early stage of training, so it is difficult to fit noisy data containing label noise. Therefore, their setting cannot be applied in our study. Additionally, the optimization within the softmax is a single step. Only at the first step, it suffices to analyze the gradient descent concerning the average of the tokens because of the weight initialization, and the complex softmax term does not need to be considered in the gradient calculation. Optimizing token selection beyond a single step introduces the softmax probabilities into the gradient calculation, resulting in a complex dependency on the current parameters. To address this, we needed to handle the relationships among attention probabilities across all training steps using an inductive approach.

Table 3: Notations used in this work.

| | |
|---|---|
| $\mathbf{X}$ | Sequence of input tokens, i.e., $\mathbf{X} = (\mathbf{x}_1, \ldots, \mathbf{x}_T)^\top$. |
| $Y$ | Training label. |
| $Y^*$ | True label. |
| $n$ | Number of training data. |
| $T$ | Length of input tokens. |
| $d$ | Dimension of token embeddings and model weights. |
| $\boldsymbol{\mu}_{+1}, \boldsymbol{\mu}_{-1}$ | Signal vectors for class 1 and $-1$, respectively. |
| $\boldsymbol{\epsilon}_t^{(i)}$ | Noise component in $\mathbf{x}_t^{(i)}$ for $i \in [n]$ and $t \in [T]$. |
| $\sigma_\epsilon^2$ | Variance of noise vector, i.e., $\boldsymbol{\epsilon} \sim N(\mathbf{0}, \sigma_\epsilon^2 \mathbf{I})$. |
| $\sigma_p^2, \sigma_w^2$ | Variances for weight initialization, i.e., $\mathbf{p}(0)_i \sim N(0, \sigma_p^2)$, $\mathbf{W}(0)_{i,j} \sim N(0, \sigma_w^2)$ for $i, j \in [d]$. |
| SNR | Signal-to-noise ratio given by $\text{SNR} := \|\boldsymbol{\mu}\|_2 / (\sigma_\epsilon \sqrt{d})$. |
| $\mathbf{s}^{(i)}(\tau)$ | Probability vector for $\mathbf{X}^{(i)}$ at $\tau$-th step, defined as $\mathbb{S}(\mathbf{X}^{(i)} \mathbf{W}(\tau)^\top \mathbf{p}(\tau))$. |
| $\gamma_t^{(i)}$ | Token score, defined as $\boldsymbol{\nu}^\top \mathbf{x}_t^{(i)}$. |
| $\mathcal{R}$ | Set of relevant tokens; $\mathcal{R} = \{1\}$ and $\mathbf{x}_1 = \boldsymbol{\mu}_{Y^*} + \boldsymbol{\epsilon}_1$. |
| $\mathcal{W}$ | Set of weakly relevant tokens; $\mathbf{x}_u = \rho \boldsymbol{\mu}_{w_u} + \boldsymbol{\epsilon}_u, u \in \mathcal{W}, w_u \in \{\pm 1\}$. |
| $\mathcal{W}_{+1}, \mathcal{W}_{-1}$ | Set of weakly relevant tokens with specific label; $\{u \in \mathcal{W} \mid w_u = +1\}$ and $\{u \in \mathcal{W} \mid w_u = -1\}$. We assume that the confusing token consists of $\mathcal{W}_{-Y^*} = \{2\}$. |
| $\mathcal{I}$ | Set of irrelevant tokens; $\mathbf{x}_v = \boldsymbol{\epsilon}_v, v \in \mathcal{I}$. |
| $\mathcal{C}$ | Set of clean data; $\{i \in [n] \mid Y^{(i)} = Y^{*(i)}\}$. |
| $\mathcal{C}_+, \mathcal{C}_-$ | Set of clean data with label 1 and $-1$, respectively. |
| $\mathcal{N}$ | Set of noisy data; $\{i \in [n] \mid Y^{(i)} \neq Y^{*(i)}\}$. |
| $\mathcal{N}_+, \mathcal{N}_-$ | Set of noisy data with training label 1 and $-1$, respectively. |
| $\alpha$ | Learning rate. |
| $\eta$ | Label noise ratio. |
| $\lambda_{+1}(\tau), \lambda_{-1}(\tau)$ | Attention scores for class signal in Definition C.1; $\lambda_{+1}(\tau) := \langle \mathbf{W}(\tau)\boldsymbol{\mu}_{+1}, \mathbf{p}(\tau) \rangle$ and $\lambda_{-1}(\tau) := \langle \mathbf{W}(\tau)\boldsymbol{\mu}_{-1}, \mathbf{p}(\tau) \rangle$. |
| $\rho_{i,t}(\tau)$ | Attention scores for noise vectors in Definition C.1; $\rho_{i,t}(\tau) := \langle \mathbf{W}(\tau)\boldsymbol{\epsilon}_t^{(i)}, \mathbf{p}(\tau) \rangle$. |
| $I_{i,+}(\tau), I_{i,-}(\tau), I_{i,j,u}(\tau),$ $I_{i,+}^W(\tau), I_{i,-}^W(\tau), I_{i,j,u}^W(\tau),$ $I_i^p(\tau)$ | Weighted inner-product terms defined in Definition C.2. |
| $\Lambda_{i,t}(\tau), \Gamma_{i,u}(\tau)$ | Attention gaps defined in Definition C.3; $\Lambda_{i,t}(\tau) := \left( \mathbf{x}_1^{(i)} - \mathbf{x}_t^{(i)} \right)^\top \mathbf{W}(\tau)^\top \mathbf{p}(\tau)$, $\Gamma_{i,u}(\tau) := \left( \mathbf{x}_2^{(i)} - \mathbf{x}_u^{(i)} \right)^\top \mathbf{W}(\tau)^\top \mathbf{p}(\tau)$. |

# B. Preliminaries

## B.1. Notation

We first list the notations used in this work in Table 3.

Furthermore, the basic computations are presented here for convenience. The gradient $\nabla_{\mathbf{W}}\widehat{\mathcal{L}}(\mathbf{W}, \mathbf{p})$ and $\nabla_{\mathbf{p}}\widehat{\mathcal{L}}(\mathbf{W}, \mathbf{p})$ used in the training can be explicitly computed as follows. Since the derivative of the softmax function is given by

$$\nabla_{\mathbf{v}}\mathbb{S}(\mathbf{v}) = \text{diag}(\mathbb{S}(\mathbf{v})) - \mathbb{S}(\mathbf{v})\mathbb{S}(\mathbf{v})^{\top},$$

where $\text{diag}(\mathbf{v}) \in \mathbb{R}^{T \times T}$ denotes the diagonal matrix whose $(i, i)$-th entry equals to $v_i$. Using the denominator layout, we have

$$\nabla_{\mathbf{W}^{\top}}\widehat{\mathcal{L}}(\mathbf{W}, \mathbf{p}) = \frac{1}{n}\sum_{i=1}^{n}\ell_i' \cdot Y^{(i)} \cdot \nabla_{\mathbf{W}^{\top}}f(\mathbf{X}^{(i)}) \tag{7}$$

$$= \frac{1}{n}\sum_{i=1}^{n}\ell_i' \cdot Y^{(i)} \cdot \mathbf{X}^{(i)\top}\left(\text{diag}(\mathbb{S}(\mathbf{X}^{(i)}\mathbf{W}^{\top}\mathbf{p})) - \mathbb{S}(\mathbf{X}^{(i)}\mathbf{W}^{\top}\mathbf{p})\mathbb{S}(\mathbf{X}^{(i)}\mathbf{W}^{\top}\mathbf{p})^{\top}\right)\mathbf{X}^{(i)}\boldsymbol{\nu}\mathbf{p}^{\top} \tag{8}$$

$$= \frac{1}{n}\sum_{i=1}^{n}\ell_i' \cdot Y^{(i)} \cdot \left(\sum_{t=1}^{T}s_t^{(i)}\left(\gamma_t^{(i)} - \sum_{u=1}^{T}s_u^{(i)}\gamma_u^{(i)}\right)\mathbf{x}_t^{(i)}\mathbf{p}^{\top}\right), \tag{9}$$

where $\ell_i'$ is abbreviation for $\ell'(Y^{(i)} \cdot \boldsymbol{\nu}^{\top}\mathbf{X}^{(i)\top}\mathbb{S}(\mathbf{X}^{(i)}\mathbf{W}^{\top}\mathbf{p}))$. From a similar calculation, we have

$$\nabla_{\mathbf{p}}\widehat{\mathcal{L}}(\mathbf{W}, \mathbf{p}) = \frac{1}{n}\sum_{i=1}^{n}\ell_i' \cdot Y^{(i)} \cdot \nabla_{\mathbf{p}}f(\mathbf{X}^{(i)}) \tag{10}$$

$$= \frac{1}{n}\sum_{i=1}^{n}\ell_i' \cdot Y^{(i)} \cdot \mathbf{W}\mathbf{X}^{(i)\top}\left(\text{diag}(\mathbb{S}(\mathbf{X}^{(i)}\mathbf{W}^{\top}\mathbf{p})) - \mathbb{S}(\mathbf{X}^{(i)}\mathbf{W}^{\top}\mathbf{p})\mathbb{S}(\mathbf{X}^{(i)}\mathbf{W}^{\top}\mathbf{p})^{\top}\right)\mathbf{X}^{(i)}\boldsymbol{\nu} \tag{11}$$

$$= \frac{1}{n}\sum_{i=1}^{n}\ell_i' \cdot Y^{(i)} \cdot \left(\sum_{t=1}^{T}s_t^{(i)}\left(\gamma_t^{(i)} - \sum_{u=1}^{T}s_u^{(i)}\gamma_u^{(i)}\right)\mathbf{W}\mathbf{x}_t^{(i)}\right). \tag{12}$$

## B.2. Proof Overview

This section provides a roadmap for the proofs in the appendix, aiming to improve readability. For the precise statements and proofs of individual lemmas and propositions, please refer to the corresponding sections.

We begin in Appendix B.3 by establishing high-probability events. Under parameter assumptions in Section 3.5, we derive bounds on norms, inner products, and class balance among samples. Lemma B.1 states the collections of these events, and their proofs are provided via standard concentration inequalities, divided across Lemmas B.3, B.4 and B.6 to B.11. As preparation for the main proof, Lemma B.12 presents an evaluation of the token score $\gamma_t$. The following Appendix C builds upon the high-probability events established in this section.

Appendix C presents the proof of the main theorem. In Appendix C.1, we first introduce technical notations (Definitions C.1 to C.3), simple calculations regarding training with gradient descent (Lemmas C.4 to C.6), and several initial properties at the start of training (Lemmas C.7 and C.8). We also provide a result regarding the balance of loss gradients across examples in Lemma C.9. Appendices C.2 and C.3 correspond to the not-overfitting and benign overfitting cases, respectively. In the not-overfitting case, Lemma C.11 is the central result. The proof proceeds inductively, incorporating desirable properties such as norm and inner product evaluations and balance in softmax probabilities. The most important proposition $C(\tau)$ characterizes the dynamics of the attention gap $\Lambda(\tau)$ introduced in Definition 4.4 (main text) and Definition C.3 (appendix), governed by the function $g(x) = 2x + 2\sinh(x - \log T)$. This inductive argument, which holds for all time steps $0 \le \tau \le T_1$, leads to the proof of the main theorem in Appendix C.2.2, establishing both the not-overfitting and generalization guarantees. The benign overfitting case follows a similar structure, but as described in Section 4.1, the analysis is divided into two stages. The inductive argument is split into Lemmas C.12 and C.13, and Appendix C.3.2 concludes with the proof of the main theorem of this case, including both the overfitting and good generalization.

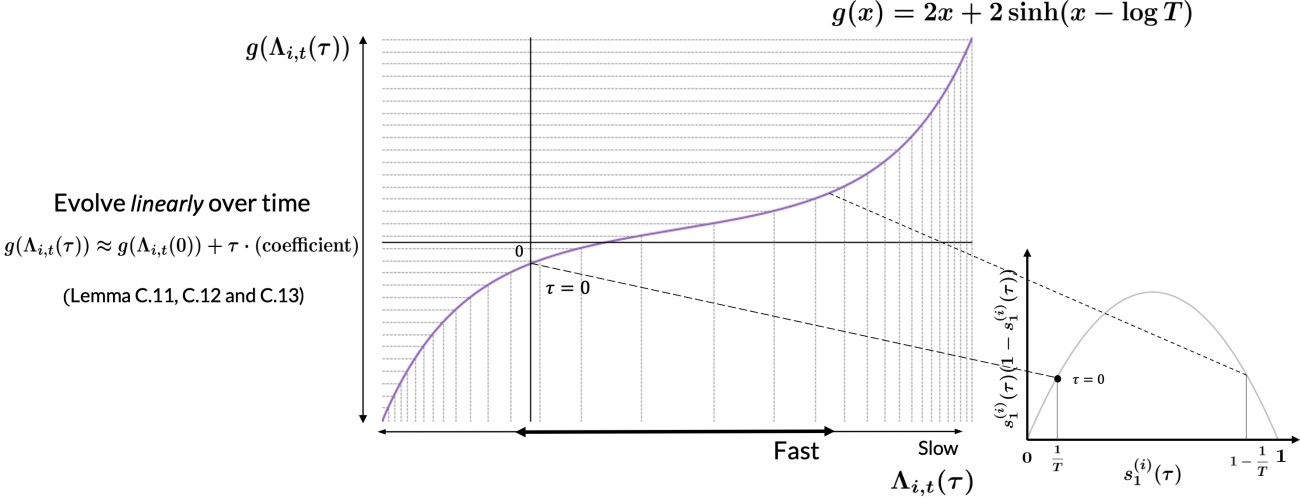

Figure 4: Illustration of the evolution of the attention gap defined in Definition C.3, based on the function $g$. The right figure is similar to Figure 2 in the main text and illustrates that as $s_1(\tau)$ approaches 0 or 1, the term $s_1(\tau)(1 - s_1(\tau))$, which determines the magnitude of the one-step update, becomes smaller—corresponding to slower training of the weights. For the relationship between $s_1^{(i)}(\tau)$ and $\Lambda_{i,t}(\tau)$, please see Lemma D.1, which shows that $s_1(\tau)(1 - s_1(\tau))$ decreases as $\Lambda(\tau)$ moves away from $\log T$.

To help illustrate the development of the attention gap driven by the function $g$, we provide a visualization in Figure 4. The function $g(x)$ behaves approximately linearly near $x = \log T$ but grows exponentially as $x$ moves away from this region. The inductive lemmas in Appendix C show that $g(\Lambda(\tau))$ and $g(\Gamma(\tau))$ evolve linearly, which implies that the growth of $\Lambda(\tau)$ and $\Gamma(\tau)$ themselves gradually slows down.

Although the main theorem is fully proved in Appendix C, Appendix D supplements the proof by presenting several auxiliary calculations. Appendix D.1 provides results linking the weight parameters to the softmax probabilities. Appendices D.2 and D.3 correspond to the not-overfitting and benign overfitting cases, respectively. These sections give one-step gradient calculations under the favorable weight conditions, showing how signal learning and noise memorization progress and supporting the induction arguments in Appendix C.

### B.3. High-probability Events

We first show that the following events occur simultaneously with high probability under the assumptions in Section 3.5.

**Lemma B.1** (High-probability events). *Suppose that the parameter assumptions in Section 3.5 hold. There exists some constant $c_1, c_2 > 0$ such that the following hold simultaneously with probability at least $1 - \delta$ over the realization of training data $S$ and model parameters $\mathbf{W}(0), \mathbf{p}(0)$:*

*(i) (Norm concentration) For all $i \in [n], t \in [T]$, we have*

$$(1 - o(1)) \sigma_\epsilon \sqrt{d} \leq \|\boldsymbol{\epsilon}_t^{(i)}\|_2 \leq (1 + o(1)) \sigma_\epsilon \sqrt{d}, \tag{13}$$

$$(1 - o(1)) \sigma_w \|\boldsymbol{\mu}\|_2 \sqrt{d} \leq \|\mathbf{W}(0)\boldsymbol{\mu}_{+1}\|_2 \leq (1 + o(1)) \sigma_w \|\boldsymbol{\mu}\|_2 \sqrt{d}, \tag{14}$$

$$(1 - o(1)) \sigma_w \|\boldsymbol{\mu}\|_2 \sqrt{d} \leq \|\mathbf{W}(0)\boldsymbol{\mu}_{-1}\|_2 \leq (1 + o(1)) \sigma_w \|\boldsymbol{\mu}\|_2 \sqrt{d}, \tag{15}$$

$$(1 - o(1)) \sigma_w \sigma_\epsilon d \leq \|\mathbf{W}(0)\boldsymbol{\epsilon}_t^{(i)}\|_2 \leq (1 + o(1)) \sigma_w \sigma_\epsilon d, \tag{16}$$

$$(1 - o(1)) \sigma_p \sqrt{d} \leq \|\mathbf{p}(0)\|_2 \leq (1 + o(1)) \sigma_p \sqrt{d}. \tag{17}$$

*(ii) (Inner-product concentration) For any $i, j \in [n], t, u \in [T]$ such that $(i, t) \neq (j, u)$, we have*

$$|\langle \boldsymbol{\epsilon}_t^{(i)}, \boldsymbol{\epsilon}_u^{(j)} \rangle| < c_1 \sigma_\epsilon^2 \sqrt{d} \log(Tn/\delta), \tag{18}$$

$$|\langle \mathbf{W}(0)\boldsymbol{\mu}_{+1}, \mathbf{W}(0)\boldsymbol{\mu}_{-1} \rangle| < c_1 \sigma_w^2 \|\boldsymbol{\mu}\|_2^2 \sqrt{d} \log(Tn/\delta), \tag{19}$$

$$|\langle \mathbf{W}(0)\boldsymbol{\mu}_{+1}, \mathbf{W}(0)\boldsymbol{\epsilon}_t^{(i)} \rangle| < c_1 (1 + o(1)) \sigma_w^2 \sigma_\epsilon \|\boldsymbol{\mu}\|_2 d \log(Tn/\delta), \tag{20}$$

$$|\langle \mathbf{W}(0)\boldsymbol{\mu}_{-1}, \mathbf{W}(0)\boldsymbol{\epsilon}_t^{(i)} \rangle| < c_1 (1 + o(1)) \sigma_w^2 \sigma_\epsilon \|\boldsymbol{\mu}\|_2 d \log(Tn/\delta), \tag{21}$$

$$|\langle \mathbf{W}(0)\boldsymbol{\epsilon}_t^{(i)}, \mathbf{W}(0)\boldsymbol{\epsilon}_u^{(j)} \rangle| < c_1 (1 + o(1)) \sigma_w^2 \sigma_\epsilon^2 d^{3/2} \log(Tn/\delta), \tag{22}$$

$$|\langle \mathbf{W}(0)\boldsymbol{\mu}_{+1}, \mathbf{p}(0) \rangle| < c_1 \sigma_w \sigma_p \|\boldsymbol{\mu}\|_2 \sqrt{d} \log(Tn/\delta), \tag{23}$$

$$|\langle \mathbf{W}(0)\boldsymbol{\mu}_{-1}, \mathbf{p}(0) \rangle| < c_1 \sigma_w \sigma_p \|\boldsymbol{\mu}\|_2 \sqrt{d} \log(Tn/\delta), \tag{24}$$

$$|\langle \mathbf{W}(0)\boldsymbol{\epsilon}_t^{(i)}, \mathbf{p}(0) \rangle| < c_1 (1 + o(1)) \sigma_w \sigma_p \sigma_\epsilon d \log(Tn/\delta), \tag{25}$$

$$|\langle \boldsymbol{\mu}_{+1}, \boldsymbol{\epsilon}_t^{(i)} \rangle| < c_2 \sigma_\epsilon \|\boldsymbol{\mu}\|_2 \sqrt{\log(Tn/\delta)}, \tag{26}$$

$$|\langle \boldsymbol{\mu}_{-1}, \boldsymbol{\epsilon}_t^{(i)} \rangle| < c_2 \sigma_\epsilon \|\boldsymbol{\mu}\|_2 \sqrt{\log(Tn/\delta)}, \tag{27}$$

$$|\langle \boldsymbol{\nu}, \boldsymbol{\epsilon}_t^{(i)} \rangle| < c_2 \sigma_\epsilon \|\boldsymbol{\nu}\|_2 \sqrt{\log(Tn/\delta)}. \tag{28}$$

*(iii) Regarding the clean and noisy samples, we have*

$$\frac{2 - 3\eta}{4} n \leq |\mathcal{C}_+|, |\mathcal{C}_-| \leq \frac{2 - \eta}{4} n, \tag{29}$$

$$\frac{\eta}{4} n \leq |\mathcal{N}_+|, |\mathcal{N}_-| \leq \frac{3\eta}{4} n. \tag{30}$$

**Definition B.2** (Good run). We denote by "good run" the trial that the events from (i) to (iii) occur.

In the proof of the main theorem, we will proceed by conditioning on these events. Lemma B.1 implies that a good run occurs with the probability at least $1 - \delta$ over the realization of training data $S$ and the weights initialization.

In the rest of this section, we will prove each high-probability event in Lemma B.1. Specifically, we can show it by combining Lemmas B.6, B.8, B.9 and B.11, using union bound argument. In the following proofs, suppose that the parameter assumptions in Section 3.5 hold. First, we show the norm concentration of the Gaussian noise vectors. The next lemma gives the lower bound for the expectation of the $L_2$ norm.

**Lemma B.3.** *For a Gaussian vector $\mathbf{x} \sim N(0, \boldsymbol{\Sigma})$, we have*

$$\sqrt{\mathrm{Tr}(\boldsymbol{\Sigma}) - 1} \leq \mathbb{E}\left[\|\mathbf{x}\|_2\right].$$

*Proof of Lemma B.3.* We use the Gaussian Poincaré Inequality ((Boucheron et al., 2003), Theorem 3.20):

$$\mathrm{Var}\left(f(\mathbf{x})\right) \leq \mathbb{E}\left[\|\nabla f(\mathbf{x})\|_2^2\right], \tag{31}$$

where $f : \mathbb{R}^d \to \mathbb{R}$ is any continuously differentiable function. By taking $f$ as $f(\mathbf{x}) = \|\mathbf{x}\|_2$, since we have $\mathbb{E}\left[\|\nabla f(\mathbf{x})\|_2^2\right] = \mathbb{E}\left[\|\mathbf{x}/\|\mathbf{x}\|_2\|_2^2\right] = 1$,

$$1 \geq \mathrm{Var}(f(\mathbf{x})) = \mathbb{E}\left[\|\mathbf{x}\|^2\right] - (\mathbb{E}\left[\|\mathbf{x}\|\right])^2 = \mathrm{Tr}(\boldsymbol{\Sigma}) - (\mathbb{E}\left[\|\mathbf{x}\|\right])^2. \tag{32}$$

Rearranging the terms, we get

$$\sqrt{\mathrm{Tr}(\boldsymbol{\Sigma}) - 1} \leq \mathbb{E}\left[\|\mathbf{x}\|_2\right], \tag{33}$$

which concludes the proof. $\square$

The following lemma is about the concentration of Lipschitz functions and is used to prove the norm concentration.

**Lemma B.4** (Rephrased from ((Wainwright, 2019), Theorem 3.16)). *For any $L$-Lipschitz function $f : \mathbb{R}^d \to \mathbb{R}$, we have*

$$\Pr_{\boldsymbol{\epsilon} \sim N(\mathbf{0}, \boldsymbol{\Sigma})} \{|f(\boldsymbol{\epsilon}) - \mathbb{E}\left[f(\boldsymbol{\epsilon})\right]| \geq w\} \leq 2 \exp\left(-\frac{w^2}{4\sigma_{max}(\boldsymbol{\Sigma}) L^2}\right). \tag{34}$$

Note that the coefficient 2 could be removed in the case of one-sided inequality.

**Remark B.5.** Generally, this holds for strongly log-concave distributions, i.e., distributions with a density $p(x) = \exp(-\psi(x))$, where $\psi : \mathbb{R}^d \to \mathbb{R}$ is a strongly convex function. Here, we used the fact that the Gaussian distribution $N(\mathbf{0}, \boldsymbol{\Sigma})$ is a strongly log-concave distribution with parameter $\sigma_{min}(\boldsymbol{\Sigma}^{-1}) = 1/\sigma_{max}(\boldsymbol{\Sigma})$.

We are now ready to prove the norm concentration as follows.

**Lemma B.6** (Norm concentration). *With probability at least $1 - \delta/4$,*

$$(1 - o(1))\,\sigma_\epsilon \sqrt{d} \leq \|\boldsymbol{\epsilon}_t^{(i)}\|_2 \leq (1 + o(1))\,\sigma_\epsilon \sqrt{d},$$
$$(1 - o(1))\,\sigma_w \|\boldsymbol{\mu}\|_2 \sqrt{d} \leq \|\mathbf{W}(0)\boldsymbol{\mu}_{+1}\|_2 \leq (1 + o(1))\,\sigma_w \|\boldsymbol{\mu}\|_2 \sqrt{d},$$
$$(1 - o(1))\,\sigma_w \|\boldsymbol{\mu}\|_2 \sqrt{d} \leq \|\mathbf{W}(0)\boldsymbol{\mu}_{-1}\|_2 \leq (1 + o(1))\,\sigma_w \|\boldsymbol{\mu}\|_2 \sqrt{d},$$
$$(1 - o(1))\,\sigma_w \|\boldsymbol{\epsilon}_t^{(i)}\|_2 \sqrt{d} \leq \|\mathbf{W}(0)\boldsymbol{\epsilon}_t^{(i)}\|_2 \leq (1 + o(1))\,\sigma_w \|\boldsymbol{\epsilon}_t^{(i)}\|_2 \sqrt{d},$$
$$(1 - o(1))\,\sigma_p \sqrt{d} \leq \|\mathbf{p}(0)\|_2 \leq (1 + o(1))\,\sigma_p \sqrt{d}.$$

*for all $i \in [n], t \in [T]$*

*Proof of Lemma B.6.* From the definition of noise distribution, $\boldsymbol{\epsilon}_t^{(i)} \sim N(\mathbf{0}, \sigma_\epsilon^2 \mathbf{I}_d)$ for all $i \in [n], t \in [T]$. To begin with, we show the norm concentration of the Gaussian vector. For $\mathbf{x}, \mathbf{y} \in \mathbb{R}^d$, since we have

$$\|\mathbf{x}\|_2 - \|\mathbf{y}\|_2 = \frac{\|\mathbf{x}\|_2 - \|\mathbf{y}\|_2}{\|\mathbf{x} - \mathbf{y}\|_2}\|\mathbf{x} - \mathbf{y}\|_2 \leq \|\mathbf{x} - \mathbf{y}\|_2, \tag{35}$$

$f(\mathbf{x}) = \|\mathbf{x}\|_2$ is 1-Lipschitz function. Using Lemma B.4, we get

$$\Pr\left\{\left|\|\boldsymbol{\epsilon}_t^{(i)}\|_2 - \mathbb{E}\left[\|\boldsymbol{\epsilon}\|_2\right]\right| > w\right\} \leq 2\exp\left(-\frac{w^2}{4\sigma_\epsilon^2}\right), \tag{36}$$

for some $i \in [n], t \in [T]$. Taking union-bound gives

$$\forall i \in [n], \forall t \in [T], \ \Pr\left\{\left|\|\boldsymbol{\epsilon}_t^{(i)}\|_2 - \mathbb{E}\left[\|\boldsymbol{\epsilon}\|_2\right]\right| > w\right\} \leq 2Tn \exp\left(-\frac{w^2}{4\sigma_\epsilon^2}\right). \tag{37}$$

Lemma B.3 and Jensen inequality lead the following bound on the expectation of the Gaussian norm:

$$\sqrt{\sigma_\epsilon^2 d - 1} \leq \mathbb{E}\left[\|\boldsymbol{\epsilon}\|_2\right] \leq \sigma_\epsilon \sqrt{d}. \tag{38}$$

Using this and Equation (37), we have with the probability at least $1 - \delta/20$,

$$\sqrt{\sigma_\epsilon^2 d - 1} - 2\sqrt{\sigma_\epsilon^2 \log\left(\frac{40Tn}{\delta}\right)} \leq \|\boldsymbol{\epsilon}_t^{(i)}\|_2 \leq \sigma_\epsilon \sqrt{d} + 2\sqrt{\sigma_\epsilon^2 \log\left(\frac{40Tn}{\delta}\right)}, \tag{39}$$

for all $i \in [n], t \in [T]$. By using $d = \omega(\log(Tn/\delta))$ in the parameter assumptions, the second term on both sides becomes $o(\sigma_\epsilon \sqrt{d})$. Combining this with $(\sigma_\epsilon \sqrt{d})\left(1 - 1/(\sigma_\epsilon^2 d)\right) < \sqrt{\sigma_\epsilon^2 d - 1}$, we have the desired result. Regarding other inequalities, by Gaussian initialization of parameters, $\mathbf{W}(0)\boldsymbol{\mu}_{+1}$ is normally distributed with mean 0 and covariance $\sigma_w^2 \|\boldsymbol{\mu}\|_2^2 \mathbf{I}_d$, and we repeat the same discussion. For $\|\mathbf{W}(0)\boldsymbol{\epsilon}_t^{(i)}\|_2$, considering the probability for parameter initialization under the realization of the training set, we can use the same argument and union bound. The inequality for $\mathbf{p}(0)$ is derived similarly. $\qquad\square$

Next, we move on to the concentration inequality for Gaussian random variables.

**Lemma B.7** (Gaussian tail bound, ((Vershynin, 2018), Prop 2.1.2)). *For a Gaussian variable $x \sim N(0, \sigma^2)$, the tail bound is given by*

$$\left( \frac{\sigma}{w} - \frac{\sigma^3}{w^3} \right) \cdot \frac{1}{\sqrt{2\pi}} \exp\left( -\frac{w^2}{2\sigma^2} \right) \leq \Pr\{x \geq w\} \leq \frac{1}{w} \cdot \frac{\sigma}{\sqrt{2\pi}} \exp\left( -\frac{w^2}{2\sigma^2} \right).$$

Using this, we can show the following result for the inner products of the noise vectors.

**Lemma B.8** (Inner-product of two Gaussian). *There exists some constant $c_1 > 0$ such that with the probability at least $1 - \delta/4$,*

$$|\langle \boldsymbol{\epsilon}_t^{(i)}, \boldsymbol{\epsilon}_u^{(j)} \rangle| < c_1 \sigma_\epsilon^2 \sqrt{d} \log(Tn/\delta),$$

$$|\langle \mathbf{W}(0)\boldsymbol{\mu}_{+1}, \mathbf{W}(0)\boldsymbol{\mu}_{-1} \rangle| < c_1 \sigma_w^2 \|\boldsymbol{\mu}\|_2^2 \sqrt{d} \log(Tn/\delta),$$

$$|\langle \mathbf{W}(0)\boldsymbol{\mu}_{+1}, \mathbf{W}(0)\boldsymbol{\epsilon}_t^{(i)} \rangle| < c_1 \sigma_w^2 \|\boldsymbol{\mu}\|_2 \|\boldsymbol{\epsilon}_t^{(i)}\|_2 \sqrt{d} \log(Tn/\delta),$$

$$|\langle \mathbf{W}(0)\boldsymbol{\mu}_{-1}, \mathbf{W}(0)\boldsymbol{\epsilon}_t^{(i)} \rangle| < c_1 \sigma_w^2 \|\boldsymbol{\mu}\|_2 \|\boldsymbol{\epsilon}_t^{(i)}\|_2 \sqrt{d} \log(Tn/\delta),$$

$$|\langle \mathbf{W}(0)\boldsymbol{\epsilon}_t^{(i)}, \mathbf{W}(0)\boldsymbol{\epsilon}_u^{(j)} \rangle| < c_1 \sigma_w^2 \|\boldsymbol{\epsilon}_t^{(i)}\|_2 \|\boldsymbol{\epsilon}_u^{(j)}\|_2 \sqrt{d} \log(Tn/\delta),$$

$$|\langle \mathbf{W}(0)\boldsymbol{\mu}_{+1}, \mathbf{p}(0) \rangle| < c_1 \sigma_w \sigma_p \|\boldsymbol{\mu}\|_2 \sqrt{d} \log(Tn/\delta),$$

$$|\langle \mathbf{W}(0)\boldsymbol{\mu}_{-1}, \mathbf{p}(0) \rangle| < c_1 \sigma_w \sigma_p \|\boldsymbol{\mu}\|_2 \sqrt{d} \log(Tn/\delta),$$

$$|\langle \mathbf{W}(0)\boldsymbol{\epsilon}_t^{(i)}, \mathbf{p}(0) \rangle| < c_1 \sigma_w \sigma_p \|\boldsymbol{\epsilon}_t^{(i)}\|_2 \sqrt{d} \log(Tn/\delta),$$

*for all $i, j \in [n], t, u \in [T]$ such that $(i, t) \neq (j, u)$.*

*Proof of Lemma B.8.* Before delving into the main part of the proof, we first show

$$\Pr\{\|\boldsymbol{\epsilon}\|_2 \geq w\} \leq 2\exp\left( -\frac{w^2}{2\sigma_\epsilon^2 d} \right), \tag{40}$$

for the Gaussian vector $\boldsymbol{\epsilon} \sim N(\mathbf{0}, \sigma_\epsilon^2 \mathbf{I}_d)$. Unlike Lemma B.6, it handles the norm itself rather than the deviation around the mean of the norm, which is more useful to prove this lemma. We have for any $\lambda > 0$ that

$$\Pr\{\|\boldsymbol{\epsilon}\|_2 \geq w\} \leq \Pr\{\|\boldsymbol{\epsilon}\|_1 \geq w\} \tag{41}$$

$$\leq \exp(-\lambda w) \cdot \prod_{k=1}^d \mathbb{E}\exp\left(\lambda|\epsilon_k|\right) \tag{42}$$

$$\leq \exp(-\lambda w) \cdot 2\prod_{k=1}^d \mathbb{E}\exp\left(\lambda\epsilon\right) \tag{43}$$

$$= 2\exp\left( \frac{\lambda^2}{2}\sigma_\epsilon^2 d - w\lambda \right), \tag{44}$$

where the second inequality follows from Markov inequality and the last follows from the moment-generating function of Gaussian distribution. Minimizing the upper bound over $\lambda$ gives the desired inequality Equation (40).

Fix $i, j \in [n]$ and $t, u \in [T]$ such that $(i, t) \neq (j, u)$. For any $v, w > 0$, we have

$$\Pr\left\{ |\langle \boldsymbol{\epsilon}_t^{(i)}, \boldsymbol{\epsilon}_u^{(j)} \rangle| > v \right\} \leq \Pr\left\{ |\langle \boldsymbol{\epsilon}_t^{(i)}, \boldsymbol{\epsilon}_u^{(j)} \rangle| > v \mid \sigma_\epsilon \|\boldsymbol{\epsilon}_u^{(j)}\|_2 \leq w \right\} + \Pr\left\{ \sigma_\epsilon \|\boldsymbol{\epsilon}_u^{(j)}\|_2 > w \right\}, \tag{45}$$

where we used the inequality $\Pr(A) = \Pr(B)\Pr(A|B) + \Pr(B^C)\Pr(A|B^C) \leq \Pr(B) + \Pr(A|B^C)$ for the event $A, B$, which gives tighter bound when outlier event $A$ and event $B$ share large common parts. Under the condition $\boldsymbol{\epsilon}_u^{(j)}$ is fixed, since $\langle \boldsymbol{\epsilon}_t^{(i)}, \boldsymbol{\epsilon}_u^{(j)} \rangle$ follows $N(0, \sigma_\epsilon^2 \|\boldsymbol{\epsilon}_u^{(j)}\|_2^2)$, Lemma B.7 gives

$$\Pr\left\{ |\langle \boldsymbol{\epsilon}_t^{(i)}, \boldsymbol{\epsilon}_u^{(j)} \rangle| > v \right\} \leq \frac{2}{v} \cdot \frac{\sigma_\epsilon \|\boldsymbol{\epsilon}_u^{(j)}\|_2}{\sqrt{2\pi}} \exp\left( -\frac{v^2}{2\sigma_\epsilon^2 \|\boldsymbol{\epsilon}_u^{(j)}\|_2^2} \right). \tag{46}$$

Thus, the conditional probability is bounded as

$$\Pr\left\{ |\langle \boldsymbol{\epsilon}_t^{(i)}, \boldsymbol{\epsilon}_u^{(j)} \rangle| > v \mid \sigma_\epsilon \|\boldsymbol{\epsilon}_u^{(j)}\|_2 \leq w \right\} \leq \frac{2}{v} \cdot \frac{w}{\sqrt{2\pi}} \exp\left(-\frac{v^2}{2w^2}\right). \tag{47}$$

Combining Equations (40) and (47), then applying union bound on Equation (45), we obtain

$$\Pr\left\{ \exists i, j \in [n], t, u \in [T], (i,t) \neq (j,u), \text{ s.t. } |\langle \boldsymbol{\epsilon}_t^{(i)}, \boldsymbol{\epsilon}_u^{(j)} \rangle| > v \right\}$$

$$\leq \Pr\left\{ \exists i, j \in [n], t, u \in [T], (i,t) \neq (j,u), \text{ s.t. } |\langle \boldsymbol{\epsilon}_t^{(i)}, \boldsymbol{\epsilon}_u^{(j)} \rangle| > v \mid \forall j \in [n], u \in [T], \sigma_\epsilon \|\boldsymbol{\epsilon}_u^{(j)}\|_2 \leq w \right\}$$

$$+ \Pr\left\{ \exists j \in [n], u \in [T], \text{ s.t. } \sigma_\epsilon \|\boldsymbol{\epsilon}_u^{(j)}\|_2 > w \right\} \tag{48}$$

$$\leq \frac{2wT^2n^2}{\sqrt{2\pi}v} \exp\left(-\frac{v^2}{2w^2}\right) + 2Tn \exp\left(-\frac{w^2}{2\sigma_\epsilon^4 d}\right). \tag{49}$$

Let $w = c_1' \left(\log(Tn/\delta)\right)^{-1/2} v$ for some constant $c_1' > 0$. By Equation (49), we have

$$\Pr\left\{ \exists i, j \in [n], t, u \in [T], (i,t) \neq (j,u), \text{ s.t. } |\langle \boldsymbol{\epsilon}_t^{(i)}, \boldsymbol{\epsilon}_u^{(j)} \rangle| > v \right\}$$

$$\leq \frac{2T^2n^2}{\sqrt{2\pi}} \cdot c_1' \left(\log(Tn/\delta)\right)^{-1/2} \left(\frac{\delta}{Tn}\right)^{1/(2c_1'^2)} + 2Tn \exp\left(-\frac{c_1'^2 v^2}{2\sigma_\epsilon^4 d \log(Tn/\delta)}\right). \tag{50}$$

Further, let $v = c_1 \sigma_\epsilon^2 \sqrt{d} \log(Tn/\delta)$, where $c_1 > 0$ is some constant. We have

$$\Pr\left\{ \exists i, j \in [n], t, u \in [T], (i,t) \neq (j,u), \text{ s.t. } |\langle \boldsymbol{\epsilon}_t^{(i)}, \boldsymbol{\epsilon}_u^{(j)} \rangle| > v \right\}$$

$$\leq \frac{2T^2n^2}{\sqrt{2\pi}} \cdot c_1' \left(\log(Tn/\delta)\right)^{-1/2} \left(\frac{\delta}{Tn}\right)^{1/(2c_1'^2)} + 2Tn \left(\frac{\delta}{Tn}\right)^{c_1'^2 c_1^2/2} \tag{51}$$

$$\leq \frac{\delta}{64} + \frac{\delta}{64} = \frac{\delta}{32}, \tag{52}$$

where the last inequality is satisfied with the appropriate choice of $c_1, c_1' > 0$. This completes the proof for the first inequality. The other inequalities are also the inner product of two Gaussian vectors and can be proved with the same argument. $\qquad\square$

**Lemma B.9** (Inner-product of signal and noise). *There exists some constant $c_2 > 0$ such that with probability at least $1 - \delta/4$,*

$$|\langle \boldsymbol{\mu}_{+1}, \boldsymbol{\epsilon}_t^{(i)} \rangle| < c_2 \sigma_\epsilon \|\boldsymbol{\mu}\|_2 \sqrt{\log(Tn/\delta)},$$

$$|\langle \boldsymbol{\mu}_{-1}, \boldsymbol{\epsilon}_t^{(i)} \rangle| < c_2 \sigma_\epsilon \|\boldsymbol{\mu}\|_2 \sqrt{\log(Tn/\delta)},$$

$$|\langle \boldsymbol{\nu}, \boldsymbol{\epsilon}_t^{(i)} \rangle| < c_2 \sigma_\epsilon \|\boldsymbol{\nu}\|_2 \sqrt{\log(Tn/\delta)}.$$

*for all $i \in [n], t \in [T]$.*

*Proof of Lemma B.9.* We will show that the inequality for $\boldsymbol{\mu}_{+1}$ holds with probability at least $1 - \delta/12$. The same discussion applies to $\boldsymbol{\mu}_{-1}$ and $\boldsymbol{\nu}$. For the fixed $i \in [n], t \in [T]$, since $\langle \boldsymbol{\mu}_{+1}, \boldsymbol{\epsilon}_t^{(i)} \rangle$ follows the Gaussian distribution $N(0, \sigma_\epsilon^2 \|\boldsymbol{\mu}\|_2^2)$, Lemma B.7 gives

$$\Pr\left\{ |\langle \boldsymbol{\mu}_{+1}, \boldsymbol{\epsilon}_t^{(i)} \rangle| > w \right\} \leq \frac{2\sigma_\epsilon \|\boldsymbol{\mu}\|_2}{\sqrt{2\pi}w} \exp\left(-\frac{w^2}{2\sigma_\epsilon^2 \|\boldsymbol{\mu}\|_2^2}\right). \tag{53}$$

Let $w = c_2 \sigma_\epsilon \|\boldsymbol{\mu}\|_2 \sqrt{\log(Tn/\delta)}$ for some constant $c_2 > 0$, then applying union bound on Equation (53) gives

$$\Pr\left\{ \exists i \in [n], t \in [T], \text{ s.t. } |\langle \boldsymbol{\mu}_{+1}, \boldsymbol{\epsilon}_t^{(i)} \rangle| > w \right\} \leq \frac{2Tn\sigma_\epsilon \|\boldsymbol{\mu}\|_2}{\sqrt{2\pi}w} \exp\left(-\frac{w^2}{2\sigma_\epsilon^2 \|\boldsymbol{\mu}\|_2^2}\right) \tag{54}$$

$$\leq \frac{2Tn}{\sqrt{2\pi}c_2 \sqrt{\log(Tn/\delta)}} \left(\frac{\delta}{Tn}\right)^{c_2^2/2} < \frac{\delta}{12}, \tag{55}$$

where the last inequality is satisfied with the appropriate choice of $c_2 > 0$. $\qquad\square$

**Lemma B.10** (Hoeffding Inequality). *Let $X_1, \ldots, X_n$ be i.i.d. random variables such that $0 \leq X \leq 1$ almost surely. Then for all $w > 0$, we have*

$$\Pr\left\{ \left| \frac{1}{n} \sum_{i=1}^{n} X_i - \mathbb{E}[X] \right| > w \right\} \leq 2 \exp\left(-2nw^2\right).$$

**Lemma B.11** (Number of Samples). *For all $c' > 0$, the following hold with probability at least $1 - \delta/4$:*

$$\left(\frac{1-\eta}{2} - c'\right) n \leq |\mathcal{C}_+| \leq \left(\frac{1-\eta}{2} + c'\right) n,$$

$$\left(\frac{1-\eta}{2} - c'\right) n \leq |\mathcal{C}_-| \leq \left(\frac{1-\eta}{2} + c'\right) n,$$

$$\left(\frac{\eta}{2} - c'\right) n \leq |\mathcal{N}_+| \leq \left(\frac{\eta}{2} + c'\right) n,$$

$$\left(\frac{\eta}{2} - c'\right) n \leq |\mathcal{N}_-| \leq \left(\frac{\eta}{2} + c'\right) n.$$

*Proof of Lemma B.11.* We show the first equation holds with probability at least $1 - \delta/16$. The proof of remaining cases follows similarly, and the desired result is achieved by using union bound.

The training data $i \in [n]$ belongs to $\mathcal{C}_+$ when its true label $Y^{(i)} = 1$ and label flip does not occur. This event occurs independently, and its probability is calculated as $(1-\eta)/2$. Since $|\mathcal{C}_+| = \sum_{i=1}^{n} \mathbf{1}_{Y^{(i)} = Y^{*(i)} = 1}$, applying Lemma B.10 to $X_i := \mathbf{1}_{Y^{(i)} = Y^{*(i)} = 1}$ leads

$$\Pr\left\{ \left| |\mathcal{C}_+| - (1-\eta)n/2 \right| > c'n \right\} \leq 2\exp\left(-2nc'^2\right) < \frac{\delta}{16}, \tag{56}$$

where the last inequality follows from $n > C \log(1/\delta)$ in parameter assumptions in Section 3.5. $\qquad\square$

At the end of this section, we prepare an evaluation of the token scores for the training data on a good run.

**Lemma B.12** (Token score). *Suppose that the linear head $\boldsymbol{\nu}$ satisfies Equation (2). Then, on a good run, there exist constants $\{c_k\}_{k \in \{\pm 1\}}$ such that $\forall k \in \{\pm 1\}$, $c_k > 0$ and we have for the clean data $i \in \mathcal{C}$ that*

$$c_{Y^{(i)}} \left(1 - o(1)\right) \|\boldsymbol{\nu}\|_2 \|\boldsymbol{\mu}\|_2 \leq Y^{(i)} \cdot \gamma_1^{(i)} \leq c_{Y^{(i)}} \left(1 + o(1)\right) \|\boldsymbol{\nu}\|_2 \|\boldsymbol{\mu}\|_2,$$

$$Y^{(i)} \cdot \gamma_t^{(i)} = \Theta\left(\rho \|\boldsymbol{\nu}\|_2 \|\boldsymbol{\mu}\|_2\right) > 0,$$

$$Y^{(i)} \cdot \gamma_u^{(i)} = -\Theta\left(\rho \|\boldsymbol{\nu}\|_2 \|\boldsymbol{\mu}\|_2\right) < 0,$$

$$|\gamma_v^{(i)}| = O\left(\sigma_\epsilon \|\boldsymbol{\nu}\|_2 \sqrt{\log(Tn/\delta)}\right),$$

*where $t \in \mathcal{W}_{Y^{(i)}}^{(i)}, u \in \mathcal{W}_{-Y^{(i)}}^{(i)}, v \in \mathcal{I}^{(i)}$. Similarly, for the noisy data $j \in \mathcal{N}$, we have*

$$-c_{-Y^{(j)}} \left(1 - o(1)\right) \|\boldsymbol{\nu}\|_2 \|\boldsymbol{\mu}\|_2 \leq Y^{(j)} \cdot \gamma_1^{(j)} \leq -c_{-Y^{(j)}} \left(1 + o(1)\right) \|\boldsymbol{\nu}\|_2 \|\boldsymbol{\mu}\|_2,$$

$$Y^{(j)} \cdot \gamma_t^{(j)} = -\Theta\left(\rho \|\boldsymbol{\nu}\|_2 \|\boldsymbol{\mu}\|_2\right) < 0,$$

$$Y^{(j)} \cdot \gamma_u^{(j)} = \Theta\left(\rho \|\boldsymbol{\nu}\|_2 \|\boldsymbol{\mu}\|_2\right) > 0,$$

$$|\gamma_v^{(j)}| = O\left(\sigma_\epsilon \|\boldsymbol{\nu}\|_2 \sqrt{\log(Tn/\delta)}\right),$$

*where $t \in \mathcal{W}_{Y^{(j)}}^{(j)}, u \in \mathcal{W}_{-Y^{(j)}}^{(j)}, v \in \mathcal{I}^{(j)}$.*

*Proof of Lemma B.12.* Using Lemma B.1 and Equation (2), we have for $i \in \mathcal{C}_+$ that

$$Y^{(i)} \cdot \gamma_1^{(i)} = \left(\boldsymbol{\mu}_{+1} + \boldsymbol{\epsilon}_1^{(i)}\right)^\top \boldsymbol{\nu} \tag{57}$$

$$= \|\boldsymbol{\nu}\|_2 \|\boldsymbol{\mu}\|_2 \cos\theta_{+1} + O(\sigma_\epsilon \|\boldsymbol{\nu}\| \sqrt{\log(Tn/\delta)}) \tag{58}$$

$$\in (1 \pm o(1)) c_{+1} \|\boldsymbol{\nu}\|_2 \|\boldsymbol{\mu}\|_2, \tag{59}$$

where recall that $\cos\theta_{+1}$ is defined as the angle between $\boldsymbol{\nu}$ and $\boldsymbol{\mu}_{+1}$. In the last line, we used $\sigma_\epsilon \sqrt{\log(Tn/\delta)} = o(\|\boldsymbol{\mu}\|_2)$ in the parameter assumption and replaced $\cos\theta_{+1}$ with a constant $c_{+1}$. The other equations are derived as well. $\qquad\square$

# C. Proof of Theorem 4.1

In this section, we present lemmas concerning the gradient descent dynamics of token selection, and provide the proof of Theorem 4.1. For clarity, the essential lemmas are included in this section, while minor and more technical lemmas are deferred to Appendix D. Appendices C.2 and C.3 provide the main lemmas and the direct proof for the not-overfitting and benign overfitting cases in Theorem 4.1, respectively.

## C.1. Preliminary Lemmas

In order to analyze the complicated dynamics of the token mechanism, we introduce the following notations.

**Definition C.1** (Attention to signal and noise). Let $\mathbf{W}(\tau)$ and $\mathbf{p}(\tau)$ denote the parameters at the $\tau$-th gradient descent step. Then, we define the attention to the signal and the noise as follows:

$$\lambda_{+1}(\tau) := \langle \mathbf{W}(\tau)\boldsymbol{\mu}_{+1}, \mathbf{p}(\tau)\rangle, \ \lambda_{-1}(\tau) := \langle \mathbf{W}(\tau)\boldsymbol{\mu}_{-1}, \mathbf{p}(\tau)\rangle, \ \rho_{i,t}(\tau) := \langle \mathbf{W}(\tau)\boldsymbol{\epsilon}_t^{(i)}, \mathbf{p}(\tau)\rangle.$$

**Definition C.2** (Weighted interaction terms). We define the following interaction terms weighted by softmax probabilities and token scores.

$$I_{i,+}(\tau) := \sum_{t=1}^{T} s_t^{(i)}(\tau)\left(\gamma_t^{(i)} - \sum_{u=1}^{T} s_u^{(i)}(\tau)\gamma_u^{(i)}\right)\langle \mathbf{x}_t^{(i)}, \boldsymbol{\mu}_{+1}\rangle,$$

$$I_{i,-}(\tau) := \sum_{t=1}^{T} s_t^{(i)}(\tau)\left(\gamma_t^{(i)} - \sum_{u=1}^{T} s_u^{(i)}(\tau)\gamma_u^{(i)}\right)\langle \mathbf{x}_t^{(i)}, \boldsymbol{\mu}_{-1}\rangle,$$

$$I_{i,j,u}(\tau) := \sum_{t=1}^{T} s_t^{(i)}(\tau)\left(\gamma_t^{(i)} - \sum_{u=1}^{T} s_u^{(i)}(\tau)\gamma_u^{(i)}\right)\langle \mathbf{x}_t^{(i)}, \boldsymbol{\epsilon}_u^{(j)}\rangle,$$

$$I_{i,+}^{W}(\tau) := \sum_{t=1}^{T} s_t^{(i)}(\tau)\left(\gamma_t^{(i)} - \sum_{u=1}^{T} s_u^{(i)}(\tau)\gamma_u^{(i)}\right)\langle \mathbf{W}(\tau)\mathbf{x}_t^{(i)}, \mathbf{W}(\tau)\boldsymbol{\mu}_{+1}\rangle,$$

$$I_{i,-}^{W}(\tau) := \sum_{t=1}^{T} s_t^{(i)}(\tau)\left(\gamma_t^{(i)} - \sum_{u=1}^{T} s_u^{(i)}(\tau)\gamma_u^{(i)}\right)\langle \mathbf{W}(\tau)\mathbf{x}_t^{(i)}, \mathbf{W}(\tau)\boldsymbol{\mu}_{-1}\rangle,$$

$$I_{i,j,u}^{W}(\tau) := \sum_{t=1}^{T} s_t^{(i)}(\tau)\left(\gamma_t^{(i)} - \sum_{u=1}^{T} s_u^{(i)}(\tau)\gamma_u^{(i)}\right)\langle \mathbf{W}(\tau)\mathbf{x}_t^{(i)}, \mathbf{W}(\tau)\boldsymbol{\epsilon}_u^{(j)}\rangle,$$

$$I_{i}^{p}(\tau) := \sum_{t=1}^{T} s_t^{(i)}(\tau)\left(\gamma_t^{(i)} - \sum_{u=1}^{T} s_u^{(i)}(\tau)\gamma_u^{(i)}\right)\langle \mathbf{W}(\tau)\mathbf{x}_t^{(i)}, \mathbf{p}(\tau)\rangle.$$

The next definition is the restatement of Definition 4.4 in the main text.

**Definition C.3** (Attention gap between significant token and other tokens). We define the following values representing the attention gap between a relevant token and other tokens:

$$\Lambda_{i,t}(\tau) := \left(\mathbf{x}_1^{(i)} - \mathbf{x}_t^{(i)}\right)^{\top} \mathbf{W}(\tau)^{\top}\mathbf{p}(\tau),$$

for $i \in [n], t \in [T] \setminus \{1\}$. Additionally, we define the attention gap between a confusing weakly relevant token and other tokens:

$$\Gamma_{i,u}(\tau) := \left(\mathbf{x}_2^{(i)} - \mathbf{x}_u^{(i)}\right)^{\top} \mathbf{W}(\tau)^{\top}\mathbf{p}(\tau),$$

for $i \in [n], u \in [T] \setminus \{2\}$.

For clarity, we provide below the results of applying the data setup in Definition 3.1 to these values. For $i \in \mathcal{C}_+ \cup \mathcal{N}_- =$

$\{i \in [n] \mid Y^{*(i)} = +1\}, j \in \mathcal{C}_- \cup \mathcal{N}_+ = \{i \in [n] \mid Y^{*(i)} = -1\}$, we have

$$\Lambda_{i,t}(\tau) = \begin{cases} (1-\rho)\lambda_{+1}(\tau) + \rho_{i,1}(\tau) - \rho_{i,t}(\tau) & \text{for } t \in \mathcal{W}_{+1}^{(i)}, \\ \lambda_{+1}(\tau) - \rho\lambda_{-1}(\tau) + \rho_{i,1}(\tau) - \rho_{i,t}(\tau) & \text{for } t \in \mathcal{W}_{-1}^{(i)} = \{2\}, \\ \lambda_{+1}(\tau) + \rho_{i,1}(\tau) - \rho_{i,t}(\tau) & \text{for } t \in \mathcal{I}^{(i)}, \end{cases}$$

$$\Lambda_{j,t}(\tau) = \begin{cases} \lambda_{-1}(\tau) - \rho\lambda_{+1}(\tau) + \rho_{j,1}(\tau) - \rho_{j,t}(\tau) & \text{for } t \in \mathcal{W}_{+1}^{(j)} = \{2\}, \\ (1-\rho)\lambda_{-1}(\tau) + \rho_{j,1}(\tau) - \rho_{j,t}(\tau) & \text{for } t \in \mathcal{W}_{-1}^{(j)}, \\ \lambda_{-1}(\tau) + \rho_{j,1}(\tau) - \rho_{j,t}(\tau) & \text{for } t \in \mathcal{I}^{(j)}, \end{cases}$$

for $t \in [T] \setminus \{1\}$, and we have

$$\Gamma_{i,u}(\tau) = \begin{cases} \rho\lambda_{-1}(\tau) - \lambda_{+1}(\tau) + \rho_{i,2}(\tau) - \rho_{i,u}(\tau) & \text{for } u \in \mathcal{R} = \{1\}, \\ \rho\lambda_{-1}(\tau) - \rho\lambda_{+1}(\tau) + \rho_{i,2}(\tau) - \rho_{i,u}(\tau) & \text{for } u \in \mathcal{W}_{+1}^{(i)}, \\ \rho\lambda_{-1}(\tau) + \rho_{i,2}(\tau) - \rho_{i,u}(\tau) & \text{for } u \in \mathcal{I}^{(i)}, \end{cases}$$

$$\Gamma_{j,u}(\tau) = \begin{cases} \rho\lambda_{+1}(\tau) - \lambda_{-1}(\tau) + \rho_{j,2}(\tau) - \rho_{j,u}(\tau) & \text{for } u \in \mathcal{R} = \{1\}, \\ \rho\lambda_{+1}(\tau) - \rho\lambda_{-1}(\tau) + \rho_{j,2}(\tau) - \rho_{j,u}(\tau) & \text{for } u \in \mathcal{W}_{-1}^{(j)}, \\ \rho\lambda_{+1}(\tau) + \rho_{j,2}(\tau) - \rho_{j,u}(\tau) & \text{for } u \in \mathcal{I}^{(j)}, \end{cases}$$

for $u \in [T] \setminus \{2\}$.

In the following, we calculate the one-step updates for various quantities that will be frequently used in subsequent proofs.

**Lemma C.4** (Updates of signal and noise attention). *The updates of $\lambda_{+1}(\tau), \lambda_{-1}(\tau),$ and $\rho_{j,u}(\tau)$ for $j \in [n], u \in [T]$, which are defined in Definition C.1, are given by*

$$\lambda_{+1}(\tau+1) - \lambda_{+1}(\tau) = \frac{\alpha}{n}\sum_{i=1}^{n}(-\ell_i'(\tau)) \cdot Y^{(i)} \cdot \left(I_{i,+}^W(\tau) + \|\mathbf{p}(\tau)\|_2^2 I_{i,+}(\tau)\right) + \alpha^2 \boldsymbol{\mu}_{+1}^\top \nabla_{\mathbf{W}^\top}\widehat{\mathcal{L}}(\tau)\nabla_{\mathbf{p}}\widehat{\mathcal{L}}(\tau),$$

$$\lambda_{-1}(\tau+1) - \lambda_{-1}(\tau) = \frac{\alpha}{n}\sum_{i=1}^{n}(-\ell_i'(\tau)) \cdot Y^{(i)} \cdot \left(I_{i,-}^W(\tau) + \|\mathbf{p}(\tau)\|_2^2 I_{i,-}(\tau)\right) + \alpha^2 \boldsymbol{\mu}_{-1}^\top \nabla_{\mathbf{W}^\top}\widehat{\mathcal{L}}(\tau)\nabla_{\mathbf{p}}\widehat{\mathcal{L}}(\tau),$$

$$\rho_{j,u}(\tau+1) - \rho_{j,u}(\tau) = \frac{\alpha}{n}\sum_{i=1}^{n}(-\ell_i'(\tau)) \cdot Y^{(i)} \cdot \left(I_{i,j,u}^W(\tau) + \|\mathbf{p}(\tau)\|_2^2 I_{i,j,u}(\tau)\right) + \alpha^2 \boldsymbol{\epsilon}_u^{(j)\top} \nabla_{\mathbf{W}^\top}\widehat{\mathcal{L}}(\tau)\nabla_{\mathbf{p}}\widehat{\mathcal{L}}(\tau).$$

*Proof of Lemma C.4.* From Equations (9) and (12), we have

$$\mathbf{W}(\tau+1)^\top\mathbf{p}(\tau+1) - \mathbf{W}(\tau)^\top\mathbf{p}(\tau) \tag{60}$$

$$= \mathbf{W}(\tau)^\top\left(-\alpha\nabla_{\mathbf{p}}\widehat{\mathcal{L}}(\tau)\right) + \left(-\alpha\nabla_{\mathbf{W}}\widehat{\mathcal{L}}(\tau)\right)^\top\mathbf{p}(\tau) + \alpha^2\left(\nabla_{\mathbf{W}}\widehat{\mathcal{L}}(\tau)\right)^\top\left(\nabla_{\mathbf{p}}\widehat{\mathcal{L}}(\tau)\right) \tag{61}$$

$$= \frac{\alpha}{n}\sum_{i=1}^{n}(-\ell_i'(\tau)) \cdot Y^{(i)} \cdot \left(\sum_{t=1}^{T}s_t^{(i)}(\tau)\left(\gamma_t^{(i)} - \sum_{u=1}^{T}s_u^{(i)}(\tau)\gamma_u^{(i)}\right)\left(\mathbf{W}(\tau)^\top\mathbf{W}(\tau) + \|\mathbf{p}(\tau)\|_2^2\right)\mathbf{x}_t^{(i)}\right)$$
$$+ \alpha^2\nabla_{\mathbf{W}^\top}\widehat{\mathcal{L}}(\tau)\nabla_{\mathbf{p}}\widehat{\mathcal{L}}(\tau). \tag{62}$$

Following the definitions of $\lambda$ and $\rho$, taking inner products with $\boldsymbol{\mu}_{+1}, \boldsymbol{\mu}_{-1},$ and $\boldsymbol{\epsilon}_u^{(j)}$ yields the desired update equations. The notations defined in Definition C.2 are applied here. $\square$

**Lemma C.5** (Updates of $\mathbf{p}$). *The update of $\|\mathbf{p}(\tau)\|_2$ is given by*

$$\|\mathbf{p}(\tau+1)\|_2^2 - \|\mathbf{p}(\tau)\|_2^2 = \frac{2\alpha}{n}\sum_{i=1}^{n}(-\ell_i'(\tau)) \cdot Y^{(i)} \cdot I_i^p(\tau) + \alpha^2\|\nabla_{\mathbf{p}}\widehat{\mathcal{L}}(\tau)\|_2^2.$$

*Proof of Lemma C.5.* From Equation (12), we have

$$\|\mathbf{p}(\tau+1)\|_2^2 - \|\mathbf{p}(\tau)\|_2^2 = 2\langle \mathbf{p}(\tau), -\alpha\nabla_{\mathbf{p}}\widehat{\mathcal{L}}(\tau)\rangle + \|\alpha\nabla_{\mathbf{p}}\widehat{\mathcal{L}}(\tau)\|_2^2 \tag{63}$$

$$= \frac{2\alpha}{n}\sum_{i=1}^{n}(-\ell_i'(\tau))\cdot Y^{(i)}\cdot I_i^p(\tau) + \alpha^2\|\nabla_{\mathbf{p}}\widehat{\mathcal{L}}(\tau)\|_2^2, \tag{64}$$

where in the last line, we used the notation defined in Definition C.2. $\qquad\square$

**Lemma C.6** (Updates of $\mathbf{W}$). *The updates of the terms related to $\mathbf{W}(\tau)$ are given by*

$$\|\mathbf{W}(\tau+1)\boldsymbol{\mu}_{+1}\|_2^2 - \|\mathbf{W}(\tau)\boldsymbol{\mu}_{+1}\|_2^2 = \frac{2\alpha}{n}\sum_{i=1}^{n}(-\ell_i'(\tau))\cdot Y^{(i)}\cdot I_{i,+}(\tau)\lambda_{+1}(\tau) + \alpha^2\|\nabla_{\mathbf{W}}\widehat{\mathcal{L}}(\tau)\boldsymbol{\mu}_{+1}\|_2^2,$$

$$\|\mathbf{W}(\tau+1)\boldsymbol{\mu}_{-1}\|_2^2 - \|\mathbf{W}(\tau)\boldsymbol{\mu}_{-1}\|_2^2 = \frac{2\alpha}{n}\sum_{i=1}^{n}(-\ell_i'(\tau))\cdot Y^{(i)}\cdot I_{i,-}(\tau)\lambda_{-1}(\tau) + \alpha^2\|\nabla_{\mathbf{W}}\widehat{\mathcal{L}}(\tau)\boldsymbol{\mu}_{-1}\|_2^2,$$

$$\|\mathbf{W}(\tau+1)\boldsymbol{\epsilon}_u^{(j)}\|_2^2 - \|\mathbf{W}(\tau)\boldsymbol{\epsilon}_u^{(j)}\|_2^2 = \frac{2\alpha}{n}\sum_{i=1}^{n}(-\ell_i'(\tau))\cdot Y^{(i)}\cdot I_{i,j,u}(\tau)\rho_{j,u}(\tau) + \alpha^2\|\nabla_{\mathbf{W}}\widehat{\mathcal{L}}(\tau)\boldsymbol{\epsilon}_u^{(j)}\|_2^2,$$

*for any $j \in [n], u \in [T]$. Additionally, we have*

$$\langle \mathbf{W}(\tau+1)\boldsymbol{\mu}_{+1}, \mathbf{W}(\tau+1)\boldsymbol{\mu}_{-1}\rangle - \langle \mathbf{W}(\tau)\boldsymbol{\mu}_{+1}, \mathbf{W}(\tau)\boldsymbol{\mu}_{-1}\rangle$$
$$= \frac{\alpha}{n}\sum_{i=1}^{n}(-\ell_i'(\tau))\cdot Y^{(i)}\cdot (I_{i,-}(\tau)\lambda_{+1}(\tau) + I_{i,+}(\tau)\lambda_{-1}(\tau)) + \alpha^2\langle\nabla_{\mathbf{W}}\widehat{\mathcal{L}}(\tau)\boldsymbol{\mu}_{+1}, \nabla_{\mathbf{W}}\widehat{\mathcal{L}}(\tau)\boldsymbol{\mu}_{-1}\rangle,$$

$$\langle \mathbf{W}(\tau+1)\boldsymbol{\mu}_{+1}, \mathbf{W}(\tau+1)\boldsymbol{\epsilon}_u^{(j)}\rangle - \langle \mathbf{W}(\tau)\boldsymbol{\mu}_{+1}, \mathbf{W}(\tau)\boldsymbol{\epsilon}_u^{(j)}\rangle$$
$$= \frac{\alpha}{n}\sum_{i=1}^{n}(-\ell_i'(\tau))\cdot Y^{(i)}\cdot (I_{i,j,u}(\tau)\lambda_{+1}(\tau) + I_{i,+}(\tau)\rho_{j,u}(\tau)) + \alpha^2\langle\nabla_{\mathbf{W}}\widehat{\mathcal{L}}(\tau)\boldsymbol{\mu}_{+1}, \nabla_{\mathbf{W}}\widehat{\mathcal{L}}(\tau)\boldsymbol{\epsilon}_u^{(j)}\rangle,$$

$$\langle \mathbf{W}(\tau+1)\boldsymbol{\mu}_{-1}, \mathbf{W}(\tau+1)\boldsymbol{\epsilon}_u^{(j)}\rangle - \langle \mathbf{W}(\tau)\boldsymbol{\mu}_{-1}, \mathbf{W}(\tau)\boldsymbol{\epsilon}_u^{(j)}\rangle$$
$$= \frac{\alpha}{n}\sum_{i=1}^{n}(-\ell_i'(\tau))\cdot Y^{(i)}\cdot (I_{i,j,u}(\tau)\lambda_{-1}(\tau) + I_{i,-}(\tau)\rho_{j,u}(\tau)) + \alpha^2\langle\nabla_{\mathbf{W}}\widehat{\mathcal{L}}(\tau)\boldsymbol{\mu}_{-1}, \nabla_{\mathbf{W}}\widehat{\mathcal{L}}(\tau)\boldsymbol{\epsilon}_u^{(j)}\rangle,$$

$$\langle \mathbf{W}(\tau+1)\boldsymbol{\epsilon}_u^{(j)}, \mathbf{W}(\tau+1)\boldsymbol{\epsilon}_v^{(k)}\rangle - \langle \mathbf{W}(\tau)\boldsymbol{\epsilon}_u^{(j)}, \mathbf{W}(\tau)\boldsymbol{\epsilon}_v^{(k)}\rangle$$
$$= \frac{\alpha}{n}\sum_{i=1}^{n}(-\ell_i'(\tau))\cdot Y^{(i)}\cdot (I_{i,k,v}(\tau)\rho_{j,u}(\tau) + I_{i,j,u}(\tau)\rho_{k,v}(\tau)) + \alpha^2\langle\nabla_{\mathbf{W}}\widehat{\mathcal{L}}(\tau)\boldsymbol{\epsilon}_u^{(j)}, \nabla_{\mathbf{W}}\widehat{\mathcal{L}}(\tau)\boldsymbol{\epsilon}_v^{(k)}\rangle,$$

*for any $j, k \in [n], u, v \in [T], (j,u) \neq (k,v)$.*

*Proof of Lemma C.6.* From Equation (9), we have

$$\|\mathbf{W}(\tau+1)\boldsymbol{\mu}_{+1}\|_2^2 - \|\mathbf{W}(\tau)\boldsymbol{\mu}_{+1}\|_2^2$$
$$= 2\alpha\langle \mathbf{W}(\tau)\boldsymbol{\mu}_{+1}, -\alpha\nabla_{\mathbf{W}}\widehat{\mathcal{L}}(\tau)\boldsymbol{\mu}_{+1}\rangle + \|\alpha\nabla_{\mathbf{W}}\widehat{\mathcal{L}}(\tau)\boldsymbol{\mu}_{+1}\|_2^2 \tag{65}$$
$$= \frac{2\alpha}{n}\sum_{i=1}^{n}(-\ell_i'(\tau))\cdot Y^{(i)}\cdot\left(\sum_{t=1}^{T}s_t^{(i)}\left(\gamma_t^{(i)} - \sum_{u=1}^{T}s_u^{(i)}\gamma_u^{(i)}\right)\cdot\left(\boldsymbol{\mu}_{+1}^{\top}\mathbf{x}_t^{(i)}\right)\cdot\mathbf{p}(\tau)^{\top}\mathbf{W}(\tau)\boldsymbol{\mu}_{+1}\right) + \alpha^2\|\nabla_{\mathbf{W}}\widehat{\mathcal{L}}(\tau)\boldsymbol{\mu}_{+1}\|_2^2 \tag{66}$$
$$= \frac{2\alpha}{n}\sum_{i=1}^{n}(-\ell_i'(\tau))\cdot Y^{(i)}\cdot I_{i,+}(\tau)\lambda_{+1}(\tau) + \alpha^2\|\nabla_{\mathbf{W}}\widehat{\mathcal{L}}(\tau)\boldsymbol{\mu}_{+1}\|_2^2, \tag{67}$$

where in the last line, we used the notations introduced in Definitions C.1 and C.2. The other two equations are shown in a

similar manner. In the same way, we have

$$\langle \mathbf{W}(\tau+1)\boldsymbol{\mu}_{+1}, \mathbf{W}(\tau+1)\boldsymbol{\mu}_{-1}\rangle - \langle \mathbf{W}(\tau)\boldsymbol{\mu}_{+1}, \mathbf{W}(\tau)\boldsymbol{\mu}_{-1}\rangle$$

$$= \langle \mathbf{W}(\tau)\boldsymbol{\mu}_{+1}, -\alpha\nabla_{\mathbf{W}}\widehat{\mathcal{L}}(\tau)\boldsymbol{\mu}_{-1}\rangle + \langle \mathbf{W}(\tau)\boldsymbol{\mu}_{-1}, -\alpha\nabla_{\mathbf{W}}\widehat{\mathcal{L}}(\tau)\boldsymbol{\mu}_{+1}\rangle + \langle \alpha\nabla_{\mathbf{W}}\widehat{\mathcal{L}}(\tau)\boldsymbol{\mu}_{+1}, \alpha\nabla_{\mathbf{W}}\widehat{\mathcal{L}}(\tau)\boldsymbol{\mu}_{-1}\rangle \qquad (68)$$

$$= \frac{\alpha}{n}\sum_{i=1}^{n}(-\ell_i'(\tau)) \cdot Y^{(i)} \cdot \left(\sum_{t=1}^{T} s_t^{(i)}\left(\gamma_t^{(i)} - \sum_{u=1}^{T} s_u^{(i)}\gamma_u^{(i)}\right) \cdot \left(\boldsymbol{\mu}_{-1}^{\top}\mathbf{x}_t^{(i)}\right) \cdot \mathbf{p}(\tau)^{\top}\mathbf{W}(\tau)\boldsymbol{\mu}_{+1}\right)$$

$$+ \frac{\alpha}{n}\sum_{i=1}^{n}(-\ell_i'(\tau)) \cdot Y^{(i)} \cdot \left(\sum_{t=1}^{T} s_t^{(i)}\left(\gamma_t^{(i)} - \sum_{u=1}^{T} s_u^{(i)}\gamma_u^{(i)}\right) \cdot \left(\boldsymbol{\mu}_{+1}^{\top}\mathbf{x}_t^{(i)}\right) \cdot \mathbf{p}(\tau)^{\top}\mathbf{W}(\tau)\boldsymbol{\mu}_{-1}\right)$$

$$+ \alpha^2\langle\nabla_{\mathbf{W}}\widehat{\mathcal{L}}(\tau)\boldsymbol{\mu}_{+1}, \nabla_{\mathbf{W}}\widehat{\mathcal{L}}(\tau)\boldsymbol{\mu}_{-1}\rangle \qquad (69)$$

$$= \frac{\alpha}{n}\sum_{i=1}^{n}(-\ell_i'(\tau)) \cdot Y^{(i)} \cdot (I_{i,-}(\tau)\lambda_{+1}(\tau) + I_{i,+}(\tau)\lambda_{-1}(\tau)) + \alpha^2\langle\nabla_{\mathbf{W}}\widehat{\mathcal{L}}(\tau)\boldsymbol{\mu}_{+1}, \nabla_{\mathbf{W}}\widehat{\mathcal{L}}(\tau)\boldsymbol{\mu}_{-1}\rangle. \qquad (70)$$

The other equations are derived as well. □

In the following, we prepare the lemmas concerning the weight initialization.

**Lemma C.7** (Softmax probabilities at initialization). *Under the parameter assumptions in Section 3.5 and on a good run, we have that for any constant $c > 1$,*

$$\frac{1 - c^{-1}}{T} \leq s_t^{(i)}(0) \leq \frac{1 + c^{-1}}{T},$$

*for all $i \in [n]$ and $t \in [T]$.*

*Proof of Lemma C.7.* By definition, we have

$$s_t^{(i)}(0) = \frac{\exp\left(\mathbf{x}_t^{(i)\top}\mathbf{W}(0)^{\top}\mathbf{p}(0)\right)}{\sum_{u=1}^{T}\exp\left(\mathbf{x}_u^{(i)\top}\mathbf{W}(0)^{\top}\mathbf{p}(0)\right)} \leq \frac{1}{T}\max_{u\in[T]}\left\{\exp\left(\left(\mathbf{x}_t^{(i)} - \mathbf{x}_u^{(i)}\right)^{\top}\mathbf{W}(0)^{\top}\mathbf{p}(0)\right)\right\}. \qquad (71)$$

From Equations (23) to (25), we have

$$\max_{u\in[T]}\left\{\left(\mathbf{x}_t^{(i)} - \mathbf{x}_u^{(i)}\right)^{\top}\mathbf{W}(0)^{\top}\mathbf{p}(0)\right\} \leq 3c_1\sigma_w\sigma_p\max\left\{\|\boldsymbol{\mu}\|_2\sqrt{d}, (1 + o(1))\sigma_\epsilon d\right\}\log(Tn/\delta) \qquad (72)$$

$$\leq \log\left(1 + c^{-1}\right), \qquad (73)$$

where the last inequality follows from Assumption A8. The same argument is applied to the lower bound, which completes the proof. □

Using a similar argument to that in Lemma C.7, we confirm that the following lemma holds.

**Lemma C.8** (Attention gap at initialization). *Under the parameter assumptions in Section 3.5 and on a good run, the attention gaps introduced in Definition C.3 at time step $\tau = 0$ are bounded as follows:*

$$|\Lambda_{i,t}(0)| = o(1), \ |\Gamma_{i,u}(0)| = o(1),$$

*for all $i \in [n], t \in [T] \setminus \{1\}$, and $u \in [T] \setminus \{2\}$.*

*Proof of Lemma C.8.* We show the case of $t \in \mathcal{I}^{(i)}$. Since $\rho < 1/C$ from the parameter assumptions, we have the same result for $t \in \mathcal{W}_{+1}^{(i)}$ and $t \in \mathcal{W}_{-1}^{(i)}$. By Definition C.3, we have

$$|\Lambda_{i,t}(0)| \leq 3\max\left\{|\lambda_{+1}(0)|, |\lambda_{-1}(0)|, \max_{i\in[n],t\in[T]}\{|\rho_{i,t}(0)|\}\right\}. \qquad (74)$$

From Equations (23) to (25), we have

$$\max\left\{|\lambda_{+1}(0)|, |\lambda_{-1}(0)|, \max_{i\in[n], t\in[T]}\{|\rho_{i,t}(0)|\}\right\} \leq c_1\sigma_w\sigma_p \max\left\{\|\boldsymbol{\mu}\|_2\sqrt{d}, (1+1/C')\sigma_\epsilon d\right\}\log(Tn/\delta) = o(1), \quad (75)$$

where the last inequality follows from Assumption A8. The exact same argument is applied to $\Gamma_{i,u}(0)$. This concludes the proof. $\qquad\square$

Next, we show that the ratio of the loss derivative can be bounded by a constant. As a byproduct, we obtain that the loss derivative itself is bounded by a constant from both sides.

**Lemma C.9** (Ratio of loss derivative). *Suppose that the assumptions in Theorem 4.1 are satisfied. There exists an absolute constant $c_\ell > 0$ such that on a good run, we have that for all time step $\tau \geq 0$,*

$$\max_{i,j\in[n]}\frac{\ell_i'(\tau)}{\ell_j'(\tau)} < c_\ell.$$

*More strongly, $-\ell_i'(\tau)$ is bounded both above and below by positive constants, uniformly over all $i \in [n]$ and $\tau \geq 0$.*

*Proof of Lemma C.9.* Recall that the derivative of the loss function is given by

$$-\ell_i'(\tau) = \frac{1}{1 + \exp\left(-Y^{(i)} \cdot \sum_{t=1}^T s_t^{(i)}(\tau)\gamma_t^{(i)}\right)}, \quad (76)$$

for $i \in [n], \tau \geq 0$. On a good run, Combining Lemma B.12 and $\|\boldsymbol{\nu}\|_2 = O(1/\|\boldsymbol{\mu}\|_2)$ gives the following token scores:

$$|\gamma_1^{(i)}| = O(1), \ |\gamma_t^{(i)}| = O(\rho), \ |\gamma_u^{(i)}| = o(1), \quad (77)$$

for all $i \in [n]$, $t \in \mathcal{W}^{(i)}$, and $u \in \mathcal{I}^{(i)}$. Thus, there exists some constant $c > 0$ such that $|\gamma_t^{(i)}| < c$ for any $i \in [n]$ and $t \in [T]$. Since $1/(1 + \exp(-x))$ is monotonically decreasing, we have

$$\frac{1}{1 + \exp(c)} < -\ell_i'(\tau) < \frac{1}{1 + \exp(-c)}. \quad (78)$$

This leads to the conclusion with the constant $c_\ell = 1 + \exp(c)/\left(1 + \exp(-c)\right)$. $\qquad\square$

**Remark C.10** (Ratio of loss derivative). This lemma, which shows that the gradients of loss function for clean data and noisy data remain within a constant factor of each other at every time step, is a critical component of the proof in the existing analyses of linear classifiers and two-layer neural networks (Chatterji & Long, 2021; Frei et al., 2022; Xu & Gu, 2023). However, in the learning of token selection, the output is always an affine combination of the token scores, and the output scale is not changed. Therefore, the training dynamics need not be considered as long as the balance of the loss derivatives in the token scores is maintained. To ensure that the derivative of the loss function for each token remains within a constant factor, a small linear head scale, as described in Lemma C.9, is required. If the scale of the linear head is too large, little gradient will be generated for clean data even at the initial weights, and learning the signal vectors will not progress.

**C.2. Analysis of Not Overfitting Case** ($\mathrm{SNR}^2 = \omega(n^{-1})$)

In this section, we provide the proof for the not-overfitting case in the main theorem. We first present the main lemma using mathematical induction and then proceed to prove the main claim.

C.2.1. MAIN LEMMA

**Lemma C.11.** *Suppose that the signal-to-noise ratio satisfies* $\mathrm{SNR}^2 = \omega(n^{-1})$. *For any time step* $T_1 = \Theta\left(\frac{1}{\alpha\|\boldsymbol{\nu}\|_2\|\boldsymbol{\mu}\|_2^3 d\max\{\sigma_w^2, \sigma_p^2\}}\right)$, *on a good run, the following propositions hold for all time step* $\tau \in [0, T_1]$:

> $A(\tau)$: *There exists a constant* $C_1 > 1$ *such that*
>
> $$(1 - 1/C_1) \cdot \sigma_p\sqrt{d} \leq \|\mathbf{p}(\tau)\|_2 \leq (1 + 1/C_1) \cdot \sigma_p\sqrt{d}.$$

$B(\tau)$**:** *There exists a constant $C_2 > 1$ such that*

$$(1 - 1/C_2) \cdot \sigma_w \|\boldsymbol{\mu}\|_2 \sqrt{d} \leq \|\mathbf{W}(\tau)\boldsymbol{\mu}_{+1}\|_2 \leq (1 + 1/C_2) \cdot \sigma_w \|\boldsymbol{\mu}\|_2 \sqrt{d},$$
$$(1 - 1/C_2) \cdot \sigma_w \|\boldsymbol{\mu}\|_2 \sqrt{d} \leq \|\mathbf{W}(\tau)\boldsymbol{\mu}_{-1}\|_2 \leq (1 + 1/C_2) \cdot \sigma_w \|\boldsymbol{\mu}\|_2 \sqrt{d},$$
$$(1 - 1/C_2) \cdot \sigma_w \sigma_\epsilon d \leq \|\mathbf{W}(\tau)\boldsymbol{\epsilon}_t^{(i)}\|_2 \leq (1 + 1/C_2) \cdot \sigma_w \sigma_\epsilon d,$$

*for all $i \in [n]$ and $t \in [T]$.*

$$|\langle \mathbf{W}(\tau)\boldsymbol{\mu}_{+1}, \mathbf{W}(\tau)\boldsymbol{\mu}_{-1}\rangle| < O(\sigma_w^2 \|\boldsymbol{\mu}\|_2^2 \sqrt{d} \log(Tn/\delta)),$$
$$|\langle \mathbf{W}(\tau)\boldsymbol{\mu}_{+1}, \mathbf{W}(\tau)\boldsymbol{\epsilon}_t^{(i)}\rangle| < O(\sigma_w^2 \sigma_\epsilon \|\boldsymbol{\mu}\|_2 d \log(Tn/\delta)),$$
$$|\langle \mathbf{W}(\tau)\boldsymbol{\mu}_{-1}, \mathbf{W}(\tau)\boldsymbol{\epsilon}_t^{(i)}\rangle| < O(\sigma_w^2 \sigma_\epsilon \|\boldsymbol{\mu}\|_2 d \log(Tn/\delta)),$$
$$|\langle \mathbf{W}(\tau)\boldsymbol{\epsilon}_t^{(i)}, \mathbf{W}(\tau)\boldsymbol{\epsilon}_u^{(j)}\rangle| < O(\sigma_w^2 \sigma_\epsilon^2 d^{3/2} \log(Tn/\delta)),$$

*for all $i, j \in [n]$ and $t, u \in [T]$ such that $(i, t) \neq (j, u)$.*

$C(\tau)$**:** *Let $g$ be a monotonically increasing function $g(x) = 2x + 2\sinh(x - \log T)$. Then, there exist constants $c_3, c_4 > 0$ with $c_3 < c_4$ such that*

$$g(\Lambda_{i,t}(\tau)) \geq g(\Lambda_{i,t}(0)) + \tau \cdot c_3 \alpha \|\boldsymbol{\nu}\|_2 \|\boldsymbol{\mu}\|_2^3 d \max\{\sigma_w^2, \sigma_p^2\},$$
$$g(\Lambda_{i,t}(\tau)) \leq g(\Lambda_{i,t}(0)) + \tau \cdot c_4 \alpha \|\boldsymbol{\nu}\|_2 \|\boldsymbol{\mu}\|_2^3 d \max\{\sigma_w^2, \sigma_p^2\},$$

*for any $i \in [n]$ and $t \in [T] \setminus \{1\}$.*

$D(\tau)$**:**

$$s_1^{(i)}(\tau) \geq \Theta(1),$$

*for all $i \in [n]$.*

$E(\tau)$**:** *There exists a constant $c_5 > 1$ such that*

$$\exp(\Lambda_{i,t}(\tau) - \Lambda_{j,u}(\tau)) < c_5,$$

*for all $i, j \in [n]$ and $t, u \in [T] \setminus \{1\}$.*

$F(\tau)$**:** *There exist constants $c_6, c_7 > 0$ such that*

$$\max_{i,j \in [n]} \frac{s_1^{(i)}(\tau)(1 - s_1^{(i)}(\tau))}{s_1^{(j)}(\tau)(1 - s_1^{(j)}(\tau))} < c_6,$$

*and*

$$\max_{t,u \in [T] \setminus \{1\}} \frac{s_t^{(i)}(\tau)}{s_u^{(i)}(\tau)} < c_6, \quad \frac{s_t^{(i)}(\tau)(1 - s_t^{(i)}(\tau))}{s_1^{(i)}(\tau)(1 - s_1^{(i)}(\tau))} < c_7,$$

*for any $i \in [n]$ and $t \in [T] \setminus \{1\}$.*

$G(\tau)$**:**

$$|\lambda_{+1}(\tau)| = O(1), \; |\lambda_{-1}(\tau)| = O(1), \; |\rho_{i,t}(\tau)| = o(1),$$

*for any $i \in [n]$ and $t \in [T]$.*

We will prove the above propositions by induction argument for $\tau \in [0, T_1]$.

**Base case.** It is obvious that $C(0)$ holds. From Lemmas B.1, C.7 and C.8, the all other base cases $A(0)$, $B(0)$, $D(0)$, $E(0)$, $F(0)$ and $G(0)$ hold.

**Proof of** $\wedge_{\tau' \leq \tau} (A(\tau') \wedge B(\tau') \wedge C(\tau') \wedge D(\tau') \wedge E(\tau') \wedge F(\tau')) \Rightarrow C(\tau + 1)$**.** We only show the case of $Y^{(i)} = 1$, i.e., $i \in \mathcal{C}_+ \cup \mathcal{N}_-$. The same argument can be applied to the case of $i \in \mathcal{C}_- \cup \mathcal{N}_+$. The conditions of Lemmas D.6 and D.7 are satisfied from $A(\tau) \wedge B(\tau) \wedge D(\tau) \wedge F(\tau)$. From these lemmas, we have that for any $i \in \mathcal{C}_+ \cup \mathcal{N}_-$,

$$\lambda_{+1}(\tau + 1) - \lambda_{+1}(\tau) \geq s_1^{(i)}(\tau)(1 - s_1^{(i)}(\tau)) \cdot c_1' \alpha \|\boldsymbol{\nu}\|_2 \|\boldsymbol{\mu}\|_2^3 d \max\{\sigma_w^2, \sigma_p^2\}, \tag{79}$$

and for any clean data $i \in \mathcal{C}_+$ and $t \in [T] \setminus \{1\}$, we have

$$\rho_{i,1}(\tau + 1) - \rho_{i,1}(\tau) \geq s_1^{(i)}(\tau)(1 - s_1^{(i)}(\tau)) \cdot c_2' \alpha n^{-1} \sigma_\epsilon^2 \|\boldsymbol{\nu}\|_2 \|\boldsymbol{\mu}\|_2 d^2 \max\{\sigma_w^2, \sigma_p^2\}, \tag{80}$$

$$\rho_{i,t}(\tau + 1) - \rho_{i,t}(\tau) \leq -s_1^{(i)}(\tau) s_t^{(i)}(\tau) \cdot c_2' \alpha n^{-1} \sigma_\epsilon^2 \|\boldsymbol{\nu}\|_2 \|\boldsymbol{\mu}\|_2 d^2 \max\{\sigma_w^2, \sigma_p^2\}, \tag{81}$$

and for any noisy data $j \in \mathcal{N}_-$ and $t \in [T] \setminus \{1\}$, we have

$$\rho_{j,1}(\tau + 1) - \rho_{j,1}(\tau) \geq -s_1^{(j)}(\tau)(1 - s_1^{(j)}(\tau)) \cdot c_3' \alpha n^{-1} \sigma_\epsilon^2 \|\boldsymbol{\nu}\|_2 \|\boldsymbol{\mu}\|_2 d^2 \max\{\sigma_w^2, \sigma_p^2\}, \tag{82}$$

$$\rho_{j,t}(\tau + 1) - \rho_{j,t}(\tau) \leq s_1^{(j)}(\tau) s_t^{(j)}(\tau) \cdot c_3' \alpha n^{-1} \sigma_\epsilon^2 \|\boldsymbol{\nu}\|_2 \|\boldsymbol{\mu}\|_2 d^2 \max\{\sigma_w^2, \sigma_p^2\}. \tag{83}$$

From the definition of $\Lambda_{i,t}$ in Definition C.3, we have for $i \in \mathcal{C}_+$ and $t \in \mathcal{I}^{(i)}$ that

$$\Lambda_{i,t}(\tau + 1) - \Lambda_{i,t}(\tau) = (\lambda_{+1}(\tau + 1) - \lambda_{+1}(\tau)) + (\rho_{i,1}(\tau + 1) - \rho_{i,1}(\tau)) - (\rho_{i,t}(\tau + 1) - \rho_{i,t}(\tau)) \tag{84}$$

$$\geq s_1^{(i)}(\tau)(1 - s_1^{(i)}(\tau)) \cdot \alpha \|\boldsymbol{\nu}\|_2 \|\boldsymbol{\mu}\|_2 d \left( c_1' \|\boldsymbol{\mu}\|_2^2 + c_2' n^{-1} \sigma_\epsilon^2 d \right) \max\{\sigma_w^2, \sigma_p^2\}$$

$$+ s_1^{(i)}(\tau) s_t^{(i)}(\tau) \cdot c_2' \alpha n^{-1} \sigma_\epsilon^2 \|\boldsymbol{\nu}\|_2 \|\boldsymbol{\mu}\|_2 d^2 \max\{\sigma_w^2, \sigma_p^2\} \tag{85}$$

$$\geq s_1^{(i)}(\tau)(1 - s_1^{(i)}(\tau)) \cdot c_1' \alpha \|\boldsymbol{\nu}\|_2 \|\boldsymbol{\mu}\|_2^3 d \max\{\sigma_w^2, \sigma_p^2\}. \tag{86}$$

Here, we bound with the signal update because it is dominant from the condition $\mathrm{SNR}^2 = \omega(n^{-1})$, i.e., $\|\boldsymbol{\mu}\|_2^2 = \omega(n^{-1} \sigma_\epsilon^2 d)$. For noisy data $j \in \mathcal{N}$, the signal update becomes dominant under the current SNR setting as follows:

$$\Lambda_{j,t}(\tau + 1) - \Lambda_{j,t}(\tau) \geq s_1^{(j)}(\tau)(1 - s_1^{(j)}(\tau)) \cdot \alpha \|\boldsymbol{\nu}\|_2 \|\boldsymbol{\mu}\|_2 d \left( c_1' \|\boldsymbol{\mu}\|_2^2 - c_3' n^{-1} \sigma_\epsilon^2 d \right) \max\{\sigma_w^2, \sigma_p^2\}$$

$$- s_1^{(j)}(\tau) s_t^{(j)}(\tau) \cdot c_3' \alpha n^{-1} \sigma_\epsilon^2 \|\boldsymbol{\nu}\|_2 \|\boldsymbol{\mu}\|_2 d^2 \max\{\sigma_w^2, \sigma_p^2\} \tag{87}$$

$$\geq s_1^{(j)}(\tau)(1 - s_1^{(j)}(\tau)) \cdot c_4' \alpha \|\boldsymbol{\nu}\|_2 \|\boldsymbol{\mu}\|_2^3 d \max\{\sigma_w^2, \sigma_p^2\}, \tag{88}$$

for some constant $c_4' > 0$. Since the same lower bound is obtained for clean and noisy data, we proceed with the discussion for $i \in \mathcal{C}_+ \cup \mathcal{N}_-$. We note that we have the same result for weakly relevant tokens $t \in \mathcal{W}_{+1}^{(i)} \cup \mathcal{W}_{-1}^{(i)}$. When $t \in \mathcal{W}_{+1}^{(i)}$, the update of $\lambda_{+1}$ is multiplied by a factor of $(1 - \rho)$. Since $\rho < 1/C$, the problem is reduced to the case of $\mathcal{I}^{(i)}$ by substituting constants. For $t \in \mathcal{W}_{-1}^{(i)}$, we also have to consider the update of $\lambda_{-1}$. However, by similarly applying Lemmas D.6 and D.7, along with the result of the softmax probability ratio in $F(\tau)$, the case for $t \in \mathcal{W}_{-1}^{(i)}$ reduces to that of $t \in \mathcal{W}_{+1}^{(i)}$. Therefore, the above inequalities hold for any $t \in [T] \setminus \{1\}$. Similarly, from Lemmas D.6 and D.7, we obtain the following upper bound:

$$\Lambda_{i,t}(\tau + 1) - \Lambda_{i,t}(\tau) \leq s_1^{(i)}(\tau)(1 - s_1^{(i)}(\tau)) \cdot c_5' \alpha \|\boldsymbol{\nu}\|_2 \|\boldsymbol{\mu}\|_2^3 d \max\{\sigma_w^2, \sigma_p^2\}, \tag{89}$$

for $i \in \mathcal{C}_+ \cup \mathcal{N}_-$ and $t \in [T] \setminus \{1\}$.

We will evaluate the softmax probabilities in the coefficients of the above inequality using Lemma D.1. As we will see later, the increase or decrease of the bounds changes at $\Lambda_{i,t}(\tau) = \log T$, so we divide the analysis into cases before and after this point. At the beginning of training $\tau = 0$, we have $|\Lambda_{i,t}(0)| = o(1)$ from Lemma C.8. Since $\Lambda_{i,t}(\tau)$ is monotonically increasing from $\wedge_{\tau' \leq \tau} C(\tau')$, there exists a time step changing to $\Lambda_{i,t}(\tau) > \log T$. We proceed with the analysis by considering cases before and after this transition.

**Case 1 ($\Lambda_{i,t}(\tau) \leq \log T$):** From Lemma D.1, we have

$$s_1^{(i)}(\tau)(1 - s_1^{(i)}(\tau)) \geq \frac{c^{-1}}{2 + 2\cosh\left(\Lambda_{i,t}(\tau) - \log T\right)}. \tag{90}$$

Substituting this to Equations (86) and (88) provides us

$$\left(2 + 2\cosh\left(\Lambda_{i,t}(\tau) - \log T\right)\right)\left(\Lambda_{i,t}(\tau + 1) - \Lambda_{i,t}(\tau)\right) \geq c_6' \alpha \|\boldsymbol{\nu}\|_2 \|\boldsymbol{\mu}\|_2^3 d \max\{\sigma_w^2, \sigma_p^2\}, \tag{91}$$

for some constant $c_6' > 0$. To apply a quadrature method to Equation (91), we will replace the coefficient with $\Lambda_{i,t}(\tau + 1)$. Using $\Lambda_{i,t}(\tau) < \Lambda_{i,t}(\tau + 1)$ and the convexity of $\cosh$, we have

$$\cosh\left(\Lambda_{i,t}(\tau + 1) - \log T\right) \geq \cosh\left(\Lambda_{i,t}(\tau) - \log T\right) + \sinh\left(\Lambda_{i,t}(\tau) - \log T\right)\left(\Lambda_{i,t}(\tau + 1) - \Lambda_{i,t}(\tau)\right) \tag{92}$$

$$\geq \cosh\left(\Lambda_{i,t}(\tau) - \log T\right)\left(1 - \left(\Lambda_{i,t}(\tau + 1) - \Lambda_{i,t}(\tau)\right)\right), \tag{93}$$

where the second inequality follows from $\sinh(x) \geq -\cosh(x)$ for all $x \in \mathbb{R}$. Using $|\Lambda_{i,t}(\tau + 1) - \Lambda_{i,t}(\tau)| = o(1)$ from Lemma D.4, we have

$$(1 + o(1))\cosh\left(\Lambda_{i,t}(\tau + 1) - \log T\right) \geq \cosh\left(\Lambda_{i,t}(\tau) - \log T\right). \tag{94}$$

By substituting this to Equation (91), we have

$$\left(2 + 2\cosh\left(\Lambda_{i,t}(\tau + 1) - \log T\right)\right)\left(\Lambda_{i,t}(\tau + 1) - \Lambda_{i,t}(\tau)\right) \geq c_7' \alpha \|\boldsymbol{\nu}\|_2 \|\boldsymbol{\mu}\|_2^3 d \max\{\sigma_w^2, \sigma_p^2\}, \tag{95}$$

where $c_7' = (1 - o(1))c_6'$. Since this inequality holds for any $0 \leq \tau' \leq \tau$ from the induction hypothesis, summing both sides from 0 to $\tau$ gives

$$\sum_{\tau'=0}^{\tau} \left(2 + 2\cosh\left(\Lambda_{i,t}(\tau' + 1) - \log T\right)\right)\left(\Lambda_{i,t}(\tau' + 1) - \Lambda_{i,t}(\tau')\right) \geq (\tau + 1) \cdot c_7' \alpha \|\boldsymbol{\nu}\|_2 \|\boldsymbol{\mu}\|_2^3 d \max\{\sigma_w^2, \sigma_p^2\}. \tag{96}$$

Recall that $g(x) = 2x + 2\sinh(x - \log T)$. Since $\Lambda_{i,t}(\tau')$ is monotonously increasing in $\tau' \in [0, \tau + 1]$, and the function $2 + 2\cosh(x - \log T)$ is monotonously decreasing in this case, we have the following lower bound by the quadrature method:

$$g\left(\Lambda_{i,t}(\tau + 1)\right) - g\left(\Lambda_{i,t}(0)\right) = \int_{\Lambda_{i,t}(0)}^{\Lambda_{i,t}(\tau+1)} \left(2 + 2\cosh\left(x - \log T\right)\right) dx \tag{97}$$

$$\geq \sum_{\tau'=0}^{\tau} \left(2 + 2\cosh\left(\Lambda_{i,t}(\tau' + 1) - \log T\right)\right)\left(\Lambda_{i,t}(\tau' + 1) - \Lambda_{i,t}(\tau')\right) \tag{98}$$

$$\geq (\tau + 1) \cdot c_7' \alpha \|\boldsymbol{\nu}\|_2 \|\boldsymbol{\mu}\|_2^3 d \max\{\sigma_w^2, \sigma_p^2\}, \tag{99}$$

which gives the desired result. Therefore, we have

$$g\left(\Lambda_{i,t}(\tau + 1)\right) \geq g\left(\Lambda_{i,t}(0)\right) + (\tau + 1) \cdot c_7' \alpha \|\boldsymbol{\nu}\|_2 \|\boldsymbol{\mu}\|_2^3 d \max\{\sigma_w^2, \sigma_p^2\}. \tag{100}$$

We also provide the upper bound. By applying Lemma D.1 to Equation (89), there exists a constant $c_8' > 0$ such that

$$\left(2 + 2\cosh\left(\Lambda_{i,t}(\tau) - \log T\right)\right)\left(\Lambda_{i,t}(\tau + 1) - \Lambda_{i,t}(\tau)\right) \leq c_8' \alpha \|\boldsymbol{\nu}\|_2 \|\boldsymbol{\mu}\|_2^3 d \max\{\sigma_w^2, \sigma_p^2\}. \tag{101}$$

Since this inequality holds for any $0 \leq \tau' \leq \tau$, summing both sides from 0 to $\tau$ and using the quadrature method, we have

$$g\left(\Lambda_{i,t}(\tau + 1)\right) - g\left(\Lambda_{i,t}(0)\right) = \int_{\Lambda_{i,t}(0)}^{\Lambda_{i,t}(\tau+1)} 2 + 2\cosh\left(x - \log T\right) dx \tag{102}$$

$$\leq \sum_{\tau'=0}^{\tau} \left(2 + 2\cosh\left(\Lambda_{i,t}(\tau') - \log T\right)\right)\left(\Lambda_{i,t}(\tau' + 1) - \Lambda_{i,t}(\tau')\right) \tag{103}$$

$$\leq (\tau + 1) \cdot c_8' \alpha \|\boldsymbol{\nu}\|_2 \|\boldsymbol{\mu}\|_2^3 d \max\{\sigma_w^2, \sigma_p^2\}. \tag{104}$$

Therefore, we have

$$g\left(\Lambda_{i,t}(\tau + 1)\right) \leq g\left(\Lambda_{i,t}(0)\right) + (\tau + 1) \cdot c_8' \alpha \|\boldsymbol{\nu}\|_2 \|\boldsymbol{\mu}\|_2^3 d \max\{\sigma_w^2, \sigma_p^2\}. \tag{105}$$

**Case 2** ($\log T < \Lambda_{i,t}(\tau)$)**:** Let $T_0$ be a first-time step falling into Case 2. Since $2 + 2\cosh(x - \log T)$ is monotonically increasing in this case, by summing both sides of Equation (91) from $T_0$ to $\tau$ and using the quadrature method, we have

$$g\left(\Lambda_{i,t}(\tau+1)\right) - g\left(\Lambda_{i,t}(T_0)\right) = \int_{\Lambda_{i,t}(T_0)}^{\Lambda_{i,t}(\tau+1)} 2 + 2\cosh\left(x - \log T\right) dx \tag{106}$$

$$\geq \sum_{\tau'=T_0}^{\tau} \left(2 + 2\cosh\left(\Lambda_{i,t}(\tau') - \log T\right)\right)\left(\Lambda_{i,t}(\tau'+1) - \Lambda_{i,t}(\tau')\right) \tag{107}$$

$$\geq (\tau - T_0 + 1) \cdot c_6'\alpha\|\boldsymbol{\nu}\|_2\|\boldsymbol{\mu}\|_2^3 d \max\{\sigma_w^2, \sigma_p^2\}. \tag{108}$$

Combining this with Equation (100), we have

$$g\left(\Lambda_{i,t}(\tau+1)\right) \geq g\left(\Lambda_{i,t}(0)\right) + (\tau+1) \cdot c_7'\alpha\|\boldsymbol{\nu}\|_2\|\boldsymbol{\mu}\|_2^3 d \max\{\sigma_w^2, \sigma_p^2\}, \tag{109}$$

which follows from $c_7' = (1 - o(1))c_6' < c_6'$.

Next, we provide the upper bound. In Case 2, to obtain an upper bound using the quadrature method, it is necessary to replace the coefficient in Equation (101) using $\Lambda_{i,t}(\tau+1)$. Since $\cosh(x - \log T)$ is convex and monotonically increasing in this case, we have

$$\cosh\left(\Lambda_{i,t}(\tau+1) - \log T\right) \leq \cosh\left(\Lambda_{i,t}(\tau) - \log T\right) + \sinh\left(\Lambda_{i,t}(\tau+1) - \log T\right)\left(\Lambda_{i,t}(\tau+1) - \Lambda_{i,t}(\tau)\right). \tag{110}$$

Using $|\Lambda_{i,t}(\tau+1) - \Lambda_{i,t}(\tau)| = o(1)$ from Lemma D.4 and $-\cosh(x) < -\sinh(x)$ for all $x \in \mathbb{R}$, we have

$$\cosh\left(\Lambda_{i,t}(\tau+1) - \log T\right)(1 - o(1)) = \cosh\left(\Lambda_{i,t}(\tau+1) - \log T\right)\left(1 - \left(\Lambda_{i,t}(\tau+1) - \Lambda_{i,t}(\tau)\right)\right) \tag{111}$$

$$\leq \cosh\left(\Lambda_{i,t}(\tau) - \log T\right). \tag{112}$$

By applying this to Equation (101), we have

$$\left(2 + 2\cosh\left(\Lambda_{i,t}(\tau+1) - \log T\right)\right)\left(\Lambda_{i,t}(\tau+1) - \Lambda_{i,t}(\tau)\right) \leq c_9'\alpha\|\boldsymbol{\nu}\|_2\|\boldsymbol{\mu}\|_2^3 d \max\{\sigma_w^2, \sigma_p^2\}, \tag{113}$$

where $c_9' = (1 + o(1))c_8'$. Again, using the quadrature method, we have

$$g\left(\Lambda_{i,t}(\tau+1)\right) - g\left(\Lambda_{i,t}(T_0)\right) = \int_{\Lambda_{i,t}(T_0)}^{\Lambda_{i,t}(\tau+1)} 2 + 2\cosh\left(x - \log T\right) dx \tag{114}$$

$$\leq \sum_{\tau'=T_0}^{\tau} \left(2 + 2\cosh\left(\Lambda_{i,t}(\tau'+1) - \log T\right)\right)\left(\Lambda_{i,t}(\tau'+1) - \Lambda_{i,t}(\tau')\right) \tag{115}$$

$$\leq (\tau - T_0 + 1) \cdot c_9'\alpha\|\boldsymbol{\nu}\|_2\|\boldsymbol{\mu}\|_2^3 d \max\{\sigma_w^2, \sigma_p^2\}. \tag{116}$$

Combining this with Equation (105), we have

$$g\left(\Lambda_{i,t}(\tau+1)\right) \leq g\left(\Lambda_{i,t}(0)\right) + (\tau+1) \cdot c_9'\alpha\|\boldsymbol{\nu}\|_2\|\boldsymbol{\mu}\|_2^3 d \max\{\sigma_w^2, \sigma_p^2\}, \tag{117}$$

which follows from $c_9' = (1 + o(1))c_8' > c_8'$. This concludes the proof.

**Proof of** $C(\tau) \Rightarrow D(\tau)$**.** For any $i \in [n]$, we have

$$s_1^{(i)}(\tau) = \frac{1}{1 + \sum_{t=2}^{T} \exp\left(-\Lambda_{i,t}(\tau)\right)} \geq \frac{1}{1 + \sum_{t=2}^{T} \exp\left(-\Lambda_{i,t}(0)\right)} = s_1^{(i)}(0) \geq \Theta(1), \tag{118}$$

where the inequality follows from $C(\tau)$, which states that $\Lambda_{i,t}(\tau)$ is greater than $\Lambda_{i,t}(0)$. The last line is derived from $D(0)$.

**Proof of** $C(\tau) \Rightarrow E(\tau)$**.** We proceed with the proof by contradiction. Suppose that $E(\tau)$ does not hold, i.e., for any constant $c > 1$, there exist $i, j \in [n]$ and $t, u \in [T] \setminus \{1\}$ such that

$$\exp\left(\Lambda_{i,t}(\tau) - \Lambda_{j,u}(\tau)\right) \geq c. \tag{119}$$

We continue the analysis, dividing it into two cases.

**Case 1 ($\Lambda_{j,u}(\tau) < 2\log T$):**  In this case, from $C(\tau)$ and the monotonic increase of $g$, we have

$$g\left(\Lambda_{j,u}(0)\right) + \tau \cdot c_3 \alpha \|\boldsymbol{\nu}\|_2 \|\boldsymbol{\mu}\|_2^3 d \max\{\sigma_w^2, \sigma_p^2\} \leq g\left(\Lambda_{j,u}(\tau)\right) \leq g\left(2\log T\right) < 4\log T + T - \frac{1}{T}. \tag{120}$$

Since $g\left(\Lambda_{j,u}(0)\right) = -(1 + o(1))T$ from the definition of $g$ and $|\Lambda_{j,u}(0)| = o(1)$ in Lemma C.8, we have

$$\tau < \frac{3T}{c_3 \alpha \|\boldsymbol{\nu}\|_2 \|\boldsymbol{\mu}\|_2^3 d \max\{\sigma_w^2, \sigma_p^2\}}. \tag{121}$$

Again from $C(\tau)$, we have

$$g\left(\Lambda_{i,t}(\tau)\right) - g\left(\Lambda_{j,u}(\tau)\right) \leq \left(g\left(\Lambda_{i,t}(0)\right) - g\left(\Lambda_{j,u}(0)\right)\right) + \tau \cdot (c_4 - c_3)\alpha \|\boldsymbol{\nu}\|_2 \|\boldsymbol{\mu}\|_2^3 d \max\{\sigma_w^2, \sigma_p^2\} \tag{122}$$

$$< o(1) + \frac{3(c_4 - c_3)}{c_3} T < \frac{3c_4}{c_3} T. \tag{123}$$

On the other hand, under Equation (119), we have

$$g\left(\Lambda_{i,t}(\tau)\right) - g\left(\Lambda_{j,u}(\tau)\right) \geq 2\log c + 2\left(\sinh\left(\Lambda_{j,u}(\tau) + \log c - \log T\right) - \sinh\left(\Lambda_{j,u}(\tau) - \log T\right)\right) \tag{124}$$

$$> 2\log c + \frac{c-1}{T}\exp\left(\Lambda_{j,u}(\tau)\right) + T\left(1 - \frac{1}{c}\right)\exp\left(-\Lambda_{j,u}(\tau)\right) \tag{125}$$

$$> 2\log c + 2\left(\sqrt{c} - \frac{1}{\sqrt{c}}\right) > \frac{3c_4}{c_3}T. \tag{126}$$

The last line is the result of AM-GM inequality, the parameter assumption $T = \Theta(1)$, and an appropriate choice of $c$. This leads to a contradiction with Equation (123).

**Case 2 ($2\log T \leq \Lambda_{j,u}(\tau)$):**  From $C(\tau)$, we have

$$\frac{g\left(\Lambda_{i,t}(\tau)\right) - g\left(\Lambda_{i,t}(0)\right)}{g\left(\Lambda_{j,u}(\tau)\right) - g\left(\Lambda_{j,u}(0)\right)} \leq \frac{c_4}{c_3}, \tag{127}$$

for $\tau \geq 1$. Additionally, from the definition of $g$ and Lemma C.8, we have that

$$\frac{g\left(\Lambda_{i,t}(0)\right)}{g\left(\Lambda_{j,u}(0)\right)} = \frac{2\Lambda_{i,t}(0) + \frac{1}{T}\exp\left(\Lambda_{i,t}(0)\right) - T\exp\left(-\Lambda_{i,t}(0)\right)}{2\Lambda_{j,u}(0) + \frac{1}{T}\exp\left(\Lambda_{j,u}(0)\right) - T\exp\left(-\Lambda_{j,u}(0)\right)} \leq 1 + o(1) < 2, \tag{128}$$

where the inequality comes from the fact that the third term is dominant and $|\Lambda_{i,t}(0) - \Lambda_{j,u}(0)| = o(1)$.

Before delving into the part of contradiction, we first show that the following inequality holds. By a simple calculation, we obtain the following for $x \geq 2\log T$:

$$2\sinh\left(x + \log c - \log T\right) = \frac{c}{T}\exp(x) - \frac{T}{c}\exp(-x) \tag{129}$$

$$= \frac{c}{3} \cdot 2\sinh\left(x - \log T\right) + \frac{2c}{3T}\exp(x) + \left(\frac{c}{3} - \frac{1}{c}\right)T\exp(-x) \tag{130}$$

$$> \frac{c}{3} \cdot 2\sinh\left(x - \log T\right) + \frac{c}{3}x + \frac{c}{3}T, \tag{131}$$

where the last line holds for a constant satisfying $c/3 > 1/c$, and we used $\exp(x) \geq T^2/(2\log T)x > Tx$ for $x \geq 2\log T$. By using Equations (119), (128) and (131), we have

$$g\left(\Lambda_{i,t}(\tau)\right) - g\left(\Lambda_{i,t}(0)\right)$$
$$= 2\Lambda_{i,t}(\tau) + 2\sinh\left(\Lambda_{i,t}(\tau) - \log T\right) - g\left(\Lambda_{i,t}(0)\right) \tag{132}$$

$$\geq 2\left(\Lambda_{j,u}(\tau) + \log c\right) + 2\sinh\left(\Lambda_{j,u}(\tau) + \log c - \log T\right) - \frac{1}{2}g\left(\Lambda_{j,u}(0)\right) \tag{133}$$

$$\geq (2 + c/3)\Lambda_{j,u}(\tau) + c/3 \cdot 2\sinh\left(\Lambda_{j,u}(\tau) - \log T\right) + ((1 - o(1))c/3 + 1/2)\left(-g\left(\Lambda_{j,u}(0)\right)\right) \tag{134}$$
$$> c_4/c_3 \cdot \left(g\left(\Lambda_{j,u}(\tau)\right) - g\left(\Lambda_{j,u}(0)\right)\right), \tag{135}$$

where in the second inequality, we also used $-g\left(\Lambda_{j,u}(0)\right) = (1 \pm o(1))T$. The last line holds sufficiently large $c > 0$ satisfying $1 + c/6 > c_4/c_3$, $c/3 > c_4/c_3$ and $(1 - o(1)) \cdot c/3 + 1/2 > c_4/c_3$. Therefore, this contradicts Equation (127).

Thus, since a contradiction arises in both Case 1 and Case 2, it follows from Equation (119) that there exists a constant $c > 1$ such that

$$\exp\left(\Lambda_{i,t}(\tau) - \Lambda_{j,u}(\tau)\right) < c, \tag{136}$$

for any $i, j \in [n]$ and $t, u \in [T] \setminus \{1\}$.

**Proof of $E(\tau) \Rightarrow F(\tau)$.** We have

$$s_1^{(i)}(\tau)(1 - s_1^{(i)}(\tau)) = \frac{1}{1 + \sum_{t=2}^T \exp\left(-\Lambda_{i,t}(\tau)\right)} \cdot \frac{\sum_{t=2}^T \exp\left(-\Lambda_{i,t}(\tau)\right)}{1 + \sum_{t=2}^T \exp\left(-\Lambda_{i,t}(\tau)\right)}. \tag{137}$$

Then, we have

$$\frac{s_1^{(i)}(\tau)(1 - s_1^{(i)}(\tau))}{s_1^{(j)}(\tau)(1 - s_1^{(j)}(\tau))} = \frac{\sum_{t=2}^T \exp\left(-\Lambda_{i,t}(\tau)\right)}{\sum_{t=2}^T \exp\left(-\Lambda_{j,t}(\tau)\right)} \cdot \left(\frac{1 + \sum_{t=2}^T \exp\left(-\Lambda_{j,t}(\tau)\right)}{1 + \sum_{t=2}^T \exp\left(-\Lambda_{i,t}(\tau)\right)}\right)^2 \tag{138}$$

$$\leq \frac{\sum_{t=2}^T \exp\left(-\Lambda_{i,t}(\tau)\right)}{\sum_{t=2}^T \exp\left(-\Lambda_{j,t}(\tau)\right)} \cdot \max\left\{1, \frac{\sum_{t=2}^T \exp\left(-\Lambda_{j,t}(\tau)\right)}{\sum_{t=2}^T \exp\left(-\Lambda_{i,t}(\tau)\right)}\right\}^2 \tag{139}$$

$$= \max\left\{\frac{\sum_{t=2}^T \exp\left(-\Lambda_{i,t}(\tau)\right)}{\sum_{t=2}^T \exp\left(-\Lambda_{j,t}(\tau)\right)}, \frac{\sum_{t=2}^T \exp\left(-\Lambda_{j,t}(\tau)\right)}{\sum_{t=2}^T \exp\left(-\Lambda_{i,t}(\tau)\right)}\right\} \tag{140}$$

$$\leq \max\left\{\max_{2 \leq t \leq T}\left\{\frac{\exp\left(-\Lambda_{i,t}(\tau)\right)}{\exp\left(-\Lambda_{j,t}(\tau)\right)}\right\}, \max_{2 \leq t \leq T}\left\{\frac{\exp\left(-\Lambda_{j,t}(\tau)\right)}{\exp\left(-\Lambda_{i,t}(\tau)\right)}\right\}\right\}, \tag{141}$$

where the inequalities are the result of mediant inequality, i.e., $(\sum_i a_i)/(\sum_i b_i) < \max_i\{a_i/b_i\}$ for $a_i, b_i > 0, \forall i$. Therefore, using $E(\tau)$, we have

$$\max_{i,j \in [n]} \frac{s_1^{(i)}(\tau)(1 - s_1^{(i)}(\tau))}{s_1^{(j)}(\tau)(1 - s_1^{(j)}(\tau))} \leq c_5. \tag{142}$$

Additionally, we have

$$\max_{t,u \in [T] \setminus \{1\}} \frac{s_t^{(i)}(\tau)}{s_u^{(i)}(\tau)} = \max_{t,u \in [T] \setminus \{1\}} \frac{\exp\left(\mathbf{x}_t^{(i)\top} \mathbf{W}(\tau)^\top \mathbf{p}(\tau)\right)}{\exp\left(\mathbf{x}_u^{(i)\top} \mathbf{W}(\tau)^\top \mathbf{p}(\tau)\right)} = \max_{t,u \in [T] \setminus \{1\}} \frac{\exp\left(-\Lambda_{i,t}(\tau)\right)}{\exp\left(-\Lambda_{i,u}(\tau)\right)} \leq c_5, \tag{143}$$

for any $i \in [n]$. Finally, since $\Lambda_{i,t}(\tau)$ is monotonically increasing for any $t \in [T] \setminus \{1\}$, Lemma D.3 concludes that $s_t^{(i)}(\tau)(1 - s_t^{(i)}(\tau))$ is dominated by $s_1^{(i)}(\tau)(1 - s_1^{(i)}(\tau))$ ignoring constants.

**Proof of $\wedge_{\tau' \leq \tau} \left(A(\tau') \wedge B(\tau') \wedge C(\tau') \wedge D(\tau') \wedge F(\tau')\right) \Rightarrow G(\tau + 1)$.** The conditions of Lemma D.6 are satisfied from $A(\tau) \wedge B(\tau) \wedge D(\tau) \wedge F(\tau)$. From Lemma D.6, we have

$$\lambda_{+1}(\tau + 1) - \lambda_{+1}(0) \leq \sum_{\tau'=0}^\tau s_1^{(i)}(\tau')(1 - s_1^{(i)}(\tau')) \cdot c'\alpha \|\boldsymbol{\nu}\|_2 \|\boldsymbol{\mu}\|_2^3 d \max\{\sigma_w^2, \sigma_p^2\}. \tag{144}$$

We use Lemma D.5 to derive the upper bound of the summation of softmax probability terms. Since Lemma D.5 follows from $\wedge_{\tau' \leq \tau} \left(C(\tau') \wedge E(\tau')\right)$ and the definition of $T_1$, we have

$$\sum_{\tau'=0}^\tau s_1^{(i)}(\tau')(1 - s_1^{(i)}(\tau')) \lesssim \frac{1}{\alpha \|\boldsymbol{\nu}\|_2 \|\boldsymbol{\mu}\|_2^3 d \max\{\sigma_w^2, \sigma_p^2\}}. \tag{145}$$

Substituting this to Equation (144), we have

$$\lambda_{+1}(\tau + 1) - \lambda_{+1}(0) = O(1). \tag{146}$$

Since $|\lambda_{+1}(0)| = o(1)$ from Lemma C.8, Equation (146) leads to $\lambda_{+1}(\tau + 1) \leq O(1)$. In the same way, we have $\lambda_{-1}(\tau + 1) \leq O(1)$. For noise memorization terms, from Lemma D.7, we have that for any $i \in [n]$,

$$|\rho_{i,1}(\tau + 1) - \rho_{i,1}(0)| \leq \sum_{\tau'=0}^{\tau} s_1^{(i)}(\tau')(1 - s_1^{(i)}(\tau')) \cdot c'\alpha n^{-1}\sigma_\epsilon^2 \|\boldsymbol{\nu}\|_2 \|\boldsymbol{\mu}\|_2 d^2 \max\{\sigma_w^2, \sigma_p^2\}. \tag{147}$$

We proceed with the same argument as in the signal update case discussed earlier. From the current SNR condition $\mathrm{SNR}^2 = \omega(n^{-1})$ and $\rho_{i,t}(0) = o(1)$, we have $\rho_{i,1}(\tau) \leq o(1)$. The bound of $\rho_{i,t}(\tau)$ is obtained as well.

**Proof of** $\wedge_{\tau' \leq \tau} (B(\tau') \wedge C(\tau') \wedge F(\tau') \wedge G(\tau')) \Rightarrow A(\tau + 1)$. It follows from Lemma C.5 that

$$\|\mathbf{p}(\tau + 1)\|_2^2 - \|\mathbf{p}(0)\|_2^2 = \sum_{\tau'=0}^{\tau} \left( \|\mathbf{p}(\tau' + 1)\|_2^2 - \|\mathbf{p}(\tau')\|_2^2 \right) \tag{148}$$

$$= \frac{2\alpha}{n} \sum_{\tau'=0}^{\tau} \sum_{i=1}^{n} (-\ell_i'(\tau')) \cdot Y^{(i)} \cdot I_i^p(\tau') + \alpha^2 \sum_{\tau'=0}^{\tau} \|\nabla_{\mathbf{p}} \widehat{\mathcal{L}}(\tau')\|_2^2. \tag{149}$$

By definition, we have

$$I_i^p(\tau') = \sum_{t=1}^{T} s_t^{(i)}(\tau') \left( \gamma_t^{(i)} - \sum_{u=1}^{T} s_u^{(i)}(\tau')\gamma_u^{(i)} \right) \langle \mathbf{W}(\tau')\mathbf{x}_t^{(i)}, \mathbf{p}(\tau') \rangle \tag{150}$$

$$\leq \sum_{t=1}^{T} s_t^{(i)}(\tau')(1 - s_t^{(i)}(\tau')) \cdot \max_{u \in [T]} \left\{ |\gamma_t^{(i)} - \gamma_u^{(i)}| \right\} \cdot \langle \mathbf{W}(\tau')\mathbf{x}_t^{(i)}, \mathbf{p}(\tau') \rangle \tag{151}$$

$$\lesssim s_1^{(i)}(\tau')(1 - s_1^{(i)}(\tau')) \cdot \max_{t \in [T]} \left\{ |\gamma_t^{(i)}| \right\} \cdot \max \left\{ |\lambda_{+1}(\tau')|, |\lambda_{-1}(\tau')|, |\rho_{i,t}(\tau')| \right\}, \tag{152}$$

where in the last line, we used the results in $F(\tau')$ and $T = \Theta(1)$ in the parameter assumptions. By substituting this into the first term of Equation (149), we obtain

$$\frac{2\alpha}{n} \sum_{\tau'=0}^{\tau} \sum_{i=1}^{n} (-\ell_i'(\tau')) \cdot Y^{(i)} \cdot I_i^p(\tau') \lesssim \alpha \|\boldsymbol{\nu}\|_2 \|\boldsymbol{\mu}\|_2 \cdot \sum_{\tau'=0}^{\tau} s_1^{(i)}(\tau')(1 - s_1^{(i)}(\tau')) \cdot \max \left\{ |\lambda_{+1}(\tau')|, |\lambda_{-1}(\tau')|, |\rho_{i,t}(\tau')| \right\} \tag{153}$$

$$\lesssim \alpha \|\boldsymbol{\nu}\|_2 \|\boldsymbol{\mu}\|_2 \cdot \frac{1}{\alpha \|\boldsymbol{\nu}\|_2 \|\boldsymbol{\mu}\|_2^3 d \max\{\sigma_w^2, \sigma_p^2\}} \tag{154}$$

$$\lesssim \frac{1}{\|\boldsymbol{\mu}\|_2^2 d \max\{\sigma_w^2, \sigma_p^2\}}, \tag{155}$$

where the inequalities follow from Lemmas B.12 and C.9, the results in $G(\tau')$, and Lemma D.5. We confirm that Equation (155) becomes $o(\sigma_p^2 d)$. From the parameter assumptions in Section 3.5, we have

$$\sigma_p^2 \max\{\sigma_w^2, \sigma_p^2\} \cdot \|\boldsymbol{\mu}\|_2^2 d^2 \gtrsim \min \left\{ d, \frac{\|\boldsymbol{\mu}\|_2^2}{\sigma_\epsilon^2} \right\} \cdot \frac{1}{\log^4(Tn/\delta)} \tag{156}$$

$$\geq \min \left\{ C^2 n \|\boldsymbol{\mu}\|_2^{1/3} d^{3/8}, \frac{C^2 d^{3/4}}{\log^2(Tn/\delta)} \right\} \tag{157}$$

$$= \omega(1), \tag{158}$$

where the first inequality follows from Assumption A8, while the next one relies on Assumptions A1 and A2. Thus, Equation (155) becomes $o(\sigma_p^2 d)$.

Finally, we confirm that the second term in Equation (149) can be ignored. From Equation (12) and the results in $B(\tau)$, we

have

$$\alpha^2 \sum_{\tau'=0}^{\tau} \|\nabla_{\mathbf{p}}\widehat{\mathcal{L}}(\tau')\|_2^2 = \alpha^2 \sum_{\tau'=0}^{\tau} \left\| \frac{1}{n}\sum_{i=1}^{n}(-\ell_i'(\tau')) \cdot Y^{(i)} \cdot \left(\sum_{t=1}^{T} s_t^{(i)}(\tau')\left(\gamma_t^{(i)} - \sum_{u=1}^{T} s_u^{(i)}(\tau')\gamma_u^{(i)}\right)\mathbf{W}(\tau')\mathbf{x}_t^{(i)}\right)\right\|_2^2 \quad (159)$$

$$\lesssim \alpha^2 \sum_{\tau'=0}^{\tau} \left(\max_{i\in[n],t\in[T]}\{s_t^{(i)}(\tau)(1-s_t^{(i)}(\tau))\} \cdot \max_{i\in[n],t\in[T]}\{|\gamma_t^{(i)}|\} \cdot \max_{i\in[n],t\in[T]}\{\|\mathbf{W}(\tau)\mathbf{x}_t^{(i)}\|_2\}\right)^2 \quad (160)$$

$$\lesssim \left(\alpha\|\boldsymbol{\nu}\|_2\|\boldsymbol{\mu}\|_2\sum_{\tau'=0}^{\tau} s_1^{(i)}(\tau)(1-s_1^{(i)}(\tau))\right) \cdot \left(\alpha\|\boldsymbol{\nu}\|_2\|\boldsymbol{\mu}\|_2\sigma_w^2\max\{\|\boldsymbol{\mu}\|_2^2 d, \sigma_\epsilon^2 d^2\}\right), \quad (161)$$

where the last inequality follows from the dominance of $s_1^{(i)}(\tau)(1 - s_1^{(i)}(\tau))$ as established by $F(\tau')$. Using $\|\boldsymbol{\nu}\|_2 = O(1/\|\boldsymbol{\mu}\|_2)$ and the parameter assumptions on $\alpha, \sigma_w^2$, the latter part of the last line becomes $o(1)$. Consequently, this quadratic term is absorbed into Equation (153). Combining these results with Equation (149), we have $\|\mathbf{p}(\tau+1)\|_2^2 \in (1 \pm 1/C_1)\sigma_p^2 d$, for some constant $C_1 > 1$, which concludes the proof.

**Proof of** $\wedge_{\tau'\leq\tau}(A(\tau') \wedge C(\tau') \wedge G(\tau')) \Rightarrow B(\tau+1)$. From Lemma C.6, we have

$$\|\mathbf{W}(\tau+1)\boldsymbol{\mu}_{+1}\|_2^2 - \|\mathbf{W}(0)\boldsymbol{\mu}_{+1}\|_2^2 = \sum_{\tau'=0}^{\tau}\left(\|\mathbf{W}(\tau'+1)\boldsymbol{\mu}_{+1}\|_2^2 - \|\mathbf{W}(\tau')\boldsymbol{\mu}_{+1}\|_2^2\right) \quad (162)$$

$$= \frac{2\alpha}{n}\sum_{\tau'=0}^{\tau}\sum_{i=1}^{n}(-\ell_i'(\tau')) \cdot Y^{(i)} \cdot I_{i,+}(\tau')\lambda_{+1}(\tau') + \sum_{\tau'=0}^{\tau}\alpha^2\|\nabla_{\mathbf{W}}\widehat{\mathcal{L}}(\tau')\boldsymbol{\mu}_{+1}\|_2^2. \quad (163)$$

We analyze the two terms separately. Recall the definition of $I_{i,+}$ in Definition C.2, and using the same discussion as Equation (152) in the previous proof for $A(\tau+1)$, we have

$$I_{i,+}(\tau') \lesssim s_1^{(i)}(\tau')(1-s_1^{(i)}(\tau')) \cdot \max_{i\in[n],t\in[T]}\{|\gamma_t^{(i)}|\} \cdot \max_{i\in[n],t\in[T]}\{|\langle\mathbf{x}_t^{(i)},\boldsymbol{\mu}_{+1}\rangle|\} \quad (164)$$

$$\lesssim s_1^{(i)}(\tau')(1-s_1^{(i)}(\tau')) \cdot \|\boldsymbol{\nu}\|_2\|\boldsymbol{\mu}\|_2^3, \quad (165)$$

which follows from Lemmas B.1 and B.12. Therefore, for the first term, we have

$$\frac{2\alpha}{n}\sum_{\tau'=0}^{\tau}\sum_{i=1}^{n}(-\ell_i'(\tau')) \cdot Y^{(i)} \cdot I_{i,+}(\tau')\lambda_{+1}(\tau') \lesssim \sum_{\tau'=0}^{\tau} s_1^{(i)}(\tau')(1-s_1^{(i)}(\tau')) \cdot \alpha\|\boldsymbol{\nu}\|_2\|\boldsymbol{\mu}\|_2^3 \quad (166)$$

$$\lesssim \frac{1}{d\max\{\sigma_w^2,\sigma_p^2\}} = o(\sigma_w^2\|\boldsymbol{\mu}\|_2^2 d), \quad (167)$$

where the inequality follows from $G(\tau')$, Lemma D.5, and the same evaluation in Equation (158).

The analysis of the second term in Equation (163) follows from a similar argument to that of Equation (161), so we omit the proof. Therefore, Equation (163) leads to

$$\|\mathbf{W}(\tau+1)\boldsymbol{\mu}_{+1}\|_2^2 = \|\mathbf{W}(0)\boldsymbol{\mu}_{+1}\|_2^2 + o(\sigma_w^2\|\boldsymbol{\mu}\|_2^2 d) = \Theta(\sigma_w^2\|\boldsymbol{\mu}\|_2^2 d). \quad (168)$$

The same result holds for $\|\mathbf{W}(\tau+1)\boldsymbol{\mu}_{-1}\|_2^2 = \Theta(\sigma_w^2\|\boldsymbol{\mu}\|_2^2 d)$. Similarly, we have

$$\|\mathbf{W}(\tau+1)\boldsymbol{\epsilon}_t^{(i)}\|_2^2 = \|\mathbf{W}(0)\boldsymbol{\epsilon}_{-1}\|_2^2 + o(\sigma_w^2\sigma_\epsilon^2 d^2) = \Theta(\sigma_w^2\sigma_\epsilon^2 d^2). \quad (169)$$

Additionally, we will show the case of the inner products as well. From Lemma C.6, we have

$$\langle\mathbf{W}(\tau+1)\boldsymbol{\mu}_{+1},\mathbf{W}(\tau+1)\boldsymbol{\mu}_{-1}\rangle - \langle\mathbf{W}(0)\boldsymbol{\mu}_{+1},\mathbf{W}(0)\boldsymbol{\mu}_{-1}\rangle$$

$$= \frac{\alpha}{n}\sum_{\tau'=0}^{\tau}\sum_{i=1}^{n}(-\ell_i'(\tau')) \cdot Y^{(i)} \cdot (I_{i,-}(\tau')\lambda_{+1}(\tau') + I_{i,+}(\tau')\lambda_{-1}(\tau')) + \alpha^2\sum_{\tau'=0}^{\tau}\langle\nabla_{\mathbf{W}}\widehat{\mathcal{L}}(\tau')\boldsymbol{\mu}_{+1},\nabla_{\mathbf{W}}\widehat{\mathcal{L}}(\tau')\boldsymbol{\mu}_{-1}\rangle. \quad (170)$$

For the first term, from a similar argument to Equation (167), we have

$$\frac{\alpha}{n}\sum_{\tau'=0}^{\tau}\sum_{i=1}^{n}(-\ell_i'(\tau'))\cdot Y^{(i)}\cdot(I_{i,-}(\tau')\lambda_{+1}(\tau')+I_{i,+}(\tau')\lambda_{-1}(\tau'))\lesssim\frac{1}{d\max\{\sigma_w^2,\sigma_p^2\}}=O(\sigma_w^2\|\boldsymbol{\mu}\|_2^2\sqrt{d}\log(Tn/\delta)),$$

(171)

where the last line is derived from the parameter assumptions; specifically, we have

$$\sigma_w^2\max\{\sigma_w^2,\sigma_p^2\}\cdot\|\boldsymbol{\mu}\|_2^2 d^{3/2}\log(Tn/\delta)\gtrsim\min\left\{d^{1/2},\frac{\|\boldsymbol{\mu}\|_2^2}{\sigma_\epsilon^2 d^{1/2}}\right\}\cdot\frac{1}{\log^3(Tn/\delta)} \tag{172}$$

$$\geq\min\left\{\frac{d^{1/2}}{\log^3(Tn/\delta)},\frac{C^2 d^{1/4}}{\log(Tn/\delta)}\right\} \tag{173}$$

$$=\Omega(1). \tag{174}$$

The second term can be ignored in a similar manner, and we have

$$\langle\mathbf{W}(\tau+1)\boldsymbol{\mu}_{+1},\mathbf{W}(\tau+1)\boldsymbol{\mu}_{-1}\rangle=O\left(\sigma_w^2\|\boldsymbol{\mu}\|_2^2\sqrt{d}\log(Tn/\delta)\right). \tag{175}$$

We will show that the other equations are shown in the same way. Since the quadratic term can be ignored by the above discussion, we denote this by $O(\alpha^2)$ in the following. Similarly, it follows from Lemmas B.1 and C.6, and $\mathrm{SNR}^2=\omega(n^{-1})$ that

$$\langle\mathbf{W}(\tau+1)\boldsymbol{\mu}_{+1},\mathbf{W}(\tau+1)\boldsymbol{\epsilon}_u^{(j)}\rangle-\langle\mathbf{W}(0)\boldsymbol{\mu}_{+1},\mathbf{W}(0)\boldsymbol{\epsilon}_u^{(j)}\rangle$$

$$=\frac{\alpha}{n}\sum_{\tau'=0}^{\tau}\sum_{i=1}^{n}(-\ell_i'(\tau'))\cdot Y^{(i)}\cdot(I_{i,j,u}(\tau')\lambda_{+1}(\tau')+I_{i,+}(\tau')\rho_{j,u}(\tau'))+O(\alpha^2) \tag{176}$$

$$\lesssim\max\left\{\frac{n^{-1}\sigma_\epsilon^2 d}{\|\boldsymbol{\mu}\|_2^2 d\max\{\sigma_w^2,\sigma_p^2\}},\frac{\|\boldsymbol{\mu}\|_2^2}{\|\boldsymbol{\mu}\|_2^2 d\max\{\sigma_w^2,\sigma_p^2\}}\right\}+O(\alpha^2) \tag{177}$$

$$\lesssim\frac{1}{d\max\{\sigma_w^2,\sigma_p^2\}}+O(\alpha^2)=O(\sigma_w^2\|\boldsymbol{\mu}\|_2 d\log(Tn/\delta)), \tag{178}$$

which follows from the parameter assumptions; specifically, we have

$$\sigma_w^2\max\{\sigma_w^2,\sigma_p^2\}\cdot\|\boldsymbol{\mu}\|_2 d^2\log(Tn/\delta)\gtrsim\min\left\{\frac{d}{\|\boldsymbol{\mu}\|_2},\frac{\|\boldsymbol{\mu}\|_2}{\sigma_\epsilon^2}\right\}\cdot\frac{1}{\log^3(Tn/\delta)} \tag{179}$$

$$\geq\min\left\{C\hat{\sigma}_\epsilon n\|\boldsymbol{\mu}\|_2^{1/3},\frac{\|\boldsymbol{\mu}\|_2}{\sigma_\epsilon^2\log^3(Tn/\delta)}\right\} \tag{180}$$

$$=\Omega(1). \tag{181}$$

The last line can be derived by combining Assumptions A1 and A2. The same result holds for $\langle\mathbf{W}(\tau+1)\boldsymbol{\mu}_{-1},\mathbf{W}(\tau+1)\boldsymbol{\epsilon}_u^{(j)}\rangle$. Finally, we have

$$\langle\mathbf{W}(\tau+1)\boldsymbol{\epsilon}_u^{(j)},\mathbf{W}(\tau+1)\boldsymbol{\epsilon}_v^{(k)}\rangle-\langle\mathbf{W}(0)\boldsymbol{\epsilon}_u^{(j)},\mathbf{W}(0)\boldsymbol{\epsilon}_v^{(k)}\rangle$$

$$=\frac{\alpha}{n}\sum_{\tau'=0}^{\tau}\sum_{i=1}^{n}(-\ell_i'(\tau'))\cdot Y^{(i)}\cdot(I_{i,k,v}(\tau')\rho_{j,u}(\tau')+I_{i,j,u}(\tau')\rho_{k,v}(\tau'))+O(\alpha^2) \tag{182}$$

$$\lesssim\frac{n^{-1}\sigma_\epsilon^2 d}{\|\boldsymbol{\mu}\|_2^2 d\max\{\sigma_w^2,\sigma_p^2\}}+O(\alpha^2)=O(\sigma_w^2\sigma_\epsilon^2 d^{3/2}\log(Tn/\delta)), \tag{183}$$

which follows from the same discussion as in Equation (172). This completes the proof.

*Proof of Lemma C.11.* At time step $\tau=0$, all propositions hold from the proof for the base case. For the next time step, $A(\tau+1),B(\tau+1),C(\tau+1)$ and $G(\tau+1)$ are proved based on the propositions up to time step $\tau$. As for $D(\tau+1)$, $E(\tau+1)$, and $F(\tau+1)$, they are derived from $C(\tau+1)$. Thus, the proof is completed by induction. $\qquad\square$

C.2.2. PROOF OF NOT OVERFITTING CASE IN THEOREM 4.1

In this section, we provide the proof of the not-overfitting case in the main theorem. We divide the proof into two parts: behavior on the training data and generalization performance.

**Training data.** The condition of Lemma C.11 is satisfied with the current SNR condition $\mathrm{SNR}^2 = \omega(n^{-1})$. The proposition $C(\tau)$ in Lemma C.11 states that there exists a time step $T_1 = \Theta\left(\frac{1}{\alpha\|\boldsymbol{\nu}\|_2\|\boldsymbol{\mu}\|_2^3 d \max\{\sigma_w^2, \sigma_p^2\}}\right)$ such that

$$g(\Lambda_{i,t}(T_1)) \geq g(\Lambda_{i,t}(0)) + \Theta(1) > c_1', \tag{184}$$

for arbitrary constant $c_1' > 0$, for any $i \in [n]$ and $t \in [T] \setminus \{1\}$. Here, we used $g(\Lambda_{i,t}(0)) = -(1 \pm o(1))T = -\Theta(1)$ from Lemma C.8. Recall that $g(x) = 2x + 2\sinh(x - \log T)$, and there exists a constant $c_2' > 1$ such that $c_2' \exp(x) > g(x)$. Therefore, for all $i \in [n]$, we have

$$s_1^{(i)}(T_1) = \frac{1}{1 + \sum_{t=2}^T \exp(-\Lambda_{i,t}(T_1))} > 1 - \sum_{t=2}^T \exp(-\Lambda_{i,t}(T_1)) > 1 - (T-1)\frac{c_2'}{c_1'} > 1 - \epsilon, \tag{185}$$

for sufficiently small constant $\epsilon > 0$. By using Equation (185) and Lemma B.12, we have that for any clean data $i \in \mathcal{C}$,

$$Y^{(i)} \cdot f_{T_1}(\mathbf{X}^{(i)}) = Y^{(i)} \cdot \boldsymbol{\nu}^\top \mathbf{X}^{(i)\top} \mathbb{S}\left(\mathbf{X}^{(i)}\mathbf{W}(T_1)^\top \mathbf{p}(T_1)\right) \tag{186}$$

$$= Y^{(i)} \cdot \gamma_1^{(i)} s_1^{(i)}(T_1) + \sum_{t=2}^T Y^{(i)} \cdot \gamma_t^{(i)} s_t^{(i)}(T_1) \tag{187}$$

$$\geq \Theta\left(\|\boldsymbol{\nu}\|_2\|\boldsymbol{\mu}\|_2\right)(1-\epsilon) - O\left(\rho\|\boldsymbol{\nu}\|_2\|\boldsymbol{\mu}\|_2\right) \cdot \epsilon \tag{188}$$

$$> 0. \tag{189}$$

Similarly, for any noise data $j \in \mathcal{N}$, we have

$$Y^{(j)} \cdot f_{T_1}(\mathbf{X}^{(j)}) \leq -\Theta\left(\|\boldsymbol{\nu}\|_2\|\boldsymbol{\mu}\|_2\right)(1-\epsilon) + O\left(\rho\|\boldsymbol{\nu}\|_2\|\boldsymbol{\mu}\|_2\right) \cdot \epsilon < 0. \tag{190}$$

Equations (189) and (190) hold deterministically on a good run. Therefore, at time step $\tau = T_1$, we have that with probability at least $1 - \delta$,

$$\forall i \in \mathcal{C}, \ f_\tau(\mathbf{X}^{(i)}) = Y^{(i)}, \ \forall j \in \mathcal{N}, \ f_\tau(\mathbf{X}^{(j)}) \neq Y^{(j)}. \tag{191}$$

**Generalization.** Let $(\mathbf{X}, Y^*) \sim P^*$ be the unseen data on which we investigate generalization performance. We first evaluate the attention values of signal vectors at time step $T_1$. From Lemmas D.5 and D.6, we have

$$\lambda_{+1}(T_1) - \lambda_{+1}(0) \geq \sum_{\tau=0}^{T_1-1} s_1^{(i)}(\tau)(1 - s_1^{(i)}(\tau)) \cdot c\alpha\|\boldsymbol{\nu}\|_2\|\boldsymbol{\mu}\|_2^3 d \max\{\sigma_w^2, \sigma_p^2\} \tag{192}$$

$$\gtrsim T_1 \cdot \alpha\|\boldsymbol{\nu}\|_2\|\boldsymbol{\mu}\|_2^3 d \max\{\sigma_w^2, \sigma_p^2\} \tag{193}$$

$$\gtrsim 1, \tag{194}$$

which follows from the definition of $T_1$. From Lemma C.8, we have $|\lambda_{+1}(0)| = o(1)$, which implies $\lambda_{+1}(T_1) \gtrsim 1$. Similarly, we have $\lambda_{-1}(T_1) \gtrsim 1$.

Next, we show that the attention scores of the noise vectors $\{\boldsymbol{\epsilon}_t\}_{t\in[T]}$ in the unseen data, i.e., $\boldsymbol{\epsilon}_t^\top \mathbf{W}(T_1)^\top \mathbf{p}(T_1)$, become sufficiently small on a good run. While it is natural to evaluate $\|\mathbf{W}(T_1)^\top \mathbf{p}(T_1)\|_2$ and use a concentration inequality, it is challenging to track the evolution of $\|\mathbf{W}(\tau)^\top \mathbf{p}(\tau)\|_2$. Therefore, following induction proof in Lemma C.11, we apply a concentration inequality at time step 0 and show that the result does not change at time step $T_1$. Let us define $\mathcal{E}$ as the event

that the following inequalities are satisfied:

$$\forall t \in [T], \ (1 - o(1)) \, \sigma_\epsilon \sqrt{d} \leq \|\boldsymbol{\epsilon}_t\|_2 \leq (1 + o(1)) \, \sigma_\epsilon \sqrt{d}, \tag{195}$$

$$\forall t \in [T], \forall k \in \{\pm 1\}, \ |\langle \boldsymbol{\epsilon}_t, \boldsymbol{\mu}_k \rangle| < c_2 \sigma_\epsilon \|\boldsymbol{\mu}\|_2 \sqrt{\log(Tn/\delta)}, \tag{196}$$

$$\forall i \in [n], \forall t, u \in [T], \ |\langle \boldsymbol{\epsilon}_t, \boldsymbol{\epsilon}_u^{(i)} \rangle| < c_1 \sigma_\epsilon^2 \sqrt{d} \log(Tn/\delta), \tag{197}$$

$$\forall t \in [T], \forall k \in \{\pm 1\}, \ |\langle \mathbf{W}(0)\boldsymbol{\epsilon}_t, \mathbf{W}(0)\boldsymbol{\mu}_k \rangle| < c_1 \sigma_w^2 \|\boldsymbol{\mu}\|_2 \|\boldsymbol{\epsilon}_t\|_2 \sqrt{d} \log(Tn/\delta), \tag{198}$$

$$\forall i \in [n], \forall t, u \in [T], \ |\langle \mathbf{W}(0)\boldsymbol{\epsilon}_t, \mathbf{W}(0)\boldsymbol{\epsilon}_u^{(i)} \rangle| < c_1 \sigma_w^2 \|\boldsymbol{\epsilon}_t\|_2 \|\boldsymbol{\epsilon}_u^{(j)}\|_2 \sqrt{d} \log(Tn/\delta), \tag{199}$$

$$\forall t \in [T], \ |\langle \mathbf{W}(0)\boldsymbol{\epsilon}_t, \mathbf{p}(0) \rangle| < c_1 \sigma_w \sigma_p \|\boldsymbol{\epsilon}_t\|_2 \sqrt{d} \log(Tn/\delta), \tag{200}$$

$$\forall t \in [T], \ |\langle \boldsymbol{\nu}, \boldsymbol{\epsilon}_t \rangle| < c_2 \sigma_\epsilon \|\boldsymbol{\nu}\|_2 \sqrt{\log(Tn/\delta)}, \tag{201}$$

where the constants $c_1, c_2$ are the same ones appeared in Lemma B.1. Applying a union bound on the modified versions of Lemmas B.6, B.8 and B.9, the probability of the occurrence of $\mathcal{E}$ can be evaluated. Since there is no need to apply the union bound over the additional $n$ training data points, the outlier probability can be reduced by $1/n$ compared to the original lemma. Therefore, we have

$$\Pr\left[\mathcal{E}\right] > 1 - \delta/n > 1 - \delta. \tag{202}$$

In the following, using the results of Lemma C.11, we will prove that the next proposition holds for all $\tau \in [0, T_1]$ under the condition $\mathcal{E}$:

$H(\tau)$:

$$|\langle \mathbf{W}(\tau)\boldsymbol{\epsilon}_t, \mathbf{p}(\tau) \rangle| < O\left(\sigma_w \sigma_p \sigma_\epsilon d \log(Tn/\delta)\right),$$
$$|\langle \mathbf{W}(\tau)\boldsymbol{\epsilon}_t, \mathbf{W}(\tau)\boldsymbol{\mu}_{+1} \rangle| < O\left(\sigma_w^2 \sigma_\epsilon \|\boldsymbol{\mu}\|_2 d \log(Tn/\delta)\right),$$
$$|\langle \mathbf{W}(\tau)\boldsymbol{\epsilon}_t, \mathbf{W}(\tau)\boldsymbol{\mu}_{-1} \rangle| < O\left(\sigma_w^2 \sigma_\epsilon \|\boldsymbol{\mu}\|_2 d \log(Tn/\delta)\right),$$
$$|\langle \mathbf{W}(\tau)\boldsymbol{\epsilon}_t, \mathbf{W}(\tau)\boldsymbol{\epsilon}_u^{(i)} \rangle| < O\left(\sigma_w^2 \sigma_\epsilon^2 d^{3/2} \log(Tn/\delta)\right),$$

for all $i \in [n]$ and $t, u \in [T]$.

*Proof of $H(\tau)$.* We proceed with the proof by induction. The base case holds from the condition $\mathcal{E}$. In the following, we suppose that $H(\tau')$ holds for any $\tau' \in [0, \tau]$. By a calculation similar to that in the proof of Lemma C.4, we have

$$\langle \mathbf{W}(\tau + 1)\boldsymbol{\epsilon}_t, \mathbf{p}(\tau + 1) \rangle - \langle \mathbf{W}(0)\boldsymbol{\epsilon}_t, \mathbf{p}(0) \rangle$$

$$= \sum_{\tau'=0}^{\tau} \left(\langle \mathbf{W}(\tau' + 1)\boldsymbol{\epsilon}_t, \mathbf{p}(\tau' + 1) \rangle - \langle \mathbf{W}(\tau')\boldsymbol{\epsilon}_t, \mathbf{p}(\tau') \rangle\right) \tag{203}$$

$$= \frac{\alpha}{n} \sum_{\tau'=0}^{\tau} \sum_{i=1}^{n} (-\ell_i'(\tau')) \cdot Y^{(i)} \cdot \sum_{t=1}^{T} s_t^{(i)}(\tau') \left(\gamma_t^{(i)} - \sum_{u=1}^{T} s_u^{(i)}(\tau')\gamma_u^{(i)}\right) \left(\langle \mathbf{W}(\tau')\boldsymbol{\epsilon}_t, \mathbf{W}(\tau')\mathbf{x}_t^{(i)} \rangle + \|\mathbf{p}(\tau')\|_2^2 \langle \boldsymbol{\epsilon}_t, \mathbf{x}_t^{(i)} \rangle\right)$$

$$+ \sum_{\tau'=0}^{\tau} \alpha^2 \boldsymbol{\epsilon}_t^\top \nabla_{\mathbf{W}^\top} \widehat{\mathcal{L}}(\tau') \nabla_{\mathbf{p}} \widehat{\mathcal{L}}(\tau'). \tag{204}$$

Using the induction hypothesis, the condition $\mathcal{E}$, and $A(\tau')$ in Lemma C.11, the first term is bounded as follows:

$$\frac{\alpha}{n} \sum_{\tau'=0}^{\tau} \sum_{i=1}^{n} (-\ell_i'(\tau')) \cdot Y^{(i)} \cdot \sum_{t=1}^{T} s_t^{(i)}(\tau') \left( \gamma_t^{(i)} - \sum_{u=1}^{T} s_u^{(i)}(\tau')\gamma_u^{(i)} \right) \left( \langle \mathbf{W}(\tau')\boldsymbol{\epsilon}_t, \mathbf{W}(\tau')\mathbf{x}_t^{(i)} \rangle + \|\mathbf{p}(\tau')\|_2^2 \langle \boldsymbol{\epsilon}_t, \mathbf{x}_t^{(i)} \rangle \right)$$

$$\lesssim \alpha \sum_{\tau'=0}^{\tau} s_1^{(i)}(\tau')(1 - s_1^{(i)}(\tau')) \cdot \max_{i \in [n], u \in [T]} \{\gamma_u^{(i)}\} \cdot \sigma_w^2 \sigma_\epsilon d \max \left\{ \|\boldsymbol{\mu}\|_2, \sigma_\epsilon \sqrt{d} \right\} \log(Tn/\delta)$$

$$+ \alpha \sum_{\tau'=0}^{\tau} s_1^{(i)}(\tau')(1 - s_1^{(i)}(\tau')) \cdot \max_{i \in [n], u \in [T]} \{\gamma_u^{(i)}\} \cdot \sigma_p^2 d \cdot \sigma_\epsilon \max \left\{ \|\boldsymbol{\mu}\|_2 \sqrt{\log(Tn/\delta)}, \sigma_\epsilon \sqrt{d} \log(Tn/\delta) \right\} \quad (205)$$

$$\lesssim \alpha \cdot \frac{1}{\alpha \|\boldsymbol{\nu}\|_2 \|\boldsymbol{\mu}\|_2^3 d \max\{\sigma_w^2, \sigma_p^2\}} \cdot \|\boldsymbol{\nu}\|_2 \|\boldsymbol{\mu}\|_2 \cdot \max\{\sigma_w^2, \sigma_p^2\}\sigma_\epsilon d \cdot \max\{\|\boldsymbol{\mu}\|_2, \sigma_\epsilon \sqrt{d}\} \log(Tn/\delta) \quad (206)$$

$$\lesssim \max\{\|\boldsymbol{\mu}\|_2^{-1}, \sigma_\epsilon \sqrt{d}\|\boldsymbol{\mu}\|_2^{-2}\} \cdot \sigma_\epsilon \log(Tn/\delta) \quad (207)$$

$$= O\left(\sigma_w \sigma_p \sigma_\epsilon d \log(Tn/\delta)\right), \quad (208)$$

where the first line follows from $F(\tau')$, in the same manner as Equation (152), and the second line follows from Lemma D.5. The last line is derived from the parameter assumptions in Section 3.5; specifically, we have

$$\sigma_w \sigma_p \|\boldsymbol{\mu}\|_2 d \gtrsim \min \left\{ \sqrt{d}, \frac{\|\boldsymbol{\mu}\|_2}{\sigma_\epsilon} \right\} \cdot \frac{1}{\log^2(Tn/\delta)} = \Omega(1), \quad (209)$$

$$\sigma_w \sigma_p \sigma_\epsilon^{-1} \|\boldsymbol{\mu}\|_2^2 \sqrt{d} \gtrsim \min \left\{ \frac{\|\boldsymbol{\mu}\|_2}{\sigma_\epsilon}, \frac{\|\boldsymbol{\mu}\|_2^2}{\sigma_\epsilon^2 \sqrt{d}} \right\} \cdot \frac{1}{\log^2(Tn/\delta)} \gtrsim \min \left\{ \frac{\|\boldsymbol{\mu}\|_2}{\sigma_\epsilon \log^2(Tn/\delta)}, C^2 d^{1/4} \right\} = \Omega(1). \quad (210)$$

For the quadratic term in Equation (204), we have

$$\sum_{\tau'=0}^{\tau} \alpha^2 \boldsymbol{\epsilon}_t^\top \nabla_{\mathbf{W}^\top} \widehat{\mathcal{L}}(\tau') \nabla_{\mathbf{p}} \widehat{\mathcal{L}}(\tau')$$

$$\leq \sum_{\tau'=0}^{\tau} \alpha^2 \|\nabla_{\mathbf{W}} \widehat{\mathcal{L}}(\tau)\boldsymbol{\epsilon}_t\|_2 \|\nabla_{\mathbf{p}} \widehat{\mathcal{L}}(\tau)\|_2 \quad (211)$$

$$\lesssim \alpha^2 \sum_{\tau'=0}^{\tau} \left( s_1^{(i)}(\tau')(1 - s_1^{(i)}(\tau')) \cdot \max_{i \in [n], t \in [T]} \{|\gamma_t^{(i)}|\} \right)^2 \cdot \max_{i \in [n], t \in [n]} \{\boldsymbol{\epsilon}_t^\top \mathbf{x}_t^{(i)}\} \|\mathbf{p}(\tau)\|_2 \cdot \max_{i \in [n], t \in [T]} \{\|\mathbf{W}(\tau)\mathbf{x}_t^{(i)}\|_2\} \quad (212)$$

$$\lesssim \alpha^2 \frac{1}{\alpha \|\boldsymbol{\nu}\|_2 \|\boldsymbol{\mu}\|_2^3 d \max\{\sigma_w^2, \sigma_p^2\}} \cdot (\|\boldsymbol{\nu}\|_2 \|\boldsymbol{\mu}\|_2)^2 \quad (213)$$

$$\cdot \sigma_\epsilon \max\{\|\boldsymbol{\mu}\|_2 \sqrt{\log(Tn/\delta)}, \sigma_\epsilon \sqrt{d} \log(Tn/\delta)\} \cdot \sigma_p \sqrt{d} \cdot \sigma_w \max\{\|\boldsymbol{\mu}\|_2 \sqrt{d}, \sigma_\epsilon d\} \quad (214)$$

$$\lesssim \left( \max\{\|\boldsymbol{\mu}\|_2, \sigma_\epsilon \sqrt{d}\} \cdot \sigma_\epsilon \log(Tn/\delta) \right) \cdot \left( \alpha \max\{\|\boldsymbol{\mu}\|_2, \sigma_\epsilon \sqrt{d}\} \right) \quad (215)$$

$$= o(\sigma_w \sigma_p \sigma_\epsilon d \log(Tn/\delta)), \quad (216)$$

where the first line is the result of the Cauchy-Schwarz inequality, and the second one follows from the gradient updates provided in Equations (9) and (12). The third inequality follows from the same discussion as in Equation (206), the condition $\mathcal{E}$, and $A(\tau')$ and $B(\tau')$ in Lemma C.11. The last inequality follows from $\|\boldsymbol{\nu}\|_2 = O(1/\|\boldsymbol{\mu}\|_2)$. Using the parameter assumption on $\alpha$, Equation (215) is absorbed in Equation (207), and the discussion is reduced to the first term analysis. Therefore, substituting Equations (208) and (216) to Equation (204) leads to

$$\langle \mathbf{W}(\tau+1)\boldsymbol{\epsilon}_t, \mathbf{p}(\tau+1) \rangle = \langle \mathbf{W}(0)\boldsymbol{\epsilon}_t, \mathbf{p}(0) \rangle + O\left(\sigma_w \sigma_p \sigma_\epsilon d \log(Tn/\delta)\right) = O\left(\sigma_w \sigma_p \sigma_\epsilon d \log(Tn/\delta)\right). \quad (217)$$

The proof for the three inequalities below in the proposition $H(\tau)$ is omitted because they can be shown in the same manner as $B(\tau+1)$ in Lemma C.11, under the induction hypothesis and the results of Lemma C.11. Since $H(\tau+1)$ holds under the condition $H(\tau)$ is valid, from induction argument, the proposition $H(\tau)$ holds for $\tau \in [0, T_1]$. $\qquad \square$

From $H(T_1)$ and Assumption A8, we have

$$|\langle \mathbf{W}(\tau)\boldsymbol{\epsilon}_t, \mathbf{p}(\tau) \rangle| \lesssim \sigma_w \sigma_p \sigma_\epsilon d \log(Tn/\delta) \leq \frac{c_3'}{\log(Tn/\delta)}, \quad (218)$$

for some constant $c_3' > 1$. Using Equation (194) and taking sufficiently large $T_1$, we have $\lambda_{+1}(T_1) > 2c_3'$ and $\lambda_{-1}(T_1) > 2c_3'$. From Equation (218), we have that for $t \in [T] \setminus \{1\}$,

$$(\mathbf{x}_t - \mathbf{x}_1)^\top \mathbf{W}(T_1)^\top \mathbf{p}(T_1) \le -(1 - \rho) \max\{\lambda_{+1}(T_1), \lambda_{-1}(T_1)\} + (\boldsymbol{\epsilon}_t - \boldsymbol{\epsilon}_1)^\top \mathbf{W}(T_1)^\top \mathbf{p}(T_1) \tag{219}$$

$$\le -(1 - 1/C) \cdot 2c_3' + \frac{2c_3'}{\log(Tn/\delta)} \tag{220}$$

$$< -c_3', \tag{221}$$

where the second inequality holds by $\rho < 1/C$. Then, the softmax probability of the relevant token is lower-bounded as:

$$s_1(T_1) = \frac{1}{1 + \sum_{t=2}^T \exp\left((\mathbf{x}_t - \mathbf{x}_1)^\top \mathbf{W}(T_1)^\top \mathbf{p}(T_1)\right)} > 1 - \frac{T-1}{\exp(c_3')} = 1 - \epsilon, \tag{222}$$

for sufficiently small $\epsilon > 0$. Consequently, we have

$$Y^* \cdot f_{T_1}(\mathbf{X}) = Y^* \cdot \gamma_1 s_1(T_1) + \sum_{t=2}^T Y^* \cdot \gamma_t s_t(T_1) \tag{223}$$

$$\ge \Theta\left(\|\boldsymbol{\nu}\|_2 \|\boldsymbol{\mu}\|_2\right) \cdot (1 - \epsilon) - O\left(\rho \|\boldsymbol{\nu}\|_2 \|\boldsymbol{\mu}\|_2\right) \cdot \epsilon > 0. \tag{224}$$

Under the conditioning on $\mathcal{E}$, the output $f_{T_1}(\mathbf{X})$ deterministically takes the same sign as the true label $Y^*$. Thus, the generalization error is bounded as:

$$\Pr_{(\mathbf{X},Y^*) \sim P^*}[\text{sign}(f_{T_1}(\mathbf{X})) \ne Y^*] = \Pr_{(\mathbf{X},Y^*) \sim P^*}[\text{sign}(f_{T_1}(\mathbf{X})) \ne Y^* \mid \mathcal{E}] + \Pr_{(\mathbf{X},Y^*) \sim P^*}[\mathcal{E}^c] < \delta, \tag{225}$$

where we used $\Pr(A) \le \Pr(A|B^C) + \Pr(B)$ and the result of Equation (202). This concludes the proof.

## C.3. Analysis of Benign Overfitting Case ($\text{SNR}^2 = o(n^{-1})$)

Next, we proceed to the analysis of benign overfitting. Although the proof by induction follows a similar structure to Lemma C.11, the proof is a two-stage analysis divided into $[0, T_1]$ and $[T_1, T_2]$ to address the behavior of noisy data. In Stage 1 ($\tau \in [0, T_1]$), we demonstrate that the probability of selecting the relevant token $\mathbf{x}_1^{(j)}$ for noisy data $j \in \mathcal{N}$ becomes sufficiently small. In Stage 2 ($\tau \in [T_1, T_2]$), we show that the model selects confusing weakly relevant tokens that can fit the noisy labels. While the coefficients differ from Lemma C.11, many arguments are shared between the not-overfitting case and benign overfitting case. To avoid redundancy and unnecessary page length, the proof repetitions are referred to and omitted.

### C.3.1. MAIN LEMMA

**Lemma C.12** (Benign Overfitting, Stage 1). *Suppose that the signal-to-noise ratio satisfies $\text{SNR}^2 = o(n^{-1})$. For some time step $T_1 = \Theta\left(\frac{\rho^{-1}}{\alpha n^{-1} \sigma_\epsilon^2 \|\boldsymbol{\nu}\|_2 \|\boldsymbol{\mu}\|_2 d^2 \max\{\sigma_w^2, \sigma_p^2\}}\right)$, on a good run, the following propositions hold for all time step $\tau \in [0, T_1]$:*

$A(\tau)$: *There exists a constant $C_1 > 1$ such that*

$$(1 - 1/C_1) \cdot \sigma_p \sqrt{d} \le \|\mathbf{p}(\tau)\|_2 \le (1 + 1/C_1) \cdot \sigma_p \sqrt{d}.$$

$B(\tau)$: *There exists a constant $C_2 > 1$ such that*

$$(1 - 1/C_2) \cdot \sigma_w \|\boldsymbol{\mu}\|_2 \sqrt{d} \le \|\mathbf{W}(\tau)\boldsymbol{\mu}_{+1}\|_2 \le (1 + 1/C_2) \cdot \sigma_w \|\boldsymbol{\mu}\|_2 \sqrt{d},$$

$$(1 - 1/C_2) \cdot \sigma_w \|\boldsymbol{\mu}\|_2 \sqrt{d} \le \|\mathbf{W}(\tau)\boldsymbol{\mu}_{-1}\|_2 \le (1 + 1/C_2) \cdot \sigma_w \|\boldsymbol{\mu}\|_2 \sqrt{d},$$

$$(1 - 1/C_2) \cdot \sigma_w \sigma_\epsilon d \le \|\mathbf{W}(\tau)\boldsymbol{\epsilon}_t^{(i)}\|_2 \le (1 + 1/C_2) \cdot \sigma_w \sigma_\epsilon d,$$

*for all $i \in [n]$ and $t \in [T]$.*

$$|\langle \mathbf{W}(\tau)\boldsymbol{\mu}_{+1}, \mathbf{W}(\tau)\boldsymbol{\mu}_{-1}\rangle| < O(\sigma_w^2 \|\boldsymbol{\mu}\|_2^2 \sqrt{d} \log(Tn/\delta)),$$

$$|\langle \mathbf{W}(\tau)\boldsymbol{\mu}_{+1}, \mathbf{W}(\tau)\boldsymbol{\epsilon}_t^{(i)}\rangle| < O(\sigma_w^2 \sigma_\epsilon \|\boldsymbol{\mu}\|_2 d \log(Tn/\delta)),$$

$$|\langle \mathbf{W}(\tau)\boldsymbol{\mu}_{-1}, \mathbf{W}(\tau)\boldsymbol{\epsilon}_t^{(i)}\rangle| < O(\sigma_w^2 \sigma_\epsilon \|\boldsymbol{\mu}\|_2 d \log(Tn/\delta)),$$

$$|\langle \mathbf{W}(\tau)\boldsymbol{\epsilon}_t^{(i)}, \mathbf{W}(\tau)\boldsymbol{\epsilon}_u^{(j)}\rangle| < O(\sigma_w^2 \sigma_\epsilon^2 d^{3/2} \log(Tn/\delta)),$$

*for all $i, j \in [n]$ and $t, u \in [T]$ such that $(i, t) \neq (j, u)$.*

$C(\tau)$**:** *Let $g$ be a monotonically increasing function $g(x) = 2x + 2\sinh(x - \log T)$. Then, there exist constants $c_3, c_4 > 0$ with $c_3 < c_4$ such that*

$$g(\Lambda_{i,t}(\tau)) \geq g(\Lambda_{i,t}(0)) + \tau \cdot c_3 \alpha n^{-1} \sigma_\epsilon^2 \|\boldsymbol{\nu}\|_2 \|\boldsymbol{\mu}\|_2 d^2 \max\{\sigma_w^2, \sigma_p^2\},$$

$$g(\Lambda_{i,t}(\tau)) \leq g(\Lambda_{i,t}(0)) + \tau \cdot c_4 \alpha n^{-1} \sigma_\epsilon^2 \|\boldsymbol{\nu}\|_2 \|\boldsymbol{\mu}\|_2 d^2 \max\{\sigma_w^2, \sigma_p^2\},$$

*for any clean data $i \in \mathcal{C}$ and $t \in [T] \setminus \{1\}$. Additionally, we have that for some constants $c_5 < c_6$,*

$$g(\Lambda_{j,t}(\tau)) \leq g(\Lambda_{j,t}(0)) - \tau \cdot c_5 \alpha n^{-1} \sigma_\epsilon^2 \|\boldsymbol{\nu}\|_2 \|\boldsymbol{\mu}\|_2 d^2 \max\{\sigma_w^2, \sigma_p^2\},$$

$$g(\Lambda_{j,t}(\tau)) \geq g(\Lambda_{j,t}(0)) - \tau \cdot c_6 \alpha n^{-1} \sigma_\epsilon^2 \|\boldsymbol{\nu}\|_2 \|\boldsymbol{\mu}\|_2 d^2 \max\{\sigma_w^2, \sigma_p^2\},$$

*for any noisy data $j \in \mathcal{N}$ and $t \in [T] \setminus \{1\}$.*

$D(\tau)$**:**

$$s_1^{(i)}(\tau) \geq \Theta(1), \ s_1^{(j)}(\tau) \geq \Theta(\rho),$$

*for all clean data $i \in \mathcal{C}$ and noisy data $j \in \mathcal{N}$.*

$E(\tau)$**:** *There exists a constant $c_7 > 1$ such that*

$$\exp(\Lambda_{i,t}(\tau) - \Lambda_{j,u}(\tau)) < c_7, \ \exp(\Lambda_{k,t}(\tau) - \Lambda_{l,u}(\tau)) < c_7,$$

*for all $i, j \in \mathcal{C}$, $k, l \in \mathcal{N}$ and $t, u \in [T] \setminus \{1\}$. Additionally, we have*

$$\exp(\Lambda_{i,t}(\tau) + \Lambda_{k,u}(\tau)) < c_7, \ \exp(-\Lambda_{i,t}(\tau) - \Lambda_{k,u}(\tau)) < c_7,$$

*for all $i \in \mathcal{C}, k \in \mathcal{N}$ and $t, u \in [T] \setminus \{1\}$.*

$F(\tau)$**:** *There exists a constant $c_8 > 1$ such that*

$$\max_{i,j \in [n]} \frac{s_1^{(i)}(\tau)(1 - s_1^{(i)}(\tau))}{s_1^{(j)}(\tau)(1 - s_1^{(j)}(\tau))} < c_8.$$

*Additionally, there exist constants $c_9, c_{10} > 1$ such that*

$$\max_{t,u \in [T] \setminus \{1\}} \frac{s_t^{(i)}(\tau)}{s_u^{(i)}(\tau)} < c_9, \ \frac{s_t^{(i')}(\tau)(1 - s_t^{(i')}(\tau))}{s_1^{(i')}(\tau)(1 - s_1^{(i')}(\tau))} < c_{10},$$

*for any $i \in [n]$, $i' \in \mathcal{C}$, and $t \in [T] \setminus \{1\}$.*

$G(\tau)$**:**

$$|\lambda_{+1}(\tau)| = o(\log \rho^{-1}), \ |\lambda_{-1}(\tau)| = o(\log \rho^{-1}), \ |\rho_{i,t}(\tau)| = O(\log \rho^{-1}),$$

*for any $i \in [n]$ and $t \in [T]$.*

We will prove the above propositions by induction argument for $\tau \in [0, T_1]$.

**Base case.** It is obvious that $C(0)$ holds. By Lemmas B.1, C.7 and C.8, the all other base cases $A(0)$, $B(0)$, $D(0)$, $E(0)$, $F(0)$ and $G(0)$ hold.

**Proof of** $\wedge_{\tau' \leq \tau} (A(\tau') \wedge B(\tau') \wedge C(\tau') \wedge D(\tau') \wedge E(\tau') \wedge F(\tau')) \Rightarrow C(\tau + 1)$**.** We only show the case of $Y^{(i)} = 1$, i.e., $i \in \mathcal{C}_+ \cup \mathcal{N}$. The same argument can be applied to the case of $i \in \mathcal{C}_- \cup \mathcal{N}_+$. In the following, we bound the attention gap introduced in Definition C.3 from both above and below, as in the proof of Lemma C.11 for the not-overfitting case. The conditions of Lemmas D.8 and D.9 are satisfied from $A(\tau) \wedge B(\tau) \wedge D(\tau) \wedge F(\tau)$. From these lemmas and the definition of $\Lambda_{i,t}$, we have that for any $i \in \mathcal{C}_+$ and $t \in \mathcal{I}^{(i)}$,

$$\Lambda_{i,t}(\tau + 1) - \Lambda_{i,t}(\tau) = (\lambda_{+1}(\tau + 1) - \lambda_{+1}(\tau)) + (\rho_{i,1}(\tau + 1) - \rho_{i,1}(\tau)) - (\rho_{i,t}(\tau + 1) - \rho_{i,t}(\tau)) \tag{226}$$

$$\geq s_1^{(i)}(\tau)(1 - s_1^{(i)}(\tau)) \cdot \alpha \|\boldsymbol{\nu}\|_2 \|\boldsymbol{\mu}\|_2 d \left( c_1' \|\boldsymbol{\mu}\|_2^2 + c_2' n^{-1} \sigma_\epsilon^2 d \right) \max\{\sigma_w^2, \sigma_p^2\}$$

$$+ s_1^{(i)}(\tau) s_t^{(i)}(\tau) \cdot c_2' \alpha n^{-1} \sigma_\epsilon^2 \|\boldsymbol{\nu}\|_2 \|\boldsymbol{\mu}\|_2 d^2 \max\{\sigma_w^2, \sigma_p^2\} \tag{227}$$

$$\geq s_1^{(i)}(\tau)(1 - s_1^{(i)}(\tau)) \cdot c_2' \alpha n^{-1} \sigma_\epsilon^2 \|\boldsymbol{\nu}\|_2 \|\boldsymbol{\mu}\|_2 d^2 \max\{\sigma_w^2, \sigma_p^2\}. \tag{228}$$

Here, we bound with the noise memorization term because it is dominant from the condition $\mathrm{SNR}^2 = o(n^{-1})$, i.e., $\|\boldsymbol{\mu}\|_2^2 = o(n^{-1} \sigma_\epsilon^2 d)$. For noisy data $j \in \mathcal{N}_-$, using $s_1^{(i)}(\tau)(1 - s_1^{(i)}(\tau)) \lesssim s_1^{(j)}(\tau)(1 - s_1^{(j)}(\tau))$ for any clean data $i \in \mathcal{C}$ from $F(\tau)$, we have

$$\Lambda_{j,t}(\tau + 1) - \Lambda_{j,t}(\tau) \leq s_1^{(i)}(\tau)(1 - s_1^{(i)}(\tau)) \cdot c_3' \alpha \|\boldsymbol{\nu}\|_2 \|\boldsymbol{\mu}\|_2^3 d$$

$$- s_1^{(j)}(\tau)(1 - s_1^{(j)}(\tau)) \cdot c_4' \alpha n^{-1} \sigma_\epsilon^2 \|\boldsymbol{\nu}\|_2 \|\boldsymbol{\mu}\|_2 d^2 \max\{\sigma_w^2, \sigma_p^2\}$$

$$+ s_1^{(j)}(\tau) s_t^{(j)}(\tau) \cdot c_4' \alpha n^{-1} \sigma_\epsilon^2 \|\boldsymbol{\nu}\|_2 \|\boldsymbol{\mu}\|_2 d^2 \max\{\sigma_w^2, \sigma_p^2\} \tag{229}$$

$$\leq s_1^{(j)}(\tau)(1 - s_1^{(j)}(\tau)) \cdot \alpha \|\boldsymbol{\nu}\|_2 \|\boldsymbol{\mu}\|_2 d \left( c_5' \|\boldsymbol{\mu}\|_2^2 - c_4' n^{-1} \sigma_\epsilon^2 d \right) \max\{\sigma_w^2, \sigma_p^2\}$$

$$+ s_1^{(j)}(\tau) s_t^{(j)}(\tau) \cdot c_4' \alpha n^{-1} \sigma_\epsilon^2 \|\boldsymbol{\nu}\|_2 \|\boldsymbol{\mu}\|_2 d^2 \max\{\sigma_w^2, \sigma_p^2\} \tag{230}$$

$$\leq -s_1^{(j)}(\tau)(1 - s_1^{(j)}(\tau)) \cdot c_6' \alpha n^{-1} \sigma_\epsilon^2 \|\boldsymbol{\nu}\|_2 \|\boldsymbol{\mu}\|_2 d^2 \max\{\sigma_w^2, \sigma_p^2\}. \tag{231}$$

We note that we have the same result for weakly relevant tokens $t \in \mathcal{W}_{+1}^{(i)} \cup \mathcal{W}_{-1}^{(i)}$. When $t \in \mathcal{W}_{+1}^{(i)}$, the update of $\lambda_{+1}$ is multiplied by a factor of $(1 - \rho)$. Since the noise update is dominant in this case, it does not affect the above bounds. For $t \in \mathcal{W}_{-1}^{(i)}$, we also have to consider the update of $\lambda_{-1}$. However, by similarly applying Lemmas D.8 and D.9, along with the result for the softmax probability ratio $F(\tau)$, the case for $t \in \mathcal{W}_{-1}^{(i)}$ reduces to that of $t \in \mathcal{W}_{+1}^{(i)}$. Therefore, the above inequalities hold for any $t \in [T] \setminus \{1\}$. Similarly, from Lemmas D.8 and D.9, we obtain the following upper bound:

$$\Lambda_{i,t}(\tau + 1) - \Lambda_{i,t}(\tau) \leq s_1^{(i)}(\tau)(1 - s_1^{(i)}(\tau)) \cdot c_7' \alpha n^{-1} \sigma_\epsilon^2 \|\boldsymbol{\nu}\|_2 \|\boldsymbol{\mu}\|_2 d^2 \max\{\sigma_w^2, \sigma_p^2\}, \tag{232}$$

$$\Lambda_{j,t}(\tau + 1) - \Lambda_{j,t}(\tau) \geq -s_1^{(j)}(\tau)(1 - s_1^{(j)}(\tau)) \cdot c_8' \alpha n^{-1} \sigma_\epsilon^2 \|\boldsymbol{\nu}\|_2 \|\boldsymbol{\mu}\|_2 d^2 \max\{\sigma_w^2, \sigma_p^2\}, \tag{233}$$

for $i \in \mathcal{C}_+, j \in \mathcal{N}_-$ and $t \in [T] \setminus \{1\}$.

The evaluation of the evolution of $\Lambda$ is identical to the proof of the not overfitting case in Lemma C.11, differing only in coefficients; therefore, we omit the discussion. For noisy data, since $\Lambda_{j,t}(\tau)$ is monotonically decreasing, the evaluation in Case 1 ($\Lambda_{j,t}(\tau) \leq \log T$) suffices to complete the argument.

**Proof of** $C(\tau) \Rightarrow D(\tau)$**.** The lower bound of clean data is derived from $C(\tau)$ and $D(0)$. To derive the lower bound for noisy data $j \in \mathcal{N}$, we use a similar technique to Equation (419) in the proof of Lemma D.5. From $C(\tau)$ and Lemma C.8, we have $\Lambda_{j,t}(\tau) < o(1)$ holds for any $t \in [T] \setminus \{1\}$. Since we can confirm that the inequality $2 + 2\cosh(x - \log T) <$

$-2x - 2\sinh(x - \log T) + 2 = -g(x) + 2$ holds for $x < o(1)$ by rearranging terms, Lemma D.1 and $C(\tau)$ give us

$$s_1^{(j)}(\tau) > s_1^{(j)}(\tau)(1 - s_1^{(j)}(\tau)) > \frac{c^{-1}}{2 + 2\cosh\left(\Lambda_{j,t}(\tau') - \log T\right)} \tag{234}$$

$$> \frac{c^{-1}}{-g\left(\Lambda_{j,t}(\tau')\right) + 2} \tag{235}$$

$$> \frac{c^{-1}}{-g\left(\Lambda_{j,t}(0)\right) + \tau \cdot c_6 \alpha n^{-1}\sigma_\epsilon^2 \|\boldsymbol{\nu}\|_2 \|\boldsymbol{\mu}\|_2 d^2 \max\{\sigma_w^2, \sigma_p^2\} + 2} \tag{236}$$

$$= \Theta(\rho), \tag{237}$$

where the last line follows from the definition of $T_1$ and $-g(\Lambda_{j,t}(0)) = (1 + o(1))T$. This completes the proof.

**Proof of** $C(\tau) \Rightarrow E(\tau)$**.** The proof for the clean data is the same as the not-benign overfitting case in Lemma C.11; therefore, we omit the discussion. We first provide the analysis of the noisy data with proof by contradiction. Suppose that $E(\tau)$ does not hold, i.e., for any constant $c > 0$, there exist $k, l \in \mathcal{N}$ and $t, u \in [T] \setminus \{1\}$ such that

$$\exp\left(\Lambda_{k,t}(\tau) - \Lambda_{l,u}(\tau)\right) \geq c. \tag{238}$$

From $C(\tau)$, we have for any $k, l \in \mathcal{N}$ and $t, u \in [T] \setminus \{1\}$ that

$$\frac{g\left(\Lambda_{l,u}(\tau)\right) - g\left(\Lambda_{l,u}(0)\right)}{g\left(\Lambda_{k,t}(\tau)\right) - g\left(\Lambda_{k,t}(0)\right)} \leq \frac{c_6}{c_5}, \tag{239}$$

for $\tau \geq 1$. Additionally, we have $g\left(\Lambda_{l,u}(0)\right)/g\left(\Lambda_{k,t}(0)\right) < 2$ by the same argument as in Equation (128). Also, $\Lambda_{k,t}(\tau) < o(1)$ and $\Lambda_{l,t}(\tau) < o(1)$ are implied by Lemma C.8 and the fact that $\Lambda(\tau)$ is monotonically decreasing as a result of $C(\tau)$. By a simple calculation, we obtain the following for $x < o(1) - \log c$:

$$2\sinh\left(x + \log c - \log T\right) = \frac{c}{T}\exp(x) - \frac{T}{c}\exp(-x) \tag{240}$$

$$= \frac{3}{c} \cdot 2\sinh\left(x - \log T\right) + \frac{2T}{c}\exp(-x) + \left(c - \frac{3}{c}\right)\frac{\exp(x)}{T} \tag{241}$$

$$> \frac{3}{c} \cdot 2\sinh\left(x - \log T\right) - 2Tx, \tag{242}$$

for $c$ satisfying $c > 3/c$. In the last line, we used $\exp(-x) > -cx$ for $x < o(1) - \log c$. By using Equations (238) and (242), we have

$$g\left(\Lambda_{k,t}(\tau)\right) - g\left(\Lambda_{k,t}(0)\right) = 2\Lambda_{k,t}(\tau) + 2\sinh\left(\Lambda_{k,t}(\tau) - \log T\right) - g\left(\Lambda_{k,t}(0)\right) \tag{243}$$

$$\geq 2\left(\Lambda_{l,u}(\tau) + \log c\right) + 2\sinh\left(\Lambda_{l,u}(\tau) + \log c - \log T\right) - \frac{1}{2}g\left(\Lambda_{l,u}(0)\right) \tag{244}$$

$$> (2 - 2T)\Lambda_{l,u}(\tau) + \frac{3}{c} \cdot 2\sinh\left(\Lambda_{l,u}(\tau) - \log T\right) + \frac{1}{2}\left(-g\left(\Lambda_{l,u}(0)\right)\right) \tag{245}$$

$$> \min\{c_5/c_6, 1/2\} \cdot \left(g\left(\Lambda_{l,u}(\tau)\right) - g\left(\Lambda_{l,u}(0)\right)\right). \tag{246}$$

In the second inequality, please note that $\Lambda_{l,u}(\tau) < \Lambda_{k,t}(\tau) - \log c = o(1) - \log c < 0$, $\sinh(\Lambda_{l,u}(\tau) - \log T) < 0$, and $-g(\Lambda_{l,u}(0)) = (1 \pm o(1))T > 0$. Thus, the last line follows from a sufficiently large $c > 0$ satisfying $3/c < \min\{c_5/c_6, 1/2\}$. Since the left-hand side of Equation (246) is negative by $C(\tau)$, Equation (246) leads to

$$\frac{g\left(\Lambda_{l,u}(\tau)\right) - g\left(\Lambda_{l,u}(0)\right)}{g\left(\Lambda_{k,t}(\tau)\right) - g\left(\Lambda_{k,t}(0)\right)} > \max\left\{\frac{c_6}{c_5}, 2\right\} \geq \frac{c_6}{c_5}, \tag{247}$$

which contradicts Equation (239).

The remaining is to show $\exp(\Lambda_{i,t}(\tau) + \Lambda_{k,u}(\tau)) < c_7$ and $\exp(-\Lambda_{i,t}(\tau) - \Lambda_{k,u}(\tau)) < c_7$. This can be shown in the same manner as the proof of Lemma C.11 for the not-overfitting case. Specifically, we divide the analysis into the initial learning phase (Case 1) and the subsequent phase (Case 2). In Case 1, the difference between $g(\Lambda_{i,t}(\tau))$ and $g(\Lambda_{k,u}(\tau))$ is shown to be of constant order in a similar way. In Case 2, since they have the same order of time evolution with opposite signs by $C(\tau)$, a similar discussion can be applied. This completes the proof.

**Proof of $E(\tau) \Rightarrow F(\tau)$.** The ratios of $s_1(\tau)(1 - s_1(\tau))$ within the clean data and within the noisy data are respectively bounded by a constant, by the same argument as in Equation (141) for the not-overfitting case. We discuss the ratio between clean data and noisy data. From the definition of $\Lambda_{i,t}$, we have

$$\frac{1}{s_1^{(i)}(\tau)(1 - s_1^{(i)}(\tau))} = \left(1 + \sum_{t=2}^{T} \exp\left(-\Lambda_{i,t}(\tau)\right)\right) \cdot \frac{1 + \sum_{t=2}^{T} \exp\left(-\Lambda_{i,t}(\tau)\right)}{\sum_{t=2}^{T} \exp\left(-\Lambda_{i,t}(\tau)\right)} \tag{248}$$

$$= 2 + \sum_{t=2}^{T} \exp\left(-\Lambda_{i,t}(\tau)\right) + \frac{1}{\sum_{t=2}^{T} \exp\left(-\Lambda_{i,t}(\tau)\right)}. \tag{249}$$

Thus, for any $i \in \mathcal{C}$ and $j \in \mathcal{N}$, there exists a constant $c_1'$ such that

$$\frac{s_1^{(j)}(\tau)(1 - s_1^{(j)}(\tau))}{s_1^{(i)}(\tau)(1 - s_1^{(i)}(\tau))} < \frac{s_1^{(j)}(\tau)}{s_1^{(i)}(\tau)(1 - s_1^{(i)}(\tau))} \tag{250}$$

$$= \frac{2 + \sum_{t=2}^{T} \exp\left(-\Lambda_{i,t}(\tau)\right)}{1 + \sum_{t=2}^{T} \exp\left(-\Lambda_{j,t}(\tau)\right)} + \frac{1}{\left(1 + \sum_{t=2}^{T} \exp\left(-\Lambda_{j,t}(\tau)\right)\right) \cdot \sum_{t=2}^{T} \exp\left(-\Lambda_{i,t}(\tau)\right)} \tag{251}$$

$$\leq \frac{2 + (T-1)o(1)}{1 + (T-1)o(1)} + \exp\left(\Lambda_{i,u}(\tau) + \Lambda_{j,v}(\tau)\right) < c_1', \tag{252}$$

for any $u, v \in [T] \setminus \{1\}$. In the last line, we used that $\Lambda_{i,t}(\tau) > -o(1)$, $\Lambda_{j,t}(\tau) < o(1)$ and the result of $E(\tau)$. Similarly, we have

$$\frac{s_1^{(i)}(\tau)(1 - s_1^{(i)}(\tau))}{s_1^{(j)}(\tau)(1 - s_1^{(j)}(\tau))} < \frac{1 - s_1^{(i)}(\tau)}{s_1^{(j)}(\tau)(1 - s_1^{(j)}(\tau))} \tag{253}$$

$$= \frac{\sum_{t=2}^{T} \exp\left(-\Lambda_{i,t}(\tau)\right)}{1 + \sum_{t=2}^{T} \exp\left(-\Lambda_{i,t}(\tau)\right)} \cdot \left(2 + \sum_{t=2}^{T} \exp\left(-\Lambda_{j,t}(\tau)\right) + \frac{1}{\sum_{t=2}^{T} \exp\left(-\Lambda_{j,t}(\tau)\right)}\right) \tag{254}$$

$$< 2 + T^2 \max_{u,v \in [T] \setminus \{1\}} \exp\left(-\Lambda_{i,u}(\tau) - \Lambda_{j,v}(\tau)\right) + \frac{(T-1)o(1)}{(T-1)o(1)} < c_2', \tag{255}$$

which follows from $\Lambda_{i,t}(\tau) > -o(1)$, $\Lambda_{j,t}(\tau) < o(1)$ and the parameter assumption $T = \Theta(1)$.

Finally, $E(\tau)$ implies

$$\max_{t,u \in [T] \setminus \{1\}} \frac{s_t^{(i)}(\tau)}{s_u^{(i)}(\tau)} = \max_{t,u \in [T] \setminus \{1\}} \frac{\exp\left(\mathbf{x}_t^{(i)\top} \mathbf{W}(\tau)^\top \mathbf{p}(\tau)\right)}{\exp\left(\mathbf{x}_u^{(i)\top} \mathbf{W}(\tau)^\top \mathbf{p}(\tau)\right)} = \max_{t,u \in [T] \setminus \{1\}} \frac{\exp\left(-\Lambda_{i,t}(\tau)\right)}{\exp\left(-\Lambda_{i,u}(\tau)\right)} < c_7, \tag{256}$$

for any $i \in [n]$. The dominance of $s_1^{(i)}(\tau)(1 - s_1^{(i)}(\tau))$ for $i \in \mathcal{C}$, relative to all tokens $t \in [T] \setminus \{1\}$, is shown by the same argument as in Lemma C.11.

**Proof of $\wedge_{\tau' \leq \tau} \left(A(\tau') \wedge B(\tau') \wedge C(\tau') \wedge D(\tau') \wedge F(\tau')\right) \Rightarrow G(\tau+1)$.** The conditions of Lemma D.8 are satisfied from $A(\tau) \wedge B(\tau) \wedge D(\tau) \wedge F(\tau)$. From Lemma D.8, we have

$$\lambda_{+1}(\tau+1) - \lambda_{+1}(0) \leq \sum_{\tau'=0}^{\tau} s_1^{(i)}(\tau')(1 - s_1^{(i)}(\tau')) \cdot c'\alpha \|\boldsymbol{\nu}\|_2 \|\boldsymbol{\mu}\|_2^3 d \max\{\sigma_w^2, \sigma_p^2\}, \tag{257}$$

for any clean data $i \in \mathcal{C}$. To derive the upper bound of the summation of softmax probability terms, we use Lemma D.5. Since Lemma D.5 follows from $\wedge_{\tau' \leq \tau} C(\tau')$ and the definition of $T_1$, we have

$$\sum_{\tau'=0}^{\tau} s_1^{(i)}(\tau')(1 - s_1^{(i)}(\tau'))$$

$$\lesssim \max \left\{ \frac{1}{\alpha n^{-1} \sigma_\epsilon^2 \|\boldsymbol{\nu}\|_2 \|\boldsymbol{\mu}\|_2 d^2 \max\{\sigma_w^2, \sigma_p^2\}}, \frac{\log \left( \tau \cdot \alpha n^{-1} \sigma_\epsilon^2 \|\boldsymbol{\nu}\|_2 \|\boldsymbol{\mu}\|_2 d^2 \max\{\sigma_w^2, \sigma_p^2\} \right)}{\alpha n^{-1} \sigma_\epsilon^2 \|\boldsymbol{\nu}\|_2 \|\boldsymbol{\mu}\|_2 d^2 \max\{\sigma_w^2, \sigma_p^2\}} \right\} \tag{258}$$

$$\leq \max \left\{ \frac{1}{\alpha n^{-1} \sigma_\epsilon^2 \|\boldsymbol{\nu}\|_2 \|\boldsymbol{\mu}\|_2 d^2 \max\{\sigma_w^2, \sigma_p^2\}}, \frac{\log \left( T_1 \cdot \alpha n^{-1} \sigma_\epsilon^2 \|\boldsymbol{\nu}\|_2 \|\boldsymbol{\mu}\|_2 d^2 \max\{\sigma_w^2, \sigma_p^2\} \right)}{\alpha n^{-1} \sigma_\epsilon^2 \|\boldsymbol{\nu}\|_2 \|\boldsymbol{\mu}\|_2 d^2 \max\{\sigma_w^2, \sigma_p^2\}} \right\} \tag{259}$$

$$= \frac{\log \rho^{-1}}{\alpha n^{-1} \sigma_\epsilon^2 \|\boldsymbol{\nu}\|_2 \|\boldsymbol{\mu}\|_2 d^2 \max\{\sigma_w^2, \sigma_p^2\}}. \tag{260}$$

Substituting this to Equation (257) and using the current SNR condition $\mathrm{SNR}^2 = o(n^{-1})$, we have

$$\lambda_{+1}(\tau+1) - \lambda_{+1}(0) = o(\log \rho^{-1}). \tag{261}$$

Since $|\lambda_{+1}(0)| = o(1)$ from Lemma C.8, Equation (261) lead to $\lambda_{+1}(\tau+1) \leq o(\log \rho^{-1})$.

In the same way, we have $\lambda_{-1}(\tau+1) \leq o(\log \rho^{-1})$. For noise memorization terms, from Lemma D.9, we have for any $i \in [n]$ that

$$|\rho_{i,1}(\tau+1) - \rho_{i,1}(0)| \leq \sum_{\tau'=0}^{\tau} s_1^{(i)}(\tau')(1 - s_1^{(i)}(\tau')) \cdot c' \alpha n^{-1} \sigma_\epsilon^2 \|\boldsymbol{\nu}\|_2 \|\boldsymbol{\mu}\|_2 d^2 \max\{\sigma_w^2, \sigma_p^2\}. \tag{262}$$

We proceed with the same argument as the previously discussed signal update case and obtain the upper bound $|\rho_{i,1}(\tau+1)| \leq O(\log \rho^{-1})$. The bound of $\rho_{i,t}$ is obtained as well.

**Proof of** $\wedge_{\tau' \leq \tau} (B(\tau') \wedge C(\tau') \wedge F(\tau') \wedge G(\tau')) \Rightarrow A(\tau+1)$**.** The proof is basically the same as the not-overfitting case, but we newly have to care about the behavior of noisy samples. It follows from Lemma C.5 that

$$\|\mathbf{p}(\tau+1)\|_2^2 - \|\mathbf{p}(0)\|_2^2 = \frac{2\alpha}{n} \sum_{\tau'=0}^{\tau} \sum_{i=1}^{n} (-\ell_i'(\tau')) \cdot Y^{(i)} \cdot I_i^p(\tau') + \alpha^2 \sum_{\tau'=0}^{\tau} \|\nabla_{\mathbf{p}} \widehat{\mathcal{L}}(\tau')\|_2^2. \tag{263}$$

By definition, we have

$$I_i^p(\tau') = \sum_{t=1}^{T} s_t^{(i)}(\tau') \left( \gamma_t^{(i)} - \sum_{u=1}^{T} s_u^{(i)}(\tau') \gamma_u^{(i)} \right) \langle \mathbf{W}(\tau') \mathbf{x}_t^{(i)}, \mathbf{p}(\tau') \rangle \tag{264}$$

$$\leq \sum_{t=1}^{T} \left| s_t^{(i)}(\tau') \left( \gamma_t^{(i)} - \sum_{u=1}^{T} s_u^{(i)}(\tau') \gamma_u^{(i)} \right) \right| \cdot \max \left\{ |\lambda_{+1}(\tau')|, |\lambda_{-1}(\tau')|, |\rho_{i,t}(\tau')| \right\}. \tag{265}$$

For clean data $i \in \mathcal{C}$, we use the argument similar to Equation (152), which follows from $F(\tau')$, and combine it with $G(\tau')$, Lemmas B.12 and D.5. Then, we have

$$\sum_{\tau'=0}^{\tau} I_i^p(\tau') \lesssim \sum_{\tau'=0}^{\tau} s_1^{(i)}(\tau')(1 - s_1^{(i)}(\tau')) \cdot \max_{t \in [T]} \left\{ |\gamma_t^{(i)}| \right\} \cdot \max \left\{ |\lambda_{+1}(\tau')|, |\lambda_{-1}(\tau')|, |\rho_{i,t}(\tau')| \right\} \tag{266}$$

$$\lesssim \frac{\log^2 \rho^{-1}}{\alpha n^{-1} \sigma_\epsilon^2 d^2 \max\{\sigma_w^2, \sigma_p^2\}}. \tag{267}$$

For noisy data $j \in \mathcal{N}$, the softmax probability $s_1^{(j)}(\tau')(1 - s_1^{(j)}(\tau'))$ is not dominant over the other tokens. Instead, we have

$$\sum_{\tau'=0}^{\tau} \sum_{t=1}^{T} \left| s_t^{(j)}(\tau') \left( \gamma_t^{(j)} - \sum_{u=1}^{T} s_u^{(j)}(\tau')\gamma_u^{(j)} \right) \right|$$

$$\leq \sum_{\tau'=0}^{\tau} \left( \left| s_1^{(j)}(\tau') \left( \gamma_1^{(j)} - \sum_{u=1}^{T} s_u^{(j)}(\tau')\gamma_u^{(j)} \right) \right| + \sum_{t=2}^{T} \left| s_t^{(j)}(\tau')s_1^{(j)}(\tau')\gamma_1^{(j)} \right| + \sum_{t=2}^{T} \left| s_t^{(j)}(\tau') \left( \gamma_t^{(j)} - \sum_{u=2}^{T} s_u^{(j)}(\tau')\gamma_u^{(j)} \right) \right| \right) \tag{268}$$

$$\lesssim \sum_{\tau'=0}^{\tau} \left( s_1^{(j)}(\tau')(1 - s_1^{(j)}(\tau')) \cdot |\gamma_1^{(j)}| + \max_{t \in [T] \setminus \{1\}} \{|\gamma_t^{(j)}|\} \right) \tag{269}$$

$$\lesssim \frac{\log \rho^{-1}}{\alpha n^{-1}\sigma_\epsilon^2 d^2 \max\{\sigma_w^2, \sigma_p^2\}} + \frac{\rho}{\alpha n^{-1}\sigma_\epsilon^2 d^2 \max\{\sigma_w^2, \sigma_p^2\}} \lesssim \frac{\log \rho^{-1}}{\alpha n^{-1}\sigma_\epsilon^2 d^2 \max\{\sigma_w^2, \sigma_p^2\}}, \tag{270}$$

where the last line follows from Lemma D.5, $\max_{t \in [T] \setminus \{1\}} \{|\gamma_t^{(j)}|\} = O(\rho\|\boldsymbol{\nu}\|_2\|\boldsymbol{\mu}\|_2)$, and the definition of $T_1$. Therefore, we have

$$\sum_{\tau'=0}^{\tau} I_j^p(\tau') \lesssim \frac{\log^2 \rho^{-1}}{\alpha n^{-1}\sigma_\epsilon^2 d^2 \max\{\sigma_w^2, \sigma_p^2\}}. \tag{271}$$

Now we have Equations (267) and (271), and the remainder of the proof follows from the same discussion as in the not-overfitting case.

**Proof of** $\wedge_{\tau' \leq \tau} (A(\tau') \wedge C(\tau') \wedge G(\tau')) \Rightarrow B(\tau + 1)$**.**  As with the proof of $A(\tau + 1)$, the contribution of the noisy data can be evaluated in the same order as the clean data, similarly to Equation (271). The proof is essentially the same as the not-overfitting case, so we omit the proof.

*Proof of Lemma C.12.*  At time step $\tau = 0$, all propositions hold from the proof for the base case. For the next time step, $A(\tau + 1), B(\tau + 1), C(\tau + 1)$ and $G(\tau + 1)$ are proved based on the propositions up to time step $\tau$. As for $D(\tau + 1)$, $E(\tau + 1)$, and $F(\tau + 1)$, they are derived from $C(\tau + 1)$. Thus, the proof is completed by induction. □

**Lemma C.13** (Benign Overfitting, Stage 2)**.**  *Let $T_1$ be the time step in Lemma C.12. For any time step $T_2 = \Theta\left(\frac{\exp(n^{-1}\mathrm{SNR}^{-2})}{\alpha n^{-1}\sigma_\epsilon^2 \|\boldsymbol{\nu}\|_2\|\boldsymbol{\mu}\|_2 d^2 \max\{\sigma_w^2, \sigma_p^2\}}\right)$, on a good run, the following propositions hold for all time step $\tau \in [T_1, T_2]$:*

$A(\tau)$**:**  *There exists a constant $C_1 > 1$ such that*

$$(1 - 1/C_1) \cdot \sigma_p \sqrt{d} \leq \|\mathbf{p}(\tau)\|_2 \leq (1 + 1/C_1) \cdot \sigma_p \sqrt{d}.$$

$B(\tau)$**:**  *There exists a constant $C_2 > 1$ such that*

$$(1 - 1/C_2) \cdot \sigma_w\|\boldsymbol{\mu}\|_2\sqrt{d} \leq \|\mathbf{W}(\tau)\boldsymbol{\mu}_{+1}\|_2 \leq (1 + 1/C_2) \cdot \sigma_w\|\boldsymbol{\mu}\|_2\sqrt{d},$$
$$(1 - 1/C_2) \cdot \sigma_w\|\boldsymbol{\mu}\|_2\sqrt{d} \leq \|\mathbf{W}(\tau)\boldsymbol{\mu}_{-1}\|_2 \leq (1 + 1/C_2) \cdot \sigma_w\|\boldsymbol{\mu}\|_2\sqrt{d},$$
$$(1 - 1/C_2) \cdot \sigma_w\sigma_\epsilon d \leq \|\mathbf{W}(\tau)\boldsymbol{\epsilon}_t^{(i)}\|_2 \leq (1 + 1/C_2) \cdot \sigma_w\sigma_\epsilon d,$$

*for all $i \in [n]$ and $t \in [T]$.*

$$|\langle \mathbf{W}(\tau)\boldsymbol{\mu}_{+1}, \mathbf{W}(\tau)\boldsymbol{\mu}_{-1} \rangle| < O(\sigma_w^2\|\boldsymbol{\mu}\|_2^2\sqrt{d}\log(Tn/\delta)),$$
$$|\langle \mathbf{W}(\tau)\boldsymbol{\mu}_{+1}, \mathbf{W}(\tau)\boldsymbol{\epsilon}_t^{(i)} \rangle| < O(\sigma_w^2\sigma_\epsilon\|\boldsymbol{\mu}\|_2 d\log(Tn/\delta)),$$
$$|\langle \mathbf{W}(\tau)\boldsymbol{\mu}_{-1}, \mathbf{W}(\tau)\boldsymbol{\epsilon}_t^{(i)} \rangle| < O(\sigma_w^2\sigma_\epsilon\|\boldsymbol{\mu}\|_2 d\log(Tn/\delta)),$$
$$|\langle \mathbf{W}(\tau)\boldsymbol{\epsilon}_t^{(i)}, \mathbf{W}(\tau)\boldsymbol{\epsilon}_u^{(j)} \rangle| < O(\sigma_w^2\sigma_\epsilon^2 d^{3/2}\log(Tn/\delta)),$$

*for all $i, j \in [n]$ and $t, u \in [T]$ such that $(i, t) \neq (j, u)$.*

$C(\tau)$**:** *Let $g$ be a monotonically increasing function $g(x) = 2x + 2\sinh\left(x - \log T\right)$. Then, there exist constants $c_3, c_4 > 0$ with $c_3 < c_4$ such that*

$$g\left(\Lambda_{i,t}(\tau)\right) \geq g\left(\Lambda_{i,t}(0)\right) + \tau \cdot c_3 \alpha n^{-1} \sigma_\epsilon^2 \|\boldsymbol{\nu}\|_2 \|\boldsymbol{\mu}\|_2 d^2 \max\{\sigma_w^2, \sigma_p^2\},$$
$$g\left(\Lambda_{i,t}(\tau)\right) \leq g\left(\Lambda_{i,t}(0)\right) + \tau \cdot c_4 \alpha n^{-1} \sigma_\epsilon^2 \|\boldsymbol{\nu}\|_2 \|\boldsymbol{\mu}\|_2 d^2 \max\{\sigma_w^2, \sigma_p^2\},$$

*for any clean data $i \in \mathcal{C}$ and $t \in [T] \setminus \{1\}$. Additionally, we have that for some constants $c_5 < c_6$,*

$$g\left(\Gamma_{j,1}(\tau) - \log \rho^{-1}\right) \geq g\left(\Gamma_{j,1}(0) - \log \rho^{-1}\right) + \tau \cdot \rho \cdot c_5 \alpha n^{-1} \sigma_\epsilon^2 \|\boldsymbol{\nu}\|_2 \|\boldsymbol{\mu}\|_2 d^2 \max\{\sigma_w^2, \sigma_p^2\},$$
$$g\left(\Gamma_{j,1}(\tau) - \log \rho^{-1}\right) \leq g\left(\Gamma_{j,1}(0) - \log \rho^{-1}\right) + \tau \cdot \rho \cdot c_6 \alpha n^{-1} \sigma_\epsilon^2 \|\boldsymbol{\nu}\|_2 \|\boldsymbol{\mu}\|_2 d^2 \max\{\sigma_w^2, \sigma_p^2\},$$

*for any noisy data $j \in \mathcal{N}$, and*

$$g\left(\Gamma_{j,t}(\tau)\right) \geq g\left(\Gamma_{j,t}(0)\right) + \tau \cdot \rho \cdot c_5 \alpha n^{-1} \sigma_\epsilon^2 \|\boldsymbol{\nu}\|_2 \|\boldsymbol{\mu}\|_2 d^2 \max\{\sigma_w^2, \sigma_p^2\},$$
$$g\left(\Gamma_{j,t}(\tau)\right) \leq g\left(\Gamma_{j,t}(0)\right) + \tau \cdot \rho \cdot c_6 \alpha n^{-1} \sigma_\epsilon^2 \|\boldsymbol{\nu}\|_2 \|\boldsymbol{\mu}\|_2 d^2 \max\{\sigma_w^2, \sigma_p^2\},$$

*for any noisy data $j \in \mathcal{N}$ and $t \in [T] \setminus \{1, 2\}$.*

$D(\tau)$**:**

$$s_1^{(i)}(\tau) \geq \Theta(1), \ s_2^{(j)}(\tau) \geq \Theta(1),$$

*for all clean data $i \in \mathcal{C}$ and noisy data $j \in \mathcal{N}$.*

$E(\tau)$**:** *There exists a constant $c_7 > 1$ such that*

$$\exp\left(\Lambda_{i,t}(\tau) - \Lambda_{j,u}(\tau)\right) < c_7, \ \exp\left(\Gamma_{k,v}(\tau) - \Gamma_{l,w}(\tau)\right) < c_7,$$
$$\exp\left(\Gamma_{k,1}(\tau) - \Gamma_{l,w}(\tau)\right) < \rho^{-1} c_7, \ \exp\left(\Gamma_{k,v}(\tau) - \Gamma_{l,1}(\tau)\right) < \rho c_7,$$
$$\exp\left(\Lambda_{i,t}(\tau) - \Gamma_{k,v}(\tau)\right) < \rho^{-1} c_7, \ \exp\left(\Gamma_{k,v}(\tau) - \Lambda_{i,t}(\tau)\right) < \rho c_7,$$
$$\exp\left(\Lambda_{i,t}(\tau) - \Gamma_{k,1}(\tau)\right) < c_7, \ \exp\left(\Gamma_{k,1}(\tau) - \Lambda_{i,t}(\tau)\right) < c_7,$$

*for all $i, j \in \mathcal{C}, k, l \in \mathcal{N}, t, u \in [T] \setminus \{1\}$ and $v, w \in [T] \setminus \{1, 2\}$.*

$F(\tau)$**:** *There exists a constant $c_8 > 1$ such that*

$$\max_{i,j \in [n]} \frac{s_1^{(i)}(\tau)(1 - s_1^{(i)}(\tau))}{s_1^{(j)}(\tau)(1 - s_1^{(j)}(\tau))} < c_8, \ \max_{k,l \in \mathcal{N}} \frac{s_2^{(k)}(\tau)(1 - s_2^{(k)}(\tau))}{s_2^{(l)}(\tau)(1 - s_2^{(l)}(\tau))} < c_8,$$
$$\max_{i \in \mathcal{C}, k \in \mathcal{N}} \frac{s_1^{(i)}(\tau)(1 - s_1^{(i)}(\tau))}{s_2^{(k)}(\tau)(1 - s_2^{(k)}(\tau))} < \rho c_8, \ \max_{i \in \mathcal{C}, k \in \mathcal{N}} \frac{s_2^{(k)}(\tau)(1 - s_2^{(k)}(\tau))}{s_1^{(i)}(\tau)(1 - s_1^{(i)}(\tau))} < \rho^{-1} c_8.$$

*Additionally, there exist constants $c_9, c_{10} > 1$ such that*

$$\max_{t,u \in [T] \setminus \{1\}} \frac{s_t^{(i)}(\tau)}{s_u^{(i)}(\tau)} < c_9, \ \max_{v,w \in [T] \setminus \{1,2\}} \frac{s_v^{(j)}(\tau)}{s_w^{(j)}(\tau)} < c_9, \ \max_{v \in [T] \setminus \{2\}} \frac{s_1^{(j)}(\tau)}{s_v^{(j)}(\tau)} < \rho c_9, \ \max_{v \in [T] \setminus \{2\}} \frac{s_v^{(j)}(\tau)}{s_1^{(j)}(\tau)} < \rho^{-1} c_9,$$
$$\frac{s_t^{(i)}(\tau)(1 - s_t^{(i)}(\tau))}{s_1^{(i)}(\tau)(1 - s_1^{(i)}(\tau))} < c_{10}, \ \frac{s_v^{(j)}(\tau)(1 - s_v^{(j)}(\tau))}{s_2^{(j)}(\tau)(1 - s_2^{(j)}(\tau))} < c_{10},$$

*for any clean data $i \in \mathcal{C}$, noisy data $j \in \mathcal{N}$, $t \in [T] \setminus \{1\}$, and $v \in [T] \setminus \{2\}$.*

$G(\tau)$**:**

$$|\lambda_{+1}(\tau)| = O(1), \ |\lambda_{-1}(\tau)| = O(1), \ |\rho_{i,t}(\tau)| = O(n^{-1}\mathrm{SNR}^{-2}),$$

*for any $i \in [n]$ and $t \in [T]$.*

We will prove the above propositions by induction argument for $\tau \in [T_1, T_2]$.

**Base case.** The propositions $A(T_1)$ and $B(T_1)$ hold from the results in Lemma C.12.

We show that $C(T_1)$ holds. It is obvious from Lemma C.12 that the inequalities for clean data $i \in \mathcal{C}$ hold. For noisy data, since $T_1$ is given by $\Theta\left(\frac{\rho^{-1}}{\alpha n^{-1}\sigma_\epsilon^2\|\boldsymbol{\nu}\|_2\|\boldsymbol{\mu}\|_2 d^2 \max\{\sigma_w^2, \sigma_p^2\}}\right)$, it is sufficient to show $g(\Gamma_{j,1}(T_1) - \log\rho^{-1}) = \Theta(1)$ and $g(\Gamma_{j,t}(T_1)) = \Theta(1)$ for $t \in [T] \setminus \{1, 2\}$. From the results in Lemma C.12, we have $g(\Lambda_{j,2}(T_1)) = -\Theta(\rho^{-1})$. Thus, we have $\Gamma_{j,1}(T_1) = -\Lambda_{j,2}(T_1) \in (1 \pm o(1))\log\rho^{-1}$, leading to $g(\Gamma_{j,1}(T_1) - \log\rho^{-1}) = \Theta(1)$. Additionally, from $E(\tau)$ in Lemma C.12, we have

$$\Gamma_{j,t}(T_1) = \left(\mathbf{x}_2^{(j)} - \mathbf{x}_t^{(j)}\right)^\top \mathbf{W}(T_1)^\top \mathbf{p}(T_1) = \Lambda_{j,t}(T_1) - \Lambda_{j,2}(T_1) < \log c_7, \tag{272}$$

where $c_7$ is a constant in Lemma C.12. Thus, we have $g(\Lambda_{j,t}(T_1)) = \Theta(1)$, which is the desired result.

Regarding $G(T_1)$, the result in Lemma C.12 and the current SNR condition $n^{-1}\text{SNR}^{-2} = \omega(1)$ conclude the base case.

**Proof of** $\wedge_{\tau' \leq \tau}\left(A(\tau') \wedge B(\tau') \wedge C(\tau') \wedge D(\tau') \wedge E(\tau') \wedge F(\tau')\right) \Rightarrow C(\tau + 1)$**.** We only show the case of $Y^{(i)} = 1$, i.e., $i \in \mathcal{C}_+ \cup \mathcal{N}_-$. The same argument can be applied to the case of $i \in \mathcal{C}_- \cup \mathcal{N}_+$. The conditions of Lemmas D.10 and D.11 are satisfied from $A(\tau) \wedge B(\tau) \wedge D(\tau) \wedge F(\tau)$. For any clean data $i \in \mathcal{C}$, since we have the same bounds for signal and noise updates as in Lemma C.12, the same argument leads to

$$\Lambda_{i,t}(\tau + 1) - \Lambda_{i,t}(\tau) \geq s_1^{(i)}(\tau)(1 - s_1^{(i)}(\tau)) \cdot c_1'\alpha n^{-1}\sigma_\epsilon^2\|\boldsymbol{\nu}\|_2\|\boldsymbol{\mu}\|_2 d^2 \max\{\sigma_w^2, \sigma_p^2\}, \tag{273}$$

$$\Lambda_{i,t}(\tau + 1) - \Lambda_{i,t}(\tau) \leq s_1^{(i)}(\tau)(1 - s_1^{(i)}(\tau)) \cdot c_2'\alpha n^{-1}\sigma_\epsilon^2\|\boldsymbol{\nu}\|_2\|\boldsymbol{\mu}\|_2 d^2 \max\{\sigma_w^2, \sigma_p^2\}, \tag{274}$$

for any $t \in [T] \setminus \{1\}$. Applying the same integral analysis, we obtain the desired inequalities for $g(\Lambda_{i,t}(\tau))$.

For noisy data $j \in \mathcal{N}_-$, from these lemmas and the definition of $\Gamma_{j,t}$, we have that for any $t \in \mathcal{I}^{(j)}$ and $i \in \mathcal{C}_+$,

$$\Gamma_{j,t}(\tau + 1) - \Gamma_{j,t}(\tau) = \rho\left(\lambda_{+1}(\tau + 1) - \lambda_{+1}(\tau)\right) + (\rho_{j,2}(\tau) - \rho_{j,2}(\tau)) - (\rho_{j,t}(\tau + 1) - \rho_{j,t}(\tau)) \tag{275}$$

$$\geq s_1^{(i)}(\tau)(1 - s_1^{(i)}(\tau)) \cdot c_3'\alpha\|\boldsymbol{\nu}\|_2\|\boldsymbol{\mu}\|_2^3 d \max\{\sigma_w^2, \sigma_p^2\} \tag{276}$$

$$+ s_2^{(j)}(\tau)(1 - s_2^{(j)}(\tau)) \cdot c_4'\rho\alpha n^{-1}\sigma_\epsilon^2\|\boldsymbol{\nu}\|_2\|\boldsymbol{\mu}\|_2 d^2 \max\{\sigma_w^2, \sigma_p^2\}$$

$$+ s_2^{(j)}(\tau)s_t^{(j)}(\tau) \cdot c_4'\rho\alpha n^{-1}\sigma_\epsilon^2\|\boldsymbol{\nu}\|_2\|\boldsymbol{\mu}\|_2 d^2 \max\{\sigma_w^2, \sigma_p^2\} \tag{277}$$

$$\geq s_2^{(j)}(\tau)(1 - s_2^{(j)}(\tau)) \cdot c_4'\rho\alpha n^{-1}\sigma_\epsilon^2\|\boldsymbol{\nu}\|_2\|\boldsymbol{\mu}\|_2 d^2 \max\{\sigma_w^2, \sigma_p^2\}. \tag{278}$$

In the last line, $s_1^{(i)}(\tau)(1 - s_1^{(i)}(\tau)) \lesssim \rho s_2^{(j)}(\tau)(1 - s_2^{(j)}(\tau))$, which is derived from $F(\tau)$, and the current SNR condition $\text{SNR}^2 = o(n^{-1})$ allow us to ignore this signal update. The same result holds for the relevant token $t \in \mathcal{R} = \{1\}$ and weakly relevant tokens $t \in \mathcal{W}^{(j)}$, as the signal update can be neglected by the same argument. In the same manner, the upper bound of $\Gamma_{j,t}(\tau + 1) - \Gamma_{j,t}(\tau)$ is derived. Using the evaluations on $s_2^{(j)}(\tau)(1 - s_2^{(j)}(\tau))$ in Lemma D.2 and applying the same integral analysis as $\Lambda_{i,t}$, which appeared in the proof of Lemma C.11, we conclude the desired results of $\Gamma_{j,t}$.

**Proof of** $C(\tau) \Rightarrow D(\tau)$**.** For any clean data $i \in \mathcal{C}$, we have

$$s_1^{(i)}(\tau) = \frac{1}{1 + \sum_{t=2}^T \exp\left(-\Lambda_{i,t}(\tau)\right)} > \frac{1}{1 + \sum_{t=2}^T \exp\left(-\Lambda_{i,t}(T_1)\right)} = s_1^{(i)}(0) \geq \Theta(1), \tag{279}$$

where the inequality follows from $C(\tau)$, which states that $\Lambda_{i,t}(\tau)$ is monotonically increasing. Similarly, since $\Gamma_{j,t}(\tau)$ for $j \in \mathcal{N}$ is monotonically increasing, we have $s_2^{(j)}(\tau) > s_2^{(j)}(T_1) \geq \Theta(1)$.

**Proof of** $C(\tau) \Rightarrow E(\tau)$**.** By the same argument as in Lemma C.11, and using the inequalities of $C(\tau)$, we can show that $\exp\left(\Lambda_{i,t}(\tau) - \Lambda_{j,u}(\tau)\right)$ is bounded by a constant for any $i, j \in \mathcal{C}$ and $t, u \in [T] \setminus \{1\}$. Similarly to clean data, it can be shown that $\exp\left(\Gamma_{k,v}(\tau) - \Gamma_{l,w}(\tau)\right)$ is bounded by a constant for any $k, l \in \mathcal{N}$ and $v, w \in [T] \setminus \{1, 2\}$ as well. At this time, since we have the result for $\Gamma_{j,1}(\tau) - \log\rho^{-1}$ instead of $\Gamma_{j,1}(\tau)$, the second line in the statement follows by factoring $\log\rho^{-1}$ out. Moreover, from $C(\tau)$, the growth order of $g(\Lambda)$ and $g(\Gamma)$ differs by $\rho$, and this ratio also appears in $\exp$ following the same proof as Lemma C.11. The last line in the statement follows similarly by factoring $\log\rho$ out for $\Gamma_{j,1}$.

**Proof of $E(\tau) \Rightarrow F(\tau)$.** We first consider the ratio of $s_1^{(i)}(\tau)(1 - s_1^{(i)}(\tau))$ across all pairs $i, j \in [n]$. Using the same argument as Lemma C.11, we can show that this ratio is bounded by a constant among clean data and noisy data, respectively. For the ratio between clean data and noisy data, from $E(\tau)$, we have

$$\exp\left(\Lambda_{i,t}(\tau) + \Lambda_{k,2}(\tau)\right) = \exp\left(\Lambda_{i,t}(\tau) - \Gamma_{k,1}(\tau)\right) < c_7, \tag{280}$$

$$\exp\left(-\Lambda_{k,2}(\tau) - \Lambda_{i,t}(\tau)\right) = \exp\left(\Gamma_{k,1}(\tau) - \Lambda_{i,t}(\tau)\right) < c_7, \tag{281}$$

for any $i \in \mathcal{C}, t \in [T] \setminus \{1\}$, and $k \in \mathcal{N}$. Additionally, from $E(\tau)$, we have that for any $v \in [T] \setminus \{1, 2\}$,

$$\Lambda_{k,v}(\tau) = \Gamma_{k,v}(\tau) - \Gamma_{k,1}(\tau) < \log(\rho c_7) < 0, \tag{282}$$

$$\Lambda_{k,2}(\tau) < \Gamma_{k,v}(\tau) + \Lambda_{k,2}(\tau) = \Lambda_{k,v}(\tau) < 0, \tag{283}$$

where the first line follows from the parameter assumption $\rho < 1/C$ for a sufficiently large constant $C$, and the second line is by $\Gamma_{k,v}(\tau) > 0$. Thus, we have $\Gamma_{k,t} < 0$ for any $t \in [T] \setminus \{1\}$. Combining this with Equations (280) and (281), and proceeding similarly to Equation (252), we obtain that there exists a constant $c_1'$ such that

$$\frac{s_1^{(k)}(\tau)(1 - s_1^{(k)}(\tau))}{s_1^{(i)}(\tau)(1 - s_1^{(i)}(\tau))} < \frac{2 + \sum_{t=2}^{T} \exp\left(-\Lambda_{i,t}(\tau)\right)}{1 + \sum_{t=2}^{T} \exp\left(-\Lambda_{k,t}(\tau)\right)} + \frac{1}{\left(1 + \sum_{t=2}^{T} \exp\left(-\Lambda_{k,t}(\tau)\right)\right) \cdot \sum_{t=2}^{T} \exp\left(-\Lambda_{i,t}(\tau)\right)} \tag{284}$$

$$< \frac{2 + (T-1)o(1)}{1 + (T-1)o(1)} + \exp\left(\Lambda_{i,u}(\tau) + \Lambda_{k,2}(\tau)\right) < c_1', \tag{285}$$

for any $u \in [T] \setminus \{1\}$. Similarly, by the same argument as in Equation (255), we have that there exists a constant $c_2'$ such that

$$\frac{s_1^{(i)}(\tau)(1 - s_1^{(i)}(\tau))}{s_1^{(k)}(\tau)(1 - s_1^{(k)}(\tau))} < \frac{\sum_{t=2}^{T} \exp\left(-\Lambda_{i,t}(\tau)\right)}{1 + \sum_{t=2}^{T} \exp\left(-\Lambda_{i,t}(\tau)\right)} \cdot \left(2 + \sum_{t=2}^{T} \exp\left(-\Lambda_{k,t}(\tau)\right) + \frac{1}{\sum_{t=2}^{T} \exp\left(-\Lambda_{k,t}(\tau)\right)}\right) \tag{286}$$

$$< 2 + T \cdot \sum_{t=2}^{T} \exp\left(-\Lambda_{i,t}(\tau) - \Lambda_{k,2}(\tau)\right) + \frac{(T-1)o(1)}{(T-1)o(1)} < c_2', \tag{287}$$

where the last line follows from Equation (283) and the parameter assumption $T = \Theta(1)$. Equations (285) and (287) completes the proof. Similarly, the ratio of $s_2^{(k)}(\tau)(1 - s_2^{(k)}(\tau))$ across $k \in \mathcal{N}$ is shown to be bounded by a constant.

We now consider the ratio between $s_1^{(i)}(\tau)(1 - s_1^{(i)}(\tau))$ and $s_2^{(k)}(\tau)(1 - s_2^{(k)}(\tau))$ for $i \in \mathcal{C}$ and $k \in \mathcal{N}$. For any $i \in \mathcal{C}$ and $k \in \mathcal{N}$, we have

$$s_1^{(i)}(\tau)(1 - s_1^{(i)}(\tau)) = \frac{\sum_{t=2}^{T} \exp\left(-\Lambda_{i,t}(\tau)\right)}{\left(1 + \sum_{t=2}^{T} \exp\left(-\Lambda_{i,t}(\tau)\right)\right)^2}, \quad s_2^{(k)}(\tau)(1 - s_2^{(k)}(\tau)) = \frac{\sum_{t \in [T] \setminus \{2\}} \exp\left(-\Gamma_{k,t}(\tau)\right)}{\left(1 + \sum_{t \in [T] \setminus \{2\}} \exp\left(-\Gamma_{k,t}(\tau)\right)\right)^2}. \tag{288}$$

Thus, there exists a constant $c_3' > 0$ such that

$$\frac{s_1^{(i)}(\tau)(1 - s_1^{(i)}(\tau))}{s_2^{(k)}(\tau)(1 - s_2^{(k)}(\tau))} = \frac{\sum_{t=2}^{T} \exp\left(-\Lambda_{i,t}(\tau)\right)}{\exp\left(-\Gamma_{k,1}(\tau)\right) + \sum_{t=3}^{T} \exp\left(-\Gamma_{k,t}(\tau)\right)} \cdot \left(\frac{1 + \exp\left(-\Gamma_{k,1}(\tau)\right) + \sum_{t=3}^{T} \exp\left(-\Gamma_{k,t}(\tau)\right)}{1 + \sum_{t=2}^{T} \exp\left(-\Lambda_{i,t}(\tau)\right)}\right)^2 \tag{289}$$

$$< \max_{t \in [T] \setminus \{1\}, u \in [T] \setminus \{1, 2\}} \left\{\exp\left(-\Lambda_{i,t}(\tau) + \Gamma_{k,u}(\tau)\right)\right\} \cdot (1 + (T-1)o(1))^2 < \rho c_3', \tag{290}$$

where in the first inequality, we use the mediant inequality excluding $\Lambda_{k,1}(\tau)$, along with the bounds $\Lambda_{i,t}(\tau), \Gamma_{k,1}(\tau), \Gamma_{k,t}(\tau) > -o(1)$. The last inequality follows from $E(\tau)$ and the parameter assumption $T = \Theta(1)$. Similarly, the desired result for the reciprocal is obtained by $E(\tau)$.

Finally, from $E(\tau)$, we have

$$\max_{t,u \in [T] \setminus \{1\}} \frac{s_t^{(i)}(\tau)}{s_u^{(i)}(\tau)} = \max_{t,u \in [T] \setminus \{1\}} \frac{\exp\left(\mathbf{x}_t^{(i)\top} \mathbf{W}(\tau)^\top \mathbf{p}(\tau)\right)}{\exp\left(\mathbf{x}_u^{(i)\top} \mathbf{W}(\tau)^\top \mathbf{p}(\tau)\right)} = \max_{t,u \in [T] \setminus \{1\}} \frac{\exp\left(-\Lambda_{i,t}(\tau)\right)}{\exp\left(-\Lambda_{i,u}(\tau)\right)} \leq c_7, \tag{291}$$

for $i \in \mathcal{C}$. Similarly, for noisy data $j \in \mathcal{N}$, the ratio of $s_v^{(j)}(\tau)$ across $v \in [T] \setminus \{2\}$ can be evaluated using the result of $E(\tau)$. The last two inequalities are derived by combining the fact that $\Lambda_{i,t}$ and $\Gamma_{j,t}$ are monotonically increasing from $C(\tau)$, and Lemma D.3.

**Proof of** $\wedge_{\tau' \leq \tau} (A(\tau') \wedge B(\tau') \wedge C(\tau') \wedge D(\tau') \wedge F(\tau')) \Rightarrow G(\tau + 1)$**.** The conditions of Lemma D.10 are satisfied from $A(\tau) \wedge B(\tau) \wedge D(\tau) \wedge F(\tau)$. Combining Lemmas D.8 and D.10, we have

$$\lambda_{+1}(\tau + 1) - \lambda_{+1}(0) \leq \sum_{\tau'=0}^{\tau} s_1^{(i)}(\tau')(1 - s_1^{(i)}(\tau')) \cdot c'\alpha \|\boldsymbol{\nu}\|_2 \|\boldsymbol{\mu}\|_2^3 d \max\{\sigma_w^2, \sigma_p^2\}, \tag{292}$$

for any clean data $i \in \mathcal{C}$. By using Lemma D.5, which is derived from $\wedge_{\tau' \leq \tau} C(\tau')$, and the definition of $T_2$, we have

$$\sum_{\tau'=0}^{\tau} s_1^{(i)}(\tau')(1 - s_1^{(i)}(\tau')) \tag{293}$$

$$\lesssim \max\left\{ \frac{1}{\alpha n^{-1}\sigma_\epsilon^2 \|\boldsymbol{\nu}\|_2 \|\boldsymbol{\mu}\|_2 d^2 \max\{\sigma_w^2, \sigma_p^2\}}, \frac{\log\left(\tau \cdot \alpha n^{-1}\sigma_\epsilon^2 \|\boldsymbol{\nu}\|_2 \|\boldsymbol{\mu}\|_2 d^2 \max\{\sigma_w^2, \sigma_p^2\}\right)}{\alpha n^{-1}\sigma_\epsilon^2 \|\boldsymbol{\nu}\|_2 \|\boldsymbol{\mu}\|_2 d^2 \max\{\sigma_w^2, \sigma_p^2\}} \right\} \tag{294}$$

$$\leq \max\left\{ \frac{1}{\alpha n^{-1}\sigma_\epsilon^2 \|\boldsymbol{\nu}\|_2 \|\boldsymbol{\mu}\|_2 d^2 \max\{\sigma_w^2, \sigma_p^2\}}, \frac{\log\left(T_2 \cdot \alpha n^{-1}\sigma_\epsilon^2 \|\boldsymbol{\nu}\|_2 \|\boldsymbol{\mu}\|_2 d^2 \max\{\sigma_w^2, \sigma_p^2\}\right)}{\alpha n^{-1}\sigma_\epsilon^2 \|\boldsymbol{\nu}\|_2 \|\boldsymbol{\mu}\|_2 d^2 \max\{\sigma_w^2, \sigma_p^2\}} \right\} \tag{295}$$

$$\lesssim \frac{n^{-1}\mathrm{SNR}^{-2}}{\alpha n^{-1}\sigma_\epsilon^2 \|\boldsymbol{\nu}\|_2 \|\boldsymbol{\mu}\|_2 d^2 \max\{\sigma_w^2, \sigma_p^2\}}. \tag{296}$$

Substituting this to Equation (292), we have

$$\lambda_{+1}(\tau + 1) - \lambda_{+1}(0) \lesssim n^{-1}\mathrm{SNR}^{-2} \cdot n\mathrm{SNR}^2 = O(1). \tag{297}$$

Since $|\lambda_{+1}(0)| = o(1)$ from Lemma C.8, Equation (297) leads to $\lambda_{+1}(\tau + 1) \leq O(1)$. In the same way, we have $\lambda_{-1}(\tau + 1) \leq O(1)$.

Next, we move on to the analysis of noise memorization terms. For clean data $i \in \mathcal{C}$, from Lemmas D.9 and D.11, similarly we have

$$|\rho_{i,1}(\tau + 1) - \rho_{i,1}(0)| \leq \sum_{\tau'=0}^{\tau} s_1^{(i)}(\tau')(1 - s_1^{(i)}(\tau')) \cdot c'\alpha n^{-1}\sigma_\epsilon^2 \|\boldsymbol{\nu}\|_2 \|\boldsymbol{\mu}\|_2 d^2 \max\{\sigma_w^2, \sigma_p^2\} = O(n^{-1}\mathrm{SNR}^{-2}). \tag{298}$$

The same upper bound is obtained for $\rho_{i,t}$ with $t \in [T] \setminus \{1\}$. For noisy data $j \in \mathcal{N}$, the upper bound for $\rho_{j,1}$ is derived in the same manner, based on Lemmas D.9 and D.11. For other tokens, we have

$$|\rho_{j,2}(\tau + 1) - \rho_{j,2}(0)| \leq |\rho_{j,2}(T_1) - \rho_{j,2}(0)| + |\rho_{j,2}(\tau + 1) - \rho_{j,2}(T_1)| \tag{299}$$

$$\leq O(\log \rho^{-1}) + \sum_{\tau'=T_1}^{\tau} s_2^{(j)}(\tau')(1 - s_2^{(j)}(\tau')) \cdot c'\rho\alpha n^{-1}\sigma_\epsilon^2 \|\boldsymbol{\nu}\|_2 \|\boldsymbol{\mu}\|_2 d^2 \max\{\sigma_w^2, \sigma_p^2\} \tag{300}$$

$$\lesssim O(\log \rho^{-1}) + \log\left(T_2 \cdot \rho\alpha n^{-1}\sigma_\epsilon^2 \|\boldsymbol{\nu}\|_2 \|\boldsymbol{\mu}\|_2 d^2 \max\{\sigma_w^2, \sigma_p^2\}\right) \tag{301}$$

$$\lesssim O(\log \rho^{-1}) + \log \rho + n^{-1}\mathrm{SNR}^{-2} \tag{302}$$

$$\lesssim n^{-1}\mathrm{SNR}^{-2}, \tag{303}$$

which leads to the same upper bound as in clean data. The upper bounds for $t \in [T] \setminus \{1, 2\}$ are derived in the same manner.

**Proof of** $\wedge_{\tau' \leq \tau} (B(\tau') \wedge C(\tau') \wedge F(\tau') \wedge G(\tau')) \Rightarrow A(\tau + 1)$**.** It follows from Lemma C.5 that

$$\|\mathbf{p}(\tau + 1)\|_2^2 - \|\mathbf{p}(T_1)\|_2^2 = \frac{2\alpha}{n} \sum_{\tau'=T_1}^{\tau} \sum_{i=1}^{n} (-\ell_i'(\tau')) \cdot Y^{(i)} \cdot I_i^p(\tau') + \alpha^2 \sum_{\tau'=T_1}^{\tau} \|\nabla_{\mathbf{p}} \widehat{\mathcal{L}}(\tau')\|_2^2. \tag{304}$$

By definition, we have

$$I_i^p(\tau') = \sum_{t=1}^{T} s_t^{(i)}(\tau') \left( \gamma_t^{(i)} - \sum_{u=1}^{T} s_u^{(i)}(\tau')\gamma_u^{(i)} \right) \langle \mathbf{W}(\tau')\mathbf{x}_t^{(i)}, \mathbf{p}(\tau') \rangle \tag{305}$$

$$\leq \sum_{t=1}^{T} \left| s_t^{(i)}(\tau') \left( \gamma_t^{(i)} - \sum_{u=1}^{T} s_u^{(i)}(\tau')\gamma_u^{(i)} \right) \right| \cdot \max \left\{ |\lambda_{+1}(\tau')|, |\lambda_{-1}(\tau')|, |\rho_{i,t}(\tau')| \right\}. \tag{306}$$

For the clean data $i \in \mathcal{C}$, we have the following upper bound through the same argument as in Equation (267):

$$\sum_{\tau'=T_1}^{\tau} \sum_{t=1}^{T} \left| s_t^{(i)}(\tau') \left( \gamma_t^{(i)} - \sum_{u=1}^{T} s_u^{(i)}(\tau')\gamma_u^{(i)} \right) \right| \lesssim s_1^{(i)}(\tau')(1 - s_1^{(i)}(\tau')) \cdot \max_{i \in [n], t \in [T]} \{\gamma_t^{(i)}\} \tag{307}$$

$$\lesssim \frac{\log \left( T_2 \cdot \alpha n^{-1}\sigma_\epsilon^2 \|\boldsymbol{\nu}\|_2 \|\boldsymbol{\mu}\|_2 d^2 \max\{\sigma_w^2, \sigma_p^2\} \right)}{\alpha n^{-1}\sigma_\epsilon^2 \|\boldsymbol{\nu}\|_2 \|\boldsymbol{\mu}\|_2 d^2 \max\{\sigma_w^2, \sigma_p^2\}} \cdot \|\boldsymbol{\nu}\|_2 \|\boldsymbol{\mu}\|_2 \tag{308}$$

$$\lesssim \frac{n^{-1}\mathrm{SNR}^{-2}}{\alpha n^{-1}\sigma_\epsilon^2 d^2 \max\{\sigma_w^2, \sigma_p^2\}}. \tag{309}$$

For the noisy data $j \in \mathcal{N}$, we evaluate separately for the token score of the relevant token $\gamma_1^{(i)}$. We have

$$\sum_{t=1}^{T} \left| s_t^{(j)}(\tau') \left( \gamma_t^{(j)} - \sum_{u=1}^{T} s_u^{(j)}(\tau')\gamma_u^{(j)} \right) \right|$$

$$\leq \left| s_1^{(j)}(\tau') \left( \gamma_1^{(j)} - \sum_{u=1}^{T} s_u^{(j)}(\tau')\gamma_u^{(j)} \right) \right| + \sum_{t=2}^{T} \left| s_t^{(j)}(\tau')s_1^{(j)}(\tau')\gamma_1^{(j)} \right| + \sum_{t=2}^{T} \left| s_t^{(j)}(\tau') \left( \gamma_t^{(j)} - \sum_{u=2}^{T} s_u^{(j)}(\tau')\gamma_u^{(j)} \right) \right| \tag{310}$$

$$\lesssim 3s_1^{(j)}(\tau')|\gamma_1^{(j)}| + (T-1) \cdot s_2^{(j)}(\tau')(1 - s_2^{(j)}(\tau')) \cdot \max_{i \in [n], t \in [T]\setminus\{1\}} \{|\gamma_t^{(i)}|\} \tag{311}$$

$$\lesssim s_2^{(j)}(\tau')(1 - s_2^{(j)}(\tau')) \cdot \left( \rho|\gamma_1^{(j)}| + \max_{i \in [n], t \in [T]\setminus\{1\}} \{|\gamma_t^{(i)}|\} \right), \tag{312}$$

where the last two inequalities follow from the ratio of softmax terms in $F(\tau')$, together with the same manipulation as in Equation (152). By summing up both sides and applying Lemmas B.12 and D.5, we have

$$\sum_{\tau'=T_1}^{\tau} \sum_{t=1}^{T} \left| s_t^{(i)}(\tau') \left( \gamma_t^{(i)} - \sum_{u=1}^{T} s_u^{(i)}(\tau')\gamma_u^{(i)} \right) \right| \lesssim \frac{\log \left( T_2 \cdot \rho \cdot \alpha n^{-1}\sigma_\epsilon^2 \|\boldsymbol{\nu}\|_2 \|\boldsymbol{\mu}\|_2 d^2 \max\{\sigma_w^2, \sigma_p^2\} \right)}{\rho \cdot \alpha n^{-1}\sigma_\epsilon^2 \|\boldsymbol{\nu}\|_2 \|\boldsymbol{\mu}\|_2 d^2 \max\{\sigma_w^2, \sigma_p^2\}} \cdot \rho \|\boldsymbol{\nu}\|_2 \|\boldsymbol{\mu}\|_2 \tag{313}$$

$$\lesssim \frac{n^{-1}\mathrm{SNR}^{-2}}{\alpha n^{-1}\sigma_\epsilon^2 d^2 \max\{\sigma_w^2, \sigma_p^2\}}, \tag{314}$$

where the first line follows from $\max_{i \in [n], t \in [T]\setminus\{1\}} \{\gamma_t^{(i)}\} = O(\rho\|\boldsymbol{\nu}\|_2\|\boldsymbol{\mu}\|_2)$ from Lemma B.12. In the last line, we ignore the small order term $\log \rho$ in the numerator.

By substituting Equations (306), (309) and (314) into the first term of Equation (304), we obtain

$$\frac{2\alpha}{n} \sum_{\tau'=T_1}^{\tau} \sum_{i=1}^{n} (-\ell_i'(\tau')) \cdot Y^{(i)} \cdot I_i^p(\tau') \lesssim \alpha \cdot \frac{n^{-1}\mathrm{SNR}^{-2}}{\alpha n^{-1}\sigma_\epsilon^2 d^2 \max\{\sigma_w^2, \sigma_p^2\}} \cdot \max \left\{ |\lambda_{+1}(\tau')|, |\lambda_{-1}(\tau')|, |\rho_{i,t}(\tau')| \right\} \tag{315}$$

$$\lesssim \frac{\mathrm{SNR}^{-4}}{n\sigma_\epsilon^2 d^2 \max\{\sigma_w^2, \sigma_p^2\}} \tag{316}$$

$$= \frac{\sigma_\epsilon^2}{n\|\boldsymbol{\mu}\|_2^4 \max\{\sigma_w^2, \sigma_p^2\}}, \tag{317}$$

which follows from Lemma C.9 and $G(\tau')$. We confirm that Equation (317) becomes $o(\sigma_p^2 d)$. From the parameter assumptions in Section 3.5, we have

$$\sigma_p^2 \max\{\sigma_w^2, \sigma_p^2\} \cdot \sigma_\epsilon^{-2} n \|\boldsymbol{\mu}\|_2^4 d \gtrsim \min\left\{\|\boldsymbol{\mu}\|_2^2, \frac{\|\boldsymbol{\mu}\|_2^4}{\sigma_\epsilon^2 d}\right\} \cdot \frac{n}{\sigma_\epsilon^2 \log^4(Tn/\delta)} \tag{318}$$

$$\gtrsim \min\left\{\frac{C^2 n d^{3/4}}{\log^2(Tn/\delta)}, C^4 n \sqrt{d}\right\} \tag{319}$$

$$= \omega(1), \tag{320}$$

where the first inequality is by Assumption A8, and the second line follows from Assumption A2. Thus, Equation (317) becomes $o(\sigma_p^2 d)$.

Finally, we confirm that the second term in Equation (304) can be ignored. From the parameter assumption on the step size $\alpha$, it can be shown in the same way as Equation (161) in the proof of the not-overfitting case. Therefore, combining Equation (304) and the results in Lemma C.12, we have $\|\mathbf{p}(\tau+1)\|_2^2 \in (1 \pm 1/C_1)\sigma_p^2 d$, for some constant $C_1 > 1$, which concludes the proof.

**Proof of** $\wedge_{\tau' \leq \tau} (A(\tau') \wedge C(\tau') \wedge F(\tau') \wedge G(\tau')) \Rightarrow B(\tau+1)$**.** In the proof of the not-overfitting case, we have already shown that the second-order term with respect to the step size in Lemma C.6 can be ignored under the small step size assumption. Therefore, we write it as $O(\alpha^2)$ in the remainder of the proof. From Lemma C.6, we have

$$\|\mathbf{W}(\tau+1)\boldsymbol{\mu}_{+1}\|_2^2 - \|\mathbf{W}(T_1)\boldsymbol{\mu}_{+1}\|_2^2 = \sum_{\tau'=T_1}^{\tau} \left(\|\mathbf{W}(\tau'+1)\boldsymbol{\mu}_{+1}\|_2^2 - \|\mathbf{W}(\tau')\boldsymbol{\mu}_{+1}\|_2^2\right) \tag{321}$$

$$= \frac{2\alpha}{n} \sum_{\tau'=T_1}^{\tau} \sum_{i=1}^{n} (-\ell_i'(\tau')) \cdot Y^{(i)} \cdot I_{i,+}(\tau')\lambda_{+1}(\tau') + O(\alpha^2). \tag{322}$$

Recalling the definition of $I_{i,+}$ in Definition C.2, and applying the same technique used in Equations (309) and (314) in the proof of $A(\tau+1)$, we have

$$\sum_{\tau'=T_1}^{\tau} I_{i,+}(\tau') = \sum_{\tau'=T_1}^{\tau} \sum_{t=1}^{T} s_t^{(i)}(\tau') \left(\gamma_t^{(i)} - \sum_{u=1}^{T} s_u^{(i)}(\tau')\gamma_u^{(i)}\right) \langle \mathbf{x}_t^{(i)}, \boldsymbol{\mu}_{+1}\rangle \tag{323}$$

$$\lesssim \frac{n^{-1}\mathrm{SNR}^{-2}}{\alpha n^{-1}\sigma_\epsilon^2 d^2 \max\{\sigma_w^2, \sigma_p^2\}} \cdot \max_{i\in[n], t\in[T]}\{|\langle \mathbf{x}_t^{(i)}, \boldsymbol{\mu}_{+1}\rangle|\} \lesssim \frac{\mathrm{SNR}^{-2}\|\boldsymbol{\mu}\|_2^2}{\alpha \sigma_\epsilon^2 d^2 \max\{\sigma_w^2, \sigma_p^2\}}. \tag{324}$$

Since we have $\lambda_{+1}(\tau') = O(1)$ from $G(\tau')$, we have

$$\frac{2\alpha}{n} \sum_{\tau'=T_1}^{\tau} \sum_{i=1}^{n} (-\ell_i'(\tau')) \cdot Y^{(i)} \cdot I_{i,+}(\tau')\lambda_{+1}(\tau') \lesssim \frac{\mathrm{SNR}^{-2}\|\boldsymbol{\mu}\|_2^2}{\sigma_\epsilon^2 d^2 \max\{\sigma_w^2, \sigma_p^2\}} \tag{325}$$

$$= \frac{1}{d\max\{\sigma_w^2, \sigma_p^2\}} = o(\sigma_w^2 \|\boldsymbol{\mu}\|_2^2 d), \tag{326}$$

where the last inequality follows from the parameter assumptions. Combining this result with the base case $\tau = 0$, we have $\|\mathbf{W}(\tau+1)\boldsymbol{\mu}_{+1}\|_2^2 \in (1 \pm 1/C_2)\sigma_w^2\|\boldsymbol{\mu}\|_2^2 d$ for some constant $C_2 > 1$, which completes the proof. The same result holds for $\|\mathbf{W}(\tau+1)\boldsymbol{\mu}_{-1}\|_2$. Similarly, it follows from Lemma C.6 and $G(\tau')$ that

$$\|\mathbf{W}(\tau+1)\boldsymbol{\epsilon}_u^{(j)}\|_2^2 - \|\mathbf{W}(T_1)\boldsymbol{\epsilon}_u^{(j)}\|_2^2 = \frac{2\alpha}{n} \sum_{\tau'=T_1}^{\tau} \sum_{i=1}^{n} (-\ell_i'(\tau')) \cdot Y^{(i)} \cdot I_{i,j,u}(\tau')\rho_{j,u}(\tau') + O(\alpha^2) \tag{327}$$

$$\lesssim \alpha \cdot \frac{n^{-1}\mathrm{SNR}^{-2} \cdot n^{-1}\sigma_\epsilon^2 d}{\alpha n^{-1}\sigma_\epsilon^2 d^2 \max\{\sigma_w^2, \sigma_p^2\}} \cdot n^{-1}\mathrm{SNR}^{-2} + O(\alpha^2) \tag{328}$$

$$\lesssim \frac{\sigma_\epsilon^4 d}{n^2 \|\boldsymbol{\mu}\|_2^4 \max\{\sigma_w^2, \sigma_p^2\}} + O(\alpha^2) = o(\sigma_w^2 \sigma_\epsilon^2 d^2), \tag{329}$$

which follows from exactly the same analysis as in Equation (317).

Additionally, we will show the case of the inner products as well. From Lemma C.6 and the above discussion, we have

$$\langle \mathbf{W}(\tau+1)\boldsymbol{\mu}_{+1}, \mathbf{W}(\tau+1)\boldsymbol{\mu}_{-1}\rangle - \langle \mathbf{W}(T_1)\boldsymbol{\mu}_{+1}, \mathbf{W}(T_1)\boldsymbol{\mu}_{-1}\rangle$$

$$= \frac{\alpha}{n} \sum_{\tau'=T_1}^{\tau} \sum_{i=1}^{n} (-\ell_i'(\tau')) \cdot Y^{(i)} \cdot (I_{i,-}(\tau')\lambda_{+1}(\tau') + I_{i,+}(\tau')\lambda_{-1}(\tau')) + O(\alpha^2) \tag{330}$$

$$\lesssim \alpha \cdot \frac{n^{-1}\mathrm{SNR}^{-2} \cdot \|\boldsymbol{\mu}\|_2^2}{\alpha n^{-1}\sigma_\epsilon^2 d^2 \max\{\sigma_w^2, \sigma_p^2\}} \cdot O(1) + O(\alpha^2) \tag{331}$$

$$\lesssim \frac{1}{d \max\{\sigma_w^2, \sigma_p^2\}} + O(\alpha^2) = O(\sigma_w^2 \|\boldsymbol{\mu}\|_2^2 \sqrt{d}\log(Tn/\delta)), \tag{332}$$

where the last line follows from the parameter assumptions; specifically, we have

$$\sigma_w^2 \max\{\sigma_w^2, \sigma_p^2\} \cdot \|\boldsymbol{\mu}\|_2^2 d^{3/2} \log(Tn/\delta) \gtrsim \min\left\{\sqrt{d}, \frac{\|\boldsymbol{\mu}\|_2^2}{\sigma_\epsilon^2 \sqrt{d}}\right\} \cdot \frac{1}{\log^3(Tn/\delta)} \tag{333}$$

$$\gtrsim \min\left\{\frac{\sqrt{d}}{\log^3(Tn/\delta)}, \frac{C^2 d^{1/4}}{\log(Tn/\delta)}\right\} \tag{334}$$

$$= \Omega(1). \tag{335}$$

Similarly, we have

$$\langle \mathbf{W}(\tau+1)\boldsymbol{\mu}_{+1}, \mathbf{W}(\tau+1)\boldsymbol{\epsilon}_u^{(j)}\rangle - \langle \mathbf{W}(T_1)\boldsymbol{\mu}_{+1}, \mathbf{W}(T_1)\boldsymbol{\epsilon}_u^{(j)}\rangle$$

$$= \frac{\alpha}{n} \sum_{\tau'=T_1}^{\tau} \sum_{i=1}^{n} (-\ell_i'(\tau')) \cdot Y^{(i)} \cdot (I_{i,j,u}(\tau')\lambda_{+1}(\tau') + I_{i,+}(\tau')\rho_{j,u}(\tau')) + O(\alpha^2) \tag{336}$$

$$\lesssim \alpha \cdot \left(\frac{n^{-1}\mathrm{SNR}^{-2} \cdot n^{-1}\sigma_\epsilon^2 d}{\alpha n^{-1}\sigma_\epsilon^2 d^2 \max\{\sigma_w^2, \sigma_p^2\}} \cdot O(1) + \frac{n^{-1}\mathrm{SNR}^{-2} \cdot \|\boldsymbol{\mu}\|_2^2}{\alpha n^{-1}\sigma_\epsilon^2 d^2 \max\{\sigma_w^2, \sigma_p^2\}} \cdot n^{-1}\mathrm{SNR}^{-2}\right) + O(\alpha^2) \tag{337}$$

$$\lesssim \frac{\sigma_\epsilon^2}{n\|\boldsymbol{\mu}\|_2^2 \max\{\sigma_w^2, \sigma_p^2\}} + \frac{\sigma_\epsilon^2}{n\|\boldsymbol{\mu}\|_2^2 \max\{\sigma_w^2, \sigma_p^2\}} + O(\alpha^2) = O(\sigma_w^2 \sigma_\epsilon \|\boldsymbol{\mu}\|_2 d\log(Tn/\delta)), \tag{338}$$

which follows from the parameter assumptions; specifically, we have

$$\sigma_w^2 \max\{\sigma_w^2, \sigma_p^2\} \cdot \sigma_\epsilon^{-1} n\|\boldsymbol{\mu}\|_2^3 d\log(Tn/\delta) \gtrsim \min\left\{\frac{\|\boldsymbol{\mu}\|_2}{\sigma_\epsilon}, \frac{\|\boldsymbol{\mu}\|_2^3}{\sigma_\epsilon^3 d}\right\} \cdot \frac{n}{\log^3(Tn/\delta)} \tag{339}$$

$$\gtrsim \min\left\{\frac{n\|\boldsymbol{\mu}\|_2}{\sigma_\epsilon \log^3(Tn/\delta)}, C^3 n d^{1/8}\right\} \tag{340}$$

$$= \Omega(1). \tag{341}$$

The same result holds for $\langle \mathbf{W}(\tau+1)\boldsymbol{\mu}_{-1}, \mathbf{W}(\tau+1)\boldsymbol{\epsilon}_u^{(j)}\rangle$. Finally, we have

$$\langle \mathbf{W}(\tau+1)\boldsymbol{\epsilon}_u^{(j)}, \mathbf{W}(\tau+1)\boldsymbol{\epsilon}_v^{(k)}\rangle - \langle \mathbf{W}(T_1)\boldsymbol{\epsilon}_u^{(j)}, \mathbf{W}(T_1)\boldsymbol{\epsilon}_v^{(k)}\rangle$$

$$= \frac{\alpha}{n} \sum_{\tau'=T_1}^{\tau} \sum_{i=1}^{n} (-\ell_i'(\tau')) \cdot Y^{(i)} \cdot (I_{i,k,v}(\tau')\rho_{j,u}(\tau') + I_{i,j,u}(\tau')\rho_{k,v}(\tau')) + O(\alpha^2) \tag{342}$$

$$\lesssim \alpha \cdot \frac{n^{-1}\mathrm{SNR}^{-2} \cdot n^{-1}\sigma_\epsilon^2 d}{\alpha n^{-1}\sigma_\epsilon^2 d^2 \max\{\sigma_w^2, \sigma_p^2\}} \cdot n^{-1}\mathrm{SNR}^{-2} + O(\alpha^2) \tag{343}$$

$$\lesssim \frac{\sigma_\epsilon^4 d}{n^2\|\boldsymbol{\mu}\|_2^4 \max\{\sigma_w^2, \sigma_p^2\}} + O(\alpha^2) = O(\sigma_w^2 \sigma_\epsilon^2 d^{3/2} \log(Tn/\delta)), \tag{344}$$

which follows from the parameter assumptions; specifically, we have

$$\sigma_w^2 \max\{\sigma_w^2, \sigma_p^2\} \cdot \sigma_\epsilon^{-2} n^2 \|\boldsymbol{\mu}\|_2^4 \sqrt{d} \log(Tn/\delta) \gtrsim \min\left\{ \frac{\|\boldsymbol{\mu}\|_2^2}{\sigma_\epsilon^2 \sqrt{d}}, \frac{\|\boldsymbol{\mu}\|_2^4}{\sigma_\epsilon^4 d^{3/2}} \right\} \cdot \frac{n^2}{\log^3(Tn/\delta)} \tag{345}$$

$$\gtrsim \min\left\{ \frac{C^2 n^2 d^{1/4}}{\log(Tn/\delta)}, C^4 n^2 \log(Tn/\delta) \right\} \tag{346}$$

$$= \Omega(1), \tag{347}$$

which completes the proof.

*Proof of Lemma C.13.* At time step $\tau = T_1$, $A(T_1)$, $B(T_1)$, $C(T_1)$, and $G(T_1)$ hold from the proof for the base case. As for $D(\tau)$, $E(\tau)$, and $F(\tau)$, they are derived from $C(\tau)$. For the next time step, $A(\tau+1), B(\tau+1), C(\tau+1)$, and $G(\tau+1)$ are proved based on the propositions up to time step $\tau$. Thus, the proof is completed by induction. □

### C.3.2. PROOF OF BENIGN OVERFITTING CASE IN THEOREM 4.1

In this section, we provide proof of the benign overfitting case in the main theorem. We divide the proof into two parts: behavior on the training data and generalization performance.

**Training data.** The conditions of Lemmas C.12 and C.13 are satisfied with the current SNR condition $\text{SNR}^2 = o(n^{-1})$. The proposition $C(\tau)$ in Lemma C.13 states that $T_2 = \Theta\left( \frac{\exp(n^{-1}\text{SNR}^{-2})}{\alpha n^{-1} \sigma_\epsilon^2 \|\boldsymbol{\nu}\|_2 \|\boldsymbol{\mu}\|_2 d^2 \max\{\sigma_w^2, \sigma_p^2\}} \right)$ satisfies

$$g(\Lambda_{i,t}(T_2)) \geq g(\Lambda_{i,t}(0)) + \Theta\left(\exp\left(n^{-1}\text{SNR}^{-2}\right)\right) = \omega(1), \tag{348}$$

for any clean data $i \in \mathcal{C}$ and $t \in [T] \setminus \{1\}$. Here, we used $g(\Lambda_{i,t}(0)) = -(1 \pm o(1))T = -\Theta(1)$ from Lemma C.8. Recall that $g(x) = 2x + 2\sinh(x - \log T)$, and there exists a constant $c_1' > 1$ such that $c_1' \exp(x) > g(x)$. Therefore, for all $i \in \mathcal{C}$, we have

$$s_1^{(i)}(T_2) = \frac{1}{1 + \sum_{t=2}^T \exp\left(-\Lambda_{i,t}(T_2)\right)} > 1 - \sum_{t=2}^T \exp\left(-\Lambda_{i,t}(T_2)\right) > 1 - (T-1)c_1' \cdot o(1) > 1 - \epsilon, \tag{349}$$

for sufficiently small constant $\epsilon > 0$. By using Equation (349) and Lemma B.12, we have for any clean data $i \in \mathcal{C}$ that

$$Y^{(i)} \cdot f_{T_2}(\mathbf{X}^{(i)}) = Y^{(i)} \cdot \boldsymbol{\nu}^\top \mathbf{X}^{(i)\top} \mathbb{S}\left( \mathbf{X}^{(i)} \mathbf{W}(T_2)^\top \mathbf{p}(T_2) \right) \tag{350}$$

$$= Y^{(i)} \cdot \gamma_1^{(i)} s_1^{(i)}(T_2) + \sum_{t=2}^T Y^{(i)} \cdot \gamma_t^{(i)} s_t^{(i)}(T_2) \tag{351}$$

$$\geq \Theta\left(\|\boldsymbol{\nu}\|_2 \|\boldsymbol{\mu}\|_2\right)(1 - \epsilon) - O\left(\rho \|\boldsymbol{\nu}\|_2 \|\boldsymbol{\mu}\|_2\right) \cdot \epsilon \tag{352}$$

$$> 0. \tag{353}$$

For noisy data, again from $C(\tau)$ in Lemma C.13, we have

$$g(\Gamma_{j,1}(\tau)) > g(\Gamma_{j,1}(\tau) - \log \rho^{-1}) \geq g(\Gamma_{j,1}(0) - \log \rho^{-1}) + \Theta\left(\rho \exp\left(n^{-1}\text{SNR}^{-2}\right)\right) = \omega(\rho^{-1}), \tag{354}$$

$$g(\Gamma_{j,t}(\tau)) \geq g(\Gamma_{j,t}(0)) + \Theta\left(\rho \exp\left(n^{-1}\text{SNR}^{-2}\right)\right) = \omega(\rho^{-1}), \tag{355}$$

for $t \in [T] \setminus \{1, 2\}$, which follows from $n^{-1}\text{SNR}^{-2} = \omega(1)$ and Remark 4.8. Similarly, we have

$$s_2^{(j)}(T_2) = \frac{1}{1 + \sum_{t=[T] \setminus \{2\}} \exp\left(-\Gamma_{j,t}(T_2)\right)} > 1 - \sum_{t=[T] \setminus \{2\}} \exp\left(-\Gamma_{j,t}(T_2)\right) > 1 - (T-1)c_1' \cdot o(\rho) > 1 - \rho\epsilon. \tag{356}$$

Thus, we have

$$Y^{(j)} \cdot f_{T_2}(\mathbf{X}^{(j)}) = Y^{(j)} \cdot \boldsymbol{\nu}^\top \mathbf{X}^{(j)\top} \mathbb{S}\left(\mathbf{X}^{(j)}\mathbf{W}(T_2)^\top \mathbf{p}(T_2)\right) \tag{357}$$

$$= Y^{(j)} \cdot \gamma_2^{(j)} s_2^{(j)}(T_2) + \sum_{t \in [T] \setminus \{2\}} Y^{(j)} \cdot \gamma_t^{(j)} s_t^{(j)}(T_2) \tag{358}$$

$$\geq \Theta\left(\rho \|\boldsymbol{\nu}\|_2 \|\boldsymbol{\mu}\|_2\right)(1 - \rho\epsilon) - O\left(\|\boldsymbol{\nu}\|_2 \|\boldsymbol{\mu}\|_2\right) \cdot \rho\epsilon \tag{359}$$

$$> 0. \tag{360}$$

Equations (353) and (360) hold deterministically on a good run. Therefore, at time step $\tau = T_2$, we have that with probability at least $1 - \delta$,

$$\forall i \in \mathcal{C}, \ f_\tau(\mathbf{X}^{(i)}) = Y^{(i)}, \ \forall j \in \mathcal{N}, \ f_\tau(\mathbf{X}^{(j)}) = Y^{(j)}. \tag{361}$$

**Generalization.** Let $(\mathbf{X}, Y^*) \sim P^*$ be the unseen data on which we investigate generalization performance. We first evaluate the attention values of signal vectors at time step $T_2$. From Lemmas D.5, D.8 and D.10, we have

$$\lambda_{+1}(T_2) - \lambda_{+1}(0) \geq \sum_{\tau=0}^{T_2-1} s_1^{(i)}(\tau)(1 - s_1^{(i)}(\tau)) \cdot c\alpha \|\boldsymbol{\nu}\|_2 \|\boldsymbol{\mu}\|_2^3 d \max\{\sigma_w^2, \sigma_p^2\} \tag{362}$$

$$\gtrsim \log\left((T_2 - 1) \cdot \alpha n^{-1} \sigma_\epsilon^2 \|\boldsymbol{\nu}\|_2 \|\boldsymbol{\mu}\|_2 d^2 \max\{\sigma_w^2, \sigma_p^2\}\right) \cdot n\text{SNR}^2 \tag{363}$$

$$\gtrsim 1, \tag{364}$$

which follows from the definition of $T_2$. From Lemma C.8, we have $|\lambda_{+1}(0)| = o(1)$, which implies $\lambda_{+1}(T_2) \gtrsim 1$. The same result holds for $\lambda_{-1}(T_2)$.

Next, we show that the attention scores of the noise vectors $\{\boldsymbol{\epsilon}_t\}_{t \in [T]}$ in the unseen data, i.e., $\boldsymbol{\epsilon}_t^\top \mathbf{W}(T_2)^\top \mathbf{p}(T_2)$, become sufficiently small on a good run. While it is natural to evaluate $\|\mathbf{W}(T_2)^\top \mathbf{p}(T_2)\|_2$ and use a concentration inequality, it is challenging to track the evolution of $\|\mathbf{W}(\tau)^\top \mathbf{p}(\tau)\|_2$. Therefore, following induction proof in Lemmas C.12 and C.13, we apply a concentration inequality at time step 0 and show that the result does not change at time step $T_2$. Let us define $\mathcal{E}$ as the event that the following inequalities are satisfied:

$$\forall t \in [T], \ (1 - o(1))\,\sigma_\epsilon\sqrt{d} \leq \|\boldsymbol{\epsilon}_t\|_2 \leq (1 + o(1))\,\sigma_\epsilon\sqrt{d}, \tag{365}$$

$$\forall t \in [T], \forall k \in \{\pm 1\}, \ |\langle \boldsymbol{\epsilon}_t, \boldsymbol{\mu}_k \rangle| < c_2 \sigma_\epsilon \|\boldsymbol{\mu}\|_2 \sqrt{\log(Tn/\delta)}, \tag{366}$$

$$\forall i \in [n], \forall t, u \in [T], \ |\langle \boldsymbol{\epsilon}_t, \boldsymbol{\epsilon}_u^{(i)} \rangle| < c_1 \sigma_\epsilon^2 \sqrt{d} \log(Tn/\delta), \tag{367}$$

$$\forall t \in [T], \forall k \in \{\pm 1\}, \ |\langle \mathbf{W}(0)\boldsymbol{\epsilon}_t, \mathbf{W}(0)\boldsymbol{\mu}_k \rangle| < c_1 \sigma_w^2 \|\boldsymbol{\mu}\|_2 \|\boldsymbol{\epsilon}_t\|_2 \sqrt{d} \log(Tn/\delta), \tag{368}$$

$$\forall i \in [n], \forall t, u \in [T], \ |\langle \mathbf{W}(0)\boldsymbol{\epsilon}_t, \mathbf{W}(0)\boldsymbol{\epsilon}_u^{(i)} \rangle| < c_1 \sigma_w^2 \|\boldsymbol{\epsilon}_t\|_2 \|\boldsymbol{\epsilon}_u^{(i)}\|_2 \sqrt{d} \log(Tn/\delta), \tag{369}$$

$$\forall t \in [T], \ |\langle \mathbf{W}(0)\boldsymbol{\epsilon}_t, \mathbf{p}(0) \rangle| < c_1 \sigma_w \sigma_p \|\boldsymbol{\epsilon}_t\|_2 \sqrt{d} \log(Tn/\delta), \tag{370}$$

$$\forall t \in [T], \ |\langle \boldsymbol{\nu}, \boldsymbol{\epsilon}_t \rangle| < c_2 \sigma_\epsilon \|\boldsymbol{\nu}\|_2 \sqrt{\log(Tn/\delta)}, \tag{371}$$

where the constants $c_1, c_2$ are the same ones appeared in Lemma B.1. Applying union bound on the modified versions of Lemmas B.6, B.8 and B.9, the probability of the occurrence of $\mathcal{E}$ can be evaluated. Since there is no need to apply the union bound over the additional $n$ training data points, the outlier probability can be reduced by $1/n$ compared to the original lemma. Therefore, we have

$$\Pr[\mathcal{E}] > 1 - \delta/n > 1 - \delta. \tag{372}$$

In the following, using the results of Lemmas C.12 and C.13, we will prove that the next proposition holds for all $\tau \in [0, T_2]$ under the condition $\mathcal{E}$:

$H(\tau)$**:**

$$|\langle \mathbf{W}(\tau)\boldsymbol{\epsilon}_t, \mathbf{p}(\tau)\rangle| < O\left(\sigma_w \sigma_p \sigma_\epsilon d \log(Tn/\delta)\right),$$
$$|\langle \mathbf{W}(\tau)\boldsymbol{\epsilon}_t, \mathbf{W}(\tau)\boldsymbol{\mu}_{+1}\rangle| < O\left(\sigma_w^2 \sigma_\epsilon \|\boldsymbol{\mu}\|_2 d \log(Tn/\delta)\right),$$
$$|\langle \mathbf{W}(\tau)\boldsymbol{\epsilon}_t, \mathbf{W}(\tau)\boldsymbol{\mu}_{-1}\rangle| < O\left(\sigma_w^2 \sigma_\epsilon \|\boldsymbol{\mu}\|_2 d \log(Tn/\delta)\right),$$
$$|\langle \mathbf{W}(\tau)\boldsymbol{\epsilon}_t, \mathbf{W}(\tau)\boldsymbol{\epsilon}_u^{(i)}\rangle| < O\left(\sigma_w^2 \sigma_\epsilon^2 d^{3/2} \log(Tn/\delta)\right),$$

for all $i \in [n]$ and $t, u \in [T]$.

*Proof of $H(\tau)$.* We proceed with the proof by induction. The base case holds from the condition $\mathcal{E}$. In the following, suppose that $H(\tau')$ holds for any $\tau' \in [0, \tau]$. By a calculation similar to that in the proof of Lemma C.4, we have

$$\langle \mathbf{W}(\tau+1)\boldsymbol{\epsilon}_t, \mathbf{p}(\tau+1)\rangle - \langle \mathbf{W}(0)\boldsymbol{\epsilon}_t, \mathbf{p}(0)\rangle$$

$$= \sum_{\tau'=0}^{\tau} \left(\langle \mathbf{W}(\tau'+1)\boldsymbol{\epsilon}_t, \mathbf{p}(\tau'+1)\rangle - \langle \mathbf{W}(\tau')\boldsymbol{\epsilon}_t, \mathbf{p}(\tau')\rangle\right) \tag{373}$$

$$= \frac{\alpha}{n} \sum_{\tau'=0}^{\tau} \sum_{i=1}^{n} (-\ell_i'(\tau')) \cdot Y^{(i)} \cdot \sum_{t=1}^{T} s_t^{(i)}(\tau') \left(\gamma_t^{(i)} - \sum_{u=1}^{T} s_u^{(i)}(\tau')\gamma_u^{(i)}\right) \left(\langle \mathbf{W}(\tau')\boldsymbol{\epsilon}_t, \mathbf{W}(\tau')\mathbf{x}_t^{(i)}\rangle + \|\mathbf{p}(\tau')\|_2^2 \langle \boldsymbol{\epsilon}_t, \mathbf{x}_t^{(i)}\rangle\right)$$

$$+ \sum_{\tau'=0}^{\tau} \alpha^2 \boldsymbol{\epsilon}_t^\top \nabla_{\mathbf{W}^\top} \widehat{\mathcal{L}}(\tau') \nabla_{\mathbf{p}} \widehat{\mathcal{L}}(\tau'). \tag{374}$$

We bound the first term using the induction hypothesis, the condition $\mathcal{E}$, and $A(\tau')$ in Lemmas C.12 and C.13. We first evaluate the softmax probability part. For $\tau \leq T_1$, using a similar argument to Equations (267) and (270), we have that for any $i \in [n]$,

$$\sum_{\tau'=0}^{\tau} \sum_{t=1}^{T} s_t^{(i)}(\tau') \left(\gamma_t^{(i)} - \sum_{u=1}^{T} s_u^{(i)}(\tau')\gamma_u^{(i)}\right) \lesssim \frac{\log \rho^{-1}}{\alpha n^{-1}\sigma_\epsilon^2 d^2 \max\{\sigma_w^2, \sigma_p^2\}}. \tag{375}$$

For $\tau \geq T_1$, it follows from Lemma C.13 and the same technique used in Equations (309) and (314) that

$$\sum_{\tau'=T_1}^{\tau} \sum_{t=1}^{T} s_t^{(i)}(\tau') \left(\gamma_t^{(i)} - \sum_{u=1}^{T} s_u^{(i)}(\tau')\gamma_u^{(i)}\right) \lesssim \frac{n^{-1}\mathrm{SNR}^{-2}}{\alpha n^{-1}\sigma_\epsilon^2 d^2 \max\{\sigma_w^2, \sigma_p^2\}}, \tag{376}$$

for any $i \in [n]$. Substituting Equations (375) and (376) to Equation (374), we have

$$\frac{\alpha}{n} \sum_{\tau'=0}^{\tau} \sum_{i=1}^{n} (-\ell_i'(\tau')) \cdot Y^{(i)} \cdot \left(\sum_{t=1}^{T} s_t^{(i)}(\tau') \left(\gamma_t^{(i)} - \sum_{u=1}^{T} s_u^{(i)}(\tau')\gamma_u^{(i)}\right) \left(\langle \mathbf{W}(\tau')\boldsymbol{\epsilon}_t, \mathbf{W}(\tau')\mathbf{x}_t^{(i)}\rangle + \|\mathbf{p}(\tau')\|_2^2 \langle \boldsymbol{\epsilon}_t, \mathbf{x}_t^{(i)}\rangle\right)\right)$$

$$\lesssim \frac{n^{-1}\mathrm{SNR}^{-2}}{n^{-1}\sigma_\epsilon^2 d^2 \max\{\sigma_w^2, \sigma_p^2\}} \cdot \sigma_w^2 \sigma_\epsilon d \max\left\{\|\boldsymbol{\mu}\|_2, \sigma_\epsilon \sqrt{d}\right\} \log(Tn/\delta)$$

$$+ \frac{n^{-1}\mathrm{SNR}^{-2}}{n^{-1}\sigma_\epsilon^2 d^2 \max\{\sigma_w^2, \sigma_p^2\}} \cdot \sigma_p^2 d \cdot \sigma_\epsilon \max\left\{\|\boldsymbol{\mu}\|_2 \sqrt{\log(Tn/\delta)}, \sigma_\epsilon \sqrt{d}\log(Tn/\delta)\right\} \tag{377}$$

$$\lesssim \frac{n^{-1}\mathrm{SNR}^{-2}}{n^{-1}\sigma_\epsilon^2 d^2 \max\{\sigma_w^2, \sigma_p^2\}} \cdot \max\{\sigma_w^2, \sigma_p^2\}\sigma_\epsilon d \cdot \max\{\|\boldsymbol{\mu}\|_2, \sigma_\epsilon \sqrt{d}\} \log(Tn/\delta) \tag{378}$$

$$\lesssim \max\{\|\boldsymbol{\mu}\|_2^{-1}, \sigma_\epsilon \sqrt{d}\|\boldsymbol{\mu}\|_2^{-2}\} \cdot \sigma_\epsilon \log(Tn/\delta) = O\left(\sigma_w \sigma_p \sigma_\epsilon d \log(Tn/\delta)\right), \tag{379}$$

where the last line follows from the parameter assumptions, using the same argument as in Equation (208) for the not-overfitting case. For the quadratic term in Equation (374), we can show that this term is small enough to ignore, similarly to Equation (216). Therefore, substituting Equation (379) to Equation (374) leads to

$$\langle \mathbf{W}(\tau+1)\boldsymbol{\epsilon}_t, \mathbf{p}(\tau+1)\rangle = \langle \mathbf{W}(0)\boldsymbol{\epsilon}_t, \mathbf{p}(0)\rangle + O\left(\sigma_w \sigma_p \sigma_\epsilon d \log(Tn/\delta)\right) = O\left(\sigma_w \sigma_p \sigma_\epsilon d \log(Tn/\delta)\right). \tag{380}$$

The proof for the three inequalities below in the proposition $H(\tau)$ is omitted because they can be shown in the same manner as $B(\tau + 1)$ in Lemmas C.12 and C.13, under the induction hypothesis and the results of Lemmas C.12 and C.13. Since $H(\tau+1)$ holds under the condition $H(\tau)$ is valid, from induction argument, the proposition $H(\tau)$ holds for $\tau \in [0, T_2]$. $\qquad\square$

From $H(T_2)$ and Assumption A8, we have

$$|\langle \mathbf{W}(\tau)\boldsymbol{\epsilon}_t, \mathbf{p}(\tau)\rangle| \lesssim \sigma_w \sigma_p \sigma_\epsilon d \log(Tn/\delta) \leq \frac{c_2'}{\log(Tn/\delta)}, \tag{381}$$

for some constant $c_2' > 0$. Using Equation (364) and taking sufficiently large $T_2$, we have $\lambda_{+1}(T_2) > 2c_2'$ and $\lambda_{-1}(T_2) > 2c_2'$. Thus, we have that for $t \in [T] \setminus \{1\}$,

$$(\mathbf{x}_t - \mathbf{x}_1)^\top \mathbf{W}(T_2)^\top \mathbf{p}(T_2) \leq -(1 - \rho) \max\{\lambda_{+1}(T_2), \lambda_{-1}(T_2)\} + (\boldsymbol{\epsilon}_t - \boldsymbol{\epsilon}_1)^\top \mathbf{W}(T_2)^\top \mathbf{p}(T_2) \tag{382}$$

$$\leq -(1 - 1/C) \cdot 2c_2' + \frac{2c_2'}{\log(Tn/\delta)} \tag{383}$$

$$< -c_2', \tag{384}$$

where the second inequality holds by $\rho < 1/C$. Then, the softmax probability of the relevant token is lower-bounded as:

$$s_1(T_2) = \frac{1}{1 + \sum_{t=2}^{T} \exp\left((\mathbf{x}_t - \mathbf{x}_1)^\top \mathbf{W}(T_2)^\top \mathbf{p}(T_2)\right)} > 1 - \epsilon, \tag{385}$$

for sufficiently small $\epsilon > 0$. Consequently, we have

$$Y^* \cdot f_{T_2}(\mathbf{X}) = Y^* \cdot \gamma_1 s_1(T_2) + \sum_{t=2}^{T} Y^* \cdot \gamma_t s_t(T_2) \tag{386}$$

$$\geq \Theta\left(\|\boldsymbol{\nu}\|_2 \|\boldsymbol{\mu}\|_2\right) \cdot (1 - \epsilon) - O\left(\rho \|\boldsymbol{\nu}\|_2 \|\boldsymbol{\mu}\|_2\right) \cdot \epsilon > 0. \tag{387}$$

Under the conditioning on $\mathcal{E}$, the output $f_{T_2}(\mathbf{X})$ deterministically takes the same sign as the true label $Y^*$. Thus, the generalization error is bounded as:

$$\Pr_{(\mathbf{X}, Y^*) \sim P^*}\left[\mathrm{sign}(f_{T_2}(\mathbf{X})) \neq Y^*\right] = \Pr_{(\mathbf{X}, Y^*) \sim P^*}\left[\mathrm{sign}(f_{T_2}(\mathbf{X})) \neq Y^* \mid \mathcal{E}\right] + \Pr_{(\mathbf{X}, Y^*) \sim P^*}\left[\mathcal{E}^c\right] < \delta, \tag{388}$$

where we used $\Pr(A) \leq \Pr(A|B^C) + \Pr(B)$ and the result of Equation (372). This concludes the proof.

## D. Technical Calculations

In this section, we provide the small lemmas that are necessary for the proof of the main theorem. The lemmas concerning the softmax probabilities are given in Appendix D.1. In Appendices D.2 and D.3, we provide lemmas for the attention updates in the not-overfitting case and the benign overfitting case, respectively.

### D.1. Softmax Probability

In the analysis of attention dynamics, the values of $s_1(\tau)(1 - s_1(\tau))$ and $s_2(\tau)(1 - s_2(\tau))$ play a significant role. We have the following lemma on the evaluation of this value.

**Lemma D.1** (Bounds for relevant token probability). *Fix arbitrary $i \in [n]$, and suppose that there exists a constant $c' > 1$ such that for all $t, u \in [T] \setminus \{1\}$, we have $\exp\left(\Lambda_{i,t} - \Lambda_{i,u}\right) < c'$. Then, there exists a constant $c > 1$ such that for any $t \in [T] \setminus \{1\}$,*

$$\frac{c^{-1}}{2 + 2\cosh\left(\Lambda_{i,t}(\tau) - \log T\right)} < s_1^{(i)}(\tau)(1 - s_1^{(i)}(\tau)) < \frac{c}{2 + 2\cosh\left(\Lambda_{i,t}(\tau) - \log T\right)}.$$

*Proof of Lemma D.1.* Using notations introduced in Definition C.3, we have

$$s_1^{(i)}(\tau)(1 - s_1^{(i)}(\tau)) = \frac{\exp\left(\mathbf{x}_1^{(i)\top}\mathbf{W}(\tau)^\top\mathbf{p}(\tau)\right)}{\sum_{t=1}^T \exp\left(\mathbf{x}_t^{(i)\top}\mathbf{W}(\tau)^\top\mathbf{p}(\tau)\right)} \cdot \frac{\sum_{t=2}^T \exp\left(\mathbf{x}_t^{(i)\top}\mathbf{W}(\tau)^\top\mathbf{p}(\tau)\right)}{\sum_{t=1}^T \exp\left(\mathbf{x}_t^{(i)\top}\mathbf{W}(\tau)^\top\mathbf{p}(\tau)\right)} \tag{389}$$

$$= \frac{1}{1 + \sum_{t=2}^T \exp(-\Lambda_{i,t}(\tau))} \cdot \frac{\sum_{t=2}^T \exp(-\Lambda_{i,t}(\tau))}{1 + \sum_{t=2}^T \exp(-\Lambda_{i,t}(\tau))}. \tag{390}$$

Using the condition of the lemma, we have

$$s_1^{(i)}(\tau)(1 - s_1^{(i)}(\tau)) \le \frac{1}{1 + c'^{-1}(T-1)\exp(-\Lambda_{i,t}(\tau))} \cdot \frac{c'(T-1)\exp(-\Lambda_{i,t}(\tau))}{1 + c'^{-1}(T-1)\exp(-\Lambda_{i,t}(\tau))}$$

$$= \frac{c'^3 T}{T-1} \cdot \frac{1}{(c'T)/(T-1) + T\exp(-\Lambda_{i,t}(\tau))} \cdot \frac{T\exp(-\Lambda_{i,t}(\tau))}{(c'T)/(T-1) + T\exp(-\Lambda_{i,t}(\tau))} \tag{391}$$

$$< c \cdot \frac{T\exp\left(-\Lambda_{i,t}(\tau)\right)}{\left(1 + T\exp\left(-\Lambda_{i,t}(\tau)\right)\right)^2} \tag{392}$$

$$= c \cdot \frac{1}{2 + 2\cosh\left(\Lambda_{i,t}(\tau) - \log T\right)}, \tag{393}$$

where the constant $c'^3 T/(T-1)$ is replaced with $c > 0$. This gives the desired result. The lower bound is shown in a similar way. $\qquad\square$

Next, we show a similar lemma that is used in the proof of the benign overfitting case.

**Lemma D.2** (Bounds for confusing token probability). *Fix arbitrary $j \in \mathcal{N}$, and suppose that there exists a constant $c' > 1$ such that for all $t, u \in [T] \setminus \{1, 2\}$, we have*

$$\exp\left(\Gamma_{j,t}(\tau) - \Gamma_{j,u}(\tau)\right) < c', \ \exp\left(\Gamma_{j,1}(\tau) - \Gamma_{j,t}(\tau)\right) < \rho^{-1}c', \ \exp\left(\Gamma_{j,t}(\tau) - \Gamma_{j,1}(\tau)\right) < \rho c'.$$

*Then, there exists a constant $c > 1$ such that for any $t \in [T] \setminus \{1, 2\}$,*

$$\frac{c^{-1}}{2 + 2\cosh\left(\Gamma_{j,t}(\tau) - \log T\right)} < s_2^{(j)}(\tau)(1 - s_2^{(j)}(\tau)) < \frac{c}{2 + 2\cosh\left(\Gamma_{j,t}(\tau) - \log T\right)},$$

$$\frac{c^{-1}}{2 + 2\cosh\left(\Gamma_{j,1}(\tau) - \log\rho^{-1} - \log T\right)} < s_2^{(j)}(\tau)(1 - s_2^{(j)}(\tau)) < \frac{c}{2 + 2\cosh\left(\Gamma_{j,1}(\tau) - \log\rho^{-1} - \log T\right)}.$$

*Proof of Lemma D.2.* From the definition of $\Gamma_{j,t}$ in Definition C.3, we have

$$s_2^{(j)}(\tau)(1 - s_2^{(j)}(\tau)) = \frac{1}{1 + \sum_{t \in [T]\setminus\{2\}} \exp(-\Gamma_{j,t}(\tau))} \cdot \frac{\sum_{t \in [T]\setminus\{2\}} \exp(-\Gamma_{j,t}(\tau))}{1 + \sum_{t \in [T]\setminus\{2\}} \exp(-\Gamma_{j,t}(\tau))}. \tag{394}$$

For $t \in [T] \setminus \{1, 2\}$, since $c'^{-1}\rho\exp(-\Gamma_{j,t}(\tau)) \le \exp(-\Gamma_{j,1}(\tau)) \le c'\rho\exp(-\Gamma_{j,t}(\tau))$ and $c'^{-1}\exp(-\Gamma_{j,t}(\tau)) \le \exp(-\Gamma_{j,u}(\tau)) \le c'\exp(-\Gamma_{j,t}(\tau))$ for any $u \in [T] \setminus \{1, 2\}$, using the same discussion as in Lemma D.1 and the parameter assumption $\rho < 1/C$, we have the desired result. For the relevant token, since $c'^{-1}\rho^{-1}\exp(-\Gamma_{j,1}(\tau)) \le \exp(-\Gamma_{j,t}(\tau)) \le c'\rho^{-1}\exp(-\Gamma_{j,1}(\tau))$ for any $t \in [T] \setminus \{1, 2\}$, similarly to Equation (393), we have

$$s_2^{(j)}(\tau)(1 - s_2^{(j)}(\tau)) < c \cdot \frac{1}{2 + 2\cosh\left(\Gamma_{j,1}(\tau) - \log\rho^{-1} - \log T\right)}. \tag{395}$$

The lower bound is derived in the same way, which completes the proof. $\qquad\square$

In the next lemma, we confirm that the significant term in tracking the gradient descent dynamics, $s(1 - s)$, is dominated by the token with the highest assigned probability.

**Lemma D.3** (Inequality for $s(1-s)$). *Let $\mathbf{s} \in \mathbb{R}^T$ be a probability vector, and let $t \in [T]$ be such that $s_t = \max_{t' \in [T]} s_{t'}$. Then, we have*

$$s_u \left(1 - s_u\right) \leq s_t \left(1 - s_t\right), \ \forall u \in [T] \setminus \{t\}.$$

*Proof of Lemma D.3.* The function $f(x) = x - x^2$ defined on $x \in [0, 1]$ is monotonically increasing in $[0, 1/2]$, and $f(x) = f(1-x)$ holds by the symmetry at $x = 1/2$. When $0 \leq s_t \leq 1/2$, the claim holds from the monotonicity over $[0, 1/2]$. For the remaining case $1/2 \leq s_t \leq 1$, since $f(s_t) = f(1 - s_t)$ and

$$s_u \leq \sum_{v \in [T] \setminus \{t\}} s_v = 1 - s_t \tag{396}$$

hold, the claim follows from the monotonicity over $[0, 1/2]$ again. $\qquad\square$

In the next lemma, we will see that with the assumption of a sufficiently small step size, the attention values do not change significantly in a single step of gradient descent.

**Lemma D.4** (One-step update of attention). *Suppose that the assumptions in Theorem 4.1 are satisfied. Then, under the condition $A(\tau)$ and $B(\tau)$, which appear in Lemmas C.11 to C.13, we have*

$$|\lambda_{+1}(\tau+1) - \lambda_{+1}(\tau)| = o(1), \ |\lambda_{-1}(\tau+1) - \lambda_{-1}(\tau)| = o(1), \ |\rho_{i,t}(\tau+1) - \rho_{i,t}(\tau)| = o(1),$$

*for any $i \in [n]$ and $t \in [T]$.*

*Proof of Lemma D.4.* From Lemmas C.4 and C.9, we have

$$|\lambda_{+1}(\tau+1) - \lambda_{+1}(\tau)| \lesssim \alpha \max\{I_{i,+}^W(\tau), \|\mathbf{p}(\tau)\|_2^2 I_{i,+}(\tau)\} + \alpha^2 \boldsymbol{\mu}_{+1}^\top \nabla_{\mathbf{W}^\top} \widehat{\mathcal{L}}(\tau) \nabla_{\mathbf{p}} \widehat{\mathcal{L}}(\tau). \tag{397}$$

For the first term, from Definition C.2, we have

$$\alpha \max\{I_{i,+}^W(\tau), \|\mathbf{p}(\tau)\|_2^2 I_{i,+}(\tau)\}$$

$$\lesssim \alpha \cdot \max_{i \in [n], t \in [T]}\{|\gamma_t^{(i)}|\} \cdot \left( \max_{i \in [n], t \in [T]} \left\{ \langle \mathbf{W}(\tau)\mathbf{x}_t^{(i)}, \mathbf{W}(\tau)\boldsymbol{\mu}_{+1} \rangle \right\} + \|\mathbf{p}(\tau)\|_2^2 \max_{i \in [n], t \in [T]} \left\{ \langle \mathbf{x}_t^{(i)}, \boldsymbol{\mu}_{+1} \rangle \right\} \right) \tag{398}$$

$$\lesssim \alpha \|\boldsymbol{\nu}\|_2 \|\boldsymbol{\mu}\|_2 \cdot \max\{\sigma_w^2, \sigma_p^2\} \|\boldsymbol{\mu}\|_2^2 d \tag{399}$$

$$= o(1), \tag{400}$$

where the third inequality follows from the propositions $A(\tau)$ and $B(\tau)$ in the conditions. The last line is by $\|\boldsymbol{\nu}\|_2 = O(\|\boldsymbol{\mu}\|_2)$ and the parameter assumptions in Section 3.5; specifically, $\alpha \leq \max\{\|\boldsymbol{\mu}\|_2 \sqrt{d}, \sigma_\epsilon d\}^{-1}/C$ and $\max\{\sigma_w^2, \sigma_p^2\} = \Theta\left(\max\{\|\boldsymbol{\mu}\|_2 \sqrt{d}, \sigma_\epsilon d\}^{-1} \log^{-2}(Tn/\delta)\right)$.

For the quadratic term, we have from Equations (9) and (12) that

$$\alpha^2 \boldsymbol{\mu}_{+1}^\top \nabla_{\mathbf{W}^\top} \widehat{\mathcal{L}}(\tau) \nabla_{\mathbf{p}} \widehat{\mathcal{L}}(\tau)$$

$$\lesssim \alpha^2 \cdot \left( \max_{i \in [n], t \in [T]}\{|\gamma_t^{(i)}|\} \right)^2 \cdot \max_{i \in [n], t \in [T]} \left\{ \langle \mathbf{x}_t^{(i)}, \boldsymbol{\mu}_{+1} \rangle \right\} \cdot \|\mathbf{p}(\tau)\|_2 \cdot \max_{i \in [n], t \in [T]} \left\{ \|\mathbf{W}(\tau)\mathbf{x}_t^{(i)}\|_2 \right\} \tag{401}$$

$$\lesssim \alpha^2 \|\boldsymbol{\nu}\|_2^2 \|\boldsymbol{\mu}\|_2^2 \cdot \|\boldsymbol{\mu}\|_2^2 \cdot \sigma_p \sqrt{d} \cdot \sigma_w \max\{\|\boldsymbol{\mu}\|_2 \sqrt{d}, \sigma_\epsilon d\} \tag{402}$$

$$= o(1), \tag{403}$$

which similarly follows from parameter assumptions. Substituting Equations (400) and (403) to Equation (397) leads to the desired result. The other inequalities for $\lambda_{-1}$ and $\rho_{i,t}$ are shown in a similar discussion. $\qquad\square$

Finally, we provide the lemma regarding the bounds for the summation of probability terms $s(\tau)(1 - s(\tau))$. This result is necessary to evaluate the attention updates.

**Lemma D.5** (Bounds for summation of probability terms).

- *(Not Overfitting Case) Let $T_1$ be the time step defined in Lemma C.11, and consider $\tau \leq T_1$. Suppose that the propositions $C(\tau')$ and $E(\tau')$ in Lemma C.11 hold for any $\tau' \in [0, \tau]$. Then, we have that for any $i \in [n]$,*

$$\sum_{\tau'=0}^{\tau} s_1^{(i)}(\tau')(1 - s_1^{(i)}(\tau')) = \Theta(\tau).$$

- *(Benign Overfitting Case) Let $T_2$ be the time step defined in Lemma C.13, and consider $\tau \leq T_2$. Suppose that the propositions $C(\tau')$ and $E(\tau')$ in Lemmas C.12 and C.13 hold for any $\tau' \in [0, \tau]$. Then, we have that for any clean data $i \in \mathcal{C}$,*

$$\sum_{\tau'=0}^{\tau} s_1^{(i)}(\tau')(1 - s_1^{(i)}(\tau')) = \begin{cases} \Theta(\tau) & \text{if } \tau = O\left(\frac{1}{\alpha n^{-1}\sigma_\epsilon^2 \|\boldsymbol{\nu}\|_2 \|\boldsymbol{\mu}\|_2 d^2 \max\{\sigma_w^2, \sigma_p^2\}}\right), \\ \Theta\left(\frac{\log(\tau \cdot \alpha n^{-1}\sigma_\epsilon^2 \|\boldsymbol{\nu}\|_2 \|\boldsymbol{\mu}\|_2 d^2 \max\{\sigma_w^2, \sigma_p^2\})}{\alpha n^{-1}\sigma_\epsilon^2 \|\boldsymbol{\nu}\|_2 \|\boldsymbol{\mu}\|_2 d^2 \max\{\sigma_w^2, \sigma_p^2\}}\right) & \text{if } \tau = \Omega\left(\frac{1}{\alpha n^{-1}\sigma_\epsilon^2 \|\boldsymbol{\nu}\|_2 \|\boldsymbol{\mu}\|_2 d^2 \max\{\sigma_w^2, \sigma_p^2\}}\right). \end{cases}$$

*For any noisy data $j \in \mathcal{N}$, we have*

$$\sum_{\tau'=0}^{\tau} s_1^{(j)}(\tau')(1 - s_1^{(j)}(\tau')) = \Theta\left(\frac{\log\left(\tau \cdot \alpha n^{-1}\sigma_\epsilon^2 \|\boldsymbol{\nu}\|_2 \|\boldsymbol{\mu}\|_2 d^2 \max\{\sigma_w^2, \sigma_p^2\}\right)}{\alpha n^{-1}\sigma_\epsilon^2 \|\boldsymbol{\nu}\|_2 \|\boldsymbol{\mu}\|_2 d^2 \max\{\sigma_w^2, \sigma_p^2\}}\right) \quad \text{if } \tau \leq T_1,$$

*and*

$$\sum_{\tau'=T_1}^{\tau} s_2^{(j)}(\tau')(1 - s_2^{(j)}(\tau')) = \Theta\left(\frac{\log\left(\tau \cdot \rho \cdot \alpha n^{-1}\sigma_\epsilon^2 \|\boldsymbol{\nu}\|_2 \|\boldsymbol{\mu}\|_2 d^2 \max\{\sigma_w^2, \sigma_p^2\}\right)}{\rho \cdot \alpha n^{-1}\sigma_\epsilon^2 \|\boldsymbol{\nu}\|_2 \|\boldsymbol{\mu}\|_2 d^2 \max\{\sigma_w^2, \sigma_p^2\}}\right) \quad \text{if } \tau \geq T_1,$$

*where $T_1$ is the time step defined in Lemma C.12.*

*Proof of Lemma D.5.* We proceed with the proof separately for the not-overfitting and benign overfitting cases.

**Not overfitting case:** Fix a single $t$ from $[T] \setminus \{1\}$. Since $\Lambda_{i,t}(\tau)$ is monotonically increasing and $|\Lambda_{i,t}(0)| = o(1)$ by $C(\tau')$ and Lemma C.8, it follows from Lemma D.1 and $T = \Theta(1)$ that $s_1^{(i)}(\tau')(1 - s_1^{(i)}(\tau')) = \Theta(1)$ for any $\tau' \in [0, \tau]$. This immediately leads to the conclusion.

**Benign overfitting case:** We first discuss the case of clean data $i \in \mathcal{C}$. Fix $t \in [T] \setminus \{1\}$, and let $T_0$ be a first-time step where $\Lambda_{i,t}(\tau)$ exceeds $\log T$. From $C(\tau)$ and the parameter assumption $T = \Theta(1)$, we have $T_0 = \Theta\left(\frac{1}{\alpha n^{-1}\sigma_\epsilon^2 \|\boldsymbol{\nu}\|_2 \|\boldsymbol{\mu}\|_2 d^2 \max\{\sigma_w^2, \sigma_p^2\}}\right)$. We first consider the case of $\tau < T_0$. In this case, the conclusion follows from the same argument as in the not-overfitting case. This leads to the first line in the statement.

Next, we consider the case of $T_0 \leq \tau$ and first derive the lower bound. From Lemma D.1, we have

$$\sum_{\tau'=0}^{\tau} s_1^{(i)}(\tau')(1 - s_1^{(i)}(\tau')) > \sum_{\tau'=0}^{\tau} \frac{c^{-1}}{2 + 2\cosh\left(\Lambda_{i,t}(\tau') - \log T\right)} > \sum_{\tau'=T_0}^{\tau} \frac{c^{-1}}{2 + 2\cosh\left(\Lambda_{i,t}(\tau') - \log T\right)} \tag{404}$$

for any $i \in [n]$. Combining the inequality $2 + 2\cosh(x - \log T) < 2x + 2\sinh(x - \log T)$ for $x > \log T$ and the proposition $C(\tau)$ in Lemmas C.12 and C.13, we have

$$\frac{1}{2 + 2\cosh\left(\Lambda_{i,t}(\tau') - \log T\right)} > \frac{1}{2\Lambda_{i,t}(\tau') + 2\sinh\left(\Lambda_{i,t}(\tau') - \log T\right)} \tag{405}$$

$$= \frac{1}{g\left(\Lambda_{i,t}(\tau')\right)} \tag{406}$$

$$\geq \frac{1}{g\left(\Lambda_{i,t}(0)\right) + \tau' \cdot c_4 \alpha n^{-1}\sigma_\epsilon^2 \|\boldsymbol{\nu}\|_2 \|\boldsymbol{\mu}\|_2 d^2 \max\{\sigma_w^2, \sigma_p^2\}}. \tag{407}$$

Since the function $1/(b + ax)$ for $a > 0$ is convex and monotonically decreasing for $b + ax > 0$, we have

$$\sum_{\tau'=T_0}^{\tau} \frac{1}{2 + 2\cosh\left(\Lambda_{i,t}(\tau') - \log T\right)}$$

$$> \int_{T_0}^{\tau+1} \frac{d\tau'}{g\left(\Lambda_{i,t}(0)\right) + \tau' \cdot c_4 \alpha n^{-1} \sigma_\epsilon^2 \|\boldsymbol{\nu}\|_2 \|\boldsymbol{\mu}\|_2 d^2 \max\{\sigma_w^2, \sigma_p^2\}} \tag{408}$$

$$= \frac{1}{c_4 \alpha n^{-1} \sigma_\epsilon^2 \|\boldsymbol{\nu}\|_2 \|\boldsymbol{\mu}\|_2 d^2 \max\{\sigma_w^2, \sigma_p^2\}} \log\left(\frac{g\left(\Lambda_{i,t}(0)\right) + (\tau + 1) \cdot c_4 \alpha n^{-1} \sigma_\epsilon^2 \|\boldsymbol{\nu}\|_2 \|\boldsymbol{\mu}\|_2 d^2 \max\{\sigma_w^2, \sigma_p^2\}}{g\left(\Lambda_{i,t}(0)\right) + T_0 \cdot c_4 \alpha n^{-1} \sigma_\epsilon^2 \|\boldsymbol{\nu}\|_2 \|\boldsymbol{\mu}\|_2 d^2 \max\{\sigma_w^2, \sigma_p^2\}}\right) \tag{409}$$

$$\gtrsim \frac{\log\left(\tau \cdot \alpha n^{-1} \sigma_\epsilon^2 \|\boldsymbol{\nu}\|_2 \|\boldsymbol{\mu}\|_2 d^2 \max\{\sigma_w^2, \sigma_p^2\}\right)}{\alpha n^{-1} \sigma_\epsilon^2 \|\boldsymbol{\nu}\|_2 \|\boldsymbol{\mu}\|_2 d^2 \max\{\sigma_w^2, \sigma_p^2\}}, \tag{410}$$

where the last line follows from the fact that the denominator inside the log is $\Theta(1)$. This is ensured by the definition of $T_0$, which guarantees that the denominator exceeds $g(\log T) = 2\log T = \Theta(1)$, and by the fact that the one-step update of attention score is $o(1)$, as shown in Lemma D.4. Consequently, by substituting this result into Equation (404), we obtain the desired lower bound.

Similarly, we provide an upper bound using the function $g$ and the result of $C(\tau)$. From Lemma D.1, we have

$$\sum_{\tau'=0}^{\tau} s_1^{(i)}(\tau')(1 - s_1^{(i)}(\tau')) < \sum_{\tau'=0}^{T_0} s_1^{(i)}(\tau')(1 - s_1^{(i)}(\tau')) + \sum_{\tau'=T_0+1}^{\tau} \frac{c}{2 + 2\cosh\left(\Lambda_{i,t}(\tau') - \log T\right)} \tag{411}$$

$$\leq \frac{T_0}{4} + \sum_{\tau'=T_0+1}^{\tau} \frac{c(2\log T + 1)}{g\left(\Lambda_{i,t}(\tau')\right)} \tag{412}$$

$$\leq \frac{T_0}{4} + \sum_{\tau'=T_0+1}^{\tau} \frac{c'}{g\left(\Lambda_{i,t}(0)\right) + \tau' \cdot c_3 \alpha n^{-1} \sigma_\epsilon^2 \|\boldsymbol{\nu}\|_2 \|\boldsymbol{\mu}\|_2 d^2 \max\{\sigma_w^2, \sigma_p^2\}}, \tag{413}$$

where the second inequality comes from $s(1-s) \leq 1/4$, $\sinh(x) < \cosh(x)$, and the inequality $2\log T \cosh(x - \log T) > x$ for all $x$, which can be verified through a straightforward evaluation of the minimum. In the last line, we used the result of $C(\tau)$, and the coefficient is replaced with $c' > 0$ from the parameter assumption $T = \Theta(1)$. Similarly to the lower bound, we have

$$\sum_{\tau'=T_0+1}^{\tau} \frac{1}{g\left(\Lambda_{i,t}(0)\right) + \tau' \cdot c_3 \alpha n^{-1} \sigma_\epsilon^2 \|\boldsymbol{\nu}\|_2 \|\boldsymbol{\mu}\|_2 d^2 \max\{\sigma_w^2, \sigma_p^2\}}$$

$$< \int_{T_0}^{\tau} \frac{d\tau'}{g\left(\Lambda_{i,t}(0)\right) + \tau' \cdot c_3 \alpha n^{-1} \sigma_\epsilon^2 \|\boldsymbol{\nu}\|_2 \|\boldsymbol{\mu}\|_2 d^2 \max\{\sigma_w^2, \sigma_p^2\}} \tag{414}$$

$$= \frac{1}{c_3 \alpha n^{-1} \sigma_\epsilon^2 \|\boldsymbol{\nu}\|_2 \|\boldsymbol{\mu}\|_2 d^2 \max\{\sigma_w^2, \sigma_p^2\}} \log\left(\frac{g\left(\Lambda_{i,t}(0)\right) + \tau \cdot c_3 \alpha n^{-1} \sigma_\epsilon^2 \|\boldsymbol{\nu}\|_2 \|\boldsymbol{\mu}\|_2 d^2 \max\{\sigma_w^2, \sigma_p^2\}}{g\left(\Lambda_{i,t}(0)\right) + T_0 \cdot c_3 \alpha n^{-1} \sigma_\epsilon^2 \|\boldsymbol{\nu}\|_2 \|\boldsymbol{\mu}\|_2 d^2 \max\{\sigma_w^2, \sigma_p^2\}}\right) \tag{415}$$

$$\lesssim \frac{\log\left(\tau \cdot \alpha n^{-1} \sigma_\epsilon^2 \|\boldsymbol{\nu}\|_2 \|\boldsymbol{\mu}\|_2 d^2 \max\{\sigma_w^2, \sigma_p^2\}\right)}{\alpha n^{-1} \sigma_\epsilon^2 \|\boldsymbol{\nu}\|_2 \|\boldsymbol{\mu}\|_2 d^2 \max\{\sigma_w^2, \sigma_p^2\}}. \tag{416}$$

Combining this result with $T_0 = \Theta(1/\alpha n^{-1} \sigma_\epsilon^2 \|\boldsymbol{\nu}\|_2 \|\boldsymbol{\mu}\|_2 d^2 \max\{\sigma_w^2, \sigma_p^2\})$ from $C(\tau)$, Equation (413) completes the proof.

For noisy data $j \in \mathcal{N}$, we first derive the lower bound for $\tau \leq T_1$. From Lemma C.8 and the proposition $C(\tau)$ in Lemma C.12, we have $\Lambda_{j,t}(\tau) < o(1)$ for any $t \in [T] \backslash \{1\}$. Since we can confirm that the inequality $2 + 2\cosh(x - \log T) <$

$-2x - 2\sinh(x - \log T) + 2 = -g(x) + 2$ holds for $x < o(1)$ by rearranging terms, Lemma D.1 and Lemma C.12 gives us

$$\sum_{\tau'=0}^{\tau} s_1^{(j)}(\tau')(1 - s_1^{(j)}(\tau')) > \sum_{\tau'=0}^{\tau} \frac{c^{-1}}{2 + 2\cosh\left(\Lambda_{j,t}(\tau') - \log T\right)} \tag{417}$$

$$> \sum_{\tau'=0}^{\tau} \frac{c^{-1}}{-g\left(\Lambda_{j,t}(\tau')\right) + 2} \tag{418}$$

$$> \sum_{\tau'=0}^{\tau} \frac{c^{-1}}{-g\left(\Lambda_{j,t}(0)\right) + \tau \cdot c_6 \alpha n^{-1}\sigma_\epsilon^2 \|\boldsymbol{\nu}\|_2 \|\boldsymbol{\mu}\|_2 d^2 \max\{\sigma_w^2, \sigma_p^2\} + 2}. \tag{419}$$

Applying the same discussion as in Equation (410), we have

$$\sum_{\tau'=0}^{\tau} s_1^{(j)}(\tau')(1 - s_1^{(j)}(\tau')) \gtrsim \frac{\log\left(\tau \cdot \alpha n^{-1}\sigma_\epsilon^2 \|\boldsymbol{\nu}\|_2 \|\boldsymbol{\mu}\|_2 d^2 \max\{\sigma_w^2, \sigma_p^2\}\right)}{\alpha n^{-1}\sigma_\epsilon^2 \|\boldsymbol{\nu}\|_2 \|\boldsymbol{\mu}\|_2 d^2 \max\{\sigma_w^2, \sigma_p^2\}}. \tag{420}$$

As for the upper bound, since we obtain the inequality $2 + 2\cosh(x - \log T) > -x - \sinh(x - \log T) = -g(x)/2$ for $x < o(1)$ by rearranging terms, similarly we have

$$\sum_{\tau'=0}^{\tau} s_1^{(j)}(\tau')(1 - s_1^{(j)}(\tau')) < \sum_{\tau'=0}^{\tau} \frac{2c}{-g\left(\Lambda_{j,t}(0)\right) + \tau \cdot c_5 \alpha n^{-1}\sigma_\epsilon^2 \|\boldsymbol{\nu}\|_2 \|\boldsymbol{\mu}\|_2 d^2 \max\{\sigma_w^2, \sigma_p^2\}}. \tag{421}$$

Using a similar evaluation with integral, we have

$$\sum_{\tau'=0}^{\tau} s_1^{(j)}(\tau')(1 - s_1^{(j)}(\tau')) \lesssim \frac{\log\left(\tau \cdot \alpha n^{-1}\sigma_\epsilon^2 \|\boldsymbol{\nu}\|_2 \|\boldsymbol{\mu}\|_2 d^2 \max\{\sigma_w^2, \sigma_p^2\}\right)}{\alpha n^{-1}\sigma_\epsilon^2 \|\boldsymbol{\nu}\|_2 \|\boldsymbol{\mu}\|_2 d^2 \max\{\sigma_w^2, \sigma_p^2\}}, \tag{422}$$

which completes the proof for $\tau \leq T_1$.

Finally, for the case of $\tau \geq T_1$, the desired result follows by repeating the same arguments as in the clean data case, using $C(\tau)$ in Lemma C.13 and Lemma D.2. $\qquad \square$

## D.2. Attention Updates for Not Overfitting Case ($\text{SNR}^2 = \omega(n^{-1})$)

**Lemma D.6** (Signal updates in Lemma C.11)**.** *Let $T_1$ be the time step defined in Lemma C.11, and let $\tau \in [0, T_1]$. Suppose that the conditions in Theorem 4.1 and $A(\tau)$, $B(\tau)$, $D(\tau)$, and $F(\tau)$ in Lemma C.11 are satisfied. Then, on a good run, there exists some constant $c > 0$ such that*

$$\lambda_{+1}(\tau + 1) - \lambda_{+1}(\tau) \geq s_1^{(i)}(\tau)(1 - s_1^{(i)}(\tau)) \cdot c\alpha \|\boldsymbol{\nu}\|_2 \|\boldsymbol{\mu}\|_2^3 d \max\{\sigma_w^2, \sigma_p^2\},$$

$$\lambda_{-1}(\tau + 1) - \lambda_{-1}(\tau) \geq s_1^{(i)}(\tau)(1 - s_1^{(i)}(\tau)) \cdot c\alpha \|\boldsymbol{\nu}\|_2 \|\boldsymbol{\mu}\|_2^3 d \max\{\sigma_w^2, \sigma_p^2\},$$

*for any $i \in [n]$. Additionally, for some constant $c' > c$, we have*

$$\lambda_{+1}(\tau + 1) - \lambda_{+1}(\tau) \leq s_1^{(i)}(\tau)(1 - s_1^{(i)}(\tau)) \cdot c'\alpha \|\boldsymbol{\nu}\|_2 \|\boldsymbol{\mu}\|_2^3 d \max\{\sigma_w^2, \sigma_p^2\},$$

$$\lambda_{-1}(\tau + 1) - \lambda_{-1}(\tau) \leq s_1^{(i)}(\tau)(1 - s_1^{(i)}(\tau)) \cdot c'\alpha \|\boldsymbol{\nu}\|_2 \|\boldsymbol{\mu}\|_2^3 d \max\{\sigma_w^2, \sigma_p^2\},$$

*for any $i \in [n]$.*

*Proof of Lemma D.6.* From Lemma C.4, the update of $\lambda_{+1}$ is given by

$$\lambda_{+1}(\tau + 1) - \lambda_{+1}(\tau) = \frac{\alpha}{n}\sum_{i=1}^{n}(-\ell_i'(\tau)) \cdot Y^{(i)} \cdot \left(I_{i,+}^W(\tau) + \|\mathbf{p}(\tau)\|_2^2 I_{i,+}(\tau)\right) + \alpha^2 \boldsymbol{\mu}_{+1}^\top \nabla_{\mathbf{W}^\top}\widehat{\mathcal{L}}(\tau)\nabla_{\mathbf{p}}\widehat{\mathcal{L}}(\tau). \tag{423}$$

Using $B(\tau)$, together with the parameter assumption $\|\boldsymbol{\mu}\|_2 = \omega(\sigma_\epsilon \log(Tn/\delta))$, we have that for $i \in \mathcal{C}_+ \cup \mathcal{N}_- = \{i \in [n] \mid Y^{*(i)} = 1\}$,

$$(1 - o(1)) \cdot \sigma_w^2 \|\boldsymbol{\mu}\|_2^2 d \leq \langle \mathbf{W}(\tau)\mathbf{x}_1^{(i)}, \mathbf{W}(\tau)\boldsymbol{\mu}_{+1}\rangle \leq (1 + o(1)) \cdot \sigma_w^2 \|\boldsymbol{\mu}\|_2^2 d, \tag{424}$$

and other terms are bounded as follows:

$$|\langle \mathbf{W}(\tau)\mathbf{x}_t^{(i)}, \mathbf{W}(\tau)\boldsymbol{\mu}_{+1}\rangle| = O\left(\rho\sigma_w^2\|\boldsymbol{\mu}\|_2^2 d\right), \tag{425}$$

$$|\langle \mathbf{W}(\tau)\mathbf{x}_u^{(i)}, \mathbf{W}(\tau)\boldsymbol{\mu}_{+1}\rangle| = O\left(\max\left\{\rho\|\boldsymbol{\mu}\|_2^2\sqrt{d}, \sigma_\epsilon\|\boldsymbol{\mu}\|_2 d\right\} \cdot \sigma_w^2 \log(Tn/\delta)\right) = O\left(\rho\sigma_w^2\|\boldsymbol{\mu}\|_2^2 d\right), \tag{426}$$

$$|\langle \mathbf{W}(\tau)\mathbf{x}_v^{(i)}, \mathbf{W}(\tau)\boldsymbol{\mu}_{+1}\rangle| = O\left(\sigma_w^2\sigma_\epsilon\|\boldsymbol{\mu}\|_2 d \log(Tn/\delta)\right) = O\left(\rho\sigma_w^2\|\boldsymbol{\mu}\|_2^2 d\right), \tag{427}$$

for $t \in \mathcal{W}_{+1}^{(i)}$, $u \in \mathcal{W}_{-1}^{(i)}$, and $v \in \mathcal{I}^{(i)}$, which follows from the lower bound of the weak signal strength $\rho$. Additionally, for any $i \in \mathcal{C}_+ \cup \mathcal{N}_-$, we see that $s_1^{(i)}(\tau) \geq \Theta(1)$ holds from $D(\tau)$. Then, we have

$$I_{i,+}^W(\tau) = \sum_{t=1}^T s_t^{(i)}(\tau)\left(\gamma_t^{(i)} - \sum_{u=1}^T s_u^{(i)}(\tau)\gamma_u^{(i)}\right)\langle \mathbf{W}(\tau)\mathbf{x}_t^{(i)}, \mathbf{W}(\tau)\boldsymbol{\mu}_{+1}\rangle \tag{428}$$

$$= \sum_{t=1}^T s_t^{(i)}(\tau) \sum_{u\in[T]\setminus\{t\}} s_u^{(i)}(\tau)\left(\gamma_t^{(i)} - \gamma_u^{(i)}\right)\langle \mathbf{W}(\tau)\mathbf{x}_t^{(i)}, \mathbf{W}(\tau)\boldsymbol{\mu}_{+1}\rangle \tag{429}$$

$$= s_1^{(i)}(\tau)(1 - s_1^{(i)}(\tau)) \cdot \gamma_1^{(i)} \cdot \langle \mathbf{W}(\tau)\mathbf{x}_1^{(i)}, \mathbf{W}(\tau)\boldsymbol{\mu}_{+1}\rangle$$

$$- \sum_{t=2}^T s_1^{(i)}(\tau)s_t^{(i)}(\tau) \cdot \gamma_t^{(i)} \cdot \langle \mathbf{W}(\tau)\mathbf{x}_1^{(i)}, \mathbf{W}(\tau)\boldsymbol{\mu}_{+1}\rangle$$

$$+ \sum_{t=2}^T s_t^{(i)}(\tau)s_1^{(i)}(\tau) \cdot (\gamma_t^{(i)} - \gamma_1^{(i)}) \cdot \langle \mathbf{W}(\tau)\mathbf{x}_t^{(i)}, \mathbf{W}(\tau)\boldsymbol{\mu}_{+1}\rangle$$

$$+ \sum_{t=2}^T s_t^{(i)}(\tau) \sum_{u\in[T]\setminus\{1,t\}} s_u^{(i)}(\tau) \cdot (\gamma_t^{(i)} - \gamma_u^{(i)}) \cdot \langle \mathbf{W}(\tau)\mathbf{x}_t^{(i)}, \mathbf{W}(\tau)\boldsymbol{\mu}_{+1}\rangle \tag{430}$$

$$\geq s_1^{(i)}(\tau)(1 - s_1^{(i)}(\tau)) \cdot \gamma_1^{(i)} \cdot \Theta\left(\sigma_w^2\|\boldsymbol{\mu}\|_2^2 d\right)$$

$$- s_1^{(i)}(\tau)(1 - s_1^{(i)}(\tau)) \cdot O(\rho\gamma_1^{(i)}) \cdot \Theta\left(\sigma_w^2\|\boldsymbol{\mu}\|_2^2 d\right)$$

$$- s_1^{(i)}(\tau)(1 - s_1^{(i)}(\tau)) \cdot (1 + O(\rho))\gamma_1^{(i)} \cdot O\left(\rho\sigma_w^2\|\boldsymbol{\mu}\|_2^2 d\right)$$

$$- \Theta(1) \cdot s_1^{(i)}(\tau)(1 - s_1^{(i)}(\tau)) \cdot O(\rho\gamma_1^{(i)}) \cdot O\left(\rho\sigma_w^2\|\boldsymbol{\mu}\|_2^2 d\right) \tag{431}$$

$$\geq \frac{1}{2} \cdot s_1^{(i)}(\tau)(1 - s_1^{(i)}(\tau)) \cdot \gamma_1^{(i)} \cdot \Theta\left(\sigma_w^2\|\boldsymbol{\mu}\|_2^2 d\right) \tag{432}$$

$$= s_1^{(i)}(\tau)(1 - s_1^{(i)}(\tau)) \cdot c_1'\sigma_w^2\|\boldsymbol{\nu}\|_2\|\boldsymbol{\mu}\|_2^3 d, \tag{433}$$

for a constant $c_1' > 0$. At the last term in the first inequality, we used $\Theta(1) \cdot s_1^{(i)}(\tau) \geq 1 \geq 1 - s_1^{(i)}(\tau) - s_t^{(i)}(\tau)$ from the condition $D(\tau)$. The second inequality follows from Lemma B.12 and $\rho < 1/C$ for sufficiently large $C$. Applying Lemma B.12 provides the last equation. Using the same reasoning, we also have the following upper bound for $i \in \mathcal{C}_+ \cup \mathcal{N}_-$:

$$I_{i,t}^W(\tau) \leq s_1^{(i)}(\tau)(1 - s_1^{(i)}(\tau)) \cdot c_2'\sigma_w^2\|\boldsymbol{\nu}\|_2\|\boldsymbol{\mu}\|_2^3 d, \tag{434}$$

for a constant $c_2' > 0$ such that $c_2' > c_1'$. In a similar way, we will provide the evaluation for $j \in \mathcal{C}_- \cup \mathcal{N}_+ = \{i \in [n] \mid Y^{*(i)} = -1\}$. The condition $B(\tau)$ gives us that

$$|\langle \mathbf{W}(\tau)\mathbf{x}_1^{(j)}, \mathbf{W}(\tau)\boldsymbol{\mu}_{+1}\rangle| = O\left(\max\left\{\|\boldsymbol{\mu}\|_2^2\sqrt{d}, \sigma_\epsilon\|\boldsymbol{\mu}\|_2 d\right\} \cdot \sigma_w^2 \log(Tn/\delta)\right) = O\left(\max\left\{\rho, \frac{\log(Tn/\delta)}{\sqrt{d}}\right\}\sigma_w^2\|\boldsymbol{\mu}\|_2^2 d\right), \tag{435}$$

$$|\langle \mathbf{W}(\tau)\mathbf{x}_t^{(j)}, \mathbf{W}(\tau)\boldsymbol{\mu}_{+1}\rangle| = O\left(\rho\sigma_w^2\|\boldsymbol{\mu}\|_2^2 d\right), \tag{436}$$

$$|\langle \mathbf{W}(\tau)\mathbf{x}_u^{(j)}, \mathbf{W}(\tau)\boldsymbol{\mu}_{+1}\rangle| = O\left(\max\left\{\rho\|\boldsymbol{\mu}\|_2^2\sqrt{d}, \sigma_\epsilon\|\boldsymbol{\mu}\|_2 d\right\} \cdot \sigma_w^2 \log(Tn/\delta)\right) = O\left(\rho\sigma_w^2\|\boldsymbol{\mu}\|_2^2 d\right), \tag{437}$$

$$|\langle \mathbf{W}(\tau)\mathbf{x}_v^{(j)}, \mathbf{W}(\tau)\boldsymbol{\mu}_{+1}\rangle| = O\left(\sigma_w^2\sigma_\epsilon\|\boldsymbol{\mu}\|_2 d \log(Tn/\delta)\right) = O\left(\rho\sigma_w^2\|\boldsymbol{\mu}\|_2^2 d\right), \tag{438}$$

for $t \in \mathcal{W}_{+1}^{(j)}$, $u \in \mathcal{W}_{-1}^{(j)}$, and $v \in \mathcal{I}^{(j)}$. Then, we have

$$|I_{j,+}^W(\tau)| = \left| \sum_{t=1}^T s_t^{(j)}(\tau) \left( \gamma_t^{(j)} - \sum_{u=1}^T s_u^{(j)}(\tau)\gamma_u^{(j)} \right) \langle \mathbf{W}(\tau)\mathbf{x}_t^{(j)}, \mathbf{W}(\tau)\boldsymbol{\mu}_{+1} \rangle \right| \tag{439}$$

$$\leq \sum_{t=1}^T \left| s_t^{(j)}(\tau)(1 - s_t^{(j)}(\tau)) \cdot \max_{u \in [T]}\{\gamma_t^{(j)} - \gamma_u^{(j)}\} \cdot \langle \mathbf{W}(\tau)\mathbf{x}_t^{(j)}, \mathbf{W}(\tau)\boldsymbol{\mu}_{+1} \rangle \right| \tag{440}$$

$$\leq T \max_{t \in [T]}\{s_t^{(j)}(\tau)(1 - s_t^{(j)}(\tau))\} \cdot O\left( \max\left\{ \rho, \frac{\log(Tn/\delta)}{\sqrt{d}} \right\} \sigma_w^2 \|\boldsymbol{\nu}\|_2 \|\boldsymbol{\mu}\|_2^3 d \right) \tag{441}$$

$$\leq s_1^{(j)}(\tau)(1 - s_1^{(j)}(\tau)) \cdot O\left( \max\{\rho, o(1)\} \sigma_w^2 \|\boldsymbol{\nu}\|_2 \|\boldsymbol{\mu}\|_2^3 d \right), \tag{442}$$

where the first inequality is the result of the triangle inequality, and the second line follows from Lemma B.12 and the evaluations just before. In the last line, we used the dominance of $s_1^{(j)}(\tau)(1 - s_1^{(j)}(\tau))$, which follows from $F(\tau)$, and the parameter assumptions $T = \Theta(1)$ and $\sqrt{d} = \omega(\log(Tn/\delta))$.

Using these evaluations, Lemma C.9, Equations (29) and (30), we have

$$\frac{\alpha}{n} \sum_{i=1}^n (-\ell_i'(\tau)) \cdot Y^{(i)} \cdot I_{i,+}^W(\tau)$$

$$= \frac{\alpha}{n} \sum_{i \in \mathcal{C}_+ \cup \mathcal{N}_- \cup (\mathcal{C}_- \cup \mathcal{N}_+)} (-\ell_i'(\tau)) \cdot Y^{(i)} \cdot I_{i,+}^W(\tau) \tag{443}$$

$$\geq \alpha \cdot (2 - 3\eta)/4 \cdot \min_{i \in \mathcal{C}_+}\{(-\ell_i'(\tau))\} \cdot \min_{i \in \mathcal{C}_+}\left\{ s_1^{(i)}(\tau)(1 - s_1^{(i)}(\tau)) \right\} \cdot c_1'\sigma_w^2 \|\boldsymbol{\nu}\|_2 \|\boldsymbol{\mu}\|_2^3 d$$

$$\quad - \alpha \cdot (3\eta)/4 \cdot c_\ell \min_{i \in \mathcal{C}_+}\{(-\ell_i'(\tau))\} \cdot \max_{j \in \mathcal{N}_-}\left\{ s_1^{(j)}(\tau)(1 - s_1^{(j)}(\tau)) \right\} \cdot c_2'\sigma_w^2 \|\boldsymbol{\nu}\|_2 \|\boldsymbol{\mu}\|_2^3 d$$

$$\quad - \alpha \cdot (1 + \eta)/2 \cdot c_\ell \min_{i \in \mathcal{C}_+}\{(-\ell_i'(\tau))\} \cdot \max_{j \in \mathcal{C}_- \cup \mathcal{N}_+}\left\{ s_1^{(j)}(\tau)(1 - s_1^{(j)}(\tau)) \right\} \cdot O\left( \max\{\rho, o(1)\} \sigma_w^2 \|\boldsymbol{\nu}\|_2 \|\boldsymbol{\mu}\|_2^3 d \right) \tag{444}$$

$$\geq s_1^{(i)}(\tau)(1 - s_1^{(i)}(\tau)) \cdot c_3'\alpha\sigma_w^2 \|\boldsymbol{\nu}\|_2 \|\boldsymbol{\mu}\|_2^3 d, \tag{445}$$

for any $i \in [n]$ and some constant $c_3' > 0$. The last inequality follows from the balance of the softmax probabilities over $i \in [n]$, as stated in $F(\tau)$. We also used $\eta < 1/C$ and $\rho < 1/C$ in the parameter assumptions. Similarly, we have

$$\frac{\alpha}{n} \sum_{i=1}^n (-\ell_i'(\tau)) \cdot Y^{(i)} \cdot I_{i,+}^W(\tau) \leq s_1^{(i)}(\tau)(1 - s_1^{(i)}(\tau)) \cdot c_4'\alpha\sigma_w^2 \|\boldsymbol{\nu}\|_2 \|\boldsymbol{\mu}\|_2^3 d, \tag{446}$$

for any $i \in [n]$ and some constant $c_4' > 0$ such that $c_4' > c_3'$.

We now turn to the analysis of $I_{i,+}(\tau)$. Lemma B.1 states that, for $i \in \mathcal{C}_+ \cup \mathcal{N}_- = \{i \in [n] \mid Y^{*(i)} = 1\}$, we have

$$(1 - o(1)) \|\boldsymbol{\mu}\|_2^2 \leq \langle \mathbf{x}_1^{(i)}, \boldsymbol{\mu}_{+1} \rangle \leq (1 + o(1)) \|\boldsymbol{\mu}\|_2^2, \tag{447}$$

and

$$|\langle \mathbf{x}_t^{(i)}, \boldsymbol{\mu}_{+1} \rangle| < \rho\|\boldsymbol{\mu}\|_2^2 + c_2\sigma_\epsilon\|\boldsymbol{\mu}\|_2\sqrt{\log(Tn/\delta)} = O\left( \rho\|\boldsymbol{\mu}\|_2^2 \right), \tag{448}$$

$$|\langle \mathbf{x}_u^{(i)}, \boldsymbol{\mu}_{+1} \rangle| < c_2\sigma_\epsilon\|\boldsymbol{\mu}\|_2\sqrt{\log(Tn/\delta)} = O\left( \rho\|\boldsymbol{\mu}\|_2^2 \right), \tag{449}$$

$$|\langle \mathbf{x}_v^{(i)}, \boldsymbol{\mu}_{+1} \rangle| < c_2\sigma_\epsilon\|\boldsymbol{\mu}\|_2\sqrt{\log(Tn/\delta)} = O\left( \rho\|\boldsymbol{\mu}\|_2^2 \right), \tag{450}$$

for $t \in \mathcal{W}_{+1}^{(i)}$, $u \in \mathcal{W}_{-1}^{(i)}$, and $v \in \mathcal{I}^{(i)}$. Here, we used $\langle \boldsymbol{\mu}_{+1}, \boldsymbol{\mu}_{-1} \rangle = 0$ in our data setup.

Then, for any $i \in \mathcal{C}_+ \cup \mathcal{N}_-$, using a calculation similar to that for $I_{i,+}^W$, there exists a constant $c_5' > 0$ such that

$$I_{i,+}(\tau) = \sum_{t=1}^T s_t^{(i)}(\tau) \left( \gamma_t^{(i)} - \sum_{u=1}^T s_u^{(i)}(\tau)\gamma_u^{(i)} \right) \langle \mathbf{x}_t^{(i)}, \boldsymbol{\mu}_{+1} \rangle \geq s_1^{(i)}(\tau)(1 - s_1^{(i)}(\tau)) \cdot c_5'\|\boldsymbol{\nu}\|_2 \|\boldsymbol{\mu}\|_2^3. \tag{451}$$

Similarly, we have the following upper bound:

$$I_{i,+}(\tau) \leq s_1^{(i)}(\tau)(1 - s_1^{(i)}(\tau)) \cdot c_6' \|\boldsymbol{\nu}\|_2 \|\boldsymbol{\mu}\|_2^3, \tag{452}$$

for some constant $c_6' > 0$ such that $c_6' > c_5'$.

Additionally, using a similar argument as in Equation (442), we give an upper bound for data with different label, i.e., for $j \in \mathcal{C}_- \cup \mathcal{N}_+ = \{i \in [n] \mid Y^{*(i)} = -1\}$, as follows:

$$|I_{j,+}(\tau)| \leq s_1^{(j)}(\tau)(1 - s_1^{(j)}(\tau)) \cdot O\left(\rho \|\boldsymbol{\nu}\|_2 \|\boldsymbol{\mu}\|_2^3\right). \tag{453}$$

Note that the term $\log(Tn/\delta)/\sqrt{d}$ does not appear in this case, due to the orthogonality of $\boldsymbol{\mu}_{+1}$ and $\boldsymbol{\mu}_{-1}$.

Thus, combining these bounds and $(1 - 1/C_1)\,\sigma_p\sqrt{d} \leq \|\mathbf{p}(\tau)\|_2 \leq (1 + 1/C_1)\,\sigma_p\sqrt{d}$ in $A(\tau)$, we have

$$\frac{\alpha}{n} \sum_{i=1}^{n} (-\ell_i'(\tau)) \cdot Y^{(i)} \cdot \|\mathbf{p}(\tau)\|_2^2 I_{i,+}(\tau)$$

$$= \frac{\alpha}{n} \|\mathbf{p}(\tau)\|_2^2 \sum_{i \in \mathcal{C}_+ \cup \mathcal{N}_- \cup (\mathcal{C}_- \cup \mathcal{N}_+)} (-\ell_i'(\tau)) \cdot Y^{(i)} \cdot I_{i,+}(\tau) \tag{454}$$

$$\geq \alpha \|\mathbf{p}(\tau)\|_2^2 \cdot (2 - 3\eta)/4 \cdot \min_{i \in \mathcal{C}_+} \{(-\ell_i'(\tau))\} \cdot \min_{i \in \mathcal{C}_+} \left\{ s_1^{(i)}(\tau)(1 - s_1^{(i)}(\tau)) \right\} \cdot c_5' \|\boldsymbol{\nu}\|_2 \|\boldsymbol{\mu}\|_2^3$$

$$- \alpha \|\mathbf{p}(\tau)\|_2^2 \cdot (3\eta)/4 \cdot c_\ell \min_{i \in \mathcal{C}_+} \{(-\ell_i'(\tau))\} \cdot \max_{j \in \mathcal{N}_-} \left\{ s_1^{(j)}(\tau)(1 - s_1^{(j)}(\tau)) \right\} \cdot c_6' \|\boldsymbol{\nu}\|_2 \|\boldsymbol{\mu}\|_2^3$$

$$- \alpha \|\mathbf{p}(\tau)\|_2^2 \cdot (1 + \eta)/2 \cdot c_\ell \min_{i \in \mathcal{C}_+} \{(-\ell_i'(\tau)\} \cdot \max_{j \in \mathcal{C}_- \cup \mathcal{N}_+} \left\{ s_1^{(j)}(\tau)(1 - s_1^{(j)}(\tau)) \right\} \cdot O\left(\rho \|\boldsymbol{\nu}\|_2 \|\boldsymbol{\mu}\|_2^3\right) \tag{455}$$

$$\geq s_1^{(i)}(\tau)(1 - s_1^{(i)}(\tau)) \cdot c_7' \alpha \sigma_p^2 \|\boldsymbol{\nu}\|_2 \|\boldsymbol{\mu}\|_2^3 d, \tag{456}$$

for any $i \in [n]$ and some constant $c_7' > 0$. We again used the results of $F(\tau)$ and $\eta, \rho < 1/C$ in the parameter assumptions. Similarly, we have the following upper bound:

$$\frac{\alpha}{n} \sum_{i=1}^{n} (-\ell_i'(\tau)) \cdot Y_i \cdot \|\mathbf{p}(\tau)\|_2^2 I_{i,+}(\tau) \leq s_1^{(i)}(\tau)(1 - s_1^{(i)}(\tau)) \cdot c_8' \alpha \sigma_p^2 \|\boldsymbol{\nu}\|_2 \|\boldsymbol{\mu}\|_2^3 d, \tag{457}$$

for any $i \in [n]$ and some constant $c_8' > 0$ such that $c_8' > c_7'$. By substituting Equations (445) and (456) to Equation (423), we have that for any $i \in [n]$,

$$\lambda_{+1}(\tau + 1) - \lambda_{+1}(\tau)$$
$$\geq s_1^{(i)}(\tau)(1 - s_1^{(i)}(\tau)) \cdot \min\{c_3', c_7'\} \alpha \|\boldsymbol{\nu}\|_2 \|\boldsymbol{\mu}\|_2^3 d \max\{\sigma_w^2, \sigma_p^2\} + \alpha^2 \boldsymbol{\mu}_{+1}^\top \nabla_{\mathbf{W}^\top} \widehat{\mathcal{L}}(\tau) \nabla_{\mathbf{p}} \widehat{\mathcal{L}}(\tau). \tag{458}$$

Finally, we show that the second term can be ignored. We have

$$\alpha^2 \boldsymbol{\mu}_{+1}^\top \nabla_{\mathbf{W}^\top} \widehat{\mathcal{L}}(\tau) \nabla_{\mathbf{p}} \widehat{\mathcal{L}}(\tau)$$
$$\leq \alpha^2 \|\nabla_{\mathbf{W}} \widehat{\mathcal{L}}(\tau) \boldsymbol{\mu}_{+1}\|_2 \|\nabla_{\mathbf{p}} \widehat{\mathcal{L}}(\tau)\|_2 \tag{459}$$

$$\lesssim \alpha^2 \left( \max_{i \in [n], t \in [T]} \{ s_t^{(i)}(\tau)(1 - s_t^{(i)}(\tau)) \} \max_{i \in [n], t \in [T]} \{ |\gamma_t^{(i)}| \} \right)^2 \cdot \max_{i \in [n], t \in [n]} \{ \boldsymbol{\mu}_{+1}^\top \mathbf{x}_t^{(i)} \} \|\mathbf{p}(\tau)\|_2 \cdot \max_{i \in [n], t \in [T]} \{ \|\mathbf{W}(\tau) \mathbf{x}_t^{(i)}\|_2 \} \tag{460}$$

$$\lesssim s_1^{(i)}(\tau)(1 - s_1^{(i)}(\tau)) \cdot \alpha^2 \|\boldsymbol{\nu}\|_2^2 \|\boldsymbol{\mu}\|_2^2 \cdot \|\boldsymbol{\mu}\|_2^2 \cdot \sigma_p \sqrt{d} \cdot \sigma_w \max\{\|\boldsymbol{\mu}\|_2 \sqrt{d}, \sigma_\epsilon d\} \tag{461}$$

$$= s_1^{(i)}(\tau)(1 - s_1^{(i)}(\tau)) \cdot \alpha \|\boldsymbol{\nu}\|_2 \|\boldsymbol{\mu}\|_2^3 d \max\{\sigma_w^2, \sigma_p^2\} \cdot \left( \alpha \|\boldsymbol{\nu}\|_2 \|\boldsymbol{\mu}\|_2 \max\{\|\boldsymbol{\mu}\|_2, \sigma_\epsilon \sqrt{d}\} \right) \tag{462}$$

$$= s_1^{(i)}(\tau)(1 - s_1^{(i)}(\tau)) \cdot o\left( \alpha \|\boldsymbol{\nu}\|_2 \|\boldsymbol{\mu}\|_2^3 d \max\{\sigma_w^2, \sigma_p^2\} \right), \tag{463}$$

where the first inequality is the result of the Cauchy-Schwarz inequality, and the second one follows from Equations (9) and (12). Here, the probability term appears in the bound from a similar argument to Equation (442). In the third line, we

used Lemma B.12, the concentration inequalities in $A(\tau)$, $B(\tau)$, and the balance of softmax probabilities in $F(\tau)$. Using $\|\boldsymbol{\nu}\|_2 = O(1/\|\boldsymbol{\mu}\|_2)$ and the learning rate assumption $\alpha = O(\max\{\|\boldsymbol{\mu}\|_2\sqrt{d}, \sigma_\epsilon d\}^{-1})$, the latter part of Equation (462) becomes $o(1)$. Therefore, the quadratic term in Equation (458) can be absorbed in the first term.

Thus, there exists a constant $c > 0$ such that

$$\lambda_{+1}(\tau+1) - \lambda_{+1}(\tau) \geq s_1^{(i)}(\tau)(1 - s_1^{(i)}(\tau)) \cdot c\alpha\|\boldsymbol{\nu}\|_2\|\boldsymbol{\mu}\|_2^3 d \max\{\sigma_w^2, \sigma_p^2\}, \tag{464}$$

for any $i \in [n]$. In the same way, from Equations (446) and (457), there exists a constant $c' > c$ such that

$$\lambda_{+1}(\tau+1) - \lambda_{+1}(\tau) \leq s_1^{(i)}(\tau)(1 - s_1^{(i)}(\tau)) \cdot c'\alpha\|\boldsymbol{\nu}\|_2\|\boldsymbol{\mu}\|_2^3 d \max\{\sigma_w^2, \sigma_p^2\}, \tag{465}$$

for any $i \in [n]$. By applying the same argument to $\lambda_{-1}$, we obtain the desired result. $\square$

Next, we analyze noise memorization in the attention update. Many equations in the proof are similar to those in Lemma D.6, so we provide the proof avoiding the redundant repetition.

**Lemma D.7** (Noise updates in Lemma C.11)**.** *Let $T_1$ be the time step defined in Lemma C.11, and let $\tau \in [0, T_1]$. Suppose that the conditions in Theorem 4.1 and $A(\tau)$, $B(\tau)$, $D(\tau)$, and $F(\tau)$ in Lemma C.11 are satisfied. Then, on a good run, there exists some constant $c > 0$ such that*

$$\rho_{i,1}(\tau+1) - \rho_{i,1}(\tau) \geq s_1^{(i)}(\tau)(1 - s_1^{(i)}(\tau)) \cdot c\alpha n^{-1}\sigma_\epsilon^2 \|\boldsymbol{\nu}\|_2 \|\boldsymbol{\mu}\|_2 d^2 \max\{\sigma_w^2, \sigma_p^2\},$$
$$\rho_{i,t}(\tau+1) - \rho_{i,t}(\tau) \leq -s_1^{(i)}(\tau) s_t^{(i)}(\tau) \cdot c\alpha n^{-1}\sigma_\epsilon^2 \|\boldsymbol{\nu}\|_2 \|\boldsymbol{\mu}\|_2 d^2 \max\{\sigma_w^2, \sigma_p^2\},$$

*for any clean data $i \in \mathcal{C}$ and $t \in [T] \setminus \{1\}$, and*

$$\rho_{j,1}(\tau+1) - \rho_{j,1}(\tau) \leq -s_1^{(j)}(\tau)(1 - s_1^{(j)}(\tau)) \cdot c\alpha n^{-1}\sigma_\epsilon^2 \|\boldsymbol{\nu}\|_2 \|\boldsymbol{\mu}\|_2 d^2 \max\{\sigma_w^2, \sigma_p^2\},$$
$$\rho_{j,t}(\tau+1) - \rho_{j,t}(\tau) \geq s_1^{(j)}(\tau) s_t^{(j)}(\tau) \cdot c\alpha n^{-1}\sigma_\epsilon^2 \|\boldsymbol{\nu}\|_2 \|\boldsymbol{\mu}\|_2 d^2 \max\{\sigma_w^2, \sigma_p^2\},$$

*for any noisy data $j \in \mathcal{N}$, and $t \in [T] \setminus \{1\}$. Additionally, for some constant $c' > c$, we have*

$$\rho_{i,1}(\tau+1) - \rho_{i,1}(\tau) \leq s_1^{(i)}(\tau)(1 - s_1^{(i)}(\tau)) \cdot c'\alpha n^{-1}\sigma_\epsilon^2 \|\boldsymbol{\nu}\|_2 \|\boldsymbol{\mu}\|_2 d^2 \max\{\sigma_w^2, \sigma_p^2\},$$
$$\rho_{i,t}(\tau+1) - \rho_{i,t}(\tau) \geq -s_1^{(i)}(\tau) s_t^{(i)}(\tau) \cdot c'\alpha n^{-1}\sigma_\epsilon^2 \|\boldsymbol{\nu}\|_2 \|\boldsymbol{\mu}\|_2 d^2 \max\{\sigma_w^2, \sigma_p^2\},$$

*for any clean data $i \in \mathcal{C}$ and $t \in [T] \setminus \{1\}$, and*

$$\rho_{j,1}(\tau+1) - \rho_{j,1}(\tau) \geq -s_1^{(j)}(\tau)(1 - s_1^{(j)}(\tau)) \cdot c'\alpha n^{-1}\sigma_\epsilon^2 \|\boldsymbol{\nu}\|_2 \|\boldsymbol{\mu}\|_2 d^2 \max\{\sigma_w^2, \sigma_p^2\},$$
$$\rho_{j,t}(\tau+1) - \rho_{j,t}(\tau) \leq s_1^{(j)}(\tau) s_t^{(j)}(\tau) \cdot c'\alpha n^{-1}\sigma_\epsilon^2 \|\boldsymbol{\nu}\|_2 \|\boldsymbol{\mu}\|_2 d^2 \max\{\sigma_w^2, \sigma_p^2\},$$

*for any noisy data $j \in \mathcal{N}$, and $t \in [T] \setminus \{1\}$.*

*Proof of Lemma D.7.* We first analyze the noise learning in the relevant token of clean data $i \in \mathcal{C}$. From Lemma C.4, we have

$$\rho_{i,1}(\tau+1) - \rho_{i,1}(\tau) = \frac{\alpha}{n}\sum_{k=1}^n (-\ell_k'(\tau)) \cdot Y^{(k)} \cdot \left(I_{k,i,1}^W(\tau) + \|\mathbf{p}(\tau)\|_2^2 I_{k,i,1}(\tau)\right) + \alpha^2 \boldsymbol{\epsilon}_1^{(i)\top}\nabla_{\mathbf{W}^\top}\widehat{\mathcal{L}}(\tau)\nabla_{\mathbf{p}}\widehat{\mathcal{L}}(\tau). \tag{466}$$

Combining Assumptions A1 and A2, we have $d \geq C\hat{\sigma}_\epsilon n \left(C\sigma_\epsilon d^{3/8}\log(Tn/\delta)\right)^{4/3}\log^3(Tn/\delta)$, from which it follows that

$$d \geq (C\sigma_\epsilon)^{14/3}n^2\log^{26/3}(Tn/\delta) = \omega(n^2\log^2(Tn/\delta)). \tag{467}$$

Additionally, from Assumption A1, we have

$$d = \omega\left(\sigma_\epsilon^{-1}n\|\boldsymbol{\mu}\|_2\log(Tn/\delta)\right). \tag{468}$$

Therefore, applying Equations (467) and (468) to the results in $B(\tau)$, we have

$$\|\mathbf{W}(\tau)\boldsymbol{\epsilon}_t^{(i)}\|_2^2 = \Theta(\sigma_w^2\sigma_\epsilon^2 d^2), \tag{469}$$

$$|\langle\mathbf{W}(\tau)\boldsymbol{\mu}_{+1}, \mathbf{W}(\tau)\boldsymbol{\epsilon}_t^{(i)}\rangle| = O(\sigma_w^2\sigma_\epsilon\|\boldsymbol{\mu}\|_2 d\log(Tn/\delta)) = o(n^{-1}\sigma_w^2\sigma_\epsilon^2 d^2), \tag{470}$$

$$|\langle\mathbf{W}(\tau)\boldsymbol{\mu}_{-1}, \mathbf{W}(\tau)\boldsymbol{\epsilon}_t^{(i)}\rangle| = O(\sigma_w^2\sigma_\epsilon\|\boldsymbol{\mu}\|_2 d\log(Tn/\delta)) = o(n^{-1}\sigma_w^2\sigma_\epsilon^2 d^2), \tag{471}$$

$$|\langle\mathbf{W}(\tau)\boldsymbol{\epsilon}_t^{(i)}, \mathbf{W}(\tau)\boldsymbol{\epsilon}_u^{(j)}\rangle| = O(\sigma_w^2\sigma_\epsilon^2 d^{3/2}\log(Tn/\delta)) = o(n^{-1}\sigma_w^2\sigma_\epsilon^2 d^2), \tag{472}$$

for any $i, j \in [n]$ and $t, u \in [T]$ such that $(i,t) \neq (j,u)$. In the case of $k = i$, using a similar argument to Equation (433), we have

$$I_{k,i,1}^W(\tau) = I_{i,i,1}^W(\tau) = \sum_{t=1}^T s_t^{(i)}(\tau)\left(\gamma_t^{(i)} - \sum_{u=1}^T s_u^{(i)}(\tau)\gamma_u^{(i)}\right)\langle\mathbf{W}(\tau)\mathbf{x}_t^{(i)}, \mathbf{W}(\tau)\boldsymbol{\epsilon}_1^{(i)}\rangle \tag{473}$$

$$\geq \frac{1}{2}\cdot s_1^{(i)}(\tau)(1 - s_1^{(i)}(\tau))\cdot\gamma_1^{(i)}\cdot\Theta\left(\sigma_w^2\sigma_\epsilon^2 d^2\right) \tag{474}$$

$$= s_1^{(i)}(\tau)(1 - s_1^{(i)}(\tau))\cdot c_1'\sigma_w^2\sigma_\epsilon^2\|\boldsymbol{\nu}\|_2\|\boldsymbol{\mu}\|_2 d^2, \tag{475}$$

for a constant $c_1' > 0$. This follows from Lemma B.12 and the dominance of Equation (469). Using the same argument, we also have the following upper bound for some constant $c_2' > 0$:

$$I_{k,i,1}^W(\tau) = I_{i,i,1}^W(\tau) \leq s_1^{(i)}(\tau)(1 - s_1^{(i)}(\tau))\cdot c_2'\sigma_w^2\sigma_\epsilon^2\|\boldsymbol{\nu}\|_2\|\boldsymbol{\mu}\|_2 d^2. \tag{476}$$

In contrast, we have that for $k \neq i$,

$$|I_{k,i,1}^W(\tau)| = \left|\sum_{t=1}^T s_t^{(k)}(\tau)\left(\gamma_t^{(k)} - \sum_{u=1}^T s_u^{(k)}(\tau)\gamma_u^{(k)}\right)\langle\mathbf{W}(\tau)\mathbf{x}_t^{(k)}, \mathbf{W}(\tau)\boldsymbol{\epsilon}_1^{(i)}\rangle\right|, \tag{477}$$

$$\leq \sum_{t=1}^T\left|s_t^{(k)}(\tau)(1 - s_t^{(k)}(\tau))\cdot\max_{u\in[T]}\left\{\gamma_t^{(k)} - \gamma_u^{(k)}\right\}\cdot\langle\mathbf{W}(\tau)\mathbf{x}_t^{(k)}, \mathbf{W}(\tau)\boldsymbol{\epsilon}_1^{(i)}\rangle\right|, \tag{478}$$

$$\leq T\max_{t\in[T]}\{s_t^{(k)}(\tau)(1 - s_t^{(k)}(\tau))\}\cdot o\left(n^{-1}\sigma_w^2\sigma_\epsilon^2\|\boldsymbol{\nu}\|_2\|\boldsymbol{\mu}\|_2 d^2\right), \tag{479}$$

$$< s_1^{(k)}(\tau)(1 - s_1^{(k)}(\tau))\cdot o(n^{-1}\sigma_w^2\sigma_\epsilon^2\|\boldsymbol{\nu}\|_2\|\boldsymbol{\mu}\|_2 d^2), \tag{480}$$

where the first inequality is the result of the triangle inequality, and the second one follows from Lemma B.12 and Equations (470) to (472). In the last line, we used the dominance of $s_1^{(k)}(\tau)(1 - s_1^{(k)}(\tau))$ from $F(\tau)$ and the parameter assumption $T = \Theta(1)$. Thus, we have

$$\frac{\alpha}{n}\sum_{k=1}^n(-\ell_k'(\tau))\cdot Y^{(k)}\cdot I_{k,i,1}^W(\tau) > \frac{\alpha}{n}(-\ell_i'(\tau))\cdot s_1^{(i)}(\tau)(1 - s_1^{(i)}(\tau))\cdot c_1'\sigma_w^2\sigma_\epsilon^2\|\boldsymbol{\nu}\|_2\|\boldsymbol{\mu}\|_2 d^2$$

$$- \frac{\alpha}{n}\sum_{k\neq i}(-\ell_k'(\tau))\cdot s_k^{(i)}(\tau)(1 - s_k^{(i)}(\tau))\cdot o\left(n^{-1}\sigma_w^2\sigma_\epsilon^2\|\boldsymbol{\nu}\|_2\|\boldsymbol{\mu}\|_2 d^2\right) \tag{481}$$

$$> s_1^{(i)}(\tau)(1 - s_1^{(i)}(\tau))\cdot c_3'\alpha n^{-1}\sigma_w^2\sigma_\epsilon^2\|\boldsymbol{\nu}\|_2\|\boldsymbol{\mu}\|_2 d^2, \tag{482}$$

for some constant $c_3' > 0$, which follows from the balance of loss derivative and softmax probabilities over training samples, as established in Lemma C.9 and $F(\tau)$. We also have the upper bound for $c_4' > 0$ as follows:

$$\frac{\alpha}{n}\sum_{k=1}^n(-\ell_k'(\tau))\cdot Y^{(k)}\cdot I_{k,i,1}^W(\tau) < s_1^{(i)}(\tau)(1 - s_1^{(i)}(\tau))\cdot c_4'\alpha n^{-1}\sigma_w^2\sigma_\epsilon^2\|\boldsymbol{\nu}\|_2\|\boldsymbol{\mu}\|_2 d^2. \tag{483}$$

Similarly, we evaluate the terms related to $I_{k,i,1}(\tau)$. Lemma B.1 and Equations (467) and (468) lead to

$$\|\boldsymbol{\epsilon}_t^{(i)}\|_2^2 = \Theta(\sigma_\epsilon^2 d), \tag{484}$$

$$|\langle \boldsymbol{\mu}_{+1}, \boldsymbol{\epsilon}_t^{(i)} \rangle| = O(\sigma_\epsilon \|\boldsymbol{\mu}\|_2 \sqrt{\log(Tn/\delta)}) = o(n^{-1}\sigma_\epsilon^2 d), \tag{485}$$

$$|\langle \boldsymbol{\mu}_{-1}, \boldsymbol{\epsilon}_t^{(i)} \rangle| = O(\sigma_\epsilon \|\boldsymbol{\mu}\|_2 \sqrt{\log(Tn/\delta)}) = o(n^{-1}\sigma_\epsilon^2 d), \tag{486}$$

$$|\langle \boldsymbol{\epsilon}_t^{(i)}, \boldsymbol{\epsilon}_u^{(j)} \rangle| = O(\sigma_\epsilon^2 \sqrt{d}\log(Tn/\delta)) = o(n^{-1}\sigma_\epsilon^2 d), \tag{487}$$

for any $i, j \in [n]$ and $t, u \in [T]$ such that $(i,t) \neq (j,u)$. Similarly to Equations (475) and (476), we have that for $k = i$,

$$I_{k,i,1}(\tau) = I_{i,i,1}(\tau) \geq s_1^{(i)}(\tau)(1 - s_1^{(i)}(\tau)) \cdot c_5' \sigma_\epsilon^2 \|\boldsymbol{\nu}\|_2 \|\boldsymbol{\mu}\|_2 d, \tag{488}$$

$$I_{k,i,1}(\tau) = I_{i,i,1}(\tau) \leq s_1^{(i)}(\tau)(1 - s_1^{(i)}(\tau)) \cdot c_6' \sigma_\epsilon^2 \|\boldsymbol{\nu}\|_2 \|\boldsymbol{\mu}\|_2 d, \tag{489}$$

for some constants $c_5', c_6' > 0$. In contrast, by the same argument as in Equation (480), we have that for $k \neq i$,

$$|I_{k,i,1}(\tau)| < s_1^{(k)}(\tau)(1 - s_1^{(k)}(\tau)) \cdot o(n^{-1}\sigma_\epsilon^2 \|\boldsymbol{\nu}\|_2 \|\boldsymbol{\mu}\|_2 d), \tag{490}$$

and similarly to Equations (482) and (483), we have

$$\frac{\alpha}{n} \sum_{k=1}^n (-\ell_k'(\tau)) \cdot Y^{(k)} \cdot I_{k,i,1}(\tau) \geq s_1^{(i)}(\tau)(1 - s_1^{(i)}(\tau)) \cdot c_7' \alpha n^{-1} \sigma_\epsilon^2 \|\boldsymbol{\nu}\|_2 \|\boldsymbol{\mu}\|_2 d, \tag{491}$$

$$\frac{\alpha}{n} \sum_{k=1}^n (-\ell_k'(\tau)) \cdot Y^{(k)} \cdot I_{k,i,1}(\tau) \leq s_1^{(i)}(\tau)(1 - s_1^{(i)}(\tau)) \cdot c_8' \alpha n^{-1} \sigma_\epsilon^2 \|\boldsymbol{\nu}\|_2 \|\boldsymbol{\mu}\|_2 d, \tag{492}$$

for some constants $c_7', c_8' > 0$. Therefore, using Equations (482) and (491), and $\|\mathbf{p}(\tau)\|_2 = \Theta(\sigma_p \sqrt{d})$ in $A(\tau)$, we have

$$\begin{aligned} &\rho_{i,1}(\tau+1) - \rho_{i,1}(\tau) \\ &\geq s_1^{(i)}(\tau)(1 - s_1^{(i)}(\tau)) \cdot \min\{c_3', c_7'\}\alpha n^{-1}\sigma_\epsilon^2 \|\boldsymbol{\nu}\|_2 \|\boldsymbol{\mu}\|_2 d^2 \max\{\sigma_w^2, \sigma_p^2\} + \alpha^2 \boldsymbol{\epsilon}_1^{(i)\top} \nabla_{\mathbf{W}^\top} \widehat{\mathcal{L}}(\tau) \nabla_{\mathbf{p}} \widehat{\mathcal{L}}(\tau). \end{aligned} \tag{493}$$

For the quadratic term in Equation (493), a similar argument to that in Equation (463) shows that it can be neglected under the small learning rate assumption. Thus, by appropriately redefining the constants, we obtain the desired lower bound for $\rho_{i,1}$. The upper bound is derived in the same way.

We now turn to the noise memorization term for $t \in [T] \setminus \{1\}$ of clean data $i \in \mathcal{C}$. Since the proof is essentially the same as for $\rho_{i,1}(\tau)$, we show only the different parts to avoid repetition. From Lemma C.4, we have

$$\rho_{i,t}(\tau+1) - \rho_{i,t}(\tau) = \frac{\alpha}{n} \sum_{k=1}^n (-\ell_k'(\tau)) \cdot Y^{(k)} \cdot \left( I_{k,i,t}^W(\tau) + \|\mathbf{p}(\tau)\|_2^2 I_{k,i,t}(\tau) \right) + \alpha^2 \boldsymbol{\epsilon}_t^{(i)\top} \nabla_{\mathbf{W}^\top} \widehat{\mathcal{L}}(\tau) \nabla_{\mathbf{p}} \widehat{\mathcal{L}}(\tau). \tag{494}$$

When $k = i$, we have

$$I_{k,i,t}^W(\tau) = I_{i,i,t}^W(\tau) = \sum_{v=1}^{T} s_v^{(i)}(\tau) \left( \gamma_v^{(i)} - \sum_{u=1}^{T} s_u^{(i)}(\tau) \gamma_u^{(i)} \right) \langle \mathbf{W}(\tau)\mathbf{x}_v^{(i)}, \mathbf{W}(\tau)\boldsymbol{\epsilon}_t^{(i)} \rangle \tag{495}$$

$$= s_1^{(i)}(\tau)(1 - s_1^{(i)}(\tau)) \cdot \gamma_1^{(i)} \cdot \langle \mathbf{W}(\tau)\mathbf{x}_1^{(i)}, \mathbf{W}(\tau)\boldsymbol{\epsilon}_t^{(i)} \rangle$$

$$- \sum_{u=2}^{T} s_1^{(i)}(\tau)s_u^{(i)}(\tau) \cdot \gamma_u^{(i)} \cdot \langle \mathbf{W}(\tau)\mathbf{x}_1^{(i)}, \mathbf{W}(\tau)\boldsymbol{\epsilon}_t^{(i)} \rangle$$

$$+ \sum_{v=2}^{T} s_v^{(i)}(\tau)s_1^{(i)}(\tau) \cdot (\gamma_v^{(i)} - \gamma_1^{(i)}) \cdot \langle \mathbf{W}(\tau)\mathbf{x}_v^{(i)}, \mathbf{W}(\tau)\boldsymbol{\epsilon}_t^{(i)} \rangle$$

$$+ \sum_{v=2}^{T} s_v^{(i)}(\tau) \sum_{u \in [T] \setminus \{1,v\}} s_u^{(i)}(\tau) \cdot (\gamma_v^{(i)} - \gamma_u^{(i)}) \cdot \langle \mathbf{W}(\tau)\mathbf{x}_v^{(i)}, \mathbf{W}(\tau)\boldsymbol{\epsilon}_t^{(i)} \rangle \tag{496}$$

$$\leq s_1^{(i)}(\tau)(1 - s_1^{(i)}(\tau)) \cdot \gamma_1^{(i)} \cdot o(n^{-1}\sigma_w^2\sigma_\epsilon^2 d^2)$$

$$+ s_1^{(i)}(\tau)(1 - s_1^{(i)}(\tau)) \cdot O(\rho\gamma_1^{(i)}) \cdot o(n^{-1}\sigma_w^2\sigma_\epsilon^2 d^2)$$

$$+ \Bigg( - s_1^{(i)}(\tau)s_t^{(i)}(\tau) \cdot (1 - O(\rho))\, \gamma_1^{(i)} \cdot \Theta(\sigma_w^2\sigma_\epsilon^2 d^2)$$

$$+ s_1^{(i)}(\tau)(1 - s_1^{(i)}(\tau) - s_t^{(i)}(\tau)) \cdot (1 + O(\rho))\, \gamma_1^{(i)} \cdot o(n^{-1}\sigma_w^2\sigma_\epsilon^2 d^2) \Bigg)$$

$$+ \Bigg( s_t^{(i)}(\tau)(1 - s_1^{(i)}(\tau) - s_t^{(i)}(\tau)) \cdot O(\rho\gamma_1^{(i)}) \cdot \Theta(\sigma_w^2\sigma_\epsilon^2 d^2)$$

$$+ \Theta(1) \cdot s_1^{(i)}(\tau)(1 - s_1^{(i)}(\tau)) \cdot O(\rho\gamma_1^{(i)}) \cdot o(n^{-1}\sigma_w^2\sigma_\epsilon^2 d^2) \Bigg) \tag{497}$$

$$\leq -s_1^{(i)}(\tau)s_t^{(i)}(\tau) \cdot c_9'\sigma_w^2\sigma_\epsilon^2 \|\boldsymbol{\nu}\|_2 \|\boldsymbol{\mu}\|_2 d^2, \tag{498}$$

for a constant $c_9' > 0$. The first inequality follows from Equations (469) to (472). In the second last line, we used $\Theta(1) \cdot s_1^{(i)}(\tau) \geq 1 \geq 1 - s_t^{(i)}(\tau)$ from the condition $D(\tau)$. The last line follows from the comparison between $s_t^{(i)}(\tau)$ and $1 - s_1^{(i)}(\tau)$, as shown in $F(\tau)$, Lemma B.12, and the parameter assumption $\rho < 1/C$ for sufficiently large $C$.

Using the same argument, we also have the following lower bound for some constant $c_{10}' > 0$:

$$I_{k,i,t}^W \geq -s_1^{(i)}(\tau)s_t^{(i)}(\tau) \cdot c_{10}'\sigma_w^2\sigma_\epsilon^2 \|\boldsymbol{\nu}\|_2 \|\boldsymbol{\mu}\|_2 d^2. \tag{499}$$

From a similar argument, we have that for $k = i$,

$$I_{k,i,t}(\tau) = I_{i,i,t}(\tau) \leq -s_1^{(i)}(\tau)s_t^{(i)}(\tau) \cdot c_{11}'\sigma_\epsilon^2 \|\boldsymbol{\nu}\|_2 \|\boldsymbol{\mu}\|_2 d, \tag{500}$$

$$I_{k,i,t}(\tau) = I_{i,i,t}(\tau) \geq -s_1^{(i)}(\tau)s_t^{(i)}(\tau) \cdot c_{12}'\sigma_\epsilon^2 \|\boldsymbol{\nu}\|_2 \|\boldsymbol{\mu}\|_2 d, \tag{501}$$

for constants $c_{11}', c_{12}' > 0$.

For $k \neq i$, $|I_{k,i,t}^W(\tau)|$ and $|I_{k,i,t}(\tau)|$ are bounded as discussed in Equations (480) and (490). Thus, using $\|\mathbf{p}(\tau)\|_2 = \Theta(\sigma_p\sqrt{d})$ in $A(\tau)$, we have

$$\rho_{i,t}(\tau + 1) - \rho_{i,t}(\tau)$$

$$\leq -s_1^{(i)}(\tau)s_t^{(i)}(\tau) \cdot \min\{c_9', c_{11}'\}\alpha n^{-1}\sigma_\epsilon^2 \|\boldsymbol{\nu}\|_2 \|\boldsymbol{\mu}\|_2 d^2 \max\{\sigma_w^2, \sigma_p^2\} + \alpha^2 \boldsymbol{\epsilon}_t^{(i)\top} \nabla_{\mathbf{W}^\top}\widehat{\mathcal{L}}(\tau) \nabla_{\mathbf{p}}\widehat{\mathcal{L}}(\tau). \tag{502}$$

We can show that the quadratic term can be ignored as in the case of $\rho_{i,1}(\tau)$. Consequently, there exists constants $c, c' > 0$ such that

$$\rho_{i,t}(\tau + 1) - \rho_{i,t}(\tau) \leq -s_1^{(i)}(\tau)s_t^{(i)}(\tau) \cdot c\alpha n^{-1}\sigma_\epsilon^2 \|\boldsymbol{\nu}\|_2 \|\boldsymbol{\mu}\|_2 d^2 \max\{\sigma_w^2, \sigma_p^2\}, \tag{503}$$

$$\rho_{i,t}(\tau + 1) - \rho_{i,t}(\tau) \geq -s_1^{(i)}(\tau)s_t^{(i)}(\tau) \cdot c'\alpha n^{-1}\sigma_\epsilon^2 \|\boldsymbol{\nu}\|_2 \|\boldsymbol{\mu}\|_2 d^2 \max\{\sigma_w^2, \sigma_p^2\}. \tag{504}$$

So far, we have discussed the updates for clean data. For noisy data $j \in \mathcal{N}$, the update equations differ only in the sign due to the flipping of $Y$. Thus, we conclude the proof. $\qquad\square$

## D.3. Attention Updates for Benign Overfitting Case ($\text{SNR}^2 = o(n^{-1})$)

### D.3.1. ANALYSIS FOR STAGE 1

We first present a lemma on the signal update under the current SNR setting. Since most of the discussion overlaps with Lemma D.6 for the not-overfitting case, particularly in basic concentration inequalities and equality evaluations, we only present the different parts. The main difference from Lemma D.6 lies in the behavior of noisy data, which shows a monotonic decrease in $s_1(\tau)$ rather than a monotonic increase as in the clean data.

**Lemma D.8** (Signal updates in Lemma C.12). *Let $T_1$ be the time step defined in Lemma C.12, and let $\tau \in [0, T_1]$. Suppose that the conditions in Theorem 4.1 and $A(\tau)$, $B(\tau)$, $D(\tau)$, and $F(\tau)$ in Lemma C.12 are satisfied. Then, on a good run, there exists some constant $c > 0$ such that*

$$\lambda_{+1}(\tau + 1) - \lambda_{+1}(\tau) \geq s_1^{(i)}(\tau)(1 - s_1^{(i)}(\tau)) \cdot c\alpha\|\boldsymbol{\nu}\|_2 \|\boldsymbol{\mu}\|_2^3 d \max\{\sigma_w^2, \sigma_p^2\},$$

$$\lambda_{-1}(\tau + 1) - \lambda_{-1}(\tau) \geq s_1^{(i)}(\tau)(1 - s_1^{(i)}(\tau)) \cdot c\alpha\|\boldsymbol{\nu}\|_2 \|\boldsymbol{\mu}\|_2^3 d \max\{\sigma_w^2, \sigma_p^2\},$$

*for any $i \in \mathcal{C}$. Additionally, for some constant $c' > c$, we have*

$$\lambda_{+1}(\tau + 1) - \lambda_{+1}(\tau) \leq s_1^{(i)}(\tau)(1 - s_1^{(i)}(\tau)) \cdot c'\alpha\|\boldsymbol{\nu}\|_2 \|\boldsymbol{\mu}\|_2^3 d \max\{\sigma_w^2, \sigma_p^2\},$$

$$\lambda_{-1}(\tau + 1) - \lambda_{-1}(\tau) \leq s_1^{(i)}(\tau)(1 - s_1^{(i)}(\tau)) \cdot c'\alpha\|\boldsymbol{\nu}\|_2 \|\boldsymbol{\mu}\|_2^3 d \max\{\sigma_w^2, \sigma_p^2\},$$

*for any $i \in \mathcal{C}$.*

*Proof of Lemma D.8.* The full proof is omitted because it follows from the reasoning similar to that in Lemma D.6. In the analysis of Equation (423), for clean data, the conditions $A(\tau)$, $B(\tau)$, and $D(\tau)$ yield the same results as in Equations (433) and (451). For noisy data, extra care is required because $s_1^{(j)}(\tau) > \Theta(1)$ does not hold; however, noting that $s_1^{(j)}(\tau) > \Theta(\rho)$ holds from $D(\tau)$, the same evaluation as in Equation (433) is obtained as follows. Let $j \in \mathcal{N}_-$, and we have

$$I_{j,+}^W(\tau) = \sum_{t=1}^{T} s_t^{(j)}(\tau) \left( \gamma_t^{(j)} - \sum_{u=1}^{T} s_u^{(j)}(\tau)\gamma_u^{(j)} \right) \langle \mathbf{W}(\tau)\mathbf{x}_t^{(j)}, \mathbf{W}(\tau)\boldsymbol{\mu}_{+1} \rangle \tag{505}$$

$$= s_1^{(j)}(\tau)(1 - s_1^{(j)}(\tau)) \cdot \gamma_1^{(j)} \cdot \langle \mathbf{W}(\tau)\mathbf{x}_1^{(j)}, \mathbf{W}(\tau)\boldsymbol{\mu}_{+1} \rangle$$

$$- \sum_{t=2}^{T} s_1^{(j)}(\tau)s_t^{(j)}(\tau) \cdot \gamma_t^{(j)} \cdot \langle \mathbf{W}(\tau)\mathbf{x}_1^{(j)}, \mathbf{W}(\tau)\boldsymbol{\mu}_{+1} \rangle$$

$$+ \sum_{t=2}^{T} s_t^{(j)}(\tau)s_1^{(j)}(\tau) \cdot (\gamma_t^{(j)} - \gamma_1^{(j)}) \cdot \langle \mathbf{W}(\tau)\mathbf{x}_t^{(j)}, \mathbf{W}(\tau)\boldsymbol{\mu}_{+1} \rangle$$

$$+ \sum_{t=2}^{T} s_t^{(j)}(\tau) \sum_{u \in [T]\setminus\{1,t\}} s_u^{(j)}(\tau) \cdot (\gamma_t^{(j)} - \gamma_u^{(j)}) \cdot \langle \mathbf{W}(\tau)\mathbf{x}_t^{(j)}, \mathbf{W}(\tau)\boldsymbol{\mu}_{+1} \rangle \tag{506}$$

$$\geq s_1^{(j)}(\tau)(1 - s_1^{(j)}(\tau)) \cdot \gamma_1^{(j)} \cdot \Theta\left( \sigma_w^2 \|\boldsymbol{\mu}\|_2^2 d \right)$$

$$- s_1^{(j)}(\tau)(1 - s_1^{(j)}(\tau)) \cdot O(\rho\gamma_1^{(j)}) \cdot \Theta\left( \sigma_w^2 \|\boldsymbol{\mu}\|_2^2 d \right)$$

$$- s_1^{(j)}(\tau)(1 - s_1^{(j)}(\tau)) \cdot (1 + O(\rho)) \gamma_1^{(j)} \cdot O\left( \rho\sigma_w^2 \|\boldsymbol{\mu}\|_2^2 d \right)$$

$$- \Theta(\rho^{-1}) \cdot s_1^{(j)}(\tau)(1 - s_1^{(j)}(\tau)) \cdot O(\rho\gamma_1^{(j)}) \cdot O\left( \rho\sigma_w^2 \|\boldsymbol{\mu}\|_2^2 d \right) \tag{507}$$

$$\geq s_1^{(j)}(\tau)(1 - s_1^{(j)}(\tau)) \cdot c_1'\sigma_w^2 \|\boldsymbol{\nu}\|_2 \|\boldsymbol{\mu}\|_2^3 d, \tag{508}$$

where the fourth term of Equation (507) is applied the result of $D(\tau)$, and since both the token score and the inner-product term have a small order, this term is negligible compared to the first term. Therefore, the same evaluation is obtained as in the clean data case. The same argument is applied to $I_{j,+}(\tau)$. Regarding data from the different class, i.e., $j \in [n]$ such that

$Y^{*(j)} = -1$, the influence can be bounded similarly to Equations (442) and (452). Here, the influence of noisy data $j \in \mathcal{N}_+$ can be shown to be in the same order as that of clean data, using a similar argument to Equation (269). For example, the part corresponding Equation (442) becomes the following:

$$|I_{j,+}^W(\tau)| = \left| \sum_{t=1}^T s_t^{(j)}(\tau) \left( \gamma_t^{(j)} - \sum_{u=1}^T s_u^{(j)}(\tau)\gamma_u^{(j)} \right) \langle \mathbf{W}(\tau)\mathbf{x}_t^{(j)}, \mathbf{W}(\tau)\boldsymbol{\mu}_{+1} \rangle \right| \tag{509}$$

$$\lesssim \left( s_1^{(j)}(\tau)(1 - s_1^{(j)}(\tau)) \cdot |\gamma_1^{(j)}| + \max_{t \in [T] \setminus \{1\}} \{|\gamma_t^{(j)}|\} \right) \cdot O\left( \max\{\rho, o(1)\} \sigma_w^2 \|\boldsymbol{\mu}\|_2^2 d \right) \tag{510}$$

$$\lesssim s_1^{(i)}(\tau)(1 - s_1^{(i)}(\tau)) \cdot O\left( \max\{\rho, o(1)\} \sigma_w^2 \|\boldsymbol{\nu}\|_2 \|\boldsymbol{\mu}\|_2^3 d \right), \tag{511}$$

for any $i \in \mathcal{C}$. Here, we used Lemma B.12, the fact that $\rho^{-1} \cdot s_1^{(i)}(\tau)(1 - s_1^{(i)}(\tau)) > \Theta(1)$ from $D(\tau)$, and that the balance of $s_1^{(k)}(\tau)(1 - s_1^{(k)}(\tau))$ holds over all examples $k \in [n]$ as provided in $F(\tau)$. Consequently, the desired result follows from the balance between the number of clean and noisy data. $\qquad \square$

The following lemma on noise memorization is basically the same as in the not-overfitting case, but the behavior of the noisy samples $j \in \mathcal{N}$ differs. In particular, $\rho_{j,t}(\tau)$ for $t \in [T] \setminus \{1, 2\}$ initially evolves in the same manner as $\rho_{j,2}(\tau)$ but gradually decreases, changing sign around $T_1$. As a result, we obtain the following evaluation that spans both positive and negative values.

**Lemma D.9** (Noise updates in Lemma C.12). *Let $T_1$ be the time step defined in Lemma C.12, and let $\tau \in [0, T_1]$. Suppose that the conditions in Theorem 4.1 and $A(\tau)$, $B(\tau)$, $D(\tau)$, and $F(\tau)$ in Lemma C.12 are satisfied. Then, on a good run, there exists some constant $c, c' > 0$ such that $c' > c$, and*

$$\rho_{i,1}(\tau+1) - \rho_{i,1}(\tau) \geq s_1^{(i)}(\tau)(1 - s_1^{(i)}(\tau)) \cdot c\alpha n^{-1}\sigma_\epsilon^2 \|\boldsymbol{\nu}\|_2 \|\boldsymbol{\mu}\|_2 d^2 \max\{\sigma_w^2, \sigma_p^2\},$$

$$\rho_{i,t}(\tau+1) - \rho_{i,t}(\tau) \leq -s_1^{(i)}(\tau)s_t^{(i)}(\tau) \cdot c\alpha n^{-1}\sigma_\epsilon^2 \|\boldsymbol{\nu}\|_2 \|\boldsymbol{\mu}\|_2 d^2 \max\{\sigma_w^2, \sigma_p^2\},$$

$$\rho_{i,1}(\tau+1) - \rho_{i,1}(\tau) \leq s_1^{(i)}(\tau)(1 - s_1^{(i)}(\tau)) \cdot c'\alpha n^{-1}\sigma_\epsilon^2 \|\boldsymbol{\nu}\|_2 \|\boldsymbol{\mu}\|_2 d^2 \max\{\sigma_w^2, \sigma_p^2\},$$

$$\rho_{i,t}(\tau+1) - \rho_{i,t}(\tau) \geq -s_1^{(i)}(\tau)s_t^{(i)}(\tau) \cdot c'\alpha n^{-1}\sigma_\epsilon^2 \|\boldsymbol{\nu}\|_2 \|\boldsymbol{\mu}\|_2 d^2 \max\{\sigma_w^2, \sigma_p^2\},$$

*for any clean data $i \in \mathcal{C}$ and $t \in [T] \setminus \{1\}$. For any noisy data $j \in \mathcal{N}$, we have*

$$\rho_{j,1}(\tau+1) - \rho_{j,1}(\tau) \leq -s_1^{(j)}(\tau)(1 - s_1^{(j)}(\tau)) \cdot c\alpha n^{-1}\sigma_\epsilon^2 \|\boldsymbol{\nu}\|_2 \|\boldsymbol{\mu}\|_2 d^2 \max\{\sigma_w^2, \sigma_p^2\},$$

$$\rho_{j,1}(\tau+1) - \rho_{j,1}(\tau) \geq -s_1^{(j)}(\tau)(1 - s_1^{(j)}(\tau)) \cdot c'\alpha n^{-1}\sigma_\epsilon^2 \|\boldsymbol{\nu}\|_2 \|\boldsymbol{\mu}\|_2 d^2 \max\{\sigma_w^2, \sigma_p^2\},$$

*and*

$$\rho_{j,2}(\tau+1) - \rho_{j,2}(\tau) \geq s_1^{(j)}(\tau)s_t^{(j)}(\tau) \cdot c\alpha n^{-1}\sigma_\epsilon^2 \|\boldsymbol{\nu}\|_2 \|\boldsymbol{\mu}\|_2 d^2 \max\{\sigma_w^2, \sigma_p^2\},$$

$$\rho_{j,2}(\tau+1) - \rho_{j,2}(\tau) \leq s_1^{(j)}(\tau)s_t^{(j)}(\tau) \cdot c'\alpha n^{-1}\sigma_\epsilon^2 \|\boldsymbol{\nu}\|_2 \|\boldsymbol{\mu}\|_2 d^2 \max\{\sigma_w^2, \sigma_p^2\},$$

*and*

$$\rho_{j,t}(\tau+1) - \rho_{j,t}(\tau) \geq -s_1^{(j)}(\tau)s_t^{(j)}(\tau) \cdot c\alpha n^{-1}\sigma_\epsilon^2 \|\boldsymbol{\nu}\|_2 \|\boldsymbol{\mu}\|_2 d^2 \max\{\sigma_w^2, \sigma_p^2\},$$

$$\rho_{j,t}(\tau+1) - \rho_{j,t}(\tau) \leq s_1^{(j)}(\tau)s_t^{(j)}(\tau) \cdot c'\alpha n^{-1}\sigma_\epsilon^2 \|\boldsymbol{\nu}\|_2 \|\boldsymbol{\mu}\|_2 d^2 \max\{\sigma_w^2, \sigma_p^2\},$$

*for any $t \in [T] \setminus \{1, 2\}$.*

*Proof of Lemma D.9.* For clean data $i \in \mathcal{C}$, the same reasoning as in Lemma D.7 is applied using the conditions of the lemma. Here, the influence of noisy samples in the summation can be treated in the same way as in the not-overfitting case, by applying the same argument as in Equation (511) in the proof of Lemma C.12.

For $\rho_{j,1}(\tau), j \in \mathcal{N}$, we can reduce the analysis to the case of clean data, as in Equation (508), using $s_1^{(j)}(\tau) > \Theta(\rho)$. In the rest of the proof, we analyze the update of $\rho_{j,t}(\tau)$ for $t \in [T] \setminus \{1\}$. From Lemma C.4, we have

$$\rho_{j,t}(\tau+1) - \rho_{j,t}(\tau) = \frac{\alpha}{n} \sum_{k=1}^n (-\ell_k'(\tau)) \cdot Y^{(k)} \cdot \left( I_{k,j,t}^W(\tau) + \|\mathbf{p}(\tau)\|_2^2 I_{k,j,t}(\tau) \right) + \alpha^2 \boldsymbol{\epsilon}_t^{(j)\top} \nabla_{\mathbf{W}^\top} \widehat{\mathcal{L}}(\tau) \nabla_{\mathbf{p}} \widehat{\mathcal{L}}(\tau). \tag{512}$$

Without loss of generality, we consider $j \in \mathcal{N}_+ = \{i \in [n] \mid Y^{*(i)} = -1, Y^{(i)} = 1\}$. Then, from the data model defined in Definition 3.1 and Lemma B.12, we have

$$\gamma_1^{(j)} = -\Theta(\|\boldsymbol{\nu}\|_2 \|\boldsymbol{\mu}\|_2), \quad \gamma_2^{(j)} = \Theta(\rho \|\boldsymbol{\nu}\|_2 \|\boldsymbol{\mu}\|_2), \tag{513}$$

$$\gamma_t^{(j)} = -\Theta(\rho \|\boldsymbol{\nu}\|_2 \|\boldsymbol{\mu}\|_2), \quad |\gamma_u^{(j)}| = O\left(\sigma_\epsilon \|\boldsymbol{\nu}\|_2 \sqrt{\log(Tn/\delta)}\right), \tag{514}$$

for $t \in \mathcal{W}_{-1}^{(j)}$ and $u \in \mathcal{I}^{(j)}$. When $k = j$, using Equations (469) to (472), we have

$$I_{k,j,t}^W(\tau) = I_{j,j,t}^W(\tau) = \sum_{v=1}^T s_v^{(j)}(\tau) \left(\gamma_v^{(j)} - \sum_{u=1}^T s_u^{(j)}(\tau)\gamma_u^{(j)}\right) \langle \mathbf{W}(\tau)\mathbf{x}_v^{(j)}, \mathbf{W}(\tau)\boldsymbol{\epsilon}_t^{(j)}\rangle \tag{515}$$

$$= s_1^{(j)}(\tau)(1 - s_1^{(j)}(\tau)) \cdot \gamma_1^{(i)} \cdot \langle \mathbf{W}(\tau)\mathbf{x}_1^{(j)}, \mathbf{W}(\tau)\boldsymbol{\epsilon}_t^{(j)}\rangle$$

$$- \sum_{u=2}^T s_1^{(j)}(\tau)s_u^{(j)}(\tau) \cdot \gamma_u^{(j)} \cdot \langle \mathbf{W}(\tau)\mathbf{x}_1^{(j)}, \mathbf{W}(\tau)\boldsymbol{\epsilon}_t^{(j)}\rangle$$

$$+ \sum_{v=2}^T s_v^{(j)}(\tau)s_1^{(j)}(\tau) \cdot (\gamma_v^{(j)} - \gamma_1^{(j)}) \cdot \langle \mathbf{W}(\tau)\mathbf{x}_v^{(j)}, \mathbf{W}(\tau)\boldsymbol{\epsilon}_t^{(j)}\rangle$$

$$+ \sum_{v=2}^T s_v^{(j)}(\tau) \sum_{u \in [T]\setminus\{1,v\}} s_u^{(j)}(\tau) \cdot (\gamma_v^{(j)} - \gamma_u^{(j)}) \cdot \langle \mathbf{W}(\tau)\mathbf{x}_v^{(j)}, \mathbf{W}(\tau)\boldsymbol{\epsilon}_t^{(j)}\rangle \tag{516}$$

$$\geq -s_1^{(j)}(\tau)(1 - s_1^{(j)}(\tau)) \cdot o\left(\|\boldsymbol{\nu}\|_2 \|\boldsymbol{\mu}\|_2 \cdot n^{-1}\sigma_w^2\sigma_\epsilon^2 d^2\right)$$

$$- s_1^{(j)}(\tau)(1 - s_1^{(j)}(\tau)) \cdot o\left(\rho \|\boldsymbol{\nu}\|_2 \|\boldsymbol{\mu}\|_2 \cdot n^{-1}\sigma_w^2\sigma_\epsilon^2 d^2\right)$$

$$+ s_1^{(j)}(\tau) \cdot s_t^{(j)}(\tau) \cdot \Theta(\|\boldsymbol{\nu}\|_2 \|\boldsymbol{\mu}\|_2 \cdot \sigma_w^2\sigma_\epsilon^2 d^2) \tag{517}$$

$$- s_1^{(j)}(\tau)(1 - s_1^{(j)}(\tau) - s_t^{(j)}(\tau)) \cdot o\left(\rho \|\boldsymbol{\nu}\|_2 \|\boldsymbol{\mu}\|_2 \cdot n^{-1}\sigma_w^2\sigma_\epsilon^2 d^2\right)$$

$$+ s_t^{(j)}(\tau) \sum_{u \in [T]\setminus\{1,t\}} s_u^{(j)}(\tau)(\gamma_t^{(j)} - \gamma_u^{(j)}) \cdot \Theta(\sigma_w^2\sigma_\epsilon^2 d^2) \tag{518}$$

$$- (1 - s_1^{(j)}(\tau) - s_t^{(j)}(\tau))^2 \cdot o\left(\rho \|\boldsymbol{\nu}\|_2 \|\boldsymbol{\mu}\|_2 \cdot n^{-1}\sigma_w^2\sigma_\epsilon^2 d^2\right). \tag{519}$$

For $t = 2$, Equation (518) becomes positive from the evaluation of token scores, and combining this with Equation (517) and $s_1^{(j)}(\tau) \geq \Theta(\rho)$ leads to

$$I_{j,j,t}^W(\tau) = I_{j,j,2}^W(\tau) \geq s_1^{(j)}(\tau)s_t^{(j)}(\tau) \cdot c_1' \sigma_w^2 \sigma_\epsilon^2 \|\boldsymbol{\nu}\|_2 \|\boldsymbol{\mu}\|_2 d. \tag{520}$$

The upper bound and the evaluation for $I_{j,j,2}(\tau)$ follow similarly. For $t \in [T] \setminus \{1,2\}$, Equation (518) is bounded from below by $-s_t^{(j)}(\tau)(1 - s_1^{(j)}(\tau) - s_t^{(j)}(\tau)) \cdot O(\rho \|\boldsymbol{\mu}\| \|\boldsymbol{\mu}\|_2 \cdot \sigma_w^2 \sigma_\epsilon^2 d^2)$. Since $s_1^{(j)}(\tau) \geq \Theta(\rho)$ from $D(\tau)$, we choose $T_1$ such that Equation (518) does not exceed Equation (517) too much. Specifically, we choose it so that we have

$$I_{j,j,t}^W(\tau) \geq -s_1^{(j)}(\tau)s_t^{(j)}(\tau) \cdot c_1' \sigma_w^2 \sigma_\epsilon^2 \|\boldsymbol{\nu}\|_2 \|\boldsymbol{\mu}\|_2 d. \tag{521}$$

The upper bound is derived trivially. The discussion for $I_{k,j,t}^W(\tau)$ and $I_{k,j,t}(\tau)$ for $k \neq j$ is proceeded in a similar way to Equation (511), and we conclude the proof by repeating the argument in Lemma D.7. $\qquad\square$

### D.3.2. ANALYSIS FOR STAGE 2

Next, we provide the results for Stage 2, i.e., $\tau \in [T_1, T_2]$ in the proof of Lemma C.13. In Stage 2, the signal updates are dominated by the $s_1^{(i)}(\tau)(1 - s^{(i)}(\tau))$ for $i \in \mathcal{C}$, as well as Stage 1 (Lemma D.10). The different part from Stage 1 is the behavior of noise memorization of noisy data, and the learning progresses in such a way that the confusing weakly relevant token, i.e., $\mathbf{x}_2^{(j)}$, would be selected (Lemma D.11).

**Lemma D.10** (Signal updates in Lemma C.13). *Let $T_1$ and $T_2$ be the time steps in Lemma C.13, and let $\tau \in [T_1, T_2]$. Suppose that the conditions in Theorem 4.1 and $A(\tau)$, $B(\tau)$, $D(\tau)$, and $F(\tau)$ in Lemma C.13 are satisfied. Then, on a good run, there exists some constant $c > 0$ such that*

$$\lambda_{+1}(\tau + 1) - \lambda_{+1}(\tau) \geq s_1^{(i)}(\tau)(1 - s_1^{(i)}(\tau)) \cdot c\alpha\|\boldsymbol{\nu}\|_2\|\boldsymbol{\mu}\|_2^3 d \max\{\sigma_w^2, \sigma_p^2\},$$

$$\lambda_{-1}(\tau + 1) - \lambda_{-1}(\tau) \geq s_1^{(i)}(\tau)(1 - s_1^{(i)}(\tau)) \cdot c\alpha\|\boldsymbol{\nu}\|_2\|\boldsymbol{\mu}\|_2^3 d \max\{\sigma_w^2, \sigma_p^2\},$$

*for any $i \in \mathcal{C}$. Additionally, for some constant $c' > c$, we have*

$$\lambda_{+1}(\tau + 1) - \lambda_{+1}(\tau) \leq s_1^{(i)}(\tau)(1 - s_1^{(i)}(\tau)) \cdot c'\alpha\|\boldsymbol{\nu}\|_2\|\boldsymbol{\mu}\|_2^3 d \max\{\sigma_w^2, \sigma_p^2\},$$

$$\lambda_{-1}(\tau + 1) - \lambda_{-1}(\tau) \leq s_1^{(i)}(\tau)(1 - s_1^{(i)}(\tau)) \cdot c'\alpha\|\boldsymbol{\nu}\|_2\|\boldsymbol{\mu}\|_2^3 d \max\{\sigma_w^2, \sigma_p^2\},$$

*for any $i \in \mathcal{C}$.*

*Proof of Lemma D.10.* We only provide the different part from Lemma D.6. Without loss of generality, we consider the case of $\lambda_{+1}(\tau)$. In the summation over $i \in [n]$ in Equation (423), we divide the terms into three groups: clean data $i \in \mathcal{C}_+$, noisy data $j \in \mathcal{N}_-$, and examples from a different class.

For clean data $i \in \mathcal{C}_+$, the conditions $A(\tau)$, $B(\tau)$, and $D(\tau)$ give the same results as Equations (433) and (451). For noisy data $j \in \mathcal{N}_-$, the ratios of softmax probabilities in $F(\tau)$ are significant. To derive the lower bound of signal updates, we use the results in $F(\tau)$; specifically, for any clean data $i \in \mathcal{C}$, $t \in [T] \setminus \{1\}$, and $u \in [T] \setminus \{2\}$, $s_t^{(j)}(\tau)(1 - s_t^{(j)}(\tau)) \lesssim \rho^{-1} s_1^{(i)}(\tau)(1 - s_1^{(i)}(\tau))$, $s_u^{(j)}(\tau)(1 - s_u^{(j)}(\tau)) \lesssim s_2^{(j)}(\tau)(1 - s_2^{(j)}(\tau))$, and the balance between $s_1^{(j)}(\tau)(1 - s_1^{(j)}(\tau))$ and $s_1^{(i)}(\tau)(1 - s_1^{(i)}(\tau))$. Under the conditions $A(\tau)$ and $B(\tau)$, we have the same lower bound of $j \in \mathcal{N}_-$ as in Equation (433) as follows:

$$I_{j,+}^W(\tau) = \sum_{t=1}^T s_t^{(j)}(\tau)\left(\gamma_t^{(j)} - \sum_{u=1}^T s_u^{(j)}(\tau)\gamma_u^{(j)}\right)\langle \mathbf{W}(\tau)\mathbf{x}_t^{(j)}, \mathbf{W}(\tau)\boldsymbol{\mu}_{+1}\rangle \tag{522}$$

$$= s_1^{(j)}(\tau)(1 - s_1^{(j)}(\tau)) \cdot \gamma_1^{(j)} \cdot \langle \mathbf{W}(\tau)\mathbf{x}_1^{(j)}, \mathbf{W}(\tau)\boldsymbol{\mu}_{+1}\rangle$$

$$- \sum_{t=2}^T s_1^{(j)}(\tau)s_t^{(j)}(\tau) \cdot \gamma_t^{(j)} \cdot \langle \mathbf{W}(\tau)\mathbf{x}_1^{(j)}, \mathbf{W}(\tau)\boldsymbol{\mu}_{+1}\rangle$$

$$+ \sum_{t=2}^T s_t^{(j)}(\tau)s_1^{(j)}(\tau) \cdot (\gamma_t^{(j)} - \gamma_1^{(j)}) \cdot \langle \mathbf{W}(\tau)\mathbf{x}_t^{(j)}, \mathbf{W}(\tau)\boldsymbol{\mu}_{+1}\rangle$$

$$+ \sum_{t=2}^T s_t^{(j)}(\tau) \sum_{u \in [T]\setminus\{1,t\}} s_u^{(j)}(\tau) \cdot (\gamma_t^{(j)} - \gamma_u^{(j)}) \cdot \langle \mathbf{W}(\tau)\mathbf{x}_t^{(j)}, \mathbf{W}(\tau)\boldsymbol{\mu}_{+1}\rangle \tag{523}$$

$$\geq s_1^{(j)}(\tau)(1 - s_1^{(j)}(\tau)) \cdot \gamma_1^{(j)} \cdot \Theta\left(\sigma_w^2\|\boldsymbol{\mu}\|_2^2 d\right)$$

$$- s_1^{(j)}(\tau)(1 - s_1^{(j)}(\tau)) \cdot O(\rho\gamma_1^{(j)}) \cdot \Theta\left(\sigma_w^2\|\boldsymbol{\mu}\|_2^2 d\right)$$

$$- s_1^{(j)}(\tau)(1 - s_1^{(j)}(\tau)) \cdot (1 + O(\rho))\gamma_1^{(j)} \cdot O\left(\rho\sigma_w^2\|\boldsymbol{\mu}\|_2^2 d\right)$$

$$- (T - 1) \cdot s_2^{(j)}(\tau)(1 - s_2^{(j)}(\tau)) \cdot O(\rho\gamma_1^{(j)}) \cdot O\left(\rho\sigma_w^2\|\boldsymbol{\mu}\|_2^2 d\right) \tag{524}$$

$$\gtrsim s_1^{(i)}(\tau)(1 - s_1^{(i)}(\tau)) \cdot \sigma_w^2\|\boldsymbol{\nu}\|_2\|\boldsymbol{\mu}\|_2^3 d, \tag{525}$$

where the second last line follows from $F(\tau)$ and $\rho < 1/C$ in the parameter assumptions. The effects of data with different

labels $k \in \mathcal{C}_- \cup \mathcal{N}_+$ are bounded similarly to Equation (442). From $B(\tau)$, we have

$$|\langle \mathbf{W}(\tau)\mathbf{x}_1^{(k)}, \mathbf{W}(\tau)\boldsymbol{\mu}_{+1}\rangle| = O\left(\max\left\{\|\boldsymbol{\mu}\|_2^2\sqrt{d}, \sigma_\epsilon\|\boldsymbol{\mu}\|_2 d\right\} \cdot \sigma_w^2 \log(Tn/\delta)\right) = O\left(\max\left\{\rho, \frac{\log(Tn/\delta)}{\sqrt{d}}\right\}\sigma_w^2\|\boldsymbol{\mu}\|_2^2 d\right),$$

(526)

$$|\langle \mathbf{W}(\tau)\mathbf{x}_t^{(k)}, \mathbf{W}(\tau)\boldsymbol{\mu}_{+1}\rangle| = O\left(\rho\sigma_w^2\|\boldsymbol{\mu}\|_2^2 d\right),$$

(527)

$$|\langle \mathbf{W}(\tau)\mathbf{x}_u^{(k)}, \mathbf{W}(\tau)\boldsymbol{\mu}_{+1}\rangle| = O\left(\max\left\{\rho\|\boldsymbol{\mu}\|_2^2\sqrt{d}, \sigma_\epsilon\|\boldsymbol{\mu}\|_2 d\right\} \cdot \sigma_w^2 \log(Tn/\delta)\right) = O\left(\rho\sigma_w^2\|\boldsymbol{\mu}\|_2^2 d\right),$$

(528)

$$|\langle \mathbf{W}(\tau)\mathbf{x}_v^{(k)}, \mathbf{W}(\tau)\boldsymbol{\mu}_{+1}\rangle| = O\left(\sigma_w^2\sigma_\epsilon\|\boldsymbol{\mu}\|_2 d \log(Tn/\delta)\right) = O\left(\rho\sigma_w^2\|\boldsymbol{\mu}\|_2^2 d\right),$$

(529)

for $t \in \mathcal{W}_{+1}^{(k)}$, $u \in \mathcal{W}_{-1}^{(k)}$, and $v \in \mathcal{I}^{(k)}$. Similarly to Equation (522), we have

$$\begin{aligned}
|I_{k,+}^W(\tau)| \leq{}& s_1^{(k)}(\tau)(1-s_1^{(k)}(\tau)) \cdot |\gamma_1^{(k)}| \cdot O\left(\max\{\rho, o(1)\}\sigma_w^2\|\boldsymbol{\mu}\|_2^2 d\right) \\
&+ s_1^{(k)}(\tau)(1-s_1^{(k)}(\tau)) \cdot O(\rho|\gamma_1^{(k)}|) \cdot O\left(\max\{\rho, o(1)\}\sigma_w^2\|\boldsymbol{\mu}\|_2^2 d\right) \\
&+ s_1^{(k)}(\tau)(1-s_1^{(k)}(\tau)) \cdot (1+O(\rho))|\gamma_1^{(k)}| \cdot O\left(\rho\sigma_w^2\|\boldsymbol{\mu}\|_2^2 d\right) \\
&+ (T-1) \cdot \max_{t\in[T]\setminus\{1\}}\left\{s_t^{(k)}(\tau)(1-s_t^{(k)}(\tau))\right\} \cdot O(\rho|\gamma_1^{(k)}|) \cdot O\left(\rho\sigma_w^2\|\boldsymbol{\mu}\|_2^2 d\right)
\end{aligned}$$

(530)

$$\lesssim s_1^{(i)}(\tau)(1-s_1^{(i)}(\tau)) \cdot \max\{\rho, o(1)\}\sigma_w^2\|\boldsymbol{\nu}\|_2\|\boldsymbol{\mu}\|_2^3 d,$$

(531)

for any clean data $i \in \mathcal{C}$, which follows from $F(\tau)$. The same discussion is applied to $|I_{k,+}(\tau)|$, and the desired result is obtained from the balance between the numbers of clean and noisy data. $\qquad\square$

**Lemma D.11** (Noise updates of clean data in Lemma C.13). *Let $T_1$ and $T_2$ be the time steps in Lemma C.13, and let $\tau \in [T_1, T_2]$. Suppose that the conditions in Theorem 4.1 and $A(\tau)$, $B(\tau)$, $D(\tau)$, and $F(\tau)$ in Lemma C.13 are satisfied. Then, on a good run, there exists some constant $c, c' > 0$ such that $c' > c$, and*

$$\begin{aligned}
\rho_{i,1}(\tau+1) - \rho_{i,1}(\tau) &\geq s_1^{(i)}(\tau)(1-s_1^{(i)}(\tau)) \cdot c\alpha n^{-1}\sigma_\epsilon^2\|\boldsymbol{\nu}\|_2\|\boldsymbol{\mu}\|_2 d^2 \max\{\sigma_w^2, \sigma_p^2\}, \\
\rho_{i,t}(\tau+1) - \rho_{i,t}(\tau) &\leq -s_1^{(i)}(\tau)s_t^{(i)}(\tau) \cdot c\alpha n^{-1}\sigma_\epsilon^2\|\boldsymbol{\nu}\|_2\|\boldsymbol{\mu}\|_2 d^2 \max\{\sigma_w^2, \sigma_p^2\}, \\
\rho_{i,1}(\tau+1) - \rho_{i,1}(\tau) &\leq s_1^{(i)}(\tau)(1-s_1^{(i)}(\tau)) \cdot c'\alpha n^{-1}\sigma_\epsilon^2\|\boldsymbol{\nu}\|_2\|\boldsymbol{\mu}\|_2 d^2 \max\{\sigma_w^2, \sigma_p^2\}, \\
\rho_{i,t}(\tau+1) - \rho_{i,t}(\tau) &\geq -s_1^{(i)}(\tau)s_t^{(i)}(\tau) \cdot c'\alpha n^{-1}\sigma_\epsilon^2\|\boldsymbol{\nu}\|_2\|\boldsymbol{\mu}\|_2 d^2 \max\{\sigma_w^2, \sigma_p^2\}.
\end{aligned}$$

*for any clean data $i \in \mathcal{C}$ and $t \in [T] \setminus \{1\}$. For any noisy data $j \in \mathcal{N}$, we have*

$$\begin{aligned}
\rho_{j,1}(\tau+1) - \rho_{j,1}(\tau) &\leq -s_1^{(j)}(\tau)(1-s_1^{(j)}(\tau)) \cdot c\alpha n^{-1}\sigma_\epsilon^2\|\boldsymbol{\nu}\|_2\|\boldsymbol{\mu}\|_2 d^2 \max\{\sigma_w^2, \sigma_p^2\}, \\
\rho_{j,1}(\tau+1) - \rho_{j,1}(\tau) &\geq -s_1^{(j)}(\tau)(1-s_1^{(j)}(\tau)) \cdot c'\alpha n^{-1}\sigma_\epsilon^2\|\boldsymbol{\nu}\|_2\|\boldsymbol{\mu}\|_2 d^2 \max\{\sigma_w^2, \sigma_p^2\},
\end{aligned}$$

*and*

$$\begin{aligned}
\rho_{j,2}(\tau+1) - \rho_{j,2}(\tau) &\geq s_2^{(j)}(\tau)(1-s_2^{(j)}(\tau)) \cdot c\rho\alpha n^{-1}\sigma_\epsilon^2\|\boldsymbol{\nu}\|_2\|\boldsymbol{\mu}\|_2 d^2 \max\{\sigma_w^2, \sigma_p^2\}, \\
\rho_{j,2}(\tau+1) - \rho_{j,2}(\tau) &\leq s_2^{(j)}(\tau)(1-s_2^{(j)}(\tau)) \cdot c'\rho\alpha n^{-1}\sigma_\epsilon^2\|\boldsymbol{\nu}\|_2\|\boldsymbol{\mu}\|_2 d^2 \max\{\sigma_w^2, \sigma_p^2\},
\end{aligned}$$

*and*

$$\begin{aligned}
\rho_{j,t}(\tau+1) - \rho_{j,t}(\tau) &\leq -s_2^{(j)}(\tau)s_t^{(j)}(\tau) \cdot c\rho\alpha n^{-1}\sigma_\epsilon^2\|\boldsymbol{\nu}\|_2\|\boldsymbol{\mu}\|_2 d^2 \max\{\sigma_w^2, \sigma_p^2\}, \\
\rho_{j,t}(\tau+1) - \rho_{j,t}(\tau) &\geq -s_2^{(j)}(\tau)s_t^{(j)}(\tau) \cdot c'\rho\alpha n^{-1}\sigma_\epsilon^2\|\boldsymbol{\nu}\|_2\|\boldsymbol{\mu}\|_2 d^2 \max\{\sigma_w^2, \sigma_p^2\},
\end{aligned}$$

*for any $t \in [T] \setminus \{1, 2\}$.*

*Proof of Lemma D.11.* The proof for clean data $i \in \mathcal{C}$ is essentially the same as the preceding proofs in Lemmas D.7 and D.9. The lower and upper bound of $I_{i,i,1}^W(\tau)$ and $I_{i,i,1}(\tau)$ is derived in the same way as Equations (475) and (488).

As for $I^W_{k,i,1}(\tau)$ and $I_{k,i,1}(\tau)$ for $k \neq i$, the difficulty arises because $s^{(i)}_1(\tau)(1 - s^{(i)}_1(\tau))$ does not dominate for noisy data $k \in \mathcal{N}$ like Equation (480). However, it is resolved using the closer analysis as in Equation (312). Specifically, using the $s^{(k)}_2(\tau)(1 - s^{(k)}_2(\tau)) \lesssim \rho^{-1} s^{(i)}_1(\tau)(1 - s^{(i)}_1(\tau))$ and the balance between $s^{(k)}_1(\tau)(1 - s^{(k)}_1(\tau))$ and $s^{(i)}_1(\tau)(1 - s^{(i)}_1(\tau))$, which are derived from $F(\tau)$, we have that for $k \neq i$,

$$|I^W_{k,i,1}(\tau)| \lesssim s^{(i)}_1(\tau)(1 - s^{(i)}_1(\tau)) \cdot o(n^{-1}\sigma^2_w\sigma^2_\epsilon\|\boldsymbol{\nu}\|_2\|\boldsymbol{\mu}\|_2 d^2), \tag{532}$$

for any $i \in \mathcal{C}$. Therefore, we obtain the desired inequalities for $\rho_{i,1}(\tau)$ following the same discussion as in Lemma D.7. The updates of $\rho_{i,t}(\tau)$ are also derived in a similar way to Lemmas D.7 and D.9.

For noisy data $j \in \mathcal{N}$, the inequalities for $I^W_{j,j,1}(\tau)$ and $I_{j,j,1}(\tau)$ are derived using a similar argument to Equation (525). Therefore, the analysis is reduced to that of clean data, and the desired results for $\rho_{j,1}(\tau)$ are obtained by flipping the sign of updates. We proceed with the analysis of $\rho_{j,t}(\tau)$, for $t \in [T] \setminus \{1\}$. From Lemma C.4, we have

$$\rho_{j,t}(\tau + 1) - \rho_{j,t}(\tau) = \frac{\alpha}{n}\sum_{k=1}^{n}(-\ell'_k(\tau)) \cdot Y^{(k)} \cdot \left(I^W_{k,j,t}(\tau) + \|\mathbf{p}(\tau)\|^2_2 I_{k,j,t}(\tau)\right) + \alpha^2 \boldsymbol{\epsilon}^{(j)\top}_t \nabla_{\mathbf{W}^\top}\widehat{\mathcal{L}}(\tau)\nabla_{\mathbf{p}}\widehat{\mathcal{L}}(\tau). \tag{533}$$

Without loss of generality, we consider $j \in \mathcal{N}_+ = \{i \in [n] \mid Y^{*(i)} = -1, Y^{(i)} = 1\}$. From the data model defined in Definition 3.1 and Lemma B.12, note that we have

$$\gamma^{(j)}_1 = -\Theta(\|\boldsymbol{\nu}\|_2\|\boldsymbol{\mu}\|_2), \ \gamma^{(j)}_2 = \Theta(\rho\|\boldsymbol{\nu}\|_2\|\boldsymbol{\mu}\|_2), \tag{534}$$

$$\gamma^{(j)}_t = -\Theta(\rho\|\boldsymbol{\nu}\|_2\|\boldsymbol{\mu}\|_2), \ |\gamma^{(j)}_u| = O(\sigma_\epsilon\|\boldsymbol{\nu}\|_2\sqrt{\log(Tn/\delta)}), \tag{535}$$

for $t \in \mathcal{W}^{(j)}_{-1}$ and $u \in \mathcal{I}^{(j)}$. Additionally, using $B(\tau)$, the current SNR condition $\mathrm{SNR}^2 = o(n^{-1})$ and the parameter assumptions, we have

$$\|\mathbf{W}(\tau)\boldsymbol{\epsilon}^{(j)}_t\|^2_2 = \Theta(\sigma^2_w\sigma^2_\epsilon d^2), \tag{536}$$

$$|\langle\mathbf{W}(\tau)\boldsymbol{\mu}_{+1}, \mathbf{W}(\tau)\boldsymbol{\epsilon}^{(j)}_t\rangle| = O(\sigma^2_w\sigma_\epsilon\|\boldsymbol{\mu}\|_2 d\log(Tn/\delta)) = o(n^{-1}\rho\sigma^2_w\sigma^2_\epsilon d^2), \tag{537}$$

$$|\langle\mathbf{W}(\tau)\boldsymbol{\mu}_{-1}, \mathbf{W}(\tau)\boldsymbol{\epsilon}^{(i)}_t\rangle| = O(\sigma^2_w\sigma_\epsilon\|\boldsymbol{\mu}\|_2 d\log(Tn/\delta)) = o(n^{-1}\rho\sigma^2_w\sigma^2_\epsilon d^2), \tag{538}$$

$$|\langle\mathbf{W}(\tau)\boldsymbol{\epsilon}^{(k)}_u, \mathbf{W}(\tau)\boldsymbol{\epsilon}^{(j)}_t\rangle| = O(\sigma^2_w\sigma^2_\epsilon d^{3/2}\log(Tn/\delta)) = o(n^{-1}\rho\sigma^2_w\sigma^2_\epsilon d^2), \tag{539}$$

for any $k \in [n]$ and $u \in [T]$ such that $(k, u) \neq (j, t)$. Therefore, when $k = j$, we have

$$I^W_{j,j,2}(\tau) = \sum_{v=1}^{T}s^{(j)}_v(\tau)\left(\gamma^{(j)}_v - \sum_{u=1}^{T}s^{(j)}_u(\tau)\gamma^{(j)}_u\right)\langle\mathbf{W}(\tau)\mathbf{x}^{(j)}_v, \mathbf{W}(\tau)\boldsymbol{\epsilon}^{(j)}_2\rangle \tag{540}$$

$$= s^{(j)}_2(\tau)(1 - s^{(j)}_2(\tau)) \cdot \gamma^{(j)}_2 \cdot \langle\mathbf{W}(\tau)\mathbf{x}^{(j)}_2, \mathbf{W}(\tau)\boldsymbol{\epsilon}^{(j)}_2\rangle$$

$$- s^{(j)}_2(\tau) \cdot \left(s^{(j)}_1(\tau)\gamma^{(j)}_1 + \sum_{u\in[T]\setminus\{1,2\}}s^{(j)}_u(\tau)\gamma^{(j)}_u\right) \cdot \langle\mathbf{W}(\tau)\mathbf{x}^{(j)}_2, \mathbf{W}(\tau)\boldsymbol{\epsilon}^{(j)}_2\rangle$$

$$+ \sum_{v=[T]\setminus\{2\}}s^{(j)}_v(\tau)s^{(j)}_2(\tau) \cdot (\gamma^{(j)}_v - \gamma^{(j)}_2) \cdot \langle\mathbf{W}(\tau)\mathbf{x}^{(j)}_v, \mathbf{W}(\tau)\boldsymbol{\epsilon}^{(j)}_2\rangle$$

$$+ \sum_{v=[T]\setminus\{2\}}s^{(j)}_v(\tau)\sum_{u\in[T]\setminus\{2,v\}}s^{(j)}_u(\tau) \cdot (\gamma^{(j)}_v - \gamma^{(j)}_u) \cdot \langle\mathbf{W}(\tau)\mathbf{x}^{(j)}_v, \mathbf{W}(\tau)\boldsymbol{\epsilon}^{(j)}_2\rangle \tag{541}$$

$$\geq s^{(j)}_2(\tau)(1 - s^{(j)}_2(\tau)) \cdot \Theta\left(\rho\|\boldsymbol{\nu}\|_2\|\boldsymbol{\mu}\|_2\sigma^2_w\sigma^2_\epsilon d^2\right)$$

$$- s^{(j)}_2(\tau) \cdot \left(-\rho(1 - s^{(j)}_2(\tau)) \cdot \Theta(\|\boldsymbol{\nu}\|_2\|\boldsymbol{\mu}\|_2) + (1 - s^{(j)}_2(\tau)) \cdot O(\sigma_\epsilon\|\boldsymbol{\nu}\|_2\sqrt{\log(Tn/\delta)})\right) \cdot \Theta(\sigma^2_w\sigma^2_\epsilon d^2)$$

$$- s^{(j)}_2(\tau)(1 - s^{(j)}_2(\tau)) \cdot o(\rho\|\boldsymbol{\nu}\|_2\|\boldsymbol{\mu}\|_2\sigma^2_w\sigma^2_\epsilon d^2) \tag{542}$$

$$\geq s^{(j)}_2(\tau)(1 - s^{(j)}_2(\tau)) \cdot c'_1\rho\sigma^2_w\sigma^2_\epsilon\|\boldsymbol{\nu}\|_2\|\boldsymbol{\mu}\|_2 d^2, \tag{543}$$

for some constant $c_1' > 0$. In the first inequality, we used $s_1^{(j)}(\tau) \lesssim \rho(1 - s_2^{(j)}(\tau))$, which is the result in $F(\tau)$, and $s_2^{(j)}(\tau) \geq \Theta(1)$ from $D(\tau)$. From the parameter assumptions, the first line and the first term of the second line in Equation (541) become dominant, leading to the result. Since $s_2^{(j)}(\tau)(1 - s_2^{(j)}(\tau))$ dominates $\rho^{-1} s_1^{(j)}(\tau)(1 - s_1^{(j)}(\tau))$ and $s_t^{(i)}(\tau)(1 - s_t^{(i)}(\tau))$ for any $t \in [T] \setminus \{1, 2\}$, which follows from $F(\tau)$, similarly to Equation (511), we have that for $k \neq j$,

$$|I_{k,j,2}^{W}(\tau)| < s_2^{(j)}(\tau)(1 - s_2^{(j)}(\tau)) \cdot o(n^{-1}\rho\sigma_w^2\sigma_\epsilon^2 \|\boldsymbol{\nu}\|_2 \|\boldsymbol{\mu}\|_2 d^2). \tag{544}$$

The arguments for $|I_{k,j,2}(\tau)|$ and the upper bounds follow similarly, and substituting them into Equation (533) yields the desired result for $\rho_{j,2}(\tau)$.

Finally, we analyze the update of $\rho_{j,t}(\tau)$ for $j \in \mathcal{N}_-$ and $t \in [T] \setminus \{1, 2\}$. Similarly, using Equations (536) to (539), there exists a constant $c_2'$ such that

$$I_{j,j,t}^{W}(\tau) = \sum_{v=1}^{T} s_v^{(j)}(\tau)\left(\gamma_v^{(j)} - \sum_{u=1}^{T} s_u^{(j)}(\tau)\gamma_u^{(j)}\right) \langle \mathbf{W}(\tau)\mathbf{x}_v^{(j)}, \mathbf{W}(\tau)\boldsymbol{\epsilon}_t^{(j)}\rangle \tag{545}$$

$$= s_2^{(j)}(\tau)(1 - s_2^{(j)}(\tau)) \cdot \gamma_2^{(i)} \cdot \langle \mathbf{W}(\tau)\mathbf{x}_2^{(j)}, \mathbf{W}(\tau)\boldsymbol{\epsilon}_t^{(j)}\rangle$$

$$- \sum_{u=[T]\setminus\{2\}} s_2^{(j)}(\tau)s_u^{(j)}(\tau) \cdot \gamma_u^{(j)} \cdot \langle \mathbf{W}(\tau)\mathbf{x}_2^{(j)}, \mathbf{W}(\tau)\boldsymbol{\epsilon}_t^{(j)}\rangle$$

$$+ \sum_{v\in[T]\setminus\{2\}} s_v^{(j)}(\tau)s_2^{(j)}(\tau) \cdot (\gamma_v^{(j)} - \gamma_2^{(j)}) \cdot \langle \mathbf{W}(\tau)\mathbf{x}_v^{(j)}, \mathbf{W}(\tau)\boldsymbol{\epsilon}_t^{(j)}\rangle$$

$$+ \sum_{v\in[T]\setminus\{2\}} s_v^{(j)}(\tau) \sum_{u\in[T]\setminus\{2,v\}} s_u^{(j)}(\tau) \cdot (\gamma_v^{(j)} - \gamma_u^{(j)}) \cdot \langle \mathbf{W}(\tau)\mathbf{x}_v^{(j)}, \mathbf{W}(\tau)\boldsymbol{\epsilon}_t^{(j)}\rangle \tag{546}$$

$$\leq s_2^{(j)}(\tau)(1 - s_2^{(j)}(\tau)) \cdot o\left(\rho\|\boldsymbol{\nu}\|_2\|\boldsymbol{\mu}\|_2 \cdot n^{-1}\rho\sigma_w^2\sigma_\epsilon^2 d^2\right)$$

$$+ s_2^{(j)}(\tau)(1 - s_2^{(j)}(\tau)) \cdot o\left(\rho\|\boldsymbol{\nu}\|_2\|\boldsymbol{\mu}\|_2 \cdot n^{-1}\rho\sigma_w^2\sigma_\epsilon^2 d^2\right)$$

$$+ s_2^{(j)}(\tau) \cdot s_t^{(j)}(\tau) \cdot (-\Theta(\rho\|\boldsymbol{\nu}\|_2\|\boldsymbol{\mu}\|_2)) \cdot \Theta(\sigma_w^2\sigma_\epsilon^2 d^2) \tag{547}$$

$$+ s_2^{(j)}(\tau)(1 - s_2^{(j)}(\tau) - s_t^{(j)}(\tau)) \cdot \left(-o\left(\|\boldsymbol{\nu}\|_2\|\boldsymbol{\mu}\|_2 \cdot n^{-1}\rho\sigma_w^2\sigma_\epsilon^2 d^2\right)\right)$$

$$+ s_t^{(j)}(\tau)s_1^{(j)}(\tau) \cdot \Theta(\|\boldsymbol{\nu}\|_2\|\boldsymbol{\mu}\|_2) \cdot \Theta(\sigma_w^2\sigma_\epsilon^2 d^2) \tag{548}$$

$$+ s_t^{(j)}(\tau)(1 - s_1^{(j)}(\tau) - s_2^{(j)}(\tau) - s_t^{(j)}(\tau)) \cdot (-\Theta(\rho\|\boldsymbol{\nu}\|_2\|\boldsymbol{\mu}\|_2)) \cdot \Theta(\sigma_w^2\sigma_\epsilon^2 d^2) \tag{549}$$

$$+ (1 - s_2^{(j)}(\tau) - s_t^{(j)}(\tau))^2 \cdot o\left(\|\boldsymbol{\nu}\|_2\|\boldsymbol{\mu}\|_2 \cdot n^{-1}\rho\sigma_w^2\sigma_\epsilon^2 d^2\right)$$

$$\leq -s_2^{(j)}(\tau)s_t^{(j)}(\tau) \cdot c_2'\rho\sigma_w^2\sigma_\epsilon^2\|\boldsymbol{\nu}\|_2\|\boldsymbol{\mu}\|_2 d^2. \tag{550}$$

In the first inequality, Equations (547) to (549) become dominant. By the choice of $T_1$ in Equation (521), $s_1^{(j)}(\tau) \lesssim \rho s_2^{(j)}(\tau)$ and Equation (548) is bounded by other two terms. This leads to the last line. The same argument is applied to the upper bound and inequalities for $I_{j,j,t}(\tau)$, we obtain the desired result. $\qquad\square$

# E. Further Discussion

### E.1. Multi-class Setting

In this section, we will see that the analysis in the multi-class setting is a straightforward extension of the binary setting studied in the main paper.

Let $K$ be the number of classes and $\mathbf{W}_V = (\boldsymbol{\nu}_1, \ldots, \boldsymbol{\nu}_K) \in \mathbb{R}^{d\times K}$ be the weight matrix. The model output is given by

$$f(\mathbf{X}) = \mathbf{W}_V^\top \mathbf{X}^\top \mathbb{S}\left(\mathbf{X}\mathbf{W}^\top \mathbf{p}\right) \in \mathbb{R}^K. \tag{551}$$

Let $\{\boldsymbol{\mu}_k\}_{k\in[K]}$ be signal vectors corresponding to each class, and we consider the data distribution $P$, which is modified for the multi-class setting from Definition 3.1. Here, we assume the orthogonality and norm equality among signal vectors. Furthermore, we assume the pretrained linear head satisfies $\cos\theta_k > \Theta(1)$, which is modified version of Equation (2), for $\boldsymbol{\nu}_k$ and the class signal $\boldsymbol{\mu}_k$, for all $k \in [K]$.

The concentration inequalities in Lemma B.1 are derived by taking a union bound for class numbers and appropriately updating the parameter assumptions to depend on $K$. The proof of Theorem 4.1 is based on the analysis of the empirical loss function on the training data $S$ sampled i.i.d. from $P$. The empirical loss function when using cross-entropy loss with softmax output is:

$$\widehat{\mathcal{L}}(\mathbf{W}, \mathbf{p}) = -\frac{1}{n} \sum_{i=1}^{n} \log \left( \frac{\exp \left( f(\mathbf{X}^{(i)})_{Y^{(i)}} \right)}{\sum_{k \in [K]} \exp \left( f(\mathbf{X}^{(i)})_k \right)} \right) \tag{552}$$

$$= \frac{1}{n} \sum_{i=1}^{n} \log \left( \sum_{k=1}^{K} \exp \left( (\boldsymbol{\nu}_k - \boldsymbol{\nu}_{Y^{(i)}})^\top \mathbf{X}^{(i)\top} \mathbb{S} \left( \mathbf{X}^{(i)} \mathbf{W}^\top \mathbf{p} \right) \right) \right). \tag{553}$$

The derivatives of the empirical loss function are given by

$$\nabla_{\mathbf{W}^\top} \widehat{\mathcal{L}}(\mathbf{W}, \mathbf{p})$$

$$= \frac{1}{n} \sum_{i=1}^{n} \frac{\sum_{k=1}^{K} \exp \left( (\boldsymbol{\nu}_k - \boldsymbol{\nu}_{Y^{(i)}})^\top \mathbf{X}^{(i)\top} \mathbb{S} \left( \mathbf{X}^{(i)} \mathbf{W}^\top \mathbf{p} \right) \right) \nabla_{\mathbf{W}^\top} (\boldsymbol{\nu}_k - \boldsymbol{\nu}_{Y^{(i)}})^\top \mathbf{X}^{(i)\top} \mathbb{S} \left( \mathbf{X}^{(i)} \mathbf{W}^\top \mathbf{p} \right)}{\sum_{l=1}^{K} \exp \left( (\boldsymbol{\nu}_l - \boldsymbol{\nu}_{Y^{(i)}})^\top \mathbf{X}^{(i)\top} \mathbb{S} \left( \mathbf{X}^{(i)} \mathbf{W}^\top \mathbf{p} \right) \right)} \tag{554}$$

$$= \frac{1}{n} \sum_{i=1}^{n} \sum_{k=1}^{K} \frac{\mathbf{X}^{(i)\top} \left( \mathrm{diag}(\mathbb{S}(\mathbf{X}^{(i)} \mathbf{W}^\top \mathbf{p})) - \mathbb{S}(\mathbf{X}^{(i)} \mathbf{W}^\top \mathbf{p}) \mathbb{S}(\mathbf{X}^{(i)} \mathbf{W}^\top \mathbf{p})^\top \right) \mathbf{X}^{(i)} (\boldsymbol{\nu}_k - \boldsymbol{\nu}_{Y^{(i)}}) \mathbf{p}^\top}{\sum_{l=1}^{K} \exp \left( (\boldsymbol{\nu}_l - \boldsymbol{\nu}_k)^\top \mathbf{X}^{(i)\top} \mathbb{S} \left( \mathbf{X}^{(i)} \mathbf{W}^\top \mathbf{p} \right) \right)}, \tag{555}$$

and

$$\nabla_{\mathbf{p}} \widehat{\mathcal{L}}(\mathbf{W}, \mathbf{p})$$

$$= \frac{1}{n} \sum_{i=1}^{n} \frac{\sum_{k=1}^{K} \exp \left( (\boldsymbol{\nu}_k - \boldsymbol{\nu}_{Y^{(i)}})^\top \mathbf{X}^{(i)\top} \mathbb{S} \left( \mathbf{X}^{(i)} \mathbf{W}^\top \mathbf{p} \right) \right) \nabla_{\mathbf{p}} (\boldsymbol{\nu}_k - \boldsymbol{\nu}_{Y^{(i)}})^\top \mathbf{X}^{(i)\top} \mathbb{S} \left( \mathbf{X}^{(i)} \mathbf{W}^\top \mathbf{p} \right)}{\sum_{l=1}^{K} \exp \left( (\boldsymbol{\nu}_l - \boldsymbol{\nu}_{Y^{(i)}})^\top \mathbf{X}^{(i)\top} \mathbb{S} \left( \mathbf{X}^{(i)} \mathbf{W}^\top \mathbf{p} \right) \right)} \tag{556}$$

$$= \frac{1}{n} \sum_{i=1}^{n} \sum_{k=1}^{K} \frac{\mathbf{W} \mathbf{X}^{(i)\top} \left( \mathrm{diag}(\mathbb{S}(\mathbf{X}^{(i)} \mathbf{W}^\top \mathbf{p})) - \mathbb{S}(\mathbf{X}^{(i)} \mathbf{W}^\top \mathbf{p}) \mathbb{S}(\mathbf{X}^{(i)} \mathbf{W}^\top \mathbf{p})^\top \right) \mathbf{X}^{(i)} (\boldsymbol{\nu}_k - \boldsymbol{\nu}_{Y^{(i)}})}{\sum_{l=1}^{K} \exp \left( (\boldsymbol{\nu}_l - \boldsymbol{\nu}_k)^\top \mathbf{X}^{(i)\top} \mathbb{S} \left( \mathbf{X}^{(i)} \mathbf{W}^\top \mathbf{p} \right) \right)}. \tag{557}$$

Since we have

$$\frac{-1}{\sum_{l=1}^{K} \exp \left( (\boldsymbol{\nu}_l - \boldsymbol{\nu}_k)^\top \mathbf{X}^{(i)\top} \mathbb{S} \left( \mathbf{X}^{(i)} \mathbf{W}^\top \mathbf{p} \right) \right)} = \frac{-1}{1 + \sum_{l \in [K] \setminus \{k\}} \exp \left( (\boldsymbol{\nu}_l - \boldsymbol{\nu}_k)^\top \mathbf{X}^{(i)\top} \mathbb{S} \left( \mathbf{X}^{(i)} \mathbf{W}^\top \mathbf{p} \right) \right)}, \tag{558}$$

this term corresponds to $\ell_i'$ in Equations (9) and (12). Under the scale condition of the linear head, this term becomes constant order as discussed in Lemma C.9. Additionally, using $\sum_{k=1}^{K} (\boldsymbol{\nu}_{Y^{(i)}} - \boldsymbol{\nu}_k) = K \left( \boldsymbol{\nu}_{Y^{(i)}} - \frac{1}{K} \sum_{k \in [K]} \boldsymbol{\nu}_k \right)$, we can confirm that Equations (555) and (557) correspond to Equations (9) and (12). The remaining proof is based on the equations of gradient updates and the results of concentration inequalities; therefore, the statement essentially does not change in the multi-class case.

### E.2. Head Optimization

In this section, we provide further discussion on Equation (2) in our problem setting. For the sake of generality, we consider the $K$-class classification setting following Appendix E.1.

The problem setting in Equation (2) specifies the alignment between the classification head $\boldsymbol{\nu}$ and the class signals. It is necessary to appropriately assign token scores, which represent the desirability of each token for solving the task, and to formulate the token selection problem. Since the analysis in this paper can generally be interpreted as an analysis of token selection dynamics under fixed token scores, our result may provide broader insights into attention mechanism across different problem settings, data and model architecture settings. However, it is an interesting question to what extent the problem setting in Equation (2) is justified from the perspective of practical scenarios.

In the following proposition, we show that the gradient direction of the expected risk at zero initialization of the weights forms an equiangular tight frame (ETF) with class vectors (Papyan et al., 2020). This corresponds to the most extreme case of the alignment condition between the linear head and the class vectors described in Equation (2). Note that the signal vectors have equal lengths and orthogonality from the definition of our data model. In Definition 3.1, we assumed that among the weakly relevant tokens, there was only one token for each class different from the true class, namely the confusing tokens. Here, we assume a more general setting that each weakly relevant token $\mathbf{x}_u$ aligns with a class $k$ uniformly sampled from $[K]$, that is, $\mathbf{x}_u = \rho \boldsymbol{\mu}_k + \epsilon_u$ for $u \in \mathcal{W}$.

**Proposition E.1.** *Suppose that the weights* $\mathbf{p}$, $\mathbf{W}$, *and* $\mathbf{W}_V$ *are initialized to zero. We also consider a generalized version of the data model in Definition 3.1, where each weakly relevant token aligns with a class $k$ uniformly sampled from $[K]$. Then, the gradient descent direction of the expected risk for $\mathbf{W}_V$ forms an ETF geometry consisting of the signal vectors, i.e., we have*

$$- (\nabla_{\mathbf{W}_V} \mathcal{L}(\mathbf{W}_V = \mathbf{0}))^\top \propto \left( \mathbf{I}_K - \frac{1}{K} \mathbf{1}_K \mathbf{1}_K^\top \right) (\boldsymbol{\mu}_1, \dots, \boldsymbol{\mu}_K)^\top, \tag{559}$$

*where* $\mathcal{L}(\mathbf{W}_V) := \mathbb{E}_{(\mathbf{X}, Y) \sim P} \left[ \widehat{\mathcal{L}}(\mathbf{W}_V) \right]$.

*Proof.* Taking the gradient of Equation (553), the gradient of expected loss at $\mathbf{W}_V = \mathbf{0}$ is given by:

$$-\nabla_{\boldsymbol{\nu}_k} \mathcal{L}(\mathbf{W}_V = \mathbf{0}) = -\mathbb{E}_{(\mathbf{X}, Y) \sim P} \left[ \sum_{k' \in [K]} \frac{\nabla_{\boldsymbol{\nu}_k} (f(\mathbf{X})_{k'} - f(\mathbf{X})_Y)}{\sum_{c \in [K]} \exp \left( (\boldsymbol{\nu}_c - \boldsymbol{\nu}_{k'})^\top \mathbf{X}^\top \mathbb{S} \left( \mathbf{X} \mathbf{W}^\top \mathbf{p} \right) \right)} \Bigg|_{\mathbf{W}_V = \mathbf{0}} \right] \tag{560}$$

$$= -\frac{1}{K} \mathbb{E}_{(\mathbf{X}, Y) \sim P} \left[ \frac{1}{T} \mathbf{X}^\top \mathbf{1} \cdot (1 - K \mathbf{1}_{k=Y}) \right] \tag{561}$$

$$= -\frac{1}{KT} \left( \frac{1}{K} (1 - K) \cdot \mathbb{E} \left[ \sum_t \mathbf{x}_t \mid Y = k \right] + \frac{K-1}{K} \mathbb{E} \left[ \sum_t \mathbf{x}_t \mid Y \neq k \right] \right) \tag{562}$$

$$= \frac{K-1}{K^2 T} \left( \mathbb{E} \left[ \sum_t \mathbf{x}_t \mid Y = k \right] - \mathbb{E} \left[ \sum_t \mathbf{x}_t \mid Y \neq k \right] \right), \tag{563}$$

where we changed the order of gradient and integral in the first line, and in the second equality, we denote by $\mathbf{1}_A$ the indicator function which returns 1 if the event $A$ is satisfied and otherwise returns 0. The third line follows that $Y^*$ is sampled from a uniform distribution over $[K]$, and label noise is added to different labels uniformly.

Let $\bar{\boldsymbol{\mu}}$ be the mean of class signals $\sum_{k \in [K]} \boldsymbol{\mu}_k / K$. We have

$$\mathbb{E} \left[ \sum_t \mathbf{x}_t \mid Y = k \right] = (1 - \eta) \mathbb{E} \left[ \sum_t \mathbf{x}_t \mid Y^* = k \right] + \frac{\eta}{K-1} \sum_{k' \in [K] \setminus \{k\}} \mathbb{E} \left[ \sum_t \mathbf{x}_t \mid Y^* = k' \right] \tag{564}$$

$$= (1 - \eta) \left( \boldsymbol{\mu}_k + \frac{|\mathcal{W}|}{T} \rho \bar{\boldsymbol{\mu}} \right) + \frac{\eta}{K-1} \sum_{k' \in [K] \setminus \{k\}} \left( \boldsymbol{\mu}_{k'} + \frac{|\mathcal{W}|}{T} \rho \bar{\boldsymbol{\mu}} \right) \tag{565}$$

$$= (1 - \eta) \boldsymbol{\mu}_k + \frac{\eta}{K-1} (K \bar{\boldsymbol{\mu}} - \boldsymbol{\mu}_k) + |\mathcal{W}| \rho \bar{\boldsymbol{\mu}} \tag{566}$$

$$= \left( 1 - \frac{K}{K-1} \eta \right) \boldsymbol{\mu}_k + \left( \frac{K}{K-1} \eta + |\mathcal{W}| \rho \right) \bar{\boldsymbol{\mu}}, \tag{567}$$

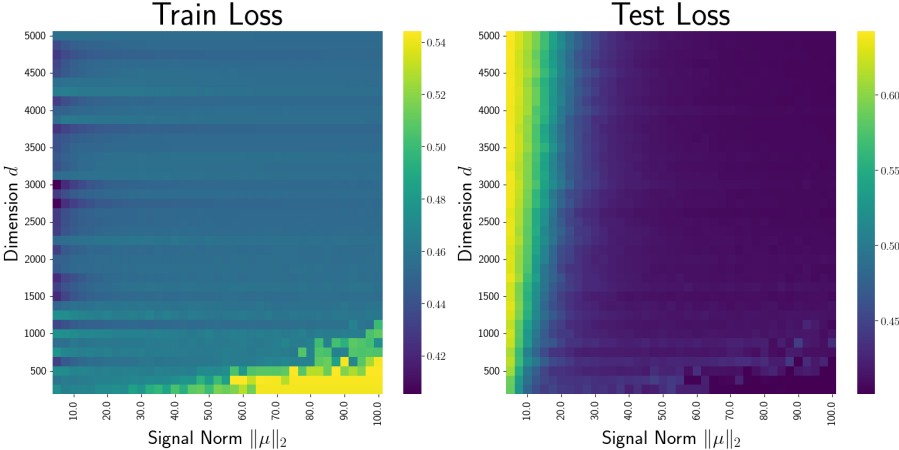

Figure 5: Heat-map of train loss and test loss under different signal norm $\|\boldsymbol{\mu}\|_2$ and the dimension $d$, after 1000 iterations. The yellow color indicates a higher loss value. The model and data follow the setup in our analysis.

where we used the definition of our data model. Therefore, Equation (563) leads to

$$
-\nabla_{\boldsymbol{\nu}_k}\mathcal{L}\left(\mathbf{W}_V=\mathbf{0}\right)=\frac{K-1}{K^2T}\left\{\left(1-\frac{K}{K-1}\eta\right)\boldsymbol{\mu}_k+\left(\frac{K}{K-1}\eta+|\mathcal{W}|\rho\right)\bar{\boldsymbol{\mu}}\right.
$$

$$
\left.-\frac{1}{K-1}\sum_{k'\in[K]\setminus\{k\}}\left(\left(1-\frac{K}{K-1}\eta\right)\boldsymbol{\mu}_{k'}+\left(\frac{K}{K-1}\eta+|\mathcal{W}|\rho\right)\bar{\boldsymbol{\mu}}\right)\right\} \tag{568}
$$

$$
=\frac{K-1}{K^2T}\cdot\left(1-\frac{K}{K-1}\eta\right)\left(\boldsymbol{\mu}_k-\frac{1}{K-1}\left(K\bar{\boldsymbol{\mu}}-\boldsymbol{\mu}_k\right)\right) \tag{569}
$$

$$
=\frac{K-1}{K^2T}\left(1-\frac{K}{K-1}\eta\right)\frac{K}{K-1}\left(\boldsymbol{\mu}_k-\bar{\boldsymbol{\mu}}\right). \tag{570}
$$

Consequently, we have $-\nabla_{\boldsymbol{\nu}_k}\mathcal{L}\left(\mathbf{W}_V=\mathbf{0}\right)\propto\boldsymbol{\mu}_k-\bar{\boldsymbol{\mu}}$, which concludes the proof. $\qquad\square$

This proposition is the result for the case of zero initialization. However, by taking the variance of the initialization sufficiently small, as in Assumption A8, similar alignment with the class signal can be achieved under random initialization.

While we have considered the gradient of the idealized expected loss, it is natural to ask about the gradient of the empirical loss. When using the same training set $S$ as in the main theorem, $\boldsymbol{\nu}$ contains the noise $\boldsymbol{\epsilon}_t^{(i)}$ from the training examples, and thus the arguments in Lemma B.12 cannot be applied, meaning that the same result does not hold. This is because noise memorization occurs even in the head, which is outside the focus of our study on token selection. On the other hand, if we use a different training set $S'$, it is possible to appropriately modify the argument and obtain results in the main theorem. Regarding the class signal, we can show that $\boldsymbol{\nu}^\top\boldsymbol{\mu}_k\gtrsim\|\boldsymbol{\mu}\|_2^2$ as in this proposition. However, when the input noise is large, specifically when $\sigma_\epsilon\sqrt{d}\gtrsim\|\boldsymbol{\mu}\|_2$, the condition $k\cdot\cos\theta_k>\Theta(1)$ in Equation (2) may no longer hold. Nevertheless, in the discussion of token scores Lemma B.12, it suffices to show that $\boldsymbol{\nu}^\top\boldsymbol{\mu}_k$ dominates $\boldsymbol{\nu}^\top\boldsymbol{\epsilon}_t^{(i)}$, which indeed holds under the parameter assumptions in Section 3.5.

## F. Additional Experimental Results

### F.1. Additional Synthetic Experiments

**Heat-map experiments.** We conducted another synthetic experiment when varying $d$ and $\|\boldsymbol{\mu}\|_2$. Figure 5 shows the train and test loss when varying the dimension $d$ and the signal norm $\|\boldsymbol{\mu}\|_2$ under the same setting as in Section 5 in the main text. This figure shows that the balance between the dimension and the signal norm is significant for achieving low train and test loss. Additionally, while our theoretical results in Theorem 4.1 propose the boundary between not-overfitting and benign

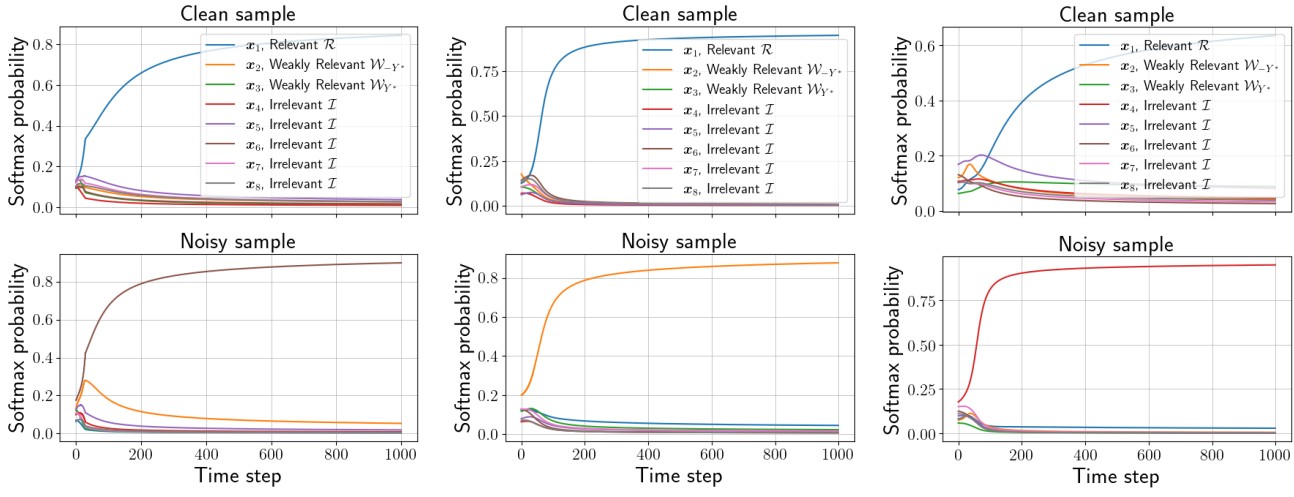

(a) Large noise setting: $d = 5000$, $\|\boldsymbol{\mu}\|_2 = 5$. Final training accuracy is $\mathbf{1.0}$ and test accuracy is $\mathbf{0.49}$ (Harmful overfitting).

(b) Balanced setting: $d = 2000$, $\|\boldsymbol{\mu}\|_2 = 20$. Final training accuracy is $\mathbf{1.0}$ and test accuracy is $\mathbf{0.99}$ (Benign overfitting).

(c) Large signal setting: $d = 1000$, $\|\boldsymbol{\mu}\|_2 = 100$. Final training accuracy is $\mathbf{1.0}$ and test accuracy is $\mathbf{1.0}$ (Benign overfitting).

Figure 6: Dynamics of softmax probability for one-layer transformer encoder. The top represents a clean sample, while the bottom represents a noisy sample. From left to right, the configurations of $d$ and $\|\boldsymbol{\mu}\|_2$ follow those used in Figure 3. While the right setting in Figure 3 corresponds to the not-overfitting case, it now shows benign overfitting.

overfitting as $\mathrm{SNR}^2 = \Theta(n^{-1})$, the training loss boundary in the left figure approximately forms a quadratic curve. This corresponds to that the ratio between $d$ and $\|\boldsymbol{\mu}\|_2^2$ remains constant at the boundary, with a fixed training size $n$.

**One-layer transformer encoder.**  We also conducted experiments under a more practical model setting using the same synthetic data. Specifically, we used a one-layer transformer encoder with a single-head attention. Compared to the model in the analysis, this model additionally incorporates non-linear feedforward layers, normalization, and skip-connections. The additional experiments here aim to provide further insights into the behavioral differences arising from joint optimization, rather than to support our analytical results. We first examined the token selection dynamics in the same way as in Figure 3, and the results are presented in Figure 6. The main findings are as follows:

- Similar benign overfitting was observed depending on the relationship between $d$ and $\|\boldsymbol{\mu}\|_2$. In a low-SNR setting, harmful overfitting was also observed in Figure 6a. However, Figure 6c shows that the model exhibited benign overfitting instead of not overfitting. This aligns with the intuition that increasing model capacity facilitates fitting to the training data.

- In the benign overfitting case, token selection for noisy samples progresses more rapidly. The bottom in Figure 6b shows that the softmax probability assigned to token $\mathbf{x}_2$, which most aligns to label noise, increases more rapidly than that of Figure 3b. The ability to learn token scores dynamically enables a cooperative interaction between the token selection mechanism and the feedforward layers, which could lead to faster token selection, as shown in the figure. Note also that, unlike the setting analyzed in the main text, fitting the label noise does not necessarily require selecting $\mathbf{x}_2$, and a different token is selected depending on the initialization.

Furthermore, Figure 7 shows the results of a heatmap-based experiment analogous to Figure 5. For the training loss, we observed very small values within the plotted range of $\|\boldsymbol{\mu}\|_2$ and $d$, which aligns with the observation of benign overfitting instead of not-overfitting in Figure 6c. As for the test loss, we obtained a similar boundary structure, but with clearer separation between the well-generalizing and poorly-generalizing regions. We suppose that this difference is due to the two main factors: i) the ability to learn output scaling and ii) the ability to distribute the effect of noise memorization not only within the attention mechanism but also to the feedforward layer.

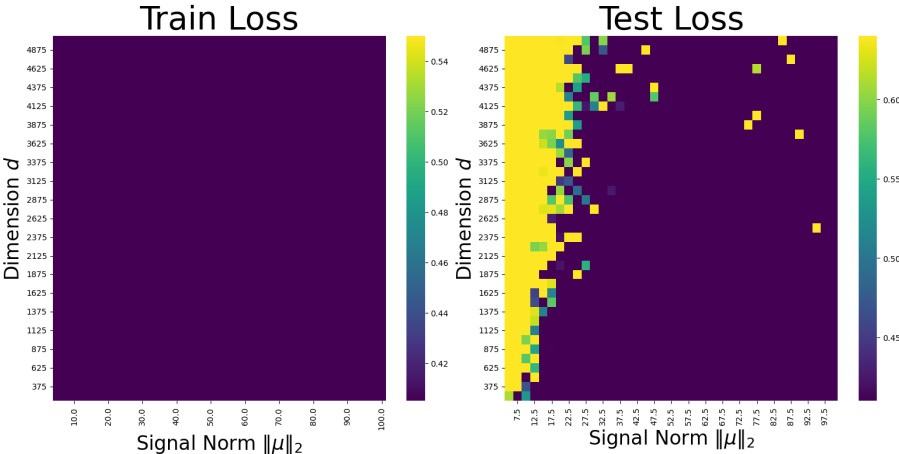

Figure 7: Heat-map of train loss and test loss under different signal norm $\|\boldsymbol{\mu}\|_2$ and the dimension $d$, after 1000 iterations. The yellow color indicates a higher loss value. The data follows the analytical setup in this paper, but the model used is a one-layer transformer encoder. For comparison with Figure 5, the range of the heat map is aligned with that of Figure 5.

Table 4: Details of image datasets.

| Dataset | Number of Class | Input Size | Training Size for Classifier Head |
|---|---|---|---|
| MNIST | 10 | $28 \times 28 \times 1$ | 10000 |
| CIFAR-10 | 10 | $32 \times 32 \times 3$ | 10000 |
| PneumoniaMNIST | 2 | $28 \times 28 \times 1$ | 3000 |
| BreastMNIST | 2 | $28 \times 28 \times 1$ | 200 |

Table 5: Details of natural language datasets.

| Dataset | Number of Class | Average Word Count | Training Size for Classifier Head |
|---|---|---|---|
| AG-news | 4 | 10 | 10000 |
| TREC | 6 | 10 | 3000 |

## F.2. Additional Information for Real-World Experiments

In this section, we provide the additional information for the experimental setup. We prepared the pre-trained ViT (Dosovitskiy et al., 2021) and BERT (Devlin et al., 2018) models using huggingface transformer library (Wolf et al., 2020). These models use the default configuration; specifically, they consist of 12 attention layers, the embedding dimension is set to 768, and the hidden dimension of the feed-forward layers is set to 3072. Dropout was not performed in any layers during the following training. Since the classifier head is initialized, we first train only the classifier head on a subset of the training data without label noise to align with Equation (2). The sizes of these sub-datasets vary due to the original dataset size, as summarized in Tables 4 and 5. The experiments in Section 5 and Appendix F.3 use training data that is a different split from the one used in the classifier head pertaining, and training is performed by initializing and updating only the attention weights in the final layer. As explained in Section 5, this setup is designed to align with our analytical setting as closely as possible, treating the pretrained model up to the final layer as a feature extractor, and training single-layer attention weights on top of these features. It should be noted, however, that the data model of course does not follow our analytical setups, and differences such as normalization, skip connections, and multi-head attention remain. During the experiments, the AdamW optimizer (Loshchilov & Hutter, 2019) without weight decay was used with a learning rate of $5e-5$, along with linear warmup and learning rate decay. The used datasets are described as follows.

**Image Dataset**   We conducted experiments on image classification on the following four datasets. The **MNIST** (LeCun et al., 2010) dataset consists of gray-scale $28 \times 28$ images with 10 classes. Each image is copied to form a 3-channel input and fed to the common image processor for the pre-trained ViT model. The **CIFAR-10** (Krizhevsky et al., 2009) is the

Table 6: Training loss and test accuracy when only the final attention query-key weights are trained on a sub-dataset for 2000 epochs, sufficiently long time. The results show the average over three different runs with the standard deviation.

(a) Label noise $\eta = 0.0$.

| Dataset | Eval | Training Size $n$ | | |
|---|---|---|---|---|
| | | 20 | 200 | 1000 |
| MNIST | train | $0.00_{\pm 0.00}$ | $0.00_{\pm 0.00}$ | $0.00_{\pm 0.00}$ |
| | test | $90.8_{\pm 1.5}$ | $91.8_{\pm 0.1}$ | $93.9_{\pm 0.1}$ |
| CIFAR-10 | train | $0.00_{\pm 0.00}$ | $0.00_{\pm 0.00}$ | $0.00_{\pm 0.00}$ |
| | test | $96.0_{\pm 0.1}$ | $95.9_{\pm 0.1}$ | $96.3_{\pm 0.1}$ |
| Pneumonia MNIST | train | $0.00_{\pm 0.00}$ | $0.00_{\pm 0.00}$ | $0.00_{\pm 0.00}$ |
| | test | $80.4_{\pm 3.1}$ | $79.0_{\pm 2.0}$ | $81.5_{\pm 0.9}$ |
| Breast MNIST | train | $0.08_{\pm 0.02}$ | $0.12_{\pm 0.01}$ | − |
| | test | $74.4_{\pm 1.4}$ | $78.4_{\pm 1.1}$ | − |
| AG-news | train | $0.00_{\pm 0.00}$ | $0.00_{\pm 0.00}$ | $0.00_{\pm 0.00}$ |
| | test | $84.0_{\pm 0.3}$ | $85.9_{\pm 0.5}$ | $85.5_{\pm 0.3}$ |
| TREC | train | $0.00_{\pm 0.00}$ | $0.00_{\pm 0.00}$ | $0.00_{\pm 0.00}$ |
| | test | $81.4_{\pm 0.3}$ | $78.7_{\pm 0.5}$ | $82.8_{\pm 0.4}$ |

(b) Label noise $\eta = 0.2$.

| Dataset | Eval | Training Size $n$ | | |
|---|---|---|---|---|
| | | 20 | 200 | 1000 |
| MNIST | train | $0.00_{\pm 0.00}$ | $0.04_{\pm 0.01}$ | $0.14_{\pm 0.02}$ |
| | test | $84.0_{\pm 7.2}$ | $87.3_{\pm 1.2}$ | $88.4_{\pm 0.1}$ |
| CIFAR-10 | train | $0.18_{\pm 0.07}$ | $0.18_{\pm 0.01}$ | $0.33_{\pm 0.02}$ |
| | test | $95.5_{\pm 0.4}$ | $94.3_{\pm 0.3}$ | $93.2_{\pm 0.1}$ |
| Pneumonia MNIST | train | $0.03_{\pm 0.04}$ | $0.02_{\pm 0.00}$ | $0.07_{\pm 0.00}$ |
| | test | $79.2_{\pm 7.0}$ | $81.2_{\pm 2.2}$ | $82.5_{\pm 0.7}$ |
| Breast MNIST | train | $0.12_{\pm 0.03}$ | $0.17_{\pm 0.02}$ | − |
| | test | $74.8_{\pm 2.0}$ | $76.3_{\pm 2.3}$ | − |
| AG-news | train | $0.00_{\pm 0.00}$ | $0.00_{\pm 0.00}$ | $0.00_{\pm 0.00}$ |
| | test | $82.5_{\pm 0.7}$ | $77.8_{\pm 0.6}$ | $69.6_{\pm 1.1}$ |
| TREC | train | $0.24_{\pm 0.33}$ | $0.11_{\pm 0.01}$ | $0.12_{\pm 0.05}$ |
| | test | $79.2_{\pm 2.4}$ | $73.3_{\pm 0.8}$ | $69.1_{\pm 1.7}$ |

dataset composed of $32 \times 32$ color images in 10 classes. These classes are mainly made up of vehicles and animals. Finally, we focused on the MedMNIST (Yang et al., 2023) dataset, specifically using **PneumoniaMNIST** and **BreastMNIST**, which are tasks for disease detection. Both datasets consist of $28 \times 28$ gray-scale images similar to MNIST, and they are binary classification settings based on the presence or absence of disease. Table 4 summarizes the details of these datasets.

Furthermore, we provide comments on how our data model, based on relevant, weakly relevant, and irrelevant tokens defined in Definition 3.1, corresponds to the input images. For example, consider the case of cancer detection using medical images such as MedMNIST. In this case, the patches directly indicating cancer, such as tumors or lesions, can considered relevant tokens. The patches showing enlarged lymph nodes or inflammatory signs, which often co-occur with cancer but are not definite, can be categorized as weakly relevant tokens. Meanwhile, the patches showing normal tissue or background anatomy, which are irrelevant to the task, correspond to irrelevant tokens.

**Natural Language Dataset** We conducted experiments on sentence classification in natural language in addition to image data. The **AG-news** (Zhang et al., 2015) is the dataset for the topic classification of news articles. It has four largest classes: "world", "sports", "business", and "science/technology". The **TREC** (Li & Roth, 2002) dataset is the dataset for question classification in 6 classes. Each question is labeled based on the content and the question type. The details of these datasets are summarized in Table 5. Note that the text data can also correspond to the data model in the paper based on the relevance of each word to the class, similar to the case of images.

### F.3. Additional Real-World Experiments

In this section, we provide additional results under the real-world experiment setting described in Appendix F.2.

Table 6 presents results corresponding to the experiment in Table 2 of the main text, where the label noise level $\eta = 0.1$ is varied to $\eta = 0.0$ and $\eta = 0.2$. In the absence of label noise, the model fits the data across all settings except for BreastMNIST. Notably, in the case of AG-news, where increasing the sample size $n$ under $\eta = 0.1$ led to worse test accuracy, no such accuracy degradation is observed when $\eta = 0.0$. The case of $\eta = 0.2$ shows similar trends to those observed with $\eta = 0.1$ in the main text.

Next, Figure 8 presents the results as the label noise ratio is varied continuously from $\eta = 0.0$ to $0.2$. In datasets such as CIFAR10 and TREC, the training loss increases as the label noise increases, whereas in MNIST, PneumoniaMNIST, and AG-news, the model maintains relatively good fitting across all noise levels. Regarding test loss, we observe that language

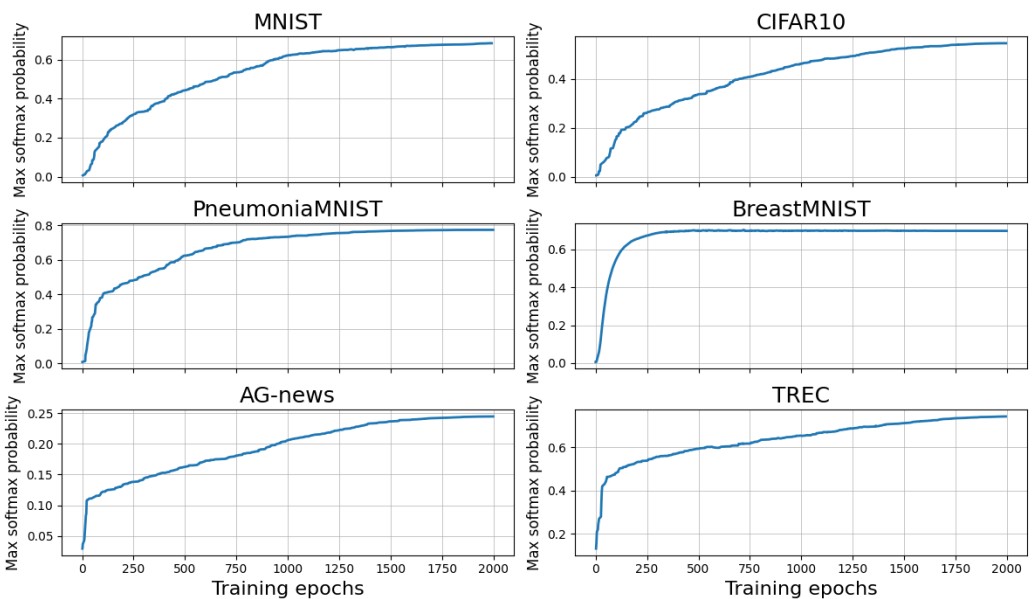

Figure 8: Heat-map of train loss and test loss when the label noise ratio $\eta$ is varied from $0.0$ to $0.2$. The training size $n$ is 200, and the training setup follows that in Appendix F.2. The yellow color indicates a higher loss value.

Figure 9: Dynamics of the maximum softmax probability at the position of the CLS token in the final attention layer. The experimental setup follows the real-world experiment described in Appendix F.2, with a label noise level $\eta = 0.1$.

datasets such as AG-news and TREC are relatively more difficult to predict, and the increase in test loss is more severe as the noise ratio increases.

Finally, Figure 9 illustrates the evolution of the maximum softmax probability at the position of the CLS token in the real-world experimental setting. For all datasets, the softmax probability increases as training progresses. In MNIST and PneumoniaMNIST, the model tends to focus on a single token, whereas in other datasets, attention does not necessarily concentrate on a single token. This behavior may be attributed to differences in the data distribution and architecture, such as skip connections and non-linear feed-forward layers, which are not captured by our analytical setting.

