# OpenReview forum: "Benign Overfitting in Token Selection of Attention Mechanism"
_ICML.cc/2025/Conference — ICML 2025 poster_

### Official Review · Reviewer_BNsU · 2025-03-11

**Overall Recommendation:** 4

**Summary:**

This paper explores benign overfitting in the token selection process of attention mechanisms, analyzing how transformers generalize despite fitting noisy training labels. It adopts feature learning framework to explain when models ignore noise (high SNR) or fit it while still generalizing well. Through theoretical analysis and experiments on both synthetic and real-world datasets, the paper demonstrates how training dynamics influence token selection.

**Claims And Evidence:**

This paper is well written, with rigorous theoretical analysis and clear experimental evidence.

**Essential References Not Discussed:**

_

**Experimental Designs Or Analyses:**

The experiment results further supported the theory.

**Methods And Evaluation Criteria:**

The methods and evaluation criteria makes sense for the problem.

**Other Comments Or Suggestions:**

See above.

**Other Strengths And Weaknesses:**

Several comments:

[1] It would be beneficial for the authors to conduct experiments with varying levels of noise in real-world datasets to further validate their findings. Additionally, a heatmap visualization on real data would provide valuable insights into the model’s behavior under different noise conditions.

[2] While the authors aim to introduce a brief proof outline in Section 4.1, the presentation feels more like a summary of multiple results. A more detailed explanation would enhance clarity. I recommend adding a section in the Appendix to provide an overview of the proof, including key derivations. For instance, is it possible to give a short discussion on how is the function  $g(x) = 2x + 2 \sinh(x - \log T)$  developed?

**Questions For Authors:**

I notice that in this paper, the authors assume that the number of iterations $T$ can grow exponentially. I am curious about the potential impact of this assumption on the conclusions. Since $\log T$ can be polynomial, are the results dependent on the exponential property in the loss function? Additionally, do the proofs establish that certain desirable properties hold for all t, allowing $\log T$ to grow polynomially instead?

**Relation To Broader Scientific Literature:**

I appreciate the authors efforts to compare their results with existing works and highlight their contributions.

**Theoretical Claims:**

The theoretical analysis is rigorous. By adopting feature learning framework, the authors analyzed the training dynamic systems and give the description of growth rate of the attention. I believe the rate is analyzed from some implicit equations but I did not check the details.

---

> ### Author Rebuttal · Authors · 2025-03-31
>
> Thank you for your detailed review. We appreciate that you highly evaluate our work.
>
> ---
> > 1: It would be beneficial for the authors to conduct experiments with varying levels of noise in real-world datasets to further validate their findings. Additionally, a heatmap visualization on real data would provide valuable insights into the model’s behavior under different noise conditions.
>
> In response to your comment, **we are making the following additions to Section G.2**:
> 1. Table similar to Table 2, presenting results for different label noise levels $\eta = 0$ and $0.2$ in addition to the $\eta = 0.1$ results in the main text
> 2. Heatmap illustrating how accuracies change as the label noise $\eta$ varies continuously within the range $[0, 0.2]$ for each dataset.
> While these results do not directly serve as an experimental validation of the SNR boundary in the theorem, we believe that they provide additional insights into the behavior of real-world datasets.
>
> ---
> > 2: While the authors aim to introduce a brief proof outline in Section 4.1, the presentation feels more like a summary of multiple results. A more detailed explanation would enhance clarity. I recommend adding a section in the Appendix to provide an overview of the proof, including key derivations. For instance, is it possible to give a short discussion on how is the function $g(x)=2x+2\sinh⁡(x−\log⁡T)$ developed?
>
> To address your concern and enhance the motivation, we have made the following improvements in Section 4.1 and Appendix:
> - To prevent the abrupt start of Section 4.1, **we have added the following overview after left column, line 308**:
> 	  "In the analysis of benign overfitting, it is necessary to track the model behavior on the training data while also its generalization ability. We first present the result of training dynamics, and the generalization result is shown at the end of this section in Lemma 4.8."
> - **We have clarified the reason for introducing Definition 4.4** by adding the following sentence after left column, line 310:
> 	  "This quantity is useful for evaluating the softmax probability at each time step $\tau$ for a given training example."
> - **We have added the motivation of the function $g(x)$ at line 329** as follows:
> 	  "This function naturally arises when expressing the evolution of the attention gap using the weight updates in Equations 4 and 5 and evaluating the dynamics via the quadrature method. Please refer to the Appendix for further details on the derivation."
> - **We have added a brief explanation of how the dynamics of the attention gap are derived after Lemma 4.5**:
> 	  "This lemma is shown by tracking the gradient descent dynamics and conducting induction argument with several desirable properties."
> - **We have added a new section titled "Proof Sketch" before Section C** to provide an overview of the proof structure and the motivation behind our analytical approach.
>
> The motivation for using the function $g(x) = 2x + 2\sinh(x - \log T)$, where $T$ is the sequence length of the inputs, naturally arises in the analysis of token selection dynamics. Specifically, the parameter updates are governed by the softmax probability term $s(\tau)(1 - s(\tau))$, which can be expressed in terms of the "attention gap" in Definition 4.4 and the equation $(2 + 2\cosh(x - \log T))^{-1}$ by Lemma E.1 and E.2. The function $g(x)$ appears naturally when applying the quadrature method to analyze the training dynamics (please refer to the discussion around Equation 96).
>
> ---
> > 3: I notice that in this paper, the authors assume that the number of iterations $T$ can grow exponentially. I am curious about the potential impact of this assumption on the conclusions. Since $\log ⁡T$ can be polynomial, are the results dependent on the exponential property in the loss function? Additionally, do the proofs establish that certain desirable properties hold for all t, allowing $\log T$ to grow polynomially instead?
>
> Thank you for your question. We answer the questions as follows.
> 1. As you correctly pointed out, the main theorem states that the required time steps for benign overfitting is in an exponential order. In our paper, please note that $T$ is used to denote sequence length rather than time steps.
> 2. This exponential time steps arises not from the shape of the loss function but from the softmax function in attention mechanism. We provide this intuition around line 376 in the main text.
> 3. As you noted, Lemmas D.11, D.12, and D.13 in the Appendix use an inductive argument to establish that some desired properties hold for all time steps. In particular, Proposition $C(\tau)$ implies that the evolution of the attention gap, as defined in Definition 4.4, follows a logarithmic order. Consequently, the number of time steps required for generalization scales exponentially.

---

### Official Review · Reviewer_vUF4 · 2025-03-13

**Overall Recommendation:** 3

**Summary:**

This paper presents theoretical analysis of benign overfitting in token selection of the attention mechanism focusing on the training dynamics and the generalization performance. The author shows that under some conditions based on signal-to-noise ratio, "benign overfitting" phenomenon occurs in the attention mechanism.

**Claims And Evidence:**

Both the theoretical analysis and experiments clearly support the claims.

**Essential References Not Discussed:**

No more.

**Experimental Designs Or Analyses:**

In the real-world experiments, the comparison among "not overfitting", "benign overfitting", and "harmful overfitting" is not shown. Maybe it needs more explanation and experiments for harmful overfitting.

**Methods And Evaluation Criteria:**

Yes.

**Other Comments Or Suggestions:**

No more.

**Other Strengths And Weaknesses:**

Strength:

1. The paper understands the token selection ability of attention mechanism from the "benign overfitting" aspect.

2. The mathematical analysis and proofs are solid and clear.

3. The paper proposes a novel method of using an auxiliary function $g(x)$ to describe the training of attention gap. This method successfully deals with the difficulty caused by the softmax probabilities.

Weakness:

1. The model may be a bit simple, involving only one attention layer without any additional nonlinearity. What if there is FFN and nonlinearity on top of the attention layer?

2. Fixing the value of $v$ and only training $W$ and $p$ during the training process is less challenging. Once $v$ is also included into the updating procedure, what will it influence the attention mechanism?

3. The assumptions in the paper seem very strong (A1-A8). It is not clear why we need these assumptions. It is recommended to contain some explanations about these assumptions.

4. The data distribution appears implausible due to exactly three distinct groups and a consistent scale $\rho$ that separates the relevant token from the weakly relevant token.

**Questions For Authors:**

Why does the author consider the function $g(x)$ (line 328) to analyze the attention gap? What's the motivation of utilizing this kind of function?

**Relation To Broader Scientific Literature:**

This paper addresses the attention mechanism from the perspective of benign overfitting, which will guide the use of attention-based models.

**Theoretical Claims:**

The proofs seem correct.

---

> ### Author Rebuttal · Authors · 2025-03-31
>
> Thank you for your detailed review. We hope that our answer has addressed all your concerns and pray this reply will change your decision.
>
> ---
> > 1: The comparison among not, benign, and harmful overfitting lacks in real-world experiments. It needs more explanation and experiments for harmful overfitting.
>
> The primary objective of real-world experiments is varying the training size $n$ to support the transition between the not overfitting and benign overfitting in Theorem 4.1. This is because we cannot control the values of $\\|\mu\\|_2$ and $d$ unlike synthetic experiments.
> However, Table 2 also provide insights into harmful overfitting. For instance, in the AG-news dataset, while the model perfectly fits the noisy samples, increasing noisy samples leads to a decline in test accuracy, which is a harmful overfitting scenario.
>
> In response to the review, **we are adding following experiments to Section G.2**: softmax dynamics in real-world settings similar to Figure 3 and new tables similar to Table 2 for different label noise $\eta$, to improve the generality of experimental results.
>
> ---
> > 2, 3: What if there is FFN and nonlinearity on top of the attention layer? Once $\nu$ is also trained, what will it influence the attention mechanism?
>
> If the FFN and nonlinearity are fixed, the problem remains a token selection problem given the token scores, and similar results hold. As explained below, **fixing the head $\nu$ is a necessary problem setting rather than an assumption or limitation of analysis**.
>
> Since our focus is analyzing benign overfitting in **the token selection mechanism**, jointly optimizing the head would unnecessarily expand the components enabling noise memorization and benign overfitting. Specifically, when we show benign overfitting, it is unclear whether this is because of the token selection inside softmax or simply a consequence of the linear model $\nu$, which has already been extensively studied (e.g., [Bartlett+, 2020]).
> Assumption 3.3 is necessary to formulate the problem of token selection given the assigned token scores. To clarify this point, **we have moved this part from Assumption to the Problem Setting**.
> Furthermore, please note that this setting is practically plausible in the area of parameter-efficient fine-tuning, such as prompt tuning and LoRA.
>
> Our study focuses on aligning with a more realistic data model instead of joint head optimization. We newly analyze the token selection in the presence of label noise (see Table 1), which highly complicates softmax dynamics by introducing different training directions within the same training run. Since the parameter updates depend on the softmax probabilities (Equations 8 and 11), addressing this difficulty requires carefully tracking and evaluating softmax dynamics, and a significant portion of our proof is dedicated to resolving this new challenge.
>
> ---
> > 4: The assumptions (A1-A8) seem very strong.
>
> In Section 3.5, before (A1-A8), we explicitly list and compare the assumptions used in prior works on benign overfitting. This comparison shows that our assumptions align with commonly used ones in the literature.
> While the assumptions may seem complicated, the essential part is the relationships among $d$, $\\|\mu\\|_2$, and $n$. It is common to express parameter assumptions using big-O notation ignoring logarithmic dependencies [Cao+, 2022; Jiang+, 2024], but we present them explicitly.
>
> Could you please share which specific assumption you find particularly strong compared to the existing studies? Based on your feedback, we can further elaborate on why it is necessary for our analysis.
>
> ---
> > 5: Three distinct token groups and a consistent scale $\rho$ are implausible.
>
> Building on previous research, we have carefully designed the analysis setting to better align with real-world scenarios.
>
> The existing benign overfitting works commonly assume that the input image is split to two parts: signal and noise [Cao+, 2022; Kou+, 2023]. Similarly, in the prior studies on the attention dynamics, it is often assumed that each input is made of a single optimal token and the other irrelevant noise vectors that are orthogonal to signal [Tarzanagh+, 2023b; Jiang+, 2024], as summarized in Table 1.
> Our study introduces several realistic elements to the analysis, including middle states termed "weakly relevant tokens" and non-orthogonality between signal and noise.
> To show the correspondence with real-world datasets and strengthen the plausibility of data model, we provided an example of medical imaging in the paragraph from line 4105.
>
> Finally, our analysis also holds if the scale $\rho$ varies across examples. **We have added this clarification after Definition 3.1.**
>
> ---
> > 6: What is the motivation for using $g(x)$?
>
> Due to the character limit of OpenReview, please refer to the second comment in Reviewer BNsU at the bottom. **We have made several updates to clarify the motivation throughout Section 4.1, not just for $g(x)$.**

---

> > ### Comment · Reviewer_vUF4 · 2025-04-06
> >
> > I appreciate your detailed response, which has addressed some of my concerns. However, I am still concerned about whether the findings can be extended to more complex models, such as multi-layer settings or incorporating additional nonlinear components (without fixing any parameters). Could the authors clarify whether their results hold in such settings, or provide experimental validation using these more realistic frameworks?
> >
> > ---
> >
> > Update (09 Apr 2025)
> >
> > Thank you for your detailed response and additional experiments in a more practical setting. Most of my concerns have been addressed, and I will increase my score to 3.

---

> > > ### Author Response · Authors · 2025-04-08
> > >
> > > Thank you for your further question. We are delighted that our reply has resolved some of your concerns. We hope that the following reply is fully satisfactory and will be reflected in your evaluation.
> > >
> > > ---
> > > > I am still concerned about whether the findings can be extended to more complex models, such as multi-layer settings or incorporating additional nonlinear components (without fixing any parameters). Could the authors clarify whether their results hold in such settings, or provide experimental validation using these more realistic frameworks?
> > >
> > > Based on your suggestion, **we have conducted additional experiments under a more practical model setting**.
> > > Specifically, we considered a one-layer Transformer encoder that includes non-linear feedforward layers and layer normalization.
> > > We conducted an experiment similar to Figure 3 to investigate the dynamics of token selection, and we observed the following:
> > > 1. Similar **benign overfitting was observed** depending on the relationship between the dimension $d$ and the signal strength $\\|\mu\\|_2$. In low-SNR settings, harmful overfitting was also observed (Figure 3(a)). However, under the same setting as in Figure 3(c), the model exhibited benign overfitting instead of not overfitting. This aligns with the intuition that increasing model capacity facilitates fitting to the training data.
> > > 2. In the benign overfitting case, **token selection for noisy samples progresses more rapidly**. The following table shows the dynamics of the softmax probability assigned to token $x_2$, which most aligns to label noise. The top row corresponds to the same run as Figure 3, b, below.
> > >
> > > | Model \ Time step (iteration) | 50 | 100 | 200 | 300 | 400 | 500 | ~ | 1000 |
> > > | - | - | - | -| - | - | - | - | - |
> > > | Model in our analysis  | 0.16 | 0.21 | 0.43 | 0.67 | 0.82 | 0.88 |  | 0.96 |
> > > | One-layer encoder | **0.44** | **0.74** | **0.86** | **0.89** | 0.89 | 0.91 |  | 0.93 |
> > >
> > > This result is consistent with the understanding from our analysis.
> > > Our theoretical analysis proves benign overfitting under a stricter setting, where only the attention mechanism can be trained. In contrast, the additional experiment here allows the training of feedforward layers and classifier, which makes noise memorization possible in broader components. The ability to learn token scores dynamically enables a cooperative interaction between the token selection mechanism and the feedforward layers, which could lead to faster token selection as shown in the table.
> > >
> > > Furthermore, **we have already conducted a heatmap-based experiment similar to Figure 4, and we have added this result to the appendix**.
> > > Regarding the training loss, we observed very small values within the range shown in the plot. For the test loss, **we observed a similar boundary structure**, and in the generalizing region, the loss was smaller than that in Figure 4. We suppose that this result is due to the two main factors: i) the ability to learn output scaling and ii) the ability to distribute the effect of noise memorization not only within the attention mechanism but also to the feedforward layer.
> > > While the experiments in the main paper, both synthetic and real-world, are designed to support our main theorem, the additional experiments here aim to provide further insights into the behavioral differences arising from joint optimization. We believe these new experiments not only address your concern but also strengthen the overall contribution of our paper.
> > >
> > > Benign overfitting provides a theoretical explanation for the empirical observation that over-parameterized models can generalize well despite memorizing the training data. However, due to the difficulty of analyzing training set fitting and generalization without relying on uniform convergence, the existing theoretical studies have been limited to simplified model settings. These include linear models [Bartlett+, 2020; Chatterji & Long, 2021] and two-layer NNs or CNNs with fixed second-layer weights [Frei+, 2022; Xu & Gu,2023; Meng+, 2024; Cao+, 2022; Xu+, 2024]. Bridging the gap between such theoretical settings and practically used models remains an important direction for future research on benign overfitting.
> > >
> > > ---
> > > **(Edit: 8 Apr, AoE)**
> > > Thank you very much for your detailed and constructive feedback. We have carefully addressed each of your comments, and we hope that our responses sufficiently resolve your concerns. We would be truly grateful if you could kindly reconsider the evaluation score. We sincerely appreciate the time and effort you have invested in reviewing our work.

---

### Official Review · Reviewer_ddf6 · 2025-03-14

**Overall Recommendation:** 2

**Summary:**

The paper studies benign overfitting in token selection within the attention mechanism, using a data model consisting of signal and noise and a one-layer attention model. This work characterizes the conditions under which benign overfitting, harmful overfitting, or no overfitting occurs.

## Update After Rebuttal
Most of my concerns were addressed during the rebuttal, and I increased my score to 2 (weak reject). However, I believe that additional reviews are necessary to determine whether the revised manuscript fully addresses the concerns raised by all reviewers. For this reason, I still lean toward rejection.

**Claims And Evidence:**

The main theorem statement is clearly presented. While I did not have time to read the full proof in the appendix, I find that the proof techniques described in the main text lack motivation. That is, it is difficult for the reader to understand the intuition behind the chosen proof techniques.

**Essential References Not Discussed:**

I believe the paper includes most of the relevant works on benign overfitting and the theoretical aspects of attention and transformers. However, I found an incorrect reference. In line 107 (left), Yun et al. (2020) is cited as a related work on the training dynamics of transformers. However, this paper does not consider training dynamics; instead, it shows that transformers are universal approximators. I recommend the authors carefully review their citations, as there may be other misreferenced works that I did not notice.

**Experimental Designs Or Analyses:**

I find the experimental results, especially the real-setting experiments, somewhat trivial and expected—showing that more data leads to better loss. I highly encourage the authors to include additional experimental results that better highlight the intuition behind their findings.

**Methods And Evaluation Criteria:**

N/A

**Other Comments Or Suggestions:**

* Running head should be fixed

* I suggest that the authors provide the gradient update equations for $W$ and$p$ for the convenience of readers. Since these equations are not included in the current draft, it is difficult to follow the technical details. Also, I encourage the authors to clarify the reason why the technical assumptions are required in their analysis.

**Other Strengths And Weaknesses:**

The problem setting is novel and more general than in previous works. Additionally, the comparison to prior studies (Table 1) is helpful. However, there are several weaknesses that should be addressed:

- Some assumptions are strong and not well-motivated. For example, Assumption 3.3, (A1)–(A8), and the first two lines in Theorem 4.1 seem overly restrictive, and their significance is unclear.
- The proof techniques lack motivation. Section 4.1 is difficult to follow because it primarily consists of lemmas without sufficient explanation or intuition in the current draft.

**Questions For Authors:**

- In Theorem 4.1, what happens in “Not Overfitting” case after the time step $\tau$?
- In Theorem 4.1, I think time step $\tau$ should depend on uncertainty $\delta$ but it seems there is no such dependency.
- Why does $g(x) = 2x+2\mathrm{sing h}(x-\log T)$ appeare in the proof techniques?

**Relation To Broader Scientific Literature:**

The paper studies benign overfitting in a more general data distribution and a more complex neural network architecture compared to previous works, such as Cao et al. (2022) and Oymak et al. (2023). This contribution broadens the understanding of benign overfitting in more general settings.

**Theoretical Claims:**

I did not check the full proof in the appendix and only read the proof techniques section in the main text. Due to its less-motivated presentation, I cannot confidently assess the correctness of the theorems.

---

> ### Author Rebuttal · Authors · 2025-03-31
>
> Thank you for your detailed review. We hope that our answer has addressed all your concerns and pray this reply will change your decision.
>
> ---
> > 1: The proof techniques lack motivation. Section 4.1 lacks sufficient explanation or intuition in the current draft.
>
> **We have added the following improvements to enhance motivation**.  Due to the character limit of OpenReview, please refer to the second comment from Reviewer BNsU at the bottom for details on the updates.
> - The overview of Section 4.1 after left column, line 308
> - The reason for introducing Definition 4.4 after left column, line 310
> - The motivation of the function $g(x)$ at line 329 (see also Comment 9 below)
> - The brief explanation of how the dynamics of the attention gap are derived after Lemma 4.5
> - The new section titled "Proof Sketch" before Section C
>
> ---
> > 2: The real-setting experiments are somewhat trivial and expected—showing that more data leads to better loss. The authors should include additional experimental results that better highlight the intuition behind their findings.
>
> Increasing the number of data containing label noise does not trivially lead to better loss. Depending on the dataset, fitting to label noise can result in worse generalization; for example, the AG-news dataset in Table 2 exhibits a harmful overfitting scenario.
> To further improve the generality of experimental results, **we are adding the following new experiments to Section G.2:** softmax dynamics in real-world settings similar to Figure 3 and new tables similar to Table 2 for different label noise $\eta$.
>
> ---
> > 3: Yun et al. (2020) is an incorrect reference.
>
> You are correct, and we have already fixed it. Additionally, we have reviewed the references again to ensure there are no errors.
>
> ---
> > 4: Assumption 3.3, (A1)–(A8), and the first two lines in Theorem 4.1 seem overly restrictive, and their significance is unclear.
>
> **Assumption 3.3**
> Our study is the analysis of benign overfitting in the **token selection mechanism**; thus, training the head $\nu$ is inappropriate as it obscures which part enables noise memorization and benign overfitting. Specifically, when we show benign overfitting, it is unclear whether this is because of the token selection inside softmax or simply a consequence of the linear model $\nu$, which has already been extensively studied (e.g., [Bartlett+, 2020]).
> Assumption 3.3 is necessary to formulate the problem of token selection given the assigned token scores.
> To clarify this point, **we have moved this part from Assumption to the Problem Setting**.
> Finally, **we have added a new proposition after Appendix E** showing that a one-step optimization from a randomly initialized $\nu$ satisfies Assumption 3.3. This result supports the validity of the setup to some extent.
>
> **(A1-A8)**
> In Section 3.5, we explicitly list and compare the assumptions used in prior works on benign overfitting. This comparison shows that our assumptions align with commonly used ones in the literature.
> Several papers express parameter assumptions using big-O notation ignoring logarithmic dependencies [Cao+, 2022; Jiang+, 2024], but we present them explicitly.
> Could you please share which specific assumption you find strong compared to the prior studies? Based on your feedback, we can further elaborate on why it is necessary for our analysis.
>
> **Assumption in Theorem 4.1**
> The assumption $\\|\nu\\|_2 = O(1 / \\|\mu\\|_2)$ is made to ensure that the token scores remain at most of constant order. This corresponds to appropriately scaling down the network output, and satisfying the assumption is straightforward.
>
> ---
> > 5: Running head should be fixed.
>
> Thank you for your comment. We've already fixed it.
>
> ---
> > 6: The authors should provide the gradient update equations for $W$ and $p$.
>
> For gradient updates, they are provided in Equations 8 and 11 in Appendix B due to space limitations in the main text. **We have added the following sentence after the gradient update equations in Section 3.4:**
> "The weight updates with specifically calculated loss gradients are provided in Appendix B."
>
> ---
> > 7: What happens in “Not Overfitting” case after the time step $\tau$?
>
> In the not-overfitting case, signal learning is dominant, and the model fits only the clean samples as stated in the theorem.
> This is discussed in detail in Figure 2, the paragraph from line 291, and Appendix D.2.
>
> ---
> > 8: Time step $\tau$ should depend on uncertainty $\delta$ but it seems there is no such dependency.
>
> Uncertainty $\delta$ indirectly affects the time step $\tau$ through the parameter assumptions (A1)-(A8).
> A similar order notation for the time step is also found in existing benign overfitting studies [Cao+, 2022; Jiang+, 2024].
>
> ---
> > 9 : Why does $g(x)=2x+2\sinh(x− \log⁡T)$ appear in the proof techniques?
>
> Due to character limit of OpenReview, please refer to the second comment from Reviewer BNsU at the bottom. **We have added an explanation in the main text after line 329**.

---

> > ### Comment · Reviewer_ddf6 · 2025-04-06
> >
> > Thank you for your response. Some of my concerns have been resolved, but I still have one remaining question.
> >
> > >> 7: What happens in the “Not Overfitting” case after the time step $\tau$?
> > > In the not-overfitting case, signal learning is dominant, and the model fits only the clean samples as stated in the theorem. This is discussed in detail in Figure 2, the paragraph from line 291, and Appendix D.2.
> >
> > I would like to clarify my understanding of this response. In the “Not Overfitting” case, is it correct that the training loss does not continue to decrease indefinitely, since the model does not fit the noisy samples?
> > In related literature, such as Kou et al. (2023), which considers a similar setting with noisy labels and Gaussian noise, it has been shown that models can eventually memorize the noisy data by fitting the noise. Could the authors clarify whether their setting fundamentally prevents this kind of memorization, or whether the training loss would continue to decrease if training were extended further?
> >
> > In addition, I would encourage the authors to consider improving the presentation of the paper in future versions, especially by providing more detailed discussion of the technical assumptions and key terms used throughout the text.
> >
> > Reference:
> > Kou et al., Benign Overfitting in Two-Layer ReLU Convolutional Neural Networks, ICML 2023.

---

> > > ### Author Response · Authors · 2025-04-07
> > >
> > > Thank you for your further questions. We sincerely hope that our response to the remaining questions below will fully resolve your concerns and be reflected in your evaluation of our paper.
> > >
> > > ---
> > > >  Could the authors clarify whether their setting fundamentally prevents this kind of memorization, or whether the training loss would continue to decrease if training were extended further?
> > >
> > > You are absolutely right that the model does not fit the noisy samples in the "Not Overfitting" case, and therefore the loss does not continue to decrease. The CNN analysis in [Kou+, 2023] demonstrates the memorization of noisy data, corresponding to our analysis in the "Benign Overfitting" case. We show that the model memorizes noisy samples by fitting the noise in this case (please see Figure 2, right).
> > >
> > > We emphasize that **our "Not Overfitting" case, characterized by $\text{SNR}^2 = \omega(n^{-1})$, is not within the scope of [Kou+, 2023]**. Specifically, Condition 4.1. 1 in their paper assumes a sufficiently large dimension $d$, imposing a stricter condition than our assumption (A1). Extracting the relationship among $d$, $n$, and $\\|\mu\\|_2$, their setting assumes $\text{SNR}^2 \lesssim n^{-1}$; thus, the regime we study in our "Not Overfitting" case: $\text{SNR}^2 = \omega(n^{-1})$ is not covered in their framework.
> > > Our results successfully demonstrate distinct token selection scenarios across different SNR regimes **under more general assumption on $d$**.
> > >
> > > ---
> > > > I would encourage the authors to consider improving the presentation of the paper in future versions, especially by providing more detailed discussion of the technical assumptions and key terms used throughout the text.
> > >
> > > Thank you for your additional suggestion. In the final version of the paper, we are allowed to include one extra page, and we have added the followings to address your comments on the technical assumptions and key terms.
> > >
> > > **Technical assumptions**
> > > - We have added the following comment after line 253, right:
> > > *"The assumption $\\|\nu\\| = O(1/\\|\mu\\|_2)$ in the theorem ensures that the token scores remain at most of constant order. This can be easily satisfied by appropriately scaling down the model output.”*
> > > - For paragraph starting line 177 in the right side, we elaborated on why the condition of head $\mu$ is required for analyzing benign overfitting in token selection, as explained in our previous reply. Due to the space limitations of OpenReview, we omit the specific update here.
> > > - To improve clarity, the discussion starting at line 189 in the right has been restructured into a new remark titled *"Relevance to practical scenarios".*
> > >
> > > **Key terms**
> > > - In response to your question about "Not overfitting", we added clarification after line 299 left:
> > > *"The middle in Figure 2 corresponds to the not overfitting case, where signal learning dominates and the model does not fit noisy data. The right figure illustrates the benign overfitting case, where noise memorization becomes dominant."*
> > > - To emphasize the inherent difficulty of the problem we study and highlight the novelty of our contributions, we added the following explanation after line 304, left:
> > > *"This makes analyzing the token selection dynamics inherently challenging. Our analysis under label noise setting must account for competing training directions between signal learning and noise memorization (as seen in Figure 2), as well as between clean and noisy samples for signal learning. These balances depend on softmax probabilities and are not determined by pre-training quantities such as SNR or label noise $\eta$. This is a specific difficulty to the attention mechanism, which is absent in existing benign overfitting studies. For instance, depending on the convergence speed—how quickly $s(\tau)$ approaches 0 or 1—it is possible that even when label noise $\eta$ is small, the actual contribution to the weight updates at some time step can be dominated by noisy samples. This motivates us to carefully analyze the dynamics of softmax probabilities to evaluate the direction of these competing relationships."*
> > >
> > > Regarding your initial concern about the less-motivated presentation of Section 4.1, we have already addressed this with concrete updates outlined in our first reply.
> > > In the revised version, we have improved our presentation to deliver our contributions and motivations as clearly as possible within the page limit, while including the problem setting, main results, proof essence, and experiments. If you have specific parts you find particularly inappropriate or unclear, we would be happy to revise further.
> > >
> > > ---
> > > **(Edit: 8 Apr, AoE)**
> > > Thank you very much for your detailed and constructive feedback. We have carefully addressed each of your comments, and we hope that our responses sufficiently resolve your concerns. We would be truly grateful if you could kindly reconsider the evaluation score. We sincerely appreciate the time and effort you have invested in reviewing our work.

---

### Official Review · Reviewer_ETcq · 2025-03-15

**Overall Recommendation:** 3

**Summary:**

The paper develops a theoretical framework to analyze the dynamics and generalization properties of token selection in attention mechanisms under label noise, focusing on a one-layer attention network for binary classification. It demonstrates that with a high signal-to-noise ratio (SNR), the model selectively fits clean samples and generalizes well (not overfitting), while with a low SNR, the model overfits noisy training data yet still achieves low test error—a phenomenon known as benign overfitting. The analysis introduces key concepts such as the evolution of softmax probabilities and the "attention gap" metric, which quantify how the mechanism distinguishes between relevant and noisy tokens during training, and it highlights a delayed generalization process reminiscent of grokking. Extensive experiments on both synthetic and real-world datasets support the theoretical findings by showcasing transitions between harmful overfitting, benign overfitting, and non-overfitting regimes.

**Claims And Evidence:**

The paper’s claims are largely supported by rigorous theoretical analysis and clear empirical evidence, with the real-world experiments being a particular highlight. However, when compared to Jiang et al. (2024), the paper appears relatively less impressive in terms of novelty and overall impact. Moreover, the comparison with Magen et al. (2024) is not sufficiently detailed; it would be beneficial to include a comprehensive comparison—preferably in Table 1—to clearly delineate the strengths and weaknesses of each approach in both the theoretical analysis and experimental validation.

**Essential References Not Discussed:**

I agree that the paper seems to have provided all the essential references needed to understand its context and contributions. The citations cover the key areas of benign overfitting, attention mechanism dynamics, and related analyses in transformers. No critical works appear to be missing.

**Experimental Designs Or Analyses:**

I examined the experimental designs for both the synthetic and real-world experiments. The synthetic experiments are well-designed to validate the theoretical predictions.

**Methods And Evaluation Criteria:**

The experiments are clear and can effectively support the theoretical results, with a primary focus on validating the theoretical claims.

**Other Comments Or Suggestions:**

I suggest that in Figure 1 the paper quantitatively indicate how large or small the values are (for example, by annotating the scales or ranges on the plot).

Additionally, in the second part of Theorem 4.1, it would be beneficial to explicitly state the lower bound for SNR to clarify the precise conditions under which benign overfitting occurs.

**Other Strengths And Weaknesses:**

Other strengths of the paper include its clear and well-organized presentation, as well as its rigorous approach that blends theoretical analysis with empirical validation. The work extends existing benign overfitting analyses to the context of attention mechanisms in transformers, which is a creative and timely direction given the widespread use of these models.

On the other hand, some weaknesses are apparent. The novelty is somewhat limited when compared to concurrent works such as Jiang et al. (2024) and Magen et al. (2024), which explore similar phenomena in vision transformers. Additionally, certain assumptions—like directly fixing $\nu$ (Assumption 3.3)—could be seen as a shortcut that may restrict the generality of the results. Clarifications regarding the definition of “good run” in the appendices and a more explicit delineation between benign and harmful overfitting regimes would also improve the paper's clarity and impact.

**Questions For Authors:**

What is the specific role of label noise in your training process, and how does its presence complicate the dynamics compared to a scenario without label noise?

**Relation To Broader Scientific Literature:**

Overall, this is an interesting topic with some novel findings. The paper builds on and extends prior work on benign overfitting—studied in linear regression, two-layer neural networks, and kernel methods (e.g., Bartlett et al., 2020; Hastie et al., 2022; Liang & Rakhlin, 2020)—by applying these ideas to the attention mechanism in transformers.

While the topic and findings are compelling, the contribution is somewhat diminished in comparison with recent works like Jiang et al. (2024) and Magen et al. (2024), which also explore similar phenomena in vision transformers.

**Theoretical Claims:**

I reviewed the theoretical proofs and did not find any major issues overall. However, I have two points for improvement. First, the appendices frequently refer to a “good run” without a precise definition; clarifying what qualifies as a good run would be beneficial. Second, while the paper distinguishes between benign and harmful overfitting, it would be clearer to provide an explicit formal threshold or result that delineates the boundary between these two regimes.

Additionally, I feel that directly assuming a fixed $\nu$ (as in Assumption 3.3) is somewhat of a “shortcut”. This assumption sidesteps the challenge of analyzing the full dynamics when $\nu$ is also learned, and may limit the generality of the results. Providing further justification or exploring a more general setting ould strengthen the paper’s theoretical contributions.

---

> ### Author Rebuttal · Authors · 2025-03-31
>
> Thank you for your detailed review. We hope that our answer has addressed all your concerns and pray this reply will change your decision.
>
> We first answer your question because it is an important point relating to our contribution.
>
> ---
> > 1: What is the specific difficulty of label noise setting?
>
> The presence of label noise introduces a significant challenge due to the existence of **competing training directions within the same training run**. This results in two key difficulties:
> 1. Signal learning vs memorization in each token selection (Figure 2).
> 2. Clean samples vs noisy samples in the learning direction of class signals​.
>
> These challenges become even more difficult because the weight updates depend on the softmax probability $s(\tau)$ (Eqs. 8 and 11) and are NOT statically determined by pre-training quantities such as SNR or label noise $\eta$. This is **a fundamental difficulty that does not appear in previous analyses of benign overfitting** (e.g., two-layer NN [Frei+, 2022]).
> For instance, depending on the convergence speed—how quickly $s(\tau)$ converges to $0$ or $1$—it is possible that even when label noise $\eta$ is very small, the actual contribution to weight updates at some time step can be dominated by noisy samples.
> Thus, it is crucial to track the whole training dynamics of each token to analyze 1 and 2, making the analysis inherently difficult. A large portion of this paper is dedicated to addressing and resolving this issue.
> In addition to the existing explanation from line 86, **we have added a new section titled "Difficulty of Label Noise Setting" in Appendix A**, incorporating an explanation similar to this reply.
>
> ---
> > 2: The novelty and impact are limited compared to [Jiang+, 2024].
>
> The above question highlights a challenge that is specific to the attention mechanism but is entirely absent in [Jiang+, 2024]. As discussed above, addressing this challenge is a major focus of our work and not a straightforward extension.
> Additionally, we discuss finer differences starting from line 627 in the Appendix.
>
> ---
> > 3: There is no sufficient comparison with [Magen+, 2024].
>
> We discuss the differences from line 143 and a more detailed difference from line 650 in the Appendix, which clarifies the novelty of our work.
> In response to the review, **we have added a row for [Magen+, 2024] in Table 1**.
> Furthermore, since the Appendix previously only provided a theoretical comparison, **we have also newly included a comparison of experimental validation**, as suggested.
>
> ---
> > 4: There is no definition of “good run”.
>
> We have already defined "good run" precisely in Definition C.2 and have confirmed that this term is not used prior to this definition, including the main text.
>
> ---
> > 5: There are some concerns about the threshold between benign and harmful overfitting.
>
> Assumption (A2) determines the lower bound on SNR for benign overfitting as $\Omega(d^{-1/4})$. **We have added a clarification regarding this point below Theorem 4.1.**
> Although this is not a continuous boundary, we discuss in Remark 4.2 that when $\text{SNR}^2 = o(d^{-1/2})$, the model exhibits harmful overfitting. This boundary also appears as a minimax generalization bound in [Xu and Gu, 2023].
> Furthermore, the boundary is demonstrated through synthetic experiments (Figure 4, right).
> In response to your comment, **we have explicitly annotated Figure 1 with the specific values** instead of "large" and "small".
>
> ---
> > 6: Fixing head $\nu$ as in Assumption 3.3 limits the generality of the results.
>
> Our study is the analysis of benign overfitting in the **token selection mechanism**; thus, enlarging the trainable components, including head $\nu$, unnecessarily would obscure which part enables noise memorization and benign overfitting. Specifically, when we show benign overfitting, it is unclear whether this is because of the token selection inside softmax or simply a consequence of the linear model $\nu$, which has already been extensively studied (e.g., [Bartlett+, 2020]). Our work establishes a novel result that benign overfitting can occur solely through the attention mechanism.
>
> Assumption 3.3 is necessary to formulate the problem of token selection given the assigned token scores, rather than being a limitation. To clarify this point, **we have moved this part from Assumption to the Problem Setting**.
> In this setting, our work is directed toward a more general token-selection problem instead of head training. We emphasize that analyzing token selection with label noise involves handling nontrivial softmax dynamics, dedicating a significant portion of our proofs to address this difficulty. The novelty and difficulty are explained in Comment 1 above and in Table 1 of the main text.
>
> Finally, in response to your comment, **we have added a new proposition after Appendix E** showing that a one-step optimization from a randomly initialized $\nu$ satisfies Assumption 3.3, which supports the validity of this setting to some extent.

---

> > ### Comment · Reviewer_ETcq · 2025-04-09
> >
> > Thank you for your detailed responses. I believe most of my concerns have been addressed, and I have accordingly increased my score to 3. I look forward to the revised version as promised by the authors.

---

### Decision · Program_Chairs · 2025-05-01

**Decision:**

Accept (poster)

**Comment:**

The authors present an analysis of the optimization and generalization dynamics of a two layer transformer in the high dimensional regime.  They demonstrate how the signal-to-noise ratio in the token generation model can determine whether or not the transformer exhibits benign overfitting (low SNR setting) or classical generalization (high SNR setting).

The reviews were mixed, with some reviewers having hesitation about the significance of the problem and with the improvement over Magen et al.  Magen et al's work was posted on arxiv on Oct 10, which was less than four months prior to the ICML submission deadline, and thus is a concurrent work and can't be used to rule against this.  On the one hand, I do think the results here could be improved by demonstrating whether or not *harmful* overfitting ever occurs in the high-SNR setting.  On the other, I think the current paper does advance our understanding of how attention mechanisms fit noisy examples, and how benign overfitting can occur with this architecture.  I think the positives of this paper clearly outweigh the negatives and I recommend acceptance.